# Reinforcement Learning under Latent Dynamics: Toward Statistical and Algorithmic Modularity

**Philip Amortila**[*]
philipa4@illinois.edu

**Dylan J. Foster**
dylanfoster@microsoft.com

**Nan Jiang**
nanjiang@illinois.edu

**Akshay Krishnamurthy**
akshaykr@microsoft.com

**Zakaria Mhammedi**
mhammedi@google.com

## Abstract

Real-world applications of reinforcement learning often involve environments where agents operate on complex, high-dimensional observations, but the underlying ("latent") dynamics are comparatively simple. However, outside of restrictive settings such as small latent spaces, the fundamental statistical requirements and algorithmic principles for reinforcement learning under latent dynamics are poorly understood.

This paper addresses the question of reinforcement learning under *general* latent dynamics from a statistical and algorithmic perspective. On the statistical side, our main negative result shows that *most* well-studied settings for reinforcement learning with function approximation become intractable when composed with rich observations; we complement this with a positive result, identifying *latent pushforward coverability* as a general condition that enables statistical tractability. Algorithmically, we develop provably efficient *observable-to-latent* reductions—that is, reductions that transform an arbitrary algorithm for the latent MDP into an algorithm that can operate on rich observations—in two settings: one where the agent has access to hindsight observations of the latent dynamics [LADZ23], and one where the agent can estimate *self-predictive* latent models [SAGHCB20]. Together, our results serve as a first step toward a unified statistical and algorithmic theory for reinforcement learning under latent dynamics.

## 1 Introduction

Many application domains for reinforcement learning (RL) require the agent to operate on rich, high-dimensional observations of the environment, such as images or text [WSD15; LFDA16; KFPM21; NRKFG22; Bak+22; Bro+22]. However, the environment itself can often be summarized by *latent dynamics* for a low-dimensional or otherwise simple latent state space. The decoupling of latent dynamics from the complex observation process naturally suggests a *modular* framework for algorithm design: first learn a representation that *decodes* the latent state from observations, then apply a reinforcement learning algorithm for the latent dynamics on top of the learned representation. This paper investigates the algorithmic and statistical foundations of this framework. We ask: *Can we take existing algorithms and sample complexity guarantees for reinforcement learning in the latent state space and lift them to the observation space in a modular fashion?*

There is a growing body of theoretical and empirical work developing algorithms that combine representation learning and reinforcement learning to develop scalable algorithms. On the empirical side, a plethora of representation learning objectives have been deployed to varying degrees of success [PAED17; Tan+17; ZMCGL21; LSA20; YFK21; Lam+24; Guo+22; HPBL23], but we lack

---

[*]The full (author-recommended) version of this paper can be found at: https://arxiv.org/pdf/2410.17904.

38th Conference on Neural Information Processing Systems (NeurIPS 2024).

a mathematical framework to systematically compare these objectives and understand when one might be preferred to another. On the theoretical side, all existing approaches suffer from three primary drawbacks: (a) they are tailored to restricted classes of latent dynamics models (tabular MDPs [KAL16; DKJADL19; MHKL20; ZSUWAS22; MFR23], LQR [DR21; Mha+20], or factored MDPs [MLJL21]), limiting generality; (b) the analyses, despite focusing on restrictive settings, are unwieldy, limiting progress in algorithm development; and (c) they are not *modular*, in the sense that the representation learning procedures are specialized to specific choices of latent reinforcement learning algorithm, limiting ease of use.

## 1.1 Contributions

We address the aforementioned limitations by introducing a new framework, *reinforcement learning under general latent dynamics*.

**Reinforcement learning under general latent dynamics (Section 2).**   In our framework, the agent performs control based on high-dimensional observations, but the dynamics of the environment are governed by an unobserved latent state space. Following prior work (particularly the so-called Block MDP formulation [DKJADL19]), we assume that the latent states can be *uniquely decoded from observations*, but that the true decoder is unknown and must be learned. To aid in the decoding process, we supply the learner with a class of representations that is *realizable* in the sense that it is powerful enough to represent the true decoder. Our point of departure from prior theoretical works is that we do not assume specific structure (e.g., tabular or linear dynamics) on the Markov decision process (MDP) that governs the latent dynamics. Instead, we make the minimal assumption that the latent dynamics belong to a *base MDP class which is statistically tractable*, in the sense that when the latent states are directly observed there exists *some* reinforcement learning algorithm with low sample complexity that is capable of learning a near-optimal policy for every MDP in the class. We take the first steps toward building a unified and modular theory for reinforcement learning in this setting.

**Contributions: Statistical modularity (Section 3).**   A central consideration for reinforcement learning under latent dynamics is that representation learning and exploration must be intertwined: an accurate decoder is required to explore the latent state space, but exploration is required to learn an accurate decoder. To develop provable sample complexity guarantees, one must prevent errors from compounding during this interleaving process, a challenging statistical problem which prior work addresses through strong structural assumptions on the base MDP [KAL16; DKJADL19; MHKL20; ZSUWAS22; MFR23; DR21; Mha+20; MLJL21]. For the general latent-dynamics setting we consider, it is unclear whether similar techniques can be applied, or whether the setting is even statistically tractable, ignoring computational considerations. Thus, our first contribution considers the question of *statistical modularity*:[2]

> *If a base MDP class is tractable when observed directly, is the corresponding latent-dynamics problem tractable?*

Statistical modularity adopts a minimax perspective by assuming that the base MDP lies in a given class, and demands that the sample complexity of the latent-dynamics setting is controlled by a natural bound on the sample complexity of the base MDP class. We show, perhaps surprisingly, that *most* well-studied reinforcement learning settings involving function approximation [RVR13; JKALS17; SJKAL19; MJTS20; AJSWY20; Li09; DVRZ19; WSY20; ZGS21; Du+21; JLM21; FKQR21] do not admit statistical modularity (Theorem 3.1). In other words, *statistical tractability of an MDP class does not extend to the latent-dynamics setting*. We complement these negative findings with a positive result, identifying *pushforward coverability* as a general structural condition on the latent dynamics that enables sample efficiency (Theorem 3.2).

**Contributions: Algorithmic modularity (Section 4).**   Beyond developing a modular understanding of the statistical landscape, we investigate *modular algorithm design principles* for RL under general latent dynamics. Specifically, we consider the question of *observable-to-latent reductions*, whereby RL under latent dynamics can be reduced to the simpler problem of RL with latent states directly observed:

*Can we generically lift algorithms for a base MDP class to solve the corresponding latent-dynamics problem?*

---

[2]This question and associated definitions are restated formally in Section 3.1.

This property, which we refer to as *algorithmic modularity*, enables modular, greatly simplified algorithm design, allowing one to use an arbitrary base algorithm for the base MDP class to solve the corresponding latent-dynamics problem. Algorithmic modularity is a stronger property than mere statistical modularity, and thus is subject to our statistical lower bound. Accordingly, we consider two settings that sidestep the lower bound through additional feedback and modeling assumptions. Our first algorithmic result considers *hindsight observability* [LADZ23], where latent states are revealed during training, but not at deployment (Theorem 4.1). Our second considers stronger function approximation conditions that enable the estimation of *self-predictive latent models* [SAGHCB20] through representation learning (Theorem A.1). Both results are *fully modular*: they transform *any* sample-efficient algorithm for the base MDP class into a sample-efficient algorithm for the latent-dynamics setting. Thus, they constitute the first *general-purpose* algorithms for RL under latent dynamics.

Together, we believe our results can serve as a foundation for further development of practical, general-purpose algorithms for RL under latent dynamics. To this end, we highlight a number of fascinating and challenging open problems for future research (Section 5).

## 2 Reinforcement Learning under General Latent Dynamics

In this section we formally introduce our framework, *reinforcement learning under general latent dynamics*.

**MDP preliminaries.** We consider an episodic finite-horizon online reinforcement learning setting. With $H$ denoting the horizon, a Markov decision process (MDP) $M^\star = \{\mathcal{X}, \mathcal{A}, \{P_h^\star\}_{h=0}^H, \{R_h^\star\}_{h=1}^H, H\}$ consists of a state space $\mathcal{X}$, an action space $\mathcal{A}$, a reward distribution $R_h^\star : \mathcal{X} \times \mathcal{A} \to \Delta([0,1])$ (with expectation $r_h^\star(x,a)$), and a transition kernel $P_h^\star : \mathcal{X} \times \mathcal{A} \to \Delta(\mathcal{X})$ (with the convention that $P_0^\star(\cdot \mid \emptyset)$ is the initial state distribution).[3]

At the beginning of the episode, the learner selects a randomized, non-stationary *policy* $\pi = (\pi_1, \ldots, \pi_H)$, where $\pi_h : \mathcal{X} \to \Delta(\mathcal{A})$; we let $\Pi_{\mathsf{rns}}$ denote the set of all such policies. The episode evolves through the following process; beginning from $x_1 \sim P_0^\star(\cdot \mid \emptyset)$, the MDP generates a trajectory $(x_1, a_1, r_1), \ldots, (x_H, a_H, r_H)$ via $a_h \sim \pi_h(x_h)$, $r_h \sim R_h^\star(x_h, a_h)$, and $x_{h+1} \sim P_h^\star(\cdot \mid x_h, a_h)$. We let $\mathbb{P}^{M^\star, \pi}$ denote the law under this process, and let $\mathbb{E}^{M^\star, \pi}$ denote the corresponding expectation, and likewise let $\mathbb{P}^{M, \pi}$ and $\mathbb{E}^{M, \pi}$ denote the analogous laws and expectations in another MDP $M$. We assume that $\sum_{h=1}^H r_h \in [0,1]$ almost surely for any trajectory in $M^\star$.

For a policy $\pi$ and MDP $M$, the expected reward for $\pi$ is given by $J^M(\pi) \coloneqq \mathbb{E}^{M,\pi}\big[\sum_{h=1}^H r_h\big]$, and the value functions are given by $V_h^{M,\pi}(x) \coloneqq \mathbb{E}^{M,\pi}\big[\sum_{h'=h}^H r_{h'} \mid x_h = x\big]$, and $Q_h^{M,\pi}(x,a) \coloneqq \mathbb{E}^{M,\pi}\big[\sum_{h'=h}^H r_{h'} \mid x_h = x, a_h = a\big]$. We let $\pi_M = \{\pi_{M,h}\}_{h=1}^H$ denote an optimal deterministic policy of $M$, which maximizes $V^{M,\pi}$ (over $\pi$) at all states (and in particular, satisfies $\pi_M \in \arg\max_{\pi \in \Pi_{\mathsf{rns}}} J^M(\pi)$), and write $Q^{M,\star} \coloneqq Q^{M,\pi_M}$. For $f : \mathcal{X} \times \mathcal{A} \to \mathbb{R}$, we write $\pi_f(x) \coloneqq \arg\max_a f(x,a)$ as well as $V_f(x) = \max_a f(x,a)$. For MDP $M$, horizon $h \in [H]$, and $g : \mathcal{X} \to \mathbb{R}$, we let $\mathcal{T}_h^M$ denote the Bellman (optimality) operator defined via $[\mathcal{T}_h^M g](x,a) = \mathbb{E}^M[r_h + g(x_{h+1}) \mid x_h = x, a_h = a]$, and we overload notation by letting $[\mathcal{T}_h^M f](x,a) = [\mathcal{T}_h^M V_f](x,a)$. We also let $\mathcal{T}_h^{M,\pi}$ denote the Bellman *evaluation* operator defined via $[\mathcal{T}_h^{M,\pi} f](x,a) = \mathbb{E}^M\big[r_h + \mathbb{E}_{a' \sim \pi_{h+1}(\cdot \mid x_{h+1})}[f(x_{h+1}, a')] \mid x_h = x, a_h = a\big]$, for any $\pi \in \Pi_{\mathsf{rns}}$. We define the *occupancy measures* for layer $h$ via $d_h^{M,\pi}(x) = \mathbb{P}^{M,\pi}[x_h = x]$ and $d_h^{M,\pi}(x,a) = \mathbb{P}^{M,\pi}[x_h = x, a_h = a]$.

**Online reinforcement learning.** In online reinforcement learning, the learning algorithm ALG repeatedly interacts with an unknown MDP $M^\star$ by executing a policy and observing the resulting trajectory. After $T$ rounds of interaction, the algorithm outputs a final policy $\widehat{\pi}$, with the goal of minimizing their *risk*, defined via

$$\mathsf{Risk}(T, \text{ALG}, M^\star) \coloneqq J^{M^\star}(\pi_{M^\star}) - J^{M^\star}(\widehat{\pi}). \tag{1}$$

---

[3]To simplify presentation, we assume that $\mathcal{X}$ and $\mathcal{A}$ are countable; our results extend to handle continuous variables with an appropriate measure-theoretic treatment.

**Framework: Reinforcement learning under general latent dynamics.** In *reinforcement learning under general latent dynamics*, we consider MDPs $M^\star$ where the dynamics are governed by the evolution of an unobserved *latent state* $s_h$, while the agent observes and acts on *observations* $x_h$ generated from these latent states. Formally, a *latent-dynamics MDP* consists of two ingredients: a *base MDP* $M_{\text{lat}} = \{\mathcal{S}, \mathcal{A}, \{P_{\text{lat},h}\}_{h=0}^{H}, \{R_{\text{lat},h}\}_{h=1}^{H}, H\}$ defined over a *latent state space* $\mathcal{S}$, and a *decodable emission process* $\psi := \{\psi_h : \mathcal{S} \to \Delta(\mathcal{X})\}_{h=1}^{H}$, which maps each latent state to a distribution over observations. The former is an arbitrary MDP defined over $\mathcal{S}$, while the latter is defined as follows.

**Definition 2.1** (Emission process). *An* emission process *is any function* $\psi := \{\psi_h : \mathcal{S} \to \Delta(\mathcal{X})\}_{h=1}^{H}$, *and is said to be* decodable *if*

$$\forall h, \forall s' \neq s \in \mathcal{S} : \quad \operatorname{supp} \psi_h(s) \cap \operatorname{supp} \psi_h(s') = \emptyset. \quad . \tag{2}$$

*When* $\psi = \{\psi_h\}_{h=1}^{H}$ *is decodable, we let* $\psi^{-1} := \{\psi_h^{-1} : \mathcal{X} \to \mathcal{S}\}_{h=1}^{H}$ *denote the associated decoder.*

With this, we can formally introduce the notion of a latent-dynamics MDP.

**Definition 2.2** (Latent-dynamics MDP). *For a base MDP* $M_{\text{lat}} = \{\mathcal{S}, \mathcal{A}, \{P_{\text{lat},h}\}_{h=0}^{H}, \{R_{\text{lat},h}\}_{h=1}^{H}, H\}$, *and a decodable emission process* $\psi$, *the* latent-dynamics MDP $\langle\!\langle M_{\text{lat}}, \psi \rangle\!\rangle := \{\mathcal{X}, \mathcal{A}, \{P_{\text{obs},h}\}_{h=0}^{H}, \{R_{\text{obs},h}\}_{h=1}^{H}, H\}$ *is defined as the MDP where the latent dynamics evolve based on the agent's action* $a_h \in \mathcal{A}$ *via the process* $s_{h+1} \sim P_{\text{lat},h}(s_h, a_h)$ *and* $r_h \sim R_{\text{lat},h}(s_h, a_h)$. *The latent state is not observed directly, and instead the agent observes* $x_h \in \mathcal{X}$ *generated by the emission process* $x_h \sim \psi_{h+1}(s_h)$.[4]

Note that under these dynamics, the decoder $\psi^{-1}$ associated with $\psi$ ensures that $\psi_h^{-1}(x_h) = s_h$ almost surely for all $h \in [H]$. That is, the latent states can be uniquely decoded from the observations. To emphasize the distinction between the latent-dynamics MDP $\langle\!\langle M_{\text{lat}}, \psi \rangle\!\rangle$ (which operates on the observable state space $\mathcal{X}$) and the MDP $M_{\text{lat}}$ (which operates on the latent state space $\mathcal{S}$), we refer to the latter as a *base MDP* rather than, for example, a "latent MDP", and apply a similar convention to other latent objects whenever possible.[5]

Departing from prior work, we do not place any inherent restrictions on the base MDP, and in particular do not assume that the latent space is small (i.e., tabular). Rather, we aim to understand—in a unified fashion—what structural assumptions on the base MDP $M_{\text{lat}}$ are required to enable learnability under latent dynamics. To this end, it will be useful to considers specific *classes* (i.e., subsets) of base MDPs $\mathcal{M}_{\text{lat}}$ and the classes of latent-dynamics MDPs they induce.

**Definition 2.3** (Latent-dynamics MDP class). *Given a set of base MDPs* $\mathcal{M}_{\text{lat}}$ *and a set of decoders* $\Phi \subset \{\mathcal{X} \to \mathcal{S}\}$, *we let*

$$\langle\!\langle \mathcal{M}_{\text{lat}}, \Phi \rangle\!\rangle := \{\langle\!\langle M_{\text{lat}}, \psi \rangle\!\rangle : M_{\text{lat}} \in \mathcal{M}_{\text{lat}}, \psi \text{ is decodable}, \psi^{-1} \in \Phi\} \tag{3}$$

*denote the class of induced latent-dynamics MDPs.*

Stated another way, $\langle\!\langle \mathcal{M}_{\text{lat}}, \Phi \rangle\!\rangle$ is the set of all latent-dynamics MDPs $\langle\!\langle M_{\text{lat}}, \psi \rangle\!\rangle$ where (i) the base MDP $M_{\text{lat}}$ lies in $\mathcal{M}_{\text{lat}}$, and (ii), the emission process $\psi$ is decodable, with the corresponding decoder belonging to $\Phi$. The class $\mathcal{M}_{\text{lat}}$ represents our prior knowledge about the underlying MDP $M_{\text{lat}}$; concrete classes considered in prior work include tabular MDPs [KAL16; DKJADL19; MHKL20; ZSUWAS22; MFR23], linear dynamical systems [DMRY20; DR21; Mha+20], and factored MDPs [MLJL21]. In particular, the class $\mathcal{M}_{\text{lat}}$ may itself warrant using function approximation. At the same time, the class $\Phi$ represents our prior knowledge or inductive bias about the emission process, enabling representation learning. In what follows, we investigate what conditions on $\mathcal{M}_{\text{lat}}$ make the induced class $\langle\!\langle \mathcal{M}_{\text{lat}}, \Phi \rangle\!\rangle$ tractable, both statistically (statistical modularity; Section 3) and via reduction (algorithmic modularity; Section 4).

## 3 Statistical Modularity: Positive and Negative Results

This section presents our main statistical results. We begin by formally defining the notion of statistical modularity introduced in Section 1, present our main impossibility result (lower bound) and its implications (Section 3.2), then give positive results for the general class of *pushforward-coverable* MDPs (Section 3.3).

---

[4]Equivalently the dynamics can be described via $R_{\text{obs},h}(x_h, a_h) = R_{\text{lat}}(\psi_h^{-1}(x_h), a_h)$ and $P_{\text{obs},h}(x_{h+1} \mid x_h, a_h) = P_{\text{lat},h}(\psi_{h+1}^{-1}(x_{h+1}) \mid \psi_h^{-1}(x_h), a_h) \cdot \psi_{h+1}(x_{h+1} \mid \psi_{h+1}^{-1}(x_{h+1}))$.

[5]For example, in Section 4 we will be concerned with reductions from observation-space algorithms to "base algorithms" that operate on the latent state space.

### 3.1 Statistical modularity: A formal definition

We first define the *statistical complexity* for a MDP class (or, *model class*) $\mathcal{M}$.

**Definition 3.1** (Statistical complexity). *We say that an MDP class $\mathcal{M}$ can be learned up to $\varepsilon$-optimality using $\mathsf{comp}(\mathcal{M}, \varepsilon, \delta)$ samples if there exists an algorithm* ALG *which, for every $M \in \mathcal{M}$, attains*

$$\mathsf{Risk}(T, \textsc{Alg}, M) \leq \varepsilon$$

*with probability at least $1 - \delta$, after $T = \mathsf{comp}(\mathcal{M}, \varepsilon, \delta)$ rounds of online interaction in $M$.*

We say that a *base MDP class* $\mathcal{M}_{\mathtt{lat}}$ admits *statistically modularity* if, for any decoder class $\Phi$, the induced latent-dynamics MDP class $\langle\!\langle \mathcal{M}_{\mathtt{lat}}, \Phi \rangle\!\rangle$ can be learned with statistical complexity that is polynomial in: (i) the statistical complexity for the base class, and (ii) the capacity of the decoder class.

**Definition 3.2** (Statistical modularity). *We say the MDP class $\mathcal{M}_{\mathtt{lat}}$ is* statistically modular *under complexity $\mathsf{comp}(\mathcal{M}_{\mathtt{lat}}, \varepsilon, \delta)$ if, for every decoder class $\Phi$, we have*

$$\mathsf{comp}(\langle\!\langle \mathcal{M}_{\mathtt{lat}}, \Phi \rangle\!\rangle, \varepsilon, \delta) = \mathtt{poly}(\mathsf{comp}(\mathcal{M}_{\mathtt{lat}}, \varepsilon, \delta), \log|\Phi|). \tag{4}$$

*We say that $\mathcal{M}_{\mathtt{lat}}$ admits* strong statistical modularity *if Eq. (4) holds when $\mathsf{comp}(\mathcal{M}_{\mathtt{lat}}, \varepsilon, \delta)$ is the minimax sample complexity for $\mathcal{M}_{\mathtt{lat}}$.*

In the sequel, we examine well-studied MDP classes $\mathcal{M}_{\mathtt{lat}}$ (e.g., those which admit low Bellman rank [JKALS17]) and choose $\mathsf{comp}(\mathcal{M}_{\mathtt{lat}}, \varepsilon, \delta)$ based on natural upper bounds on their optimal sample complexity; in this case we will simply say they are (or are not) statistical modular, leaving the complexity upper bound comp implicit. Following prior work [KAL16; DKJADL19; MHKL20; ZSUWAS22; MFR23; DR21; Mha+20; MLJL21], we use $\log|\Phi|$ as a proxy for the statistical complexity of supervised learning with the decoder class $\Phi$.[6]

The two most notable examples of statistical modularity covered by prior work are: (i) taking $\mathcal{M}_{\mathtt{lat}}$ as the set of tabular MDPs admits strong statistical modularity [DKJADL19; MHKL20; MFR23], and (ii) taking $\mathcal{M}_{\mathtt{lat}}$ as the set of linear MDPs admits statistical modularity with complexity $\mathtt{poly}(d, H, |\mathcal{A}|, \varepsilon^{-1}, \log(\delta^{-1}))$ [AKKS20; UZS22; MCKJA24; MBFR23].[7] Interestingly, the latter does not admit strong statistical modularity, because the optimal rate for $\mathcal{M}_{\mathtt{lat}}$ does not scale with $|\mathcal{A}|$, but the rate for $\langle\!\langle \mathcal{M}_{\mathtt{lat}}, \Phi \rangle\!\rangle$ necessarily does [LS20; HLSW21]. The results of Mhammedi et al.; Misra et al.; Song et al. [Mha+20; MLJL21; SWFK24] can also be viewed as instances of statistical modularity for other base MDP classes.

### 3.2 Lower bounds: Impossibility of statistical modularity

Our main result in this section is to show that for most MDP classes $\mathcal{M}_{\mathtt{lat}}$ considered in the literature on sample-efficient reinforcement learning with function approximation [RVR13; JKALS17; SJKAL19; MJTS20; AJSWY20; Li09; DVRZ19; WSY20; ZGS21; Du+21; JLM21; FKQR21], statistical modularity (under the natural complexity upper bound for the class of interest) is *impossible*. Our central technical result is the following lower bound, which shows that statistical modularity can be impossible *even when the base MDP is known to the learner a-priori*. The lower bound is a significant generalization of the result from Song et al. [SWFK24]; we first state the lower bound, then discuss implications.

**Theorem 3.1** (Impossibility of statistical modularity). *For every $N \geq 4$, there exists a decoder class $\Phi$ with $|\Phi| = N$ and a family of base MDPs $\mathcal{M}_{\mathtt{lat}}$ satisfying (i) $|\mathcal{M}_{\mathtt{lat}}| = 1$, (ii) $H \leq \mathcal{O}(\log(N))$, (iii) $|\mathcal{S}| = |\mathcal{X}| \leq N^2$, (iv) $|\mathcal{A}| = 2$, and such that*

*1. For all $\varepsilon, \delta > 0$, we have $\mathsf{comp}(\mathcal{M}_{\mathtt{lat}}, \varepsilon, \delta) = 0$.*

*2. For an absolute constant $c > 0$, $\mathsf{comp}(\langle\!\langle \mathcal{M}_{\mathtt{lat}}, \Phi \rangle\!\rangle, c, c) \geq \Omega(N/\log(N))$.*

In other words, even when the base dynamics are fully known, strong statistical modularity (in this case, $\mathtt{poly}(\log|\Phi|)$ complexity) is impossible; any algorithm will require at least $\min\{\sqrt{S}, 2^{\Omega(H)}/H, |\Phi|/\log|\Phi|\}$ episodes to learn a near-optimal policy for a latent-dynamics MDP $\langle\!\langle M_{\mathtt{lat}}, \psi \rangle\!\rangle \in \langle\!\langle \mathcal{M}_{\mathtt{lat}}, \Phi \rangle\!\rangle$.

---

[6]Our main results easily extend to infinite classes through standard arguments.

[7]In the latter case, the latent-dynamics class $\langle\!\langle \mathcal{M}_{\mathtt{lat}}, \Phi \rangle\!\rangle$ may be seen to be a set of low-rank MDPs (that is, linear MDPs with unknown features), so that low-rank MDP algorithms may be applied directly on the observations (Appendix E.2).

| Base MDP class $\mathcal{M}_{\text{lat}}$ | Statistical Modularity? |
|---|---|
| Tabular | ✓ |
| Contextual Bandits | ✓ |
| Low-Rank MDP | ✓ |
| Known Deterministic MDP ($|\mathcal{M}_{\text{lat}}| = 1$) | ✓ |
| Low State Occupancy ($\forall \pi : \mathcal{S} \to \Delta(\mathcal{A})$) | ✓ |
| Model Class + Pushforward Coverability | ✓ |
| Linear CB/MDP | ✗* |
| Model Class + Coverability ($\forall \pi_M : M \in \mathcal{M}$) | ✗ |
| Known Stochastic MDP ($|\mathcal{M}_{\text{lat}}| = 1$) | ✗ |
| Bellman Rank ($Q$-type or $V$-type) | ✗ |
| Eluder Dimension + Bellman Completeness | ✗ |
| $Q^\star$-Irrelevant State Abstraction | ✗ |
| Linear Mixture MDP | ✗ |
| Linear $Q^\star/V^\star$ | ✗ |
| Low State/State-Action Occupancy ($\forall \pi_M : M \in \mathcal{M}$) | ✗ |
| Bisimulation | ? |
| Low State-Action Occupancy ($\forall \pi : \mathcal{S} \to \Delta(\mathcal{A})$) | ?* |
| Model Class + Coverability ($\forall \pi : \mathcal{S} \to \Delta(\mathcal{A})$) | ? |

Figure 1: Summary of statistical modularity (SM) results.
✓: SM is possible for a natural choice of comp($\cdot$) (e.g., poly($|\mathcal{S}|, |\mathcal{A}|, H, \varepsilon^{-1}, \log(\delta^{-1})$) for tabular MDPs).
✗: SM is not possible with natural choices of comp($\cdot$).
?: open.
*: SM is possible if willing to pay for (suboptimal) $|\mathcal{A}|$ complexity.
See Appendix E.2 for precise descriptions of each setting and our choices for their complexities.

**Intuition for lower bound.** The intuition behind the lower bound in Theorem 3.1 is as follows: the unobserved latent state space consists of $N = |\Phi|$ binary trees (indexed from 1 to $N$), each with $N$ leaf nodes. The starting distribution is uniform over the roots of the $N$ trees, and the agent receives a reward of 1 if and only if they navigate to the leaf node that corresponds to the index of their current tree. The observed state space is identical to the latent state space, but the emission process shifts the index of the tree by an amount which is unknown to the agent. Despite the base MDP being known and the decoder class satisfying realizability, the agent requires near-exhaustive search to identify the value of the shift and recover a near-optimal policy.

**A taxonomy of statistical modularity.** As a corollary, we prove that many (but not all) well-studied function approximation settings do not admit statistical modularity by embedding them into the lower bound construction of Theorem 3.1 (as well as a variant of the result, Theorem E.1). Our results are summarized in Figure 1. Our impossibility results highlight the following phenomenon: many MDP classes $\mathcal{M}_{\text{lat}}$ that place structural assumptions via the value functions (e.g., MDPs with linear-$Q^\star/V^\star$ [Du+21] or MDPs with a Bellman complete value function class of bounded eluder dimension [JLM21; WSY20]) become intractable under latent dynamics. Intuitively, this is because it is not possible to take advantage of structure in value functions without learning a good representation, and, simultaneously, these assumptions are too weak by themselves to enable learning such a representation. Meanwhile, MDP classes $\mathcal{M}_{\text{lat}}$ that place structural assumptions on the transition distribution (e.g., MDPs with low state occupancy complexity [Du+21] or low-rank MDPs [AKKS20]) are sometimes (but not always) tractable under latent dynamics.[8]

We point to Appendix E.2 for background on all the settings in Figure 1 and proofs that they are (or are not) statistically modular. We remark that it is fairly straightforward to embed most of the MDP classes of Figure 1 into the construction of Theorem 3.1 since it only uses only a single base MDP $M_{\text{lat}}$, and we expect that many other base MDP classes can similarly be shown to be intractable. However, proving the *positive* results in Figure 1 requires establishing several new results showing that certain base classes are tractable under latent dynamics; most notably, we next discuss the case of *pushforward coverability*.

### 3.3 Upper bounds: Pushforward-coverable MDPs are statistically modular

Our main postive result concerning statistical modularity is to highlight *pushforward coverability* [XJ21; AFK24; MFR24]—a strengthened version of the *coverability* parameter introduced in Xie et al. [XFBJK23]—as a general structural parameter that enables sample-efficient reinforcement learning under latent dynamics.

---

[8]If one is willing to pay for suboptimal $|\mathcal{A}|$ factors, then more (but not all) classes become statistically tractable (e.g., linear MDPs [JYWJ20] and MDPs with low state-action occupancy [Du+21]).

**Definition 3.3** (Pushforward coverability). *The pushforward coverability coefficient $C_{\text{push}}$ for an MDP $M_{\text{lat}}$ with transition kernel $P_{\text{lat}}$ is defined by*

$$C_{\text{push}}(M_{\text{lat}}) = \max_{h \in [H]} \inf_{\mu \in \Delta(\mathcal{S})} \sup_{(s,a,s') \in \mathcal{S} \times \mathcal{A} \times \mathcal{S}} \frac{P_{\text{lat},h-1}(s' \mid s, a)}{\mu(s')}. \tag{5}$$

Concrete examples [AFK24; MFR24] include: (i) tabular MDPs $M_{\text{lat}}$ admit $C_{\text{push}}(M_{\text{lat}}) \leq |\mathcal{S}|$; and (ii) Low-Rank MDPs $M_{\text{lat}}$ (with or without known features) in dimension $d$ admit $C_{\text{push}}(M_{\text{lat}}) \leq d$. Further examples include analytically sparse Low-Rank MDPs [GMR24] and Exogenous Block MDPs with weakly correlated noise [MFR24]. Our main result is as follows.

**Theorem 3.2** (Pushforward-coverable MDPs are statistically modular). *Let $\mathcal{M}_{\text{lat}}$ be a base MDP class such that each $M_{\text{lat}} \in \mathcal{M}_{\text{lat}}$ has pushforward coverability bounded by $C_{\text{push}}(M_{\text{lat}}) \leq C_{\text{push}}$. Then, for any decoder class $\Phi$, we have:*

1. $\text{comp}(\mathcal{M}_{\text{lat}}, \varepsilon, \delta) \leq \text{poly}(C_{\text{push}}, |\mathcal{A}|, H, \log|\mathcal{M}_{\text{lat}}|, \varepsilon^{-1}, \log(\delta^{-1}))$*, and*

2. $\text{comp}(\langle\!\langle \mathcal{M}_{\text{lat}}, \Phi \rangle\!\rangle, \varepsilon, \delta) \leq \text{poly}(C_{\text{push}}, |\mathcal{A}|, H, \log|\mathcal{M}_{\text{lat}}|, \log|\Phi|, \varepsilon^{-1}, \log(\delta^{-1}), \log\log|\mathcal{S}|)$*.*

Theorem 3.2 shows that, modulo a term that is doubly-logarithmic in $|\mathcal{S}|$, latent pushforward coverability enables statistical modularity. That is, when the base (latent) dynamics satisfy pushforward coverability, there exists an algorithm for the latent-dynamics setting which scales with the statistical complexity of the base MDP class and $\log|\Phi|$. We suspect that the additional $\log\log|\mathcal{S}|$ factor is not essential and can be removed with a more sophisticated analysis. We note that the complexity comp chosen above is not the minimax complexity for $\mathcal{M}_{\text{lat}}$, since every set of pushforward coverable MDPs is also a set of *coverable* MDPs with a potentially smaller coverability parameter [AFK24].

Let us provide some intuition for this result. We firstly note that when $M_{\text{lat}}^\star$ has pushforward coverability parameter $C_{\text{push}}$, it holds that for any emission process $\psi^\star$, the observation-level MDP $M_{\text{obs}}^\star := \langle\!\langle M_{\text{lat}}^\star, \psi^\star \rangle\!\rangle$ also satisfies pushforward coverability with the same parameter $C_{\text{push}}$ (Lemma D.5). Yet, despite access to realizable base MDP class $\mathcal{M}_{\text{lat}}$ and decoder class $\Phi$, it is unclear whether the latent-dynamics MDP $M_{\text{obs}}^\star$ satisfies any of the *observation-level* function approximation conditions required by existing approaches that provide sample complexity guarantees under pushforward coverability. In particular, known algorithms for this setting either require a Bellman-complete value function class [XFBJK23], a class realizing certain density ratios [AFJSX24; AFK24], or a realizable model class [AFK24], and it is highly nontrivial to construct these *for the latent-dynamics MDP $M_{\text{obs}}^\star = \langle\!\langle M_{\text{lat}}^\star, \psi^\star \rangle\!\rangle$* given only the base MDP class $\mathcal{M}_{\text{lat}}$ and the decoder class $\Phi$. Intuitively, this is because the former observation-level function approximation classes capture properties of the observation-level dynamics which cannot be obtained without some knowledge of the emission process.

Our main technical contribution is to establish a new structural property for pushforward-coverable MDPs (Lemma F.1): low-dimensional linear embeddings of their latent models can approximate the Bellman updates for an *arbitrary* set of test functions (as long as the set is not too large). We use this property to construct low-dimensional linear features that can approximate Bellman backups in *observation-space*, allowing us to (approximately) satisfy the Bellman completeness assumption required to apply GOLF [JLM21] to the latent-dynamics MDP. A fascinating open question is whether a similar approach can be used to establish that standard (as opposed to pushforward) coverable MDPs are statistically modular, which would encompass all other known positive cases of statistical modularity (cf. Figure 1). We refer interested readers to a more detailed technical overview in Appendix F.1, as well as the full proof in Appendix F.2.

## 4 Algorithmic Modularity

We now turn our attention to *algorithmic modularity*. Specifically, we aim for *observable-to-latent reductions*, whereby—via representation learning—RL under latent dynamics can be efficiently reduced to the simpler problem of RL with latent states directly observed. Since algorithmic modularity is a stronger property than statistical modularity, we sidestep the previous lower bounds in Section 3 through additional feedback and modeling assumptions. Our main result for this section is a new meta-algorithm, O2L, which, under these assumptions (and when equipped with an appropriately designed representation learning oracle), acts as a *universal* reduction in the sense that, whenever the representation learning oracle has low risk, the reduction transforms *any* sample-efficient algorithm for *any* base MDP class into a sample-efficient algorithm for the induced latent-dynamics MDP class.

---

**Algorithm 1** O2L: Observable-to-Latent Reduction

---

1: **input**: Epochs $T$, episodes $K$, decoder set $\Phi$, rep. learning oracle REPLEARN, base alg. ALG$_{\text{lat}}$.
2: **for** $t = 1, 2, \cdots, T$ **do**
3:      REPLEARN chooses a representation $\widehat{\phi}^{(t)} : \mathcal{X} \to \mathcal{S} \in \Phi$ based on data collected so far.
4:      Initialize new instance of ALG$_{\text{lat}}$.
5:      **for** $k = 1, 2, \cdots, K$ **do**            // ALG$_{\text{lat}}$ plays $K$ rounds in the "$\widehat{\phi}^{(t)}$-compressed dynamics."
6:          ALG$_{\text{lat}}$ chooses policy $\pi_{\text{lat}}^{(t,k)} : \mathcal{S} \times [H] \to \Delta(\mathcal{A})$.
7:          Deploy $\pi_{\text{lat}} \circ \widehat{\phi}^{(t)}$ to collect trajectory $\{x_h^{(t,k)}, a_h^{(t,k)}, r_h^{(t,k)}\}_{h=1}^H$.
8:          Update ALG$_{\text{lat}}$ with compressed trajectory $\{\widehat{\phi}_h^{(t)}(x_h^{(t,k)}), a_h^{(t,k)}, r_h^{(t,k)}\}_{h=1}^H$.
9:      **end for**
10:     ALG$_{\text{lat}}$ returns final policy $\widehat{\pi}^{(t)} : \mathcal{S} \times [H] \to \Delta(\mathcal{A})$, deploy $\widehat{\pi}^{(t)} \circ \widehat{\phi}^{(t)}$ to collect one trajectory.
11: **end for**
12: **return** $\widehat{\pi} = \texttt{Unif}(\widehat{\pi}^{(1)} \circ \widehat{\phi}^{(1)}, \ldots, \widehat{\pi}^{(T)} \circ \widehat{\phi}^{(T)})$.

---

**Setup and O2L meta-algorithm.** For the results in this section, we denote the (unknown) latent-dynamics MDP of interest by $M_{\text{obs}}^\star := \langle\!\langle M_{\text{lat}}^\star, \psi^\star \rangle\!\rangle$, and use $\phi^\star := (\psi^\star)^{-1}$ to denote the true decoder. The O2L meta-algorithm (Algorithm 1) learns a near-optimal policy for $M_{\text{obs}}^\star$ by alternating between performing representation learning and executing a black-box "base" RL algorithm (designed for the base MDP) on the learned representation; this approach is inspired by empirical methods that blend representation learning and RL in the latent space (e.g., [GKBNB19; SAGHCB20; Ni+24]).

Concretely, the algorithm takes as input a *representation learning oracle* REPLEARN and a *base RL algorithm* ALG$_{\text{lat}}$ that operates in the latent space. In each *epoch* $t \in [T]$, REPLEARN produces a new representation $\widehat{\phi}^{(t)} : \mathcal{X} \to \mathcal{S}$ based on data observed so far (potentially using additional side information, which we will elaborate on in the sequel). Then, the reduction invokes ALG$_{\text{lat}}$, using $\widehat{\phi}^{(t)}$ to simulate access to the true latent states. In particular, ALG$_{\text{lat}}$ runs for $K$ episodes, where at each episode $k$: (i) ALG$_{\text{lat}}$ produces a latent policy $\pi_{\text{lat}}^{(t,k)} : \mathcal{S} \times [H] \to \Delta(\mathcal{A})$, (ii) the latent policy is transformed into an observation-level policy via composition with $\widehat{\phi}^{(t)}$, i.e. $\pi_{\text{lat}}^{(t,k)} \circ \widehat{\phi}^{(t)}$, which is then deployed to produce a trajectory $\{x_h^{(t,k)}, a_h^{(t,k)}, r_h^{(t,k)}\}_{h=1}^H$, and (iii) the trajectory is *compressed through* $\widehat{\phi}^{(t)}$ and used to update ALG$_{\text{lat}}$ via $\{\widehat{\phi}_h^{(t)}(x_h^{(t,k)}), a_h^{(t,k)}, r_h^{(t,k)}\}_{h=1}^H$ (cf. Line 8 of Algorithm 1).[9] After the $K$ rounds conclude, ALG$_{\text{lat}}$ produces a final latent policy $\widehat{\pi}_{\text{lat}}^{(t)} : \mathcal{S} \times [H] \to \Delta(\mathcal{A})$. The final policy $\widehat{\pi}$ chosen by the O2L algorithm is a uniform mixture of $\widehat{\pi}_{\text{lat}}^{(t)} \circ \widehat{\phi}^{(t)}$ over all the epochs.

The central assumption behind O2L is that the base algorithm ALG$_{\text{lat}}$ can achieve low-risk in the underlying base MDP $M_{\text{lat}}^\star$ *if given access to the true latent states* $s_h = \phi^\star(x_h)$. Beyond this assumption, we require that the representation learning oracle REPLEARN can learn a sufficiently high-quality representation. In our applications, this will be made possible by assuming access to a realizable decoder class $\Phi$ and two distinct assumptions: *hindsight observability* (Section 4.1) and conditions enabling *self-predictive representation learning* (Section 4.2). We will show that under these conditions, we can instantiate a representation learning oracle such that O2L inherits the sample complexity guarantee for ALG$_{\text{lat}}$, thereby achieving algorithmic modularity.

## 4.1 Algorithmic modularity via hindsight observability

Our first algorithmic result bypasses the hardness in Section 3 by considering the setting of *hindsight observability*, which has garnered recent interest in the context of POMDPs [LADZ23; GCWXWB24; SLS23; LXJZV24]. Here, we assume that at training time (but not during deployment), the algorithm has access to additional feedback in the form of the true latent states, which are revealed at the end of each episode.

**Assumption 4.1** (Hindsight Observability [LADZ23])**.** *The latent states* $(\phi_1^\star(x_1), \ldots, \phi_H^\star(x_H))$ *are revealed to the learner after each episode* $(x_1, a_1, r_1, \ldots, x_H, a_H, r_H)$ *concludes.*

We emphasize that in the hindsight observability framework, the learner must still execute *observation-space policies* $\pi_{\text{obs}} : \mathcal{X} \times [H] \to \Delta(\mathcal{A})$, as the latent states are only revealed *at the end of each episode*. Under hindsight observability, we can instantiate the representation learning oracle in O2L so that the

---

[9]Note that, if $\widehat{\phi}$ is inaccurate, the compressed trajectory cannot necessarily be viewed as being generated by a latent MDP, and must instead be viewed as coming from a *Partially Observed* MDP (Appendix I.1.1).

reduction achieves low risk for *any choice of black-box base algorithm* $\text{ALG}_{\text{lat}}$. In particular, we make use of *online* classification oracles, which use the revealed latent states to achieve low classification loss with respect to $\phi^\star$ under adaptively generated data. We first state a guarantee based on generic classification oracles, then instantiate it to give a concrete end-to-end sample complexity bound.

Formally, at each step $t$, the online classification oracle, denoted via $\text{REP}_{\text{class}}$, is given the states and hindsight observations collected so far and produces a deterministic estimate $\widehat{\phi}^{(t)} = \text{REP}_{\text{class}}(\{x_h^{(i)}, \phi_h^\star(x_h^{(i)})\}_{i<t,h\leq H})$ for the true decoder $\phi^\star$. We measure the regret of the oracle via the $0/1$ loss for classification:

$$\text{Reg}_{\text{class}}(T) := \sum_{t=1}^{T} \sum_{h=1}^{H} \mathbb{E}_{\pi^{(t)} \sim p^{(t)}} \mathbb{E}^{\pi^{(t)}} \left[ \mathbb{I}\{\widehat{\phi}_h^{(t)}(x_h) \neq \phi_h^\star(x_h)\} \right],$$

where $p^{(t)}$ represents a randomization distribution over the policy $\pi^{(t)}$. Our reduction succeeds under the assumption that the oracle has low expected regret.

**Assumption 4.2.** *For any (possibly adaptive) sequence $\pi^{(t)}$, with $\pi^{(t)} \sim p^{(t)}$, the online classification oracle $\text{REP}_{\text{class}}$ has expected regret bounded by*

$$\mathbb{E}[\text{Reg}_{\text{class}}(T)] \leq \text{Est}_{\text{class}}(T),$$

*where $\text{Est}_{\text{class}}(T)$ is a known upper bound.*

We apply such an oracle within O2L as follows: at the end of each iteration $t \in [T]$ in O2L, we sample $k \sim [K]$ uniformly, and update the classification oracle with the trajectory $(x_1^{(t,k)}, a_1^{(t,k)}, r_1^{(t,k)}), \ldots, (x_H^{(t,k)} a_H^{(t,k)}, r_H^{(t,k)})$; see the proof of [Theorem 4.1](#) for details. We let $\text{Risk}_{\text{obs}}(TK)$ denote the risk of the O2L reduction when run for $T$ epochs of $K$ episodes, and we let $\text{Risk}_\star(K) := \mathbb{E}[\text{Risk}(K, \text{ALG}_{\text{lat}}, M_{\text{lat}}^\star)]$ denote the expected risk of $\text{ALG}_{\text{lat}}$ when executed on $M_{\text{lat}}^\star$ with access to the true latent states $s_h = \phi^\star(x_h)$ for $K$ episodes.

**Theorem 4.1** (Risk bound for O2L under hindsight observability). *Let $\text{ALG}_{\text{lat}}$ be a base algorithm with base risk $\text{Risk}_\star(K)$, and $\text{REP}_{\text{class}}$ a representation learning oracle satisfying [Assumption 4.2](#). Then [Algorithm 1](#), with inputs $T, K, \Phi, \text{REP}_{\text{class}}$, and $\text{ALG}_{\text{lat}}$, has expected risk*

$$\mathbb{E}[\text{Risk}_{\text{obs}}(TK)] \leq \text{Risk}_\star(K) + \frac{2K}{T}\text{Est}_{\text{class}}(T).$$

This result shows that we can achieve sublinear risk under latent dynamics as long as (i) the base algorithm achieves sublinear risk $\text{Risk}_\star(K)$ given access to the true latent states, and (ii) the classification oracle achieves sublinear regret $\text{Est}_{\text{class}}(T)$. Notably, the result is fully modular, meaning we require no explicit conditions on the latent dynamics or the base algorithm, and is computationally efficient whenever the base algorithm and classification oracle are efficient.

To make [Theorem 4.1](#) concrete, we next provide a representation learning oracle (EXPWEIGHTS.DR; [Algorithm 3](#) in [Appendix G.1](#)) based on a derandomization of the classical exponential weights mechanism, which satisfies [Assumption 4.2](#) with $\text{Est}_{\text{class}} \lesssim H \log|\Phi|$ whenever it has access to a class $\Phi$ that satisfies decoder realizability.

**Lemma 4.1** (Online classification via EXPWEIGHTS.DR). *Under decoder realizability ($\phi^\star \in \Phi$), EXPWEIGHTS.DR ([Algorithm 3](#)) satisfies [Assumption 4.2](#) with[10]*

$$\text{Est}_{\text{class}}(T) = \widetilde{\mathcal{O}}(H \log|\Phi|).$$

Instantiating [Theorem 4.1](#) with the above representation learning oracle, we obtain the following algorithmic modularity result.

**Corollary 4.1** (Algorithmic modularity under hindsight observability). *For any base algorithm $\text{ALG}_{\text{lat}}$, under decoder realizability ($\phi^\star \in \Phi$), O2L with inputs $T, K, \Phi, \text{EXPWEIGHTS.DR}$, and $\text{ALG}_{\text{lat}}$ achieves*

$$\mathbb{E}[\text{Risk}_{\text{obs}}(TK)] \lesssim \text{Risk}_\star(K) + \frac{HK \log|\Phi|}{T}.$$

*Consequently, for any $\text{ALG}_{\text{lat}}$, setting $T \approx KH \log|\Phi|/\text{Risk}_\star(K)$ achieves $\mathbb{E}[\text{Risk}_{\text{obs}}(TK)] \lesssim \text{Risk}_\star(K)$ with a number of trajectories $TK = \widetilde{\mathcal{O}}(K^2 H \log|\Phi|/\text{Risk}_\star(K))$.*

---

[10]In this section, the notations $\widetilde{\mathcal{O}}, \approx$, and $\lesssim$ ignore only constants and logarithmic factors of $H$.

Beyond achieving algorithmic modularity, this result shows that under hindsight observability, we can achieve strong statistical modularity (modulo possible $H$ factors) for *every* base MDP class $\mathcal{M}_{\mathtt{lat}}$, an important result in its own right.[11] As an example, suppose that $\mathtt{Risk}_\star(K) = \mathcal{O}(K^{-1/2})$, which is satisfied by many standard algorithms of interest [JKALS17; JYWJ20; JLM21; FKQR21]. Then, setting $T$ according to Corollary 4.1 obtains an expected risk bound of $\varepsilon$ using $\mathcal{O}\big(H\log|\Phi|/\varepsilon^5\big)$ trajectories.

**Remark 4.1** (Online versus offline oracles)**.** Theorem 4.1 critically uses that assumption that $\mathtt{REP}_{\mathtt{class}}$ satisfies an *online* classification error bound to handle the fact that data is generated adaptively based on the estimators $\widehat{\phi}^{(1)}, \ldots, \widehat{\phi}^{(T)}$ it produces, which is by now a relatively standard technique in the design of interactive decision making algorithms [FR20; FKQR21; FR23]. We note that under coverability and other exploration conditions, online oracles for classification can be directly obtained from *offline* (i.e. supervised) classification oracles [XFBJK23; BRS24; FHQR24].

## 4.2 Algorithmic modularity via self-predictive estimation

We complement the above results by studying the general online RL setting *without* hindsight observations. To address this more challenging setting, we design an *optimistic self-predictive estimation* objective (Eq. (7)), which learns a representation by jointly fitting a decoder together with a latent model. We prove that any representation learning oracle that attains low regret with respect to this objective can be used in O2L to obtain observable-to-latent reductions for any low-risk base algorithm $\mathtt{ALG}_{\mathtt{lat}}$ (for a formal statement, see Theorem A.1). We provide a (computationally inefficient) estimator (SELFPREDICT.OPT; Algorithm 4 in Appendix H.1) which we show attains low optimistic self-regret under certain statistical conditions (namely, coverability of the base MDP and a function approximation condition enabling us to express the self-prediction target as a latent model, see Lemma A.1 for a formal statement), thereby obtaining an end-to-end reduction for the general online RL setting. For lack of space, these results are deferred to Appendix A.

## 5 Discussion

Our work initiates the study of statistical and algorithmic modularity for reinforcement learning under general latent dynamics. Our positive and negative results serve as a first step toward a unified theory for reinforcement learning in the presence of high-dimensional observations. To this end, we close with some important future directions and open problems.

**Statistical modularity.** Can we obtain a unified characterization for the statistical complexity of RL under latent dynamics with a given class of base MDPs $\mathcal{M}_{\mathtt{lat}}$? Our results in Section 3 suggest that this will require new tools that go beyond existing notions of statistical complexity. Toward resolving this problem, concrete questions that are not yet understood include: (i) Is coverability [XFBJK23] (as opposed to pushforward coverability) sufficient for learnability under latent dynamics? (ii) Is the *Exogenous Block MDP* problem [EMKAL22; MFR24]—a special case of our general framework— statistically tractable? Lastly, are there additional types of feedback that are weaker than hindsight observability, yet suffice to bypass the hardness results in Section 3?

**Algorithmic modularity.** Can we derive a unified representation learning objective that enables algorithmic modularity whenever statistical modularity is possible? Ideally, such an objective would be computationally tractable. Alternatively, can we show that algorithmic modularity fundamentally requires stronger modeling assumptions than statistical modularity? Toward addressing the problems above, a first step might be to understand: (i) What are the minimal statistical assumptions under which we can minimize the self-predictive objective in Section 4.2? (ii) How can we encourage finding good representations via self-prediction beyond the use of optimism over the base (latent) models; and (iii) when can we minimize self-prediction in a computationally efficient fashion?

### Acknowledgements

Nan Jiang acknowledges funding support from NSF IIS-2112471, NSF CAREER IIS-2141781, Google Scholar Award, and Sloan Fellowship.

---

[11] Formally, while we have defined the statistical modularity condition in terms of *high-probability* risk bounds, it is straightforward to extend it to instead consider *expected* risk bounds.

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

| [MFR23] | Zakaria Mhammedi, Dylan J Foster, and Alexander Rakhlin. "Representation Learning With Multi-Step Inverse Kinematics: An Efficient and Optimal Approach to Rich-Observation RL". In: *International Conference on Machine Learning*. 2023. |
|---|---|
| [MFR24] | Zakaria Mhammedi, Dylan J Foster, and Alexander Rakhlin. "The Power of Resets in Online Reinforcement Learning". In: *arXiv:2404.15417*. 2024. |
| [Mha+20] | Zakaria Mhammedi, Dylan J Foster, Max Simchowitz, Dipendra Misra, Wen Sun, Akshay Krishnamurthy, Alexander Rakhlin, and John Langford. "Learning the Linear Quadratic Regulator From Nonlinear Observations". In: *Neural Information Processing Systems*. 2020. |
| [MHKL20] | Dipendra Misra, Mikael Henaff, Akshay Krishnamurthy, and John Langford. "Kinematic State Abstraction and Provably Efficient Rich-Observation Reinforcement Learning". In: *International Conference on Machine Learning*. 2020. |
| [MJTS20] | Aditya Modi, Nan Jiang, Ambuj Tewari, and Satinder Singh. "Sample Complexity of Reinforcement Learning Using Linearly Combined Model Ensembles". In: *International Conference on Artificial Intelligence and Statistics*. 2020. |
| [MLJL21] | Dipendra Misra, Qinghua Liu, Chi Jin, and John Langford. "Provable Rich Observation Reinforcement Learning With Combinatorial Latent States". In: *International Conference on Learning Representations*. 2021. |
| [Ni+24] | Tianwei Ni, Benjamin Eysenbach, Erfan Seyedsalehi, Michel Ma, Clement Gehring, Aditya Mahajan, and Pierre-Luc Bacon. "Bridging State and History Representations: Understanding Self-Predictive RL". In: *arXiv:2401.08898*. 2024. |
| [NRKFG22] | Suraj Nair, Aravind Rajeswaran, Vikash Kumar, Chelsea Finn, and Abhinav Gupta. "R3M: A Universal Visual Representation for Robot Manipulation". In: *arXiv:2203.12601*. 2022. |
| [OVR16] | Ian Osband and Benjamin Van Roy. "On Lower Bounds for Regret in Reinforcement Learning". In: *arXiv:1608.02732*. 2016. |
| [PAED17] | Deepak Pathak, Pulkit Agrawal, Alexei A Efros, and Trevor Darrell. "Curiosity-Driven Exploration by Self-Supervised Prediction". In: *International Conference on Machine Learning*. 2017, pp. 2778–2787. |
| [RH23] | Philippe Rigollet and Jan-Christian Hütter. "High-dimensional statistics". In: *arXiv:2310.19244*. 2023. |
| [RVR13] | Daniel Russo and Benjamin Van Roy. "Eluder Dimension and the Sample Complexity of Optimistic Exploration". In: *Neural Information Processing Systems*. 2013. |
| [SAGHCB20] | Max Schwarzer, Ankesh Anand, Rishab Goel, R Devon Hjelm, Aaron Courville, and Philip Bachman. "Data-Efficient Reinforcement Learning with Self-Predictive Representations". In: *International Conference on Learning Representations*. 2020. |
| [Sch+20] | Julian Schrittwieser, Ioannis Antonoglou, Thomas Hubert, Karen Simonyan, Laurent Sifre, Simon Schmitt, Arthur Guez, Edward Lockhart, Demis Hassabis, Thore Graepel, Timothy Lillicrap, and David Silver. "Mastering Atari, Go, Chess and Shogi by Planning With a Learned Model". In: *Nature*. 2020. |
| [SJKAL19] | Wen Sun, Nan Jiang, Akshay Krishnamurthy, Alekh Agarwal, and John Langford. "Model-Based RL in Contextual Decision Processes: PAC Bounds and Exponential Improvements Over Model-Free Approaches". In: *Conference on Learning Theory*. 2019. |
| [SLS23] | Ming Shi, Yingbin Liang, and Ness Shroff. "Theoretical Hardness and Tractability of POMDPs in RL with Partial Hindsight State Information". In: *arXiv:2306.08762*. 2023. |
| [SWFK24] | Yuda Song, Lili Wu, Dylan J Foster, and Akshay Krishnamurthy. "Rich-Observation Reinforcement Learning with Continuous Latent Dynamics". In: *International Conference on Machine Learning*. 2024. |
| [SZSBKS23] | Yuda Song, Yifei Zhou, Ayush Sekhari, J Andrew Bagnell, Akshay Krishnamurthy, and Wen Sun. "Hybrid RL: Using Both Offline and Online Data Can Make RL Efficient". In: *International Conference on Learning Representations*. 2023. |

[Tan+17]      Haoran Tang, Rein Houthooft, Davis Foote, Adam Stooke, OpenAI Xi Chen, Yan Duan, John Schulman, Filip DeTurck, and Pieter Abbeel. "# Exploration: A Study of Count-Based Exploration for Deep Reinforcement Learning". In: *Neural Information Processing Systems*. 2017.

[Tan+23]      Yunhao Tang, Zhaohan Daniel Guo, Pierre Harvey Richemond, Bernardo Avila Pires, Yash Chandak, Remi Munos, Mark Rowland, Mohammad Gheshlaghi Azar, Charline Le Lan, Clare Lyle, András György, Shantanu Thakoor, Will Dabney, Bilal Piot, Daniele Calandriello, and Michal Valko. "Understanding Self-Predictive Learning for Reinforcement Learning". In: *International Conference on Machine Learning*. 2023.

[UZS22]       Masatoshi Uehara, Xuezhou Zhang, and Wen Sun. "Representation Learning for Online and Offline RL in Low-rank MDPs". In: *The Tenth International Conference on Learning Representations*. 2022.

[WSD15]       Niklas Wahlström, Thomas B Schön, and Marc Peter Deisenroth. "From Pixels to Torques: Policy Learning With Deep Dynamical Models". In: *International Conference on Machine Learning*. 2015.

[WSY20]       Ruosong Wang, Russ R Salakhutdinov, and Lin Yang. "Reinforcement Learning with General Value Function Approximation: Provably Efficient Approach via Bounded Eluder Dimension". In: *Neural Information Processing Systems*. 2020.

[WYDW21]      Tianhao Wu, Yunchang Yang, Simon Du, and Liwei Wang. "On Reinforcement Learning With Adversarial Corruption and Its Application to Block MDP". In: *International Conference on Machine Learning*. 2021.

[XFBJK23]     Tengyang Xie, Dylan J Foster, Yu Bai, Nan Jiang, and Sham M Kakade. "The Role of Coverage in Online Reinforcement Learning". In: *International Conference on Learning Representations*. 2023.

[XJ21]        Tengyang Xie and Nan Jiang. "Batch Value-Function Approximation With Only Realizability". In: *International Conference on Machine Learning*. 2021.

[YFK21]       Denis Yarats, Rob Fergus, and Ilya Kostrikov. "Image Augmentation Is All You Need: Regularizing Deep Reinforcement Learning From Pixels". In: *International Conference on Learning Representations*. 2021.

[ZGS21]       Dongruo Zhou, Quanquan Gu, and Csaba Szepesvari. "Nearly Minimax Optimal Reinforcement Learning for Linear Mixture Markov Decision Processes". In: *Conference on Learning Theory*. 2021.

[Zha06]       Tong Zhang. "From $\epsilon$-entropy to KL-entropy: Analysis of minimum information complexity density estimation". In: *The Annals of Statistics*. Vol. 34. 5. Institute of Mathematical Statistics, 2006, pp. 2180–2210.

[Zha22]       Tong Zhang. "Feel-Good Thompson Sampling for Contextual Bandits and Reinforcement Learning". In: *SIAM Journal on Mathematics of Data Science*. 2022.

[ZMCGL21]     Amy Zhang, Rowan McAllister, Roberto Calandra, Yarin Gal, and Sergey Levine. "Learning Invariant Representations for Reinforcement Learning Without Reconstruction". In: *International Conference on Learning Representations*. 2021.

[ZSUWAS22]    Xuezhou Zhang, Yuda Song, Masatoshi Uehara, Mengdi Wang, Alekh Agarwal, and Wen Sun. "Efficient Reinforcement Learning in Block MDPs: A Model-Free Representation Learning Approach". In: *International Conference on Machine Learning*. 2022.

# Contents

# A Omitted Results from Section 4: Algorithmic Modularity via Self-predictive Estimation

In this section, we remove the assumption of hindsight observability used in Section 4.1 and instantiate O2L in the general *online RL* setting. Rather than assume access to additional side-information, we adopt a *model-based representation learning* approach, and augment our ability to perform representation learning by equipping the representation learning algorithm with a *set of base MDPs* $\mathcal{M}_{\text{lat}}$ in addition to the decoder class $\Phi$. We will learn a representation by jointly fitting a decoder and the base (latent) dynamics, which is a common approach in practice [GKBNB19; HLBN19; Haf+19; HLNB21; Sch+20; SAGHCB20; Guo+22]. We firstly present in Appendix A.1 a new notion of *optimistic self-predictive regret* which combines self-predictive representation learning with a form of optimism over a learned latent model. We then show in Appendix A.2 that any representation learning oracle that attains low regret, when used within O2L (Algorithm 1), leads to observable-to-latent reductions that ensure low risk for *any* base algorithm $\text{ALG}_{\text{lat}}$, thereby achieving algorithmic modularity. Lastly, in Appendix A.3, we instantiate this oracle under natural structural and function approximation conditions, yielding end-to-end modularity and sample complexity guarantees.

## A.1 Self-predictive estimation

Our self-predictive representation learning oracles learn to fit a representation $\phi$ such that the *induced latent transitions* ($\phi_h(x_h)$ to $\phi_{h+1}(x_{h+1})$) can be accurately modeled by some base (latent) MDP $M_{\text{lat}} \in \mathcal{M}_{\text{lat}}$. To describe the objective, let us first introduce some notation. For a given MDP $M$ over either $\mathcal{S}$ (resp. $\mathcal{X}$), we write $M_h(r_h, s_{h+1} \mid s_h, a_h)$ (resp. $M_h(r_h, x_{h+1} \mid x_h, a_h)$) for the joint conditional distribution over rewards and next states. Next, for any $\phi \in \Phi$, we define the *pushforward model* for $M^\star_{\text{obs},h}$ induced by $\phi$ via:

$$\big[\phi_{h+1} \sharp M^\star_{\text{obs},h}\big](r, s' \mid x, a) := \sum_{x':\phi_{h+1}(x')=s'} M^\star_{\text{obs},h}(r, x' \mid x, a). \tag{6}$$

The *pushforward model* for $\phi$ captures the forward probability of the estimated latent state $\phi(x')$ given a current observation $x$. To measure distance between models, we will use squared Hellinger distance (e.g, Foster et al. [FKQR21]), defined via $D^2_{\mathsf{H}}(\mathbb{P}, \mathbb{Q}) = \int \big(\sqrt{\frac{d\mathbb{P}}{d\nu}} - \sqrt{\frac{d\mathbb{Q}}{d\nu}}\big)^2 d\nu$ for a common dominating measure $\nu$. Then, for a base model $M_{\text{lat}}$ and a decoder $\phi$, the *self-predictive error* of $(M_{\text{lat}}, \phi)$, at state-action pair $x_h, a_h$, is given by

$$[\Delta_h(M_{\text{lat}}, \phi)](x_h, a_h) := D^2_{\mathsf{H}}\big(M_{\text{lat},h}(\phi_h(x_h), a_h), \big[\phi_{h+1} \sharp M^\star_{\text{obs},h}\big](x_h, a_h)\big).$$

This term captures the ability of $M_{\text{lat},h}(\phi_h(x_h), a_h)$ to predict the next latent state $\phi_{h+1}(x_{h+1})$ which is obtained by the pushforward model $\big[\phi_{h+1} \sharp M^\star_{\text{obs},h}\big](x_h, a_h)$. Formally, in our model-based representation learning setup, we consider oracles which, for each iteration $t$ within O2L, take as input the trajectories collected so far and produce an estimate $(\widehat{M}^{(t)}_{\text{lat}}, \widehat{\phi}^{(t)})$ for the decoder and base model. The representation learning oracle's *self-predictive regret*, for the sequence $(\widehat{M}^{(t)}_{\text{lat}}, \widehat{\phi}^{(t)})$, is then defined as

$$\text{Reg}_{\text{self}}(T) = \sum_{t=1}^{T} \sum_{h=0}^{H} \mathbb{E}_{\pi^{(t)} \sim p^{(t)}} \mathbb{E}^{\pi^{(t)}} \Big[[\Delta_h(\widehat{M}^{(t)}_{\text{lat}}, \widehat{\phi}^{(t)})](x_h, a_h)\Big],$$

where $p^{(t)}$ represents a randomization distribution over the policy $\pi^{(t)}$.

On its own, minimizing this regret may lead to degenerate solutions, a widely observed phenomenon in practice [Tan+23]. For example, in a standard combination lock MDP (e.g., Agarwal et al.; Misra et al. [AJKS22; MHKL20]), a degenerate decoder-model pair that maps all observations to a single latent state will have zero self-predictive loss until we reach the goal, which can take exponentially long.[12] We address this via the notion of *optimistic estimation* used in Zhang; Foster et al. [Zha22; FGQRS23], which biases the objective towards latent models with high return. This leads to the

---

[12]This is similar to the observation that naive value function approximation methods, such as Fitted Q-Iteration, can fail to explore in online RL without optimism. We expect that given access to additional exploratory data (e.g., in the *Hybrid RL* setting of Song et al. [SZSBKS23]), the latent optimism term can be removed.

following *optimistic self-predictive regret*, defined for a parameter $\gamma > 0$, via

$$\text{Reg}_{\text{self;opt}}(T, \gamma) = \sum_{t=1}^{T} \sum_{h=0}^{H} \mathbb{E}_{\pi^{(t)} \sim p^{(t)}} \mathbb{E}^{\pi^{(t)}} \left[ [\Delta_h(\widehat{M}_{\text{lat}}^{(t)}, \widehat{\phi}^{(t)})](x_h, a_h) \right]$$

$$+ \gamma^{-1} \left( J^{M_{\text{lat}}^\star}(\pi_{M_{\text{lat}}^\star}) - J^{\widehat{M}_{\text{lat}}^{(t)}}(\pi_{\widehat{M}_{\text{lat}}^{(t)}}) \right). \tag{7}$$

We assume going forward that $\text{REP}_{\text{self;opt}}$ obtains low optimistic self-predictive regret; in Appendix A.3 we provide a maximum-likelihood-type estimator and conditions under which this holds.

**Assumption A.1.** *For a parameter $\gamma > 0$ and any (possibly adaptive) sequence $\pi^{(t)}$, with $\pi^{(t)} \sim p^{(t)}$, the online representation learning oracle $\text{REP}_{\text{self;opt}}$ is proper (i.e. outputs $\widehat{M}_{\text{lat}}^{(t)} \in \mathcal{M}_{\text{lat}}$ for all $t \in [T]$) and satisfies*

$$\mathbb{E}\left[\text{Reg}_{\text{self;opt}}(T, \gamma)\right] \leq \text{Est}_{\text{self;opt}}(T, \gamma),$$

*where $\text{Est}_{\text{self;opt}}(T, \gamma)$ is a known upper bound.*

We note that only the decoder $\widehat{\phi}^{(t)}$ is used within O2L; the model $\widehat{M}_{\text{lat}}^{(t)}$ is only used for analysis (and possibly within the representation learner $\text{REP}_{\text{self;opt}}$).

## A.2  Main result

We now state the main guarantee for O2L with self-predictive representation learning. Recall that $\text{Risk}_{\text{obs}}(TK)$ denotes the risk of the O2L reduction. Compared to the hindsight-observable setting, we require a slightly stronger performance guarantee from the base algorithm $\text{ALG}_{\text{lat}}$: our result scales with the *worst-case* expected risk for $\text{ALG}_{\text{lat}}$ over all $M_{\text{lat}} \in \mathcal{M}_{\text{lat}}$, defined via $\text{Risk}_{\text{base}}(K) := \sup_{M_{\text{lat}} \in \mathcal{M}_{\text{lat}}} \mathbb{E}[\text{Risk}(K, \text{ALG}_{\text{lat}}, M_{\text{lat}}))]$.

**Theorem A.1** (Risk bound for O2L under self-predictive estimation)**.** *Suppose $\text{REP}_{\text{self;opt}}$ satisfies Assumption A.1 with parameter $\gamma > 0$. Then Algorithm 1, with inputs $T, K, \Phi$, $\text{REP}_{\text{self;opt}}$, and $\text{ALG}_{\text{lat}}$ has expected risk*

$$\mathbb{E}[\text{Risk}_{\text{obs}}(TK)] \leq c_1 \cdot \text{Risk}_{\text{base}}(K) + c_2 \gamma \cdot \frac{K}{T} \text{Est}_{\text{self;opt}}(T, \gamma) + c_3 \gamma^{-1} \cdot KH,$$

*for absolute constants $c_1, c_2, c_3 > 0$.*

Theorem A.1 achieves sublinear risk as long as (i) the latent algorithm achieves sublinear risk $\text{Risk}_{\text{base}}(K)$ given access to the true states, and (ii) the self-predictive representation learning oracle achieves sublinear regret $\text{Est}_{\text{self;opt}}(T, \gamma)$ for an appropriate choice of $\gamma$.[13] Intuitively, our result scales with $\text{Risk}_{\text{base}}(K)$ instead of $\text{Risk}_\star(K)$ due to potential symmetries in the self-predictive objective. For example, there might be a representation-model pair $(\widehat{M}_{\text{lat}}, \widehat{\phi})$ that is identical to $(M_{\text{lat}}^\star, \phi^\star)$ *up to* permutations of the latent state space; these cannot be distinguished by a representation learning oracle that does not observe the latent states directly, and thus the base algorithm may be tasked with solving either of these base MDPs. As with Theorem 4.1, this result achieves algorithmic modularity (since O2L inherits the risk of the base algorithm), and is computationally efficient whenever the base algorithm and self-predictive representation learning oracle are efficient.

Let us provide some intuition behind the proof of Theorem A.1. Recall that, within the inner loop of O2L, the latent algorithm $\text{ALG}_{\text{lat}}$ interacts with the $\widehat{\phi}^{(t)}$-compressed dynamics generated by compressing the observations $x_h, a_h$ through the current decoder $\widehat{\phi}_h^{(t)}$ (Line 8). The crux of the analysis is the following observation: by the self-predictive representation learning guarantee, these dynamics, despite being possibly non-Markovian and generated from a POMDP (Definition I.1), are well approximated in squared Hellinger distance by the base model $\widehat{M}_{\text{lat}}^{(t)}$ estimated by $\text{REP}_{\text{self;opt}}$ (cf. Lemma I.2). We can then show that $\text{ALG}_{\text{lat}}$, when given data from the $\widehat{\phi}^{(t)}$-compressed dynamics, has risk (for solving $\widehat{M}_{\text{lat}}^{(t)}$) that is proportional to: i) its base risk if it were to observe states from $\widehat{M}_{\text{lat}}^{(t)}$, and ii) the Hellinger distance between $\widehat{M}_{\text{lat}}^{(t)}$ and the process induced by its $\widehat{\phi}^{(t)}$-compressed

---

[13]For example, in our estimator of Appendix A.3, we can first set $\gamma \approx KH/\text{Risk}_{\text{base}}(K)$ so that the third term matches $\text{Risk}_{\text{base}}(K)$, and then set $T$ so that the second term does.

dynamics. The last ingredient is the use of latent optimism in Eq. (7), through which the risk on $M_{\mathrm{lat}}^\star$ is upper bounded by the risk on $\widehat{M}_{\mathrm{lat}}^{(t)}$.

In the above, showing that $\mathrm{ALG}_{\mathrm{lat}}$ obtains low risk for $\widehat{M}_{\mathrm{lat}}^{(t)}$ (despite given data from a different process) is done by establishing a certain form of *corruption robustness* (Definition I.2). Indeed, Theorem A.1 is a special case of a more general theorem (Theorem H.1), which provides a bound that adapts to $\mathrm{ALG}_{\mathrm{lat}}$'s level of robustness. We obtain Theorem A.1 by showing that *any algorithm* satisfies the property we require (for a suitably slow rate), but we further show that tighter rates can be achieved by analyzing the specifics of various algorithms of interest (Appendix I.1.4).

### A.3 Instantiating the self-predictive estimation oracle

We now present an algorithm, SELFPREDICT.OPT (Algorithm 4 in Appendix H.1), which satisfies Assumption A.1 under additional technical conditions, allowing us to instantiate Theorem A.1 to give end-to-end guarantees. Before stating the main guarantee, we highlight a few technical difficulties regarding obtaining finite-sample guarantees for (online) self-predictive estimation, and use them to motivate our statistical assumptions and algorithm design.

**The statistics of (online) self-predictive estimation.** The first challenge is a realizability issue: when $\phi \neq \phi^\star$, we may not even be able to represent the objective $\phi \sharp M_{\mathrm{obs}}^\star$ as a latent model using only decoder and latent model realizability. Since we can never guarantee that $\phi = \phi^\star$ exactly in the presence of statistical errors, we must introduce a modelling assumption which lets us capture the pushforward models $\phi \sharp M_{\mathrm{obs}}^\star$. To this end, we introduce the mismatch functions, which are defined as follows.

**Definition A.1** (Mismatch functions). *For a decodable emission process $\psi^\star$ and decoder $\phi \in \Phi$, the* mismatch function *for $\phi$, $\Gamma_\phi = \{\Gamma_{\phi,h} : \mathcal{S} \to \Delta(\mathcal{S})\}_{h=1}^H$, is defined, for every $h \in [H]$, as the probability kernel*

$$\Gamma_{\phi,h}(s_h' \mid s_h) := \mathbb{P}_{x_h \sim \psi_h^\star(s_h)}(\phi_h(x_h) = s_h').$$

In the context of self-prediction, we show that the following *mismatch completeness* assumption suffices to capture the pushforward models $\phi \sharp M_{\mathrm{obs}}^\star$.

**Assumption A.2** (Mismatch completeness). *We have a model class $\mathcal{L}$ such that, for each $\phi \in \Phi$, and $M_{\mathrm{lat}} \in \mathcal{M}_{\mathrm{lat}}$, we have $\Gamma_\phi \circ M_{\mathrm{lat}} \in \mathcal{L}$, where*

$$[\Gamma_\phi \circ M_{\mathrm{lat}}]_h(r_h, s_{h+1} \mid s_h, a_h) := \sum_{s_{h+1}' \in \mathcal{S}} M_{\mathrm{lat},h}(r_h, s_{h+1}' \mid s_h, a_h) \Gamma_{\phi,h+1}(s_{h+1} \mid s_{h+1}').$$

In particular, Lemma D.8 establishes that

$$[\phi_{h+1} \sharp M_{\mathrm{obs},h}^\star](\cdot \mid x, a) = [\Gamma_\phi \circ M_{\mathrm{lat}}^\star]_h(\cdot \mid \phi_h^\star(x), a).$$

Accordingly, we view this assumption as a minimal way to realize the pushforward models $\phi \sharp M_{\mathrm{obs}}^\star$.

The second challenge is a *double-sampling* issue, which appears because the decoders in Eq. (7) are coupled at different horizons. We address this with a novel "debiased" maximum likelihood procedure that subtracts a form of excess risk (cf. Eq. (60)) to recover an unbiased estimator [Jia24]. Our debiased estimator and the mismatch completeness assumption can be viewed as analogous to the techniques and assumptions that are required for squared Bellman error minimization in the context of value function approximation [CJ19; JLM21].

The last issue stems from seeking an *online* estimation guarantee: the policies chosen by the latent algorithm are a function of the estimated decoders, which precludes the use of randomized estimators (e.g. exponential weights). We bypass this issue by appealing to the structural condition of coverability [XFBJK23], which allows us to restrict our attention to estimators that achieve low *offline* estimation error (via Lemma C.7).[14]

**Definition A.2** (State Coverability). *The state coverability coefficient for an MDP $M$ and a policy class $\Pi$ defined over a state space $\mathcal{Z}$, $C_{\mathrm{cov,st}}(M, \Pi)$, is given by*

$$C_{\mathrm{cov,st}}(M, \Pi) := \max_{h \in [H]} \min_{\mu \in \Delta(\mathcal{Z})} \max_{\pi \in \Pi} \max_{z \in \mathcal{Z}} \left\{ \frac{d_h^{M,\pi}(z)}{\mu(z)} \right\}. \tag{8}$$

---

[14]More generally, we expect that our results can be extended to any "decoupling coefficient" [Zha22; AZ22].

We require coverability in $M_{\mathrm{obs}}^{\star}$ over the set of (observation-space) policies played by the O2L reduction (cf. Line 7). Again appealing to the mismatch functions, we can express this as an assumption about the base dynamics $M_{\mathrm{lat}}^{\star}$; we show (Lemma D.1) that the latter is *equivalent* to assuming coverability in $M_{\mathrm{lat}}^{\star}$ over the set of stochastic policies

$$
\Gamma_{\Phi} \circ \Pi_{\mathrm{lat}} \coloneqq \left\{ [\Gamma_{\phi} \circ \pi_{\mathrm{lat}}]_h(a \mid s) = \sum_{s' \in \mathcal{S}} \Gamma_{\phi,h}(s' \mid s) \pi_{\mathrm{lat},h}(a \mid s') \mid \phi \in \Phi, \pi_{\mathrm{lat}} \in \Pi_{\mathrm{lat}} \right\}, \quad (9)
$$

where $\Pi_{\mathrm{lat}}$ denotes the set of policies that $\mathrm{ALG}_{\mathrm{lat}}$ may execute. While this set may appear complicated, it is sufficient to assume coverability over the set of all *deterministic* non-stationary policies on $M_{\mathrm{lat}}^{\star}$.[15]

**Guarantee for our self-predictive estimation oracle.**   With these prerequisites, the main guarantee for our estimator, SELFPREDICT.OPT (Algorithm 4), is as follows.

**Lemma A.1** (Optimistic self-predictive estimation via SELFPREDICT.OPT). *Let $\Pi_{\mathrm{lat}}$ denote the set of policies played by* $\mathrm{ALG}_{\mathrm{lat}}$, *and* $C_{\mathrm{cov,st}} = C_{\mathrm{cov,st}}(M_{\mathrm{lat}}^{\star}, \Gamma_{\Phi} \circ \Pi_{\mathrm{lat}})$ *be the state coverability parameter on* $M_{\mathrm{lat}}^{\star}$ *over the set of stochastic policies* $\Gamma_{\Phi} \circ \Pi_{\mathrm{lat}}$ *(Eq. (9)). Then, for any $\gamma > 0$, under decoder realizability ($\phi^{\star} \in \Phi$), base model realizability ($M_{\mathrm{lat}}^{\star} \in \mathcal{M}_{\mathrm{lat}}$), and mismatch function completeness with class $\mathcal{L}_{\mathrm{lat}}$ (Assumption A.2), the estimator in Algorithm 4 with inputs $\Phi, \mathcal{M}_{\mathrm{lat}}, \mathcal{L}_{\mathrm{lat}}$, and $\gamma$ satisfies Assumption A.1 with*[16]

$$
\mathsf{Est}_{\mathrm{self;opt}}(T, \gamma) = \widetilde{\mathcal{O}}\left( \sqrt{H C_{\mathrm{cov,st}} |\mathcal{A}| T \log(|\mathcal{M}_{\mathrm{lat}}||\mathcal{L}_{\mathrm{lat}}||\Phi|)} \right).
$$

Instantiating Theorem A.1 with the above representation learning oracle, we obtain the following algorithmic modularity result.

**Corollary A.1** (Algorithmic modularity via SELFPREDICT.OPT). *Under the same conditions as in Lemma A.1, and for any base algorithm* $\mathrm{ALG}_{\mathrm{lat}}$, *O2L with inputs $T, K, \Phi$,* SELFPREDICT.OPT, *and* $\mathrm{ALG}_{\mathrm{lat}}$ *achieves*

$$
\mathbb{E}[\mathsf{Risk}_{\mathrm{obs}}(TK)] \lesssim c_1 \cdot \mathsf{Risk}_{\mathrm{base}}(K) + c_2 \gamma \cdot \frac{K}{\sqrt{T}} \sqrt{H C_{\mathrm{cov,st}} |\mathcal{A}|} \log(|\mathcal{M}_{\mathrm{lat}}||\mathcal{L}_{\mathrm{lat}}||\Phi|) + c_3 \gamma^{-1} \cdot KH,
$$

*for absolute constants $c_1, c_2, c_3$. Consequently, for any* $\mathrm{ALG}_{\mathrm{lat}}$ *with base risk* $\mathsf{Risk}_{\mathrm{base}}(K)$, *setting $\gamma$ and $T$ appropriately gives*

$$
\mathbb{E}[\mathsf{Risk}_{\mathrm{obs}}(TK)] \lesssim \mathsf{Risk}_{\mathrm{base}}(K),
$$

*with a number of trajectories* $TK = \widetilde{\mathcal{O}}\big( K^5 H^3 C_{\mathrm{cov,st}} |\mathcal{A}| \log^2(|\mathcal{M}_{\mathrm{lat}}||\mathcal{L}_{\mathrm{lat}}||\Phi|) / (\mathsf{Risk}_{\mathrm{base}}(K))^4 \big)$.

For example, if $\mathrm{ALG}_{\mathrm{lat}}$ is a base algorithm with $\mathsf{Risk}_{\mathrm{base}}(K) = \mathcal{O}(K^{-1/2})$, setting $\gamma$ and $T$ appropriately gives an expected risk of $\varepsilon$ with a number of trajectories $TK = \widetilde{\mathcal{O}}\big( H^3 C_{\mathrm{cov,st}} |\mathcal{A}| (\log(|\mathcal{M}_{\mathrm{lat}}||\mathcal{L}_{\mathrm{lat}}||\Phi|))^2 / \varepsilon^{14} \big)$. This result shows that statistical modularity can be achieved *up to* $\log(|\mathcal{L}_{\mathrm{lat}}|)$ *factors* for every base MDP class $\mathcal{M}_{\mathrm{lat}}$ which is subsumed by coverability, including tabular MDPs and low-rank MDPs.[17] Compared to our positive result for the case of pushforward coverability (Section 3.3), this imposes less dynamics assumptions (since coverability is implied by pushforward coverability) but requires more representational assumptions (namely, access to the mismatch-complete class $\mathcal{L}_{\mathrm{lat}}$). We further remark that the mismatch completeness assumption always holds for i) the Block MDP setting, since we can always construct $\mathcal{L}_{\mathrm{lat}}$ such that $\log(|\mathcal{L}_{\mathrm{lat}}|) = \mathcal{O}(HS^2)$, and ii) every MDP class $\mathcal{M}_{\mathrm{lat}}$ whenever we also have a realizable set of emission processes ($\psi^{\star} \in \Psi$), since we can construct $\mathcal{L}_{\mathrm{lat}}$ such that $\log(|\mathcal{L}_{\mathrm{lat}}|) = \log(|\Phi||\mathcal{M}_{\mathrm{lat}}||\Psi|)$. However, the mismatch completeness assumption may be more general than either of these settings.

---

[15]This follows from Lemma D.3 by noting that each maximum on the right hand side of Eq. (13) is attained by a deterministic non-stationary policy.

[16]In this section, the notations $\widetilde{\mathcal{O}}$ and $\lesssim$ ignores constants and logarithmic factors of: $H, C_{\mathrm{cov,st}}, |\mathcal{A}|, T$, and $\log(|\mathcal{M}_{\mathrm{lat}}||\mathcal{L}_{\mathrm{lat}}||\Phi|)$.

[17]This provides a partial answer to the "Model Class + Coverability" open question of Figure 1.

Our results can be viewed as providing a theoretical justification for self-predictive representation learning, which has been widely used in empirical works [GKBNB19; SAGHCB20]. We consider self-prediction's ability to obtain universal observable-to-latent reductions as a strong indicator that it merits further theoretical study. In particular, many empirical works propose heuristics to alleviate the degeneracy/non-uniqueness issues inherent with self-prediction [GKBNB19; SAGHCB20; HPBL23; Tan+23]. Our methods provide a principled way to address these, and it would be interesting to investigate whether this is also *empirically* effective. In general, however, it is unclear whether our loss admits a computationally efficient implementation, due to the presence of optimism. Towards this, a fascinating direction for future work is understanding how self-predictive estimation can be used to obtain algorithmic modularity *without* the addition of optimism over the base (latent) models.

# B  Additional Discussion of Related Work

In this section, we discuss aspects of related work not already covered in greater detail.

**Reinforcement learning under latent dynamics (or, with rich observations).**  Reinforcement learning under latent dynamics (or, with rich observations) has received extensive investigation in recent years, however most works have been focused on the *Block MDP* model in which the latent state space is tabular/finite [KAL16; DKJADL19; MHKL20; ZSUWAS22; MFR23] (see also the the closely related framework of *Low-Rank MDPs* [AKKS20; MCKJA24; ZSUWAS22; UZS22; MBFR23]). Beyond tabular spaces, Dean et al.; Dean et al.; Mhammedi et al. [DMRY20; DR21; Mha+20] consider continuous *linear* dynamics, Misra et al. [MLJL21] considers factored (but discrete) latent dynamics, Efroni et al.; Efroni et al.; Mhammedi et al. [EMKAL22; EFMKL22; MFR24] consider the Exogenous Block MDP problem in which a tabular latent state space is augmented with a non-controllable ("exogenous") factor, and Song et al. [SWFK24] consider Lipshitz continuous dynamics. To our knowledge, our work is the first to: i) explore reinforcement learning under general latent dynamics, in particular in settings where the latent space itself admits function approximation, and ii) take a more modular approach (cf. the taxonomy of Section 3).

On the algorithmic side, the works of Uehara et al. [UZS22] and Zhang et al. [ZSUWAS22], which consider Low-Rank MDPs and Block MDPs respectively, can be viewed as interleaving representation learning with "latent" reinforcement learning algorithms that assume access to a good representation, and were an inspiration for this work. However, the algorithmic details and analyses are highly specialized to Block/Low-Rank MDPs, and unlikely to be directly applicable to reinforcement learning under general latent dynamics. Other works with a modular flavor include:

- Feng et al. [FWYDY20] solve tabular Block MDPs by combining a black-box latent algorithm with an "unsupervised learning oracle" for representation learning. This approach only leads to guarantees for tabular Block MDPs, and it is unclear whether the unsupervised learning oracle their approach requires can be constructed in natural settings.

- Wu et al. [WYDW21] solve tabular block MDPs by combining a corruption-robust latent algorithm with a representation learning procedure based on clustering. Again, this work is restricted to the tabular setting, and requires a separation condition which may not be satisfied in general.

**General complexity measures for reinforcement learning.**  Another line of research provides general complexity measures that enable sample-efficient reinforcement learning, including Bellman rank [JKALS17; SJKAL19; Du+21; JLM21], eluder dimension [RVR13], coverability [XFBJK23], and the Decision-Estimation Coefficient (DEC) [FKQR21; FGH23; FGQRS23]. Bellman rank and other complexity measures based on average Bellman error [JKALS17; SJKAL19; Du+21; JLM21] are insufficient to characterize learnability under general latent dynamics, as there are classes $\mathcal{M}_{\mathrm{lat}}$ that are known to be learnable, yet do not have bounded Bellman rank or Bellman-Eluder dimension [EMKAL22; XFBJK23]. Meanwhile, variants of Bellman rank based on squared Bellman error or related notions of error can [XFBJK23; AFJSX24] address this problem for some settings, but satisfying the modeling/realizability assumptions (e.g., Bellman completeness) required by these methods in the latent-dynamics setting is non-trivial. For example, the crux of our sample complexity bounds under latent pushforward coverability in Section 3 (Theorem 3.2) is to prove a rather involved structural result which shows that Bellman completeness can indeed be satisfied under this assumption, but it is unclear whether these techniques can be applied to more general latent dynamics classes. We expect that it is possible to bound the Decision-Estimation Coefficient [FKQR21; FGH23; FGQRS23] for the framework, but deriving efficient algorithms using this framework is non-trivial.

## C  Technical Tools

**Lemma C.1.** *For any sequence of real-valued random variables $(X_t)_{t \leq T}$ adapted to a filtration $(\mathscr{F}_t)_{t \leq T}$, it holds that with probability at least $1 - \delta$,*

$$\sum_{t=1}^{T} X_t \leq \sum_{t=1}^{T} \log\big(\mathbb{E}_{t-1}\big[e^{X_t}\big]\big) + \log\big(\delta^{-1}\big).$$

**Lemma C.2** (Freedman's inequality (e.g., Agarwal et al. [AHKLLS14])). *Let $(X_t)_{t \leq T}$ be a real-valued martingale difference sequence adapted to a filtration $(\mathscr{F}_t)_{t \leq T}$. If $|X_t| \leq R$ almost surely, then for any $\eta \in (0, 1/R)$, with probability at least $1 - \delta$,*

$$\sum_{t=1}^{T} X_t \leq \eta \sum_{t=1}^{T} \mathbb{E}_{t-1}\big[X_t^2\big] + \frac{\log\big(\delta^{-1}\big)}{\eta}.$$

**Lemma C.3** (Corollary of Lemma C.2). *Let $(X_t)_{t \leq T}$ be a sequence of random variables adapted to a filtration $(\mathscr{F}_t)_{t \leq T}$. If $0 \leq X_t \leq R$ almost surely, then with probability at least $1 - \delta$,*

$$\sum_{t=1}^{T} X_t \leq \frac{3}{2} \sum_{t=1}^{T} \mathbb{E}_{t-1}[X_t] + 4R \log\big(2\delta^{-1}\big),$$

*and*

$$\sum_{t=1}^{T} \mathbb{E}_{t-1}[X_t] \leq 2 \sum_{t=1}^{T} X_t + 8R \log\big(2\delta^{-1}\big).$$

**Lemma C.4** (Lemma D.2 of Foster et al. [FHQR24]). *Let $(\mathcal{X}_1, \mathfrak{F}_1), \ldots, (\mathcal{X}_n, \mathfrak{F}_n)$ be a sequence of measurable spaces, and let $\mathcal{X}^{(i)} = \prod_{t=1}^{i} \mathcal{X}_t$ and $\mathfrak{F}^{(i)} = \otimes_{t=1}^{i} \mathfrak{F}_t$. For each $i$, let $P^{(i)}$ and $Q^{(i)}$ be probability kernels from $(\mathcal{X}^{(i-1)}, \mathfrak{F}^{(i-1)})$ to $(\mathcal{X}_i, \mathfrak{F}_i)$. Let $P$ and $Q$ be the laws of $X_1, \ldots, X_n$ under $X_i \sim P^{(i)}(\cdot \mid X_{1:i-1})$ and $X_i \sim Q^{(i)}(\cdot \mid X_{1:i-1})$, respectively. Then it holds that*

$$D_{\mathsf{H}}^2(P, Q) \leq 7 \,\mathbb{E}_P \left[ \sum_{i=1}^{n} D_{\mathsf{H}}^2(P^{(i)}(\cdot \mid X_{1:i-1}), Q^{(i)}(\cdot \mid X_{1:i-1})) \right]$$

**Lemma C.5** (Lemma A.11 of Foster et al. [FKQR21]). *Let $\mathbb{P}$ and $\mathbb{Q}$ be probability measures on $(\mathcal{X}, \mathfrak{F})$. For all $h : \mathcal{X} \to \mathbb{R}$ with $0 \leq h(X) \leq R$ almost surely under $\mathbb{P}$ and $\mathbb{Q}$, we have*

$$\mathbb{E}_{\mathbb{P}}[h(X)] \leq 3 \,\mathbb{E}_{\mathbb{Q}}[h(X)] + 4RD_{\mathsf{H}}^2(\mathbb{P}, \mathbb{Q}).$$

**Lemma C.6** (Lemma 1 of Jiang et al. [JKALS17]). *For any $f : \mathcal{X} \times \mathcal{A} \to [0, 1]$, $\pi : \mathcal{S} \times [H] \to \Delta(\mathcal{A})$, we have*

$$\mathbb{E}_{x_1}[f(x_1, \pi(x_1))] - J(\pi) = \sum_{h=1}^{H} \mathbb{E}^{\pi}[f(x_h, a_h) - \mathcal{T}^{\pi} f(x_h, a_h)].$$

**Lemma C.7** (Offline-to-online conversion under coverability [XFBJK23; FHQR24]). *Let $M$ be an MDP over state space $\mathcal{Z}$, $\Pi$ be a policy set, and $C_{\mathsf{cov}} = C_{\mathsf{cov}}(M, \Pi)$ be the (state-action) coverability coefficient for $M$ and $\Pi$ (Definition D.3). Let $p^{(t)} \in \Delta(\Pi)$ be a sequence of distributions over $\Pi$, and $g_h^{(t)} : \mathcal{Z} \times \mathcal{A} \to [0, 1]$ be a sequence of functions. Then we have that*

$$\sum_{t=1}^{T} \sum_{h=1}^{H} \mathbb{E}_{\pi^{(t)} \sim p^{(t)}} \mathbb{E}^{\pi^{(t)}} \big[g_h^{(t)}(x_h, a_h)\big]$$

$$\leq \mathcal{O}\left( \sqrt{HC_{\mathsf{cov}} \log(T) \sum_{t=1}^{T} \sum_{h=1}^{H} \sum_{i=1}^{t-1} \mathbb{E}_{\pi^{(i)} \sim p^{(i)}} \mathbb{E}^{\pi^{(i)}} \big[g_h^{(t)}(x_h, a_h)\big]} + HC_{\mathsf{cov}} \right).$$

# D  Structural Properties of Coverability and Mismatch Functions

This appendix contains structural results regarding coverability and the mismatch functions. We firstly recall the definition of the mismatch functions.

**Definition D.1** (Mismatch functions). *For decodable emission process $\psi^\star$, decoder $\phi \in \Phi$ and $h \in [H]$, we define the* mismatch function *for $\phi$, $\Gamma_{\phi,h} : \mathcal{S} \to \Delta(\mathcal{S})$, as the probability kernel*

$$\Gamma_{\phi,h}(s'_h \mid s_h) := \mathbb{P}_{x_h \sim \psi_h^\star(s_h)}(\phi_h(x_h) = s'_h).$$

We also recall the definition of state coverability.

**Definition D.2** (State Coverability). *The coverability coefficient for an MDP $M$ and a policy class $\Pi$ defined over a state space $\mathcal{Z}$, $C_{\mathrm{cov,st}}(M, \Pi)$, is given by*

$$C_{\mathrm{cov,st}}(M, \Pi) := \max_{h \in [H]} \min_{\mu \in \Delta(\mathcal{Z})} \max_{\pi \in \Pi} \max_{z \in \mathcal{Z}} \left\{ \frac{d_h^{M,\pi}(z)}{\mu(z)} \right\}. \tag{10}$$

We also define the related notion of state-action coverability.

**Definition D.3** (State-Action Coverability). *The coverability coefficient for an MDP $M$ and a policy class $\Pi$ defined over a state space $\mathcal{Z}$ and action space $\mathcal{A}$, $C_{\mathrm{cov}}(M, \Pi)$, is given by*

$$C_{\mathrm{cov}}(M, \Pi) := \max_{h \in [H]} \min_{\mu \in \Delta(\mathcal{Z} \times \mathcal{A})} \max_{\pi \in \Pi} \max_{z,a \in \mathcal{Z} \times \mathcal{A}} \left\{ \frac{d_h^{M,\pi}(z,a)}{\mu(z,a)} \right\}. \tag{11}$$

In the remainder of the section, we let $\Pi_{\mathrm{lat}} \subseteq \{\mathcal{S} \times [H] \to \Delta(\mathcal{A})\}$ denote an arbitrary set of latent policies, and

$$\Gamma_\Phi \circ \Pi_{\mathrm{lat}} = \left\{ [\Gamma_\phi \circ \pi_{\mathrm{lat}}]_h(a \mid s) := \sum_{s' \in \mathcal{S}} \Gamma_{\phi,h}(s' \mid s)\pi_{\mathrm{lat},h}(a \mid s') \mid \phi \in \Phi, \pi_{\mathrm{lat}} \in \Pi_{\mathrm{lat}} \right\}. \tag{12}$$

**Lemma D.1** (State coverability is invariant to rich observations). *Let $M_{\mathrm{obs}}^\star = \langle\!\langle M_{\mathrm{lat}}^\star, \psi^\star \rangle\!\rangle$. Then, we have*

$$C_{\mathrm{cov,st}}(M_{\mathrm{obs}}^\star, \Pi_{\mathrm{lat}} \circ \Phi) = C_{\mathrm{cov,st}}(M_{\mathrm{lat}}^\star, \Gamma_\Phi \circ \Pi_{\mathrm{lat}}).$$

*Furthermore, letting $\{\mu_{\mathrm{lat},h} \in \Delta(\mathcal{S})\}_{h \in [H]}$ denote the distribution which witnesses the right-hand-side, the left-hand-side is witnessed by the distribution*

$$\mu_{\mathrm{obs},h}(x) = \psi_h^\star(x \mid \phi_h^\star(x))\mu_{\mathrm{lat},h}(\phi_h^\star(x)).$$

The lemma follows from the following two observations.

**Lemma D.2.** *Let $\{\Gamma_\phi\}_{\phi \in \Phi}$ denote the mismatch functions for emission $\psi^\star$, and let $M_{\mathrm{obs}} = \langle\!\langle M_{\mathrm{lat}}, \psi^\star \rangle\!\rangle$. Then, for any $\pi_{\mathrm{lat}} \in \Pi_{\mathrm{lat}}$, $\phi \in \Phi$, $h \in [H]$, $x \in \mathcal{X}$, we have*

$$d_h^{M_{\mathrm{obs}}, \pi_{\mathrm{lat}} \circ \phi}(x) = \psi_h^\star(x \mid \phi_h^\star(x))d_h^{M_{\mathrm{lat}}, \Gamma_\phi \circ \pi_{\mathrm{lat}}}(\phi_h^\star(x)).$$

**Proof of Lemma D.2.** Below, we write $s_h = \phi^\star(x_h)$. We proceed by induction, simply writing $d_{\mathrm{obs},h} := d_h^{M_{\mathrm{obs}}, \pi_{\mathrm{lat}} \circ \phi}$ and $d_{\mathrm{lat},h} := d_h^{M_{\mathrm{lat}}, \Gamma_\phi \circ \pi_{\mathrm{lat}}}$. The base case ($h = 1$) is obtained by noting that $d_{\mathrm{lat},1}(s) = P_{\mathrm{lat},1}(s \mid \emptyset)$ while $d_{\mathrm{obs},1}(x) = P_{\mathrm{obs},1}(x \mid \emptyset) = \psi_1^\star(x \mid s)P_{\mathrm{lat},1}(s \mid \emptyset)$. For the general case, via the Bellman flow equations, we have

$$
\begin{aligned}
d_{\mathrm{obs},h}(x_h) &= \sum_{x_{h-1},a_{h-1} \in \mathcal{X} \times \mathcal{A}} P_{\mathrm{obs},h}(x_h \mid x_{h-1}, a_{h-1})d_{\mathrm{obs},h-1}(x_{h-1})\pi_{\mathrm{lat}}(a_{h-1} \mid \phi(x_{h-1})) \\
&= \psi(x_h \mid s_h) \sum_{x_{h-1},a_{h-1} \in \mathcal{X} \times \mathcal{A}} P_{\mathrm{lat},h}(s_h \mid s_{h-1}, a_{h-1})d_{\mathrm{lat},h-1}(s_{h-1})\psi(x_{h-1} \mid s_{h-1}) \\
&\qquad\qquad\qquad\qquad\qquad \times \pi_{\mathrm{lat}}(a_{h-1} \mid \phi(x_{h-1})) \\
&= \psi(x_h \mid s_h) \sum_{s_{h-1},a_{h-1} \in \mathcal{S} \times \mathcal{A}} P_{\mathrm{lat},h}(s_h \mid s_{h-1}, a_{h-1})d_{\mathrm{lat},h-1}(s_{h-1}) \\
&\qquad\qquad\qquad\qquad \times \sum_{x_{h-1}:\phi^\star(x_{h-1})=s_{h-1}} \psi(x_{h-1} \mid s_{h-1})\pi_{\mathrm{lat}}(a_{h-1} \mid \phi(x_{h-1})).
\end{aligned}
$$

The result is obtained by noting that

$$\Gamma_\phi \circ \pi_{\text{lat}}(a_{h-1} \mid s_{h-1}) = \sum_{s' \in \mathcal{S}} \Gamma_\phi(s' \mid s_{h-1})\pi_{\text{lat}}(a_{h-1} \mid s')$$

$$= \sum_{s' \in \mathcal{S}} \sum_{x_{h-1}:\phi^\star(x_{h-1})=s_{h-1}} \psi(x_{h-1} \mid s_{h-1})\mathbb{I}\{\phi(x_{h-1}) = s'\}\pi_{\text{lat}}(a_h \mid s')$$

$$= \sum_{x_{h-1}:\phi^\star(x_{h-1})=s_{h-1}} \psi(x_{h-1} \mid s_{h-1})\pi_{\text{lat}}(a_{h-1} \mid \phi(x_{h-1})),$$

where the second line follows from the definition of the mismatch functions. □

**Lemma D.3** (Equivalence of state coverability and cumulative state reachability). *Let $M$ be an MDP defined over a state space $\mathcal{Z}$. The following definition is equivalent to Definition D.2:*

$$C_{\text{cov,st}}(M, \Pi) := \max_{h \in [H]} \sum_{z \in \mathcal{Z}} \max_{\pi \in \Pi} d_h^{M,\pi}(z). \tag{13}$$

**Proof of Lemma D.3.** Straightforward adaptation of the proof of Lemma 3 from Xie et al. [XFBJK23]. □

**Proof of Lemma D.1.** Using Lemma D.2 and Lemma D.3, we have

$$C_{\text{cov,st}}(M_{\text{obs}}, \Pi_{\text{lat}} \circ \Phi) = \max_{h \in [H]} \sum_{x \in \mathcal{X}} \max_{\pi_{\text{lat}}, \phi} d_{\text{obs}}^{\pi_{\text{lat}} \circ \phi}(x)$$

$$= \max_{h \in [H]} \sum_{x \in \mathcal{X}} \max_{\pi_{\text{lat}}, \phi} \psi^\star(x \mid \phi^\star(x)) d_{\text{lat}}^{\Gamma_\phi \circ \pi_{\text{lat}}}(\phi^\star(x))$$

$$= \max_{h \in [H]} \sum_{s \in \mathcal{S}} \sum_{x:\phi^\star(x)=s} \max_{\pi_{\text{lat}}, \phi} \psi^\star(x \mid s) d_{\text{lat}}^{\Gamma_\phi \circ \pi_{\text{lat}}}(s)$$

$$= \max_{h \in [H]} \sum_{s \in \mathcal{S}} \max_{\pi_{\text{lat}}, \phi} d_{\text{lat}}^{\Gamma_\phi \circ \pi_{\text{lat}}}(s) \sum_{x:\phi^\star(x)=s} \psi^\star(x \mid s)$$

$$= C_{\text{cov,st}}(M_{\text{lat}}, \Gamma_\Phi \circ \Pi_{\text{lat}}).$$

□

Lastly, we show that state-action coverability is bounded by state coverability times the size of the action set.

**Lemma D.4** (State-action coverability bound). *For any MDP $M$ and policy set $\Pi$, we have*

$$C_{\text{cov}}(M, \Pi) \le C_{\text{cov,st}}(M, \Pi)|\mathcal{A}|.$$

**Proof of Lemma D.4.** Let $\mu_s \in \Delta(\mathcal{Z})$ witness $C_{\text{cov,st}}(M, \Pi)$. Fix $h \in [H]$, which we omit below for cleanliness. Then, we have

$$\min_{\mu_{s,a} \in \Delta(\mathcal{Z} \times \mathcal{A})} \max_{\pi \in \Pi} \max_{z,a \in \mathcal{Z} \times \mathcal{A}} \left\{ \frac{d^{M,\pi}(z,a)}{\mu_{s,a}(z,a)} \right\} \le \max_{\pi \in \Pi} \max_{z,a \in \mathcal{Z} \times \mathcal{A}} \left\{ \frac{d^{M,\pi}(z)\pi(a \mid z)}{\mu_s(z)^{1/|\mathcal{A}|}} \right\}$$

$$\le |\mathcal{A}| \max_{\pi \in \Pi} \max_{z \in \mathcal{Z}} \left\{ \frac{d^{M,\pi}(z)}{\mu_s(z)} \right\}$$

$$= C_{\text{cov,st}}(M, \Pi)|\mathcal{A}|.$$

□

**Lemma D.5** (Pushforward coverability is invariant to rich observations). *Let $C_{\text{push}}(M)$ denote the pushforward coverability parameter for an MDP $M$ (Definition 3.3), and $M_{\text{obs}}^\star := \langle\!\langle M_{\text{lat}}^\star, \psi^\star \rangle\!\rangle$. Then, we have*

$$C_{\text{push}}(M_{\text{obs}}^\star) = C_{\text{push}}(M_{\text{lat}}^\star).$$

*Furthermore, letting $\{\mu_{\mathsf{lat},h} \in \Delta(\mathcal{S})\}_{h \in [H]}$ denote the distribution which witnesses the right-hand-side, the left-hand-side is witnessed by the distribution*

$$\mu_{\mathsf{obs},h}(x) = \psi_h^\star(x \mid \phi_h^\star(x))\mu_{\mathsf{lat},h}(\phi_h^\star(x)).$$

This follows from an analogous equivalence of pushforward coverability and cumulative *conditional reachability*.

**Lemma D.6** (Equivalence of pushforward coverability and cumulative conditional reachability)**.** *Let $M$ be an MDP defined over a state space $\mathcal{Z}$ with transition kernel $P$. The following definition is equivalent to pushforward coverability (Definition 3.3):*

$$C_{\mathsf{push}}(M) := \max_{h \in [H]} \sum_{z' \in \mathcal{Z}} \max_{z,a \in \mathcal{Z} \times \mathcal{A}} P_h(z' \mid z, a).$$

**Proof of Lemma D.6.** Fix $h \in [H]$, whose dependence we omit below. For the first direction, letting $\mu$ denote the pushforward coverability distribution, we have:

$$\sum_{z' \in \mathcal{Z}} \max_{z,a \in \mathcal{Z} \times \mathcal{A}} P(z' \mid z, a) = \sum_{z' \in \mathcal{Z}} \max_{z,a \in \mathcal{Z} \times \mathcal{A}} \frac{P(z' \mid z, a)}{\mu(z')}\mu(z') \leq C_{\mathsf{push}} \sum_{z' \in \mathcal{Z}} \mu(z') = C_{\mathsf{push}}.$$

For the second direction, taking $\mu(z') \propto \max_{z,a} P(z' \mid z, a)$, we have

$$\min_{\mu \in \Delta(\mathcal{Z})} \max_{z,a,z' \in \mathcal{Z} \times \mathcal{A} \times \mathcal{Z}} \frac{P(z' \mid z, a)}{\mu(z')} \leq \max_{z,a,z' \in \mathcal{Z} \times \mathcal{A} \times \mathcal{Z}} \frac{P(z' \mid z, a)}{\max_{\tilde{z},\tilde{a}} P(z' \mid \tilde{z}, \tilde{a})} \sum_{z'} \max_{\tilde{z},\tilde{a}} P(\tilde{z}' \mid \tilde{z}, \tilde{a})$$

$$\leq \sum_{z'} \max_{z,a} P(z' \mid z, a).$$

$\square$

**Proof of Lemma D.5.** This result follows by Lemma D.6 since,

$$\begin{aligned}
C_{\mathsf{push}}(M_{\mathsf{obs}}) &= \sum_{x' \in \mathcal{X}} \max_{x,a} P_{\mathsf{obs}}(x' \mid x, a) \\
&= \sum_{s' \in \mathcal{S}} \sum_{x':\phi^\star(x')=s'} \max_{x,a} \psi^\star(x' \mid s')P_{\mathsf{lat}}(s' \mid \phi^\star(x), a) \\
&= \sum_{s' \in \mathcal{S}} \max_{x,a} P_{\mathsf{lat}}(s' \mid \phi^\star(x), a) \sum_{x':\phi^\star(x')=s'} \psi^\star(x' \mid s') \\
&= \sum_{s' \in \mathcal{S}} \max_{s,a} P_{\mathsf{lat}}(s' \mid s, a) = C_{\mathsf{push}}(M_{\mathsf{lat}}).
\end{aligned}$$

$\square$

We next show that the mismatch functions can be used to express the observation-level backups for any function of the decoders. For any $g : \mathcal{S} \to \mathbb{R}$, $h \in [H]$, we define the function $[\Gamma_{\phi,h} \circ g] : \mathcal{S} \to \mathbb{R}$

$$[\Gamma_{\phi,h} \circ g](s) := \sum_{s' \in \mathcal{S}} \Gamma_{\phi,h}(s' \mid s)g(s').$$

We further overload the Bellman operator notation and define, for any $g : \mathcal{S} \to \mathbb{R}$ and $M_{\mathsf{lat}} = (r_{\mathsf{lat}}, P_{\mathsf{lat}})$,

$$[\mathcal{T}_h^{M_{\mathsf{lat}}} g](s, a) = r_{\mathsf{lat}}(s, a) + \mathbb{E}_{s' \sim P_{\mathsf{lat}}(s,a)}[g(s')].$$

**Lemma D.7.** *Let $M_{\mathsf{obs}} = \langle\!\langle M_{\mathsf{lat}}, \psi^\star \rangle\!\rangle$, $\phi^\star := (\psi^\star)^{-1}$, $\phi \in \Phi$, and $\Gamma_\phi$ be the mismatch function for emission $\psi^\star$ (Definition D.1). Then, for any $f_{\mathsf{lat}} : \mathcal{S} \times \mathcal{A} \to \mathbb{R}$, $h \in [H]$, and $(x, a) \in \mathcal{X} \times \mathcal{A}$, we have*

$$\left[\mathcal{T}_h^{M_{\mathsf{obs}}}(f_{\mathsf{lat}} \circ \phi_{h+1})\right](x, a) = \left[\mathcal{T}_h^{M_{\mathsf{lat}}}(\Gamma_{\phi,h+1} \circ V_{f_{\mathsf{lat}}})\right](\phi_h^\star(x), a).$$

**Proof of Lemma D.7.** Let $f := f_{\texttt{lat}}$, $h \in [H]$, and $(x, a) \in \mathcal{X} \times \mathcal{A}$ be given. Then, we have:

$$\left[\mathcal{T}_h^{M_{\texttt{obs}}}(f \circ \phi_{h+1})\right](x, a)$$

$$= r_{\texttt{lat},h}(\phi_h^\star(x), a) + \mathbb{E}_{s_{h+1} \sim P_{\texttt{lat},h}(\phi_h^\star(x),a)} \mathbb{E}_{x_{h+1} \sim \psi_{h+1}^\star(s_{h+1})}[V_f(\phi(x_{h+1}))]$$

$$= r_{\texttt{lat},h}(\phi_h^\star(x), a) + \mathbb{E}_{s_{h+1} \sim P_{\texttt{lat},h}(\phi_h^\star(x),a)}\left[\sum_{x_{h+1} \in \mathcal{X}} \psi^\star(x_{h+1} \mid s_{h+1}) V_f(\phi(x_{h+1}))\right]$$

$$= r_{\texttt{lat},h}(\phi_h^\star(x), a) + \mathbb{E}_{s_{h+1} \sim P_{\texttt{lat},h}(\phi_h^\star(x),a)}\left[\sum_{s' \in \mathcal{S}} \Gamma_\phi(s' \mid s_{h+1}) V_f(s')\right]$$

$$= r_{\texttt{lat},h}(\phi_h^\star(x), a) + \mathbb{E}_{s_{h+1} \sim P_{\texttt{lat},h}(\phi_h^\star(x),a)}[\Gamma_\phi \circ V_f(s_{h+1})]$$

$$= \left[\mathcal{T}_h^{M_{\texttt{lat}}}(\Gamma_\phi \circ V_f)\right](\phi_h^\star(x), a),$$

where the third line follows from the definition of the mismatch function $\Gamma_\phi$. $\square$

We next show that the mismatch functions can be used to realize the pushforward dynamics $\phi \sharp M_{\texttt{obs}}^\star$, which we recall are defined as:

$$\left[\phi \sharp M_{\texttt{obs},h}^\star\right](r, s' \mid x, a) = \sum_{x' : \phi(x') = s'} M_{\texttt{obs},h}^\star(r, x' \mid x, a). \tag{14}$$

We also recall the notation $[\Gamma_{\phi,h+1} \circ M_{\texttt{lat}}]_h$, defined via:

$$[\Gamma_\phi \circ M_{\texttt{lat}}]_h(r_h, s_{h+1} \mid s_h, a_h) := \sum_{s'_{h+1} \in \mathcal{S}} M_{\texttt{lat},h}(r_h, s'_{h+1} \mid s_h, a_h)\Gamma_{\phi,h+1}(s_{h+1} \mid s'_{h+1}).$$

**Lemma D.8** (Pushforward model realizability via mismatch functions). *For all $\phi \in \Phi$, $h \in [H]$, we have:*

$$[\phi_{h+1} \sharp M_{\texttt{obs},h}^\star](\cdot \mid x, a) = \left[[\Gamma_\phi \circ M_{\texttt{lat}}^\star]_h \circ \phi_h^\star\right](\cdot \mid x, a) \tag{15}$$

**Proof of Lemma D.8.** Note that $\Gamma_\phi$ can alternatively be written as:

$$\Gamma_{\phi,h}(s'_h \mid s_h) = \sum_{x_h : \phi(x_h) = s'_h} \psi_h^\star(x_h \mid s_h).$$

We have

$$\phi_{h+1} \sharp M_{\texttt{obs},h}^\star(r_{h+1}, s_{h+1} \mid x_h, a_h)$$

$$= \sum_{x_{h+1} : \phi_{h+1}(x_{h+1}) = s_{h+1}} M_{\texttt{obs},h}^\star(r_{h+1}, x_{h+1} \mid x_h, a_h)$$

$$= \sum_{x_{h+1} : \phi_{h+1}(x_{h+1}) = s_{h+1}} \left(\sum_{r,s' \in \mathbb{R} \times \mathcal{S}} M_{\texttt{lat},h}^\star(r, s' \mid \phi_h^\star(x_h), a_h)\psi_{h+1}^\star(x_{h+1} \mid s')\right)$$

$$= \sum_{r,s' \in \mathbb{R} \times \mathcal{S}} M_{\texttt{lat},h}^\star(r, s' \mid \phi_h^\star(x_h), a_h) \sum_{x_{h+1} : \phi_{h+1}(x_{h+1}) = s_{h+1}} \psi_{h+1}^\star(x_{h+1} \mid s')$$

$$= \sum_{r,s' \in \mathbb{R} \times \mathcal{S}} M_{\texttt{lat},h}^\star(r, s' \mid \phi_h^\star(x_h), a_h)\Gamma_{\phi,h+1}(s' \mid s_{h+1})$$

$$= [\Gamma_\phi \circ M_{\texttt{lat}}^\star]_h(r, s_{h+1} \mid \phi_h^\star(x_h), a_h),$$

as desired. $\square$

# E  Proofs and Additional Results for Section 3.2: Impossibility Results

This section contains additional information and proofs related to our impossibility results regarding statistical modularity (Section 3.2), and is organized as follows:

- Appendix E.1 contains the statement for an additional lower bound that is useful for establishing the impossibility results of Figure 1.

- Appendix E.2 contains details for each entry of Figure 1.

- Appendix E.3 contains for proofs for our main lower bound (Theorem 3.1) and the additional lower bound (Theorem E.1).

## E.1  Additional Lower Bound

**Theorem E.1** (Alternative lower bound). *For every $N \geq 4$, there exists an emission class $\Psi$ and a decoder class $\Phi$ with $|\Psi| = |\Phi| = N$ and a family of latent MDPs $\mathcal{M}_{\mathrm{lat}}$ satisfying (i) $|\mathcal{M}_{\mathrm{lat}}| = 1$, (ii) $H = 1$, (iii) $|\mathcal{S}| = |\mathcal{X}| = N$, (iv) $|\mathcal{A}| = N$, and such that*

*1. For all $\varepsilon, \delta > 0$, we have $\mathrm{comp}(\mathcal{M}_{\mathrm{lat}}, \varepsilon, \delta) = 0$.*

*2. For an absolute constant $c > 0$, $\mathrm{comp}(\langle\!\langle \mathcal{M}_{\mathrm{lat}}, \Phi \rangle\!\rangle, c, c) \geq \Omega(N/\log(N))$.*

**Proof of Theorem E.1.** See Appendix E.3.2. ☐

## E.2  Details for Figure 1

Below, we provide details on each entry in Figure 1. More precisely, for each latent class $\mathcal{M}_{\mathrm{lat}}$, we will give a (brief) description of the MDP class $\mathcal{M}_{\mathrm{lat}}$, give our choice of latent complexity comp for $\mathcal{M}_{\mathrm{lat}}$, and prove that the class is or is not statistically modular for that choice of latent complexity. We view our choices of latent complexities as natural complexities for the respective classes.

**Tabular MDPs (✓).**

- Latent class $\mathcal{M}_{\mathrm{lat}}$: Tabular MDPs $M_{\mathrm{lat}} = (\mathcal{S}, \mathcal{A}, P_{\mathrm{lat}}, R_{\mathrm{lat}}, H)$. [AOM17]

- Latent complexity comp: We take $\mathrm{comp}(\mathcal{M}_{\mathrm{lat}}, \varepsilon, \delta) = \mathrm{poly}(|\mathcal{S}|, |\mathcal{A}|, H, \varepsilon^{-1}, \log \delta^{-1})$, which is attainable, for example, via the UCB-VI algorithm of Azar et al. [AOM17]

- Statistical modularity (✓): Known Block MDP algorithms (e.g. MUSIK [MFR23], BRIEE [ZSUWAS22]) have sample complexities of $\mathrm{poly}(|\mathcal{S}|, |\mathcal{A}|, H, \varepsilon^{-1}, \log \delta^{-1}, \log |\Phi|)$.

**Contextual Bandits (✓).**

- Latent class $\mathcal{M}_{\mathrm{lat}}$: Contextual bandits with context space $\mathcal{S}$, action space $\mathcal{A}$, reward function $r^\star_{\mathrm{lat}} : \mathcal{S} \times \mathcal{A} \to [0,1]$ and a finite realizable function class satisfying $r^\star \in \mathcal{F}_{\mathrm{lat}}$.

- Latent complexity comp: We take $\mathrm{comp}(\mathcal{M}_{\mathrm{lat}}, \varepsilon, \delta) = \mathrm{poly}(|\mathcal{A}|, \log|\mathcal{F}_{\mathrm{lat}}|, \varepsilon^{-1}, \log \delta^{-1})$, attainable via, e.g., the SQUARE-CB algorithm [FR20].

- Statistical modularity (✓): We note that $\mathcal{F}_{\mathrm{lat}} \circ \Phi = \{[f \circ \phi] \mid f \in \mathcal{F}, \phi \in \Phi\}$ is a realizable function class for the observation-level reward function $r^\star_{\mathrm{obs}}$, since $r^\star_{\mathrm{obs}} = [r^\star_{\mathrm{lat}} \circ \phi^\star] \in \mathcal{F}_{\mathrm{lat}} \circ \Phi$. Thus, applying the SQUARE-CB algorithm directly on the observations $x^{(t)}, a^{(t)}, r^{(t)}$ will give complexity $\mathrm{poly}(|\mathcal{A}| \log(|\mathcal{F}_{\mathrm{lat}}||\Phi|), \varepsilon^{-1}, \log \delta^{-1}) = \mathrm{poly}(|\mathcal{A}|, \log |\mathcal{F}_{\mathrm{lat}}|, \log |\Phi|, \varepsilon^{-1}, \log \delta^{-1})$.

**Low-rank MDP (✓).**

- Latent class $\mathcal{M}_{\mathrm{lat}}$: MDPs $M_{\mathrm{lat}} = (\mathcal{S}, \mathcal{A}, H, P_{\mathrm{lat}}, r_{\mathrm{lat}})$ such that there exists $\mu^\star_{\mathrm{lat},h} \in \mathbb{R}^d$, $\theta^\star_{\mathrm{lat},h} \in \mathbb{R}^d$, and a known set of features $\Xi_{\mathrm{lat}} = \left\{ \xi_{\mathrm{lat}} = \left\{ \xi_{\mathrm{lat},h} : \mathcal{S} \times \mathcal{A} \to \mathbb{R}^d \right\}_{h=1}^H \right\}$ such that for all $h \in [H]$ we have $r_{\mathrm{lat}}(s_h, a_h) = \langle \xi^\star_{\mathrm{lat},h}(s_h, a_h), \theta^\star_{\mathrm{lat},h} \rangle$ as well as

$$P_{\mathrm{lat},h}(s_{h+1} \mid s_h, a_h) = \langle \xi^\star_{\mathrm{lat},h}(s_h, a_h), \mu^\star_{\mathrm{lat},h+1}(s_{h+1}) \rangle \tag{16}$$

for some $\xi^\star_{\mathrm{lat}} \in \Xi_{\mathrm{lat}}$.

- Latent complexity comp: We take $\mathrm{comp}(\mathcal{M}_{\mathrm{lat}}, \varepsilon, \delta) = \mathrm{poly}(d, |\mathcal{A}|, H, \log |\Xi_{\mathrm{lat}}|, \varepsilon^{-1}, \log \delta^{-1})$, which is attainable via the VOX algorithm of Mhammedi et al. [MBFR23].

- Statistical modularity (✓): This is obtained by noting that the observation-level dynamics also satisfy the low-rank property with the same dimension. Formally, letting $P_{\mathrm{obs}}$ be the transition kernel for $\langle\!\langle M_{\mathrm{lat}}, \psi^\star \rangle\!\rangle$ and $\phi^\star = (\psi^\star)^{-1}$, we have

$$P_{\mathrm{obs},h}(x_{h+1} \mid x_h, a_h) = \sum_{s_{h+1}\in\mathcal{S}} P_{\mathrm{lat},h}(s_{h+1} \mid \phi_h^\star(x_h), a_h)\psi_{h+1}^\star(x_{h+1} \mid s_{h+1})$$

$$= \sum_{s_{h+1}\in\mathcal{S}} \left\langle \xi_{\mathrm{lat},h}^\star(\phi_h^\star(x), a), \mu_{\mathrm{lat},h+1}^\star(s_{h+1}) \right\rangle \psi_{h+1}^\star(x_{h+1} \mid s_{h+1})$$

$$= \left\langle \xi_{\mathrm{lat},h}^\star(\phi_h^\star(x), a), \sum_{s_{h+1}\in\mathcal{S}} \mu_{\mathrm{lat},h+1}^\star(s_{h+1})\psi_{h+1}^\star(x_{h+1} \mid s_{h+1}) \right\rangle.$$

Thus, the transition kernel $P_{\mathrm{obs}}$ is a low-rank MDP with $\mu_{\mathrm{obs},h+1}(x_{h+1}) := \sum_{s_{h+1}} \mu_{\mathrm{lat},h+1}^\star(s_{h+1})\psi_{h+1}^\star(x_{h+1} \mid s_{h+1})$ and feature class

$$\Xi_{\mathrm{lat}} \circ \Phi = \left\{ \xi_{\mathrm{lat}} \circ \phi = \{\xi_h \circ \phi_h : x, a \mapsto \xi_h(\phi_h(x), a)\}_{h=1}^{H} \mid \xi_{\mathrm{lat}} \in \Xi_{\mathrm{lat}}, \phi \in \Phi \right\}.$$

Lastly, since $r_{\mathrm{obs}} = [r_{\mathrm{lat}} \circ \phi^\star]$, the reward function is also linear with the same unknown feature class. Thus we can apply VOX directly on top of the observations, with the feature class $\Xi_{\mathrm{lat}} \circ \Phi$, which will achieve a complexity $\mathtt{poly}(d, |\mathcal{A}|, H, \log|\Xi_{\mathrm{lat}}|, \log|\Phi|, \varepsilon^{-1}, \log(\delta^{-1}))$.

## Known Deterministic MDP ($|\mathcal{M}_{\mathtt{lat}}| = 1$) (✓).

- Latent class $\mathcal{M}_{\mathrm{lat}}$: $\mathcal{M}_{\mathrm{lat}} = \{M_{\mathrm{lat}} = (\mathcal{S}, \mathcal{A}, P_{\mathrm{lat}}, R_{\mathrm{lat}}, H)\}$ is a set of MDPs of size 1 with both deterministic rewards and deterministic transitions.

- Latent complexity comp: We take $\mathtt{comp}(\mathcal{M}_{\mathrm{lat}}, \varepsilon, \delta) = 0$, which is attainable as $M_{\mathrm{lat}}$ is known and we can simply deploy its optimal policy.

- Statistical modularity (✓): We note that, due to determinism, the latent optimal policy can be chosen to be *open-loop* without loss of generality, and thus will always experience the same trajectory $(s_1^\star, a_1^\star, \ldots, s_H^\star, a_H^\star)$. We can define the observation-level policy which commits to this same sequence of actions, i.e. $\pi_{\mathrm{obs},h}(x_h) = a_h^\star$ for all $x_h$. This will be an optimal policy for any $M_{\mathrm{obs}} = \langle\!\langle M_{\mathrm{lat}}, \psi \rangle\!\rangle$, and can also be learned in 0 samples.

## Low State Occupancy ($\forall \pi : \mathcal{S} \to \Delta(\mathcal{A})$) (✓).

- Latent class $\mathcal{M}_{\mathrm{lat}}$: $\mathcal{M}_{\mathrm{lat}} = \{M_{\mathrm{lat}} = (\mathcal{S}, \mathcal{A}, P_{\mathrm{lat}}, R_{\mathrm{lat}}, H)\}$ is a set of MDPs for which we have a realizable value function class, and such that there exists a feature map $\zeta_{\mathrm{lat}} = \{\zeta_{\mathrm{lat},h} : \mathcal{S} \to \mathbb{R}^d\}_{h=1}^{H}$ such that for all $\pi : \mathcal{S} \to \Delta(\mathcal{A})$ and for all $M_{\mathrm{lat}} \in \mathcal{M}_{\mathrm{lat}}$, we have

$$\forall h \in [H] \ \exists \theta_h^{M_{\mathrm{lat}},\pi} : \quad d_h^{M_{\mathrm{lat}},\pi}(s) = \left\langle \zeta_{\mathrm{lat},h}(s), \theta_h^{M_{\mathrm{lat}},\pi} \right\rangle.$$

Note that the feature map does not need to be known.

- Latent complexity comp: We take $\mathtt{comp}(\mathcal{M}_{\mathrm{lat}}, \varepsilon, \delta) = \mathtt{poly}(d, |\mathcal{A}|, H, \log|\mathcal{F}_{\mathrm{lat}}|, \varepsilon^{-1}, \log(\delta^{-1}))$, which is attainable by the BILIN-UCB algorithm of Du et al., since i) MDPs with this property have Bilinear rank bounded by $d|\mathcal{A}|$ (see Definition 4.3 and Lemma 4.6 of Du et al. [Du+21]), and ii) one can construct the value function class $\mathcal{F}_{\mathrm{lat}} = \{Q^{M_{\mathrm{lat}},\star} \mid M_{\mathrm{lat}} \in \mathcal{M}_{\mathrm{lat}}\}$, which is realizable and has size $\log|\mathcal{F}_{\mathrm{lat}}| = \log|\mathcal{M}_{\mathrm{lat}}|$.

- Statistical modularity (✓): We firstly note that one can construct a realizable value function class for the set $\langle\!\langle \mathcal{M}_{\mathrm{lat}}, \Phi \rangle\!\rangle$, via the set $\mathcal{F}_{\mathrm{obs}} = \{Q^{M_{\mathrm{lat}},\star} \circ \phi \mid M_{\mathrm{lat}} \in \mathcal{M}_{\mathrm{lat}}, \phi \in \Phi\}$. This is realizable since, for any $M_{\mathrm{obs}} := \langle\!\langle M_{\mathrm{lat}}, \psi \rangle\!\rangle$, letting $\phi^\star = \psi^{-1}$, we have $Q^{M_{\mathrm{obs}},\star} = Q^{M_{\mathrm{lat}},\star} \circ \phi^\star$, and that this class has size $\log|\mathcal{M}_{\mathrm{lat}}||\Phi|$. We can then show that the occupancies $d^{M_{\mathrm{obs}},\pi_{f_{\mathrm{obs}}}}$, for $f_{\mathrm{obs}} \in \mathcal{F}_{\mathrm{obs}}$, can also be expressed as $d$-dimensional linear function for an appropriate choice of features, which will imply that the BILIN-UCB algorithm run directly on $M_{\mathrm{obs}}$ will attain a complexity of $\mathtt{poly}(d, |\mathcal{A}|, H, \log\mathcal{M}_{\mathrm{lat}}, \log\Phi, \varepsilon^{-1}, \log(\delta^{-1}))$. To obtain this, we recall the following lemma:

**Lemma D.2.** *Let $\{\Gamma_\phi\}_{\phi\in\Phi}$ denote the mismatch functions for emission $\psi^\star$, and let $M_{\mathrm{obs}} = \langle\!\langle M_{\mathrm{lat}}, \psi^\star \rangle\!\rangle$. Then, for any $\pi_{\mathrm{lat}} \in \Pi_{\mathrm{lat}}$, $\phi \in \Phi$, $h \in [H]$, $x \in \mathcal{X}$, we have*

$$d_h^{M_{\mathrm{obs}},\pi_{\mathrm{lat}}\circ\phi}(x) = \psi_h^\star(x \mid \phi_h^\star(x))d_h^{M_{\mathrm{lat}},\Gamma_\phi\circ\pi_{\mathrm{lat}}}(\phi_h^\star(x)).$$

Thanks to the above lemma, we have

$$
\begin{aligned}
d_{\mathrm{obs}}^{\pi_{f \circ \phi}}(x_h) &= \psi(x_h \mid \phi^\star(x_h)) d_{\mathrm{lat}}^{\Gamma_\phi \circ \pi_f}(\phi^\star(x_h)) \\
&= \psi(x_h \mid \phi^\star(x_h)) \left\langle [\zeta_{\mathrm{lat},h} \circ \phi_h^\star](x_h), \theta_h^{M_{\mathrm{lat}}, \Gamma_\phi \circ \pi_f} \right\rangle \\
&= \left\langle \psi(x_h \mid \phi^\star(x_h))[\zeta_{\mathrm{lat},h} \circ \phi_h^\star](x_h), \theta_h^{M_{\mathrm{lat}}, \Gamma_\phi \circ \pi_f} \right\rangle
\end{aligned}
$$

and so $d_{\mathrm{obs}}^{\pi_{f \circ \phi}}$ is linear with feature mapping $\psi(x_h \mid \phi^\star(x_h))[\zeta_{\mathrm{lat},h} \circ \phi_h^\star]$ and parameter $\theta^{M_{\mathrm{lat}}, \Gamma_\phi \circ \pi_f}$. Recall that the feature map need not be known, so that BILIN-UCB can still be applied despite not knowing $\psi$ and $\phi^\star$.

**Model class + Pushforward Coverability (✓).**

- Latent class $\mathcal{M}_{\mathrm{lat}}$: $\mathcal{M}_{\mathrm{lat}} = \{M_{\mathrm{lat}} = (\mathcal{S}, \mathcal{A}, P_{\mathrm{lat}}, R_{\mathrm{lat}}, H)\}$ is a set of MDPs that all satisfy pushforward coverability $C_{\mathrm{push}}(M_{\mathrm{lat}}) \leq C_{\mathrm{push}}$ (cf. Eq. (28) for the definition).

- Latent complexity comp: We take $\mathrm{comp}(\mathcal{M}_{\mathrm{lat}}, \varepsilon, \delta) = \mathrm{poly}(C_{\mathrm{push}}, |\mathcal{A}|, H, \log|\mathcal{M}_{\mathrm{lat}}|, \varepsilon^{-1}, \log(\delta^{-1}))$, which is attainable by the GOLF algorithm via the results of Xie et al. [XFBJK23] (see also Lemma F.3). We obtain this by noting that i) $C_{\mathrm{cov}} \leq C_{\mathrm{push}}|\mathcal{A}|$, where $C_{\mathrm{cov}}$ is defined in Definition 2 of Xie et al. [XFBJK23], and ii) a realizable model class can be used to construct a realizable value function class $\mathcal{F}$ and a Bellman-complete value function helper class $\mathcal{G}$ with sizes $\log|\mathcal{F}| = \log|\mathcal{M}|$ and $\log|\mathcal{G}| = \mathcal{O}(\log|\mathcal{M}|)$.

- Statistical modularity (✓): This is obtained via Theorem 3.2.

**Linear CB/MDP (✗⋆).**

- Latent class $\mathcal{M}_{\mathrm{lat}}$: MDPs $M_{\mathrm{lat}} = (\mathcal{S}, \mathcal{A}, P_{\mathrm{lat}}, R_{\mathrm{lat}}, H)$ that are linear with respect to a known feature map $\xi_{\mathrm{lat}}^\star : \mathcal{S} \times \mathcal{A} \to \mathbb{R}^d$ (i.e. such that Eq. (16) holds for $\xi_{\mathrm{lat}}^\star$).

- Latent complexity comp: We take $\mathrm{comp}(\mathcal{M}_{\mathrm{lat}}, \varepsilon, \delta) = \mathrm{poly}(d, H, \varepsilon^{-1}, \log(\delta^{-1}))$, which is attainable via the LSVI-UCB algorithm of Jin et al. [JYWJ20]. Note that this guarantee does not depend on the number of actions.

- Statistical intractability (✗): The latent model used in the construction of Theorem E.1 is a set (of size 1) of linear MDPs with $d = 1$. In particular, that construction was a contextual bandit so we only have to realize a reward function, and since there is only one latent model so we can trivially embed this with $d = 1$ via $\xi_{\mathrm{lat}}^\star(s, a) = r_{\mathrm{lat}}(s, a)$, where $r_{\mathrm{lat}}$ is the reward function of the MDP used in Theorem E.1.

- Statistical modularity with additional $|\mathcal{A}|$-dependence: As in the Low-rank MDP case above, $\langle\!\langle M_{\mathrm{lat}}, \psi \rangle\!\rangle$ is low-rank with unknown feature set $\Phi' = \{\xi_{\mathrm{lat}}^\star \circ \phi \mid \phi \in \Phi\}$. Thus, by the same conclusion, a the VOX algorithm will have complexity $\mathrm{poly}(d, |\mathcal{A}|, H, \log|\Phi|)$, which is of the desired form if we allow suboptimal dependence on $|\mathcal{A}|$.

**Model class + Coverability ($\forall \pi_M : M \in \mathcal{M}$) (✗).**

- Latent assumption: $\mathcal{M}_{\mathrm{lat}} = \{M_{\mathrm{lat}} = (\mathcal{S}, \mathcal{A}, P_{\mathrm{lat}}, R_{\mathrm{lat}}, H)\}$ is a set of MDPs that all satisfy coverability with respect to the policy class $\Pi_{\mathcal{M}} = \{\pi_M \mid M \in \mathcal{M}\}$, i.e. we have

$$
\forall M_{\mathrm{lat}} \in \mathcal{M}_{\mathrm{lat}} : \quad C_{\mathrm{cov}}(M_{\mathrm{lat}}) = \inf_{\mu_h \in \Delta(\mathcal{S} \times \mathcal{A})} \sup_{h \in [H]} \sup_{\pi \in \Pi_{\mathcal{M}}} \left\| \frac{d_h^{M_{\mathrm{lat}}, \pi}}{\mu_h} \right\|_\infty < \infty
$$

- Latent complexity comp: We take $\mathrm{comp}(\mathcal{M}_{\mathrm{lat}}, \varepsilon, \delta) = \mathrm{poly}(C_{\mathrm{cov}}, H, \log|\mathcal{M}_{\mathrm{lat}}|, \varepsilon^{-1}, \log(\delta^{-1}))$, which is attainable by the GOLF algorithm via the results of Xie et al. [XFBJK23] (see also Lemma F.3). We obtain this by noting that a realizable model class can be used to construct a realizable value function class $\mathcal{F}$ and a complete value function class $\mathcal{G}$ of sizes $\log|\mathcal{F}| = \log|\mathcal{M}|$ and $\log|\mathcal{G}| = \mathcal{O}(\log|\mathcal{M}|)$.

- Statistical intractability (✗): The latent models used in the construction of Theorem 3.1 are a set of coverable MDPs – in particular, these are trivially coverable with $C_{\mathrm{cov}} = 1$ since there is a single latent model and we can take $\mu = d^{M_{\mathrm{lat}}^\star, \pi_{M_{\mathrm{lat}}^\star}}$. We remark that it is an interesting open question whether this impossibility result continues to hold if we require coverability with respect to the class $\Pi$ of all possible latent policies.

**Known Stochastic MDP ($|\mathcal{M}_{\mathtt{lat}}| = 1$) (✗).**

- Latent class $\mathcal{M}_{\mathtt{lat}}$: $\mathcal{M}_{\mathtt{lat}} = \{M_{\mathtt{lat}} = (\mathcal{S}, \mathcal{A}, P_{\mathtt{lat}}, R_{\mathtt{lat}}, H)\}$ is a set of MDPs of size 1.

- Latent complexity comp: We take $\mathsf{comp}(\mathcal{M}_{\mathtt{lat}}, \varepsilon, \delta) = 0$, which is attainable as $M_{\mathtt{lat}}$ is known and we can simply deploy its optimal policy.

- Statistical intractability (✗): This is precisely the setting of Theorem 3.1, which shows that at least $\Omega(N/\log(N))$ samples will be needed, where $N = |\Phi|$.

**Bellman rank ($Q$-type or $V$-type) (✗)**

- Latent assumption: $\mathcal{M}_{\mathtt{lat}} = \{M_{\mathtt{lat}} = (\mathcal{S}, \mathcal{A}, P_{\mathtt{lat}}, R_{\mathtt{lat}}, H)\}$ is a set of latent models such that each $M_{\mathtt{lat}} \in \mathcal{M}_{\mathtt{lat}}$ has $Q$-type Bellman rank $d$ or $V$-type Bellman rank $d$ [JLM21]. Letting $\mathcal{F}$ be a realizable value function class for $\mathcal{M}_{\mathtt{lat}}$, in the $Q$-type case, this means that the $|\Pi_{\mathcal{F}}| \times |\mathcal{F}|$ matrix

$$\mathcal{E}_h^Q(\pi, f) = \mathbb{E}^\pi \Big[ f_h(s_h, a_h) - r_h - \max_{a'} f_{h+1}(s_{h+1}, a') \Big],$$

admits a rank $d$ factorization. In the $V$-type case, the matrix

$$\mathcal{E}_h^V(\pi, f) = \mathbb{E}_{s_h \sim d_h^\pi, a_h \sim \pi_f} \Big[ f_h(s_h, a_h) - r_h - \max_{a'} f_{h+1}(s_{h+1}, a') \Big]$$

admits a rank-$d$ matrix factorization.

- Latent complexity comp: We take $\mathsf{comp}(\mathcal{M}_{\mathtt{lat}}, \varepsilon, \delta) = \mathsf{poly}(d, H, |\mathcal{A}|\log|\mathcal{F}|, \varepsilon^{-1}, \log(\delta^{-1}))$ for the $V$-type Bellman rank case, which is achievable by the OLIVE algorithm of Jiang et al. [JKALS17], and $\mathsf{comp}(\mathcal{M}_{\mathtt{lat}}, \varepsilon, \delta) = \mathsf{poly}(d, H, \log|\mathcal{F}|, \varepsilon^{-1}, \log(\delta^{-1}))$ for $Q$-type Bellman rank, which is achievable by the BILIN-UCB algorithm of Du et al. [Du+21].

- Statistical intractability (✗): We note that the construction in Theorem 3.1 has $|\mathcal{M}_{\mathtt{lat}}| = 1$, which trivially has Bellman rank equal to 1, so Theorem 3.1 precludes statistical modularity with complexity comp.

**Eluder dimension + Bellman Completeness (✗)**

- Latent class $\mathcal{M}_{\mathtt{lat}}$: $\mathcal{M}_{\mathtt{lat}} = \{M_{\mathtt{lat}} = (\mathcal{S}, \mathcal{A}, P_{\mathtt{lat}}, R_{\mathtt{lat}}, H)\}$ is a set of MDPs such that there is a function class $\mathcal{F}_{\mathtt{lat}}$ satisfying

$$\forall f_{\mathtt{lat}} \in \mathcal{F}_{\mathtt{lat}}, M_{\mathtt{lat}} \in \mathcal{M}_{\mathtt{lat}} : \quad \mathcal{T}^{M_{\mathtt{lat}}} f_{\mathtt{lat}} \in \mathcal{F}_{\mathtt{lat}}.$$

Furthermore, each $M_{\mathtt{lat}} \in \mathcal{M}_{\mathtt{lat}}$ has Bellman-Eluder dimension bounded by $d$ (see Definition 8 of [JLM21]).

- Latent complexity comp: We take $\mathsf{comp}(\mathcal{M}_{\mathtt{lat}}, \varepsilon, \delta) = \mathsf{poly}(d, H, \log|\mathcal{F}|, \varepsilon^{-1}, \log(\delta^{-1}))$, which is attainable by the GOLF algorithm of Jin et al. [JLM21].

- Statistical intractability (✗): As in the Bellman rank case, the construction in Theorem 3.1 has $|\mathcal{M}_{\mathtt{lat}}| = 1$, so we can take $\mathcal{F}_{\mathtt{lat}} = \{Q^{M_{\mathtt{lat}}, \star} \mid M_{\mathtt{lat}} \in \mathcal{M}_{\mathtt{lat}}\}$ which is evidently complete for $\mathcal{T}^{M_{\mathtt{lat}}}$, and has Eluder dimension 1, so Theorem 3.1 precludes statistical modularity with complexity comp.

**$Q^\star$-irrelevant State Abstraction (✗)**

- Latent class $\mathcal{M}_{\mathtt{lat}}$: $M_{\mathtt{lat}} = (\mathcal{S}, \mathcal{A}, P_{\mathtt{lat}}, R_{\mathtt{lat}}, H)$ such that there is a known state abstraction function $\zeta_{\mathtt{lat}} : \mathcal{S} \to \mathcal{Z}$ such that $\zeta_{\mathtt{lat}}(s) = \zeta_{\mathtt{lat}}(s')$ implies that $Q^{M_{\mathtt{lat}}, \star}(s, a) = Q^{M_{\mathtt{lat}}, \star}(s', a)$ for all $a \in \mathcal{A}$.

- Latent complexity comp: We take $\mathsf{comp}(\mathcal{M}_{\mathtt{lat}}, \varepsilon, \delta) = \mathsf{poly}(|\mathcal{Z}|, |\mathcal{A}|, H, \varepsilon^{-1}, \log(\delta^{-1}))$ which is attainable by the OLIVE algorithm of Jiang et al. [JKALS17].

- Statistical intractability (✗): We take $\mathcal{M}_{\mathtt{lat}} = \{M_{\mathtt{lat}}\}$ as the MDP class from the construction of Theorem 3.1. Let $Q_{\mathtt{lat}}^\star := Q^{M_{\mathtt{lat}}, \star}$. Note that we have $Q_{\mathtt{lat}}^\star(s, a) \in \{0, 1\}$ for all $s, a$, so we can take a latent abstract state space $\mathcal{Z} = \{(0, 0), (0, 1), (1, 0), (1, 1)\}$ and a state abstraction function $\zeta_{\mathtt{lat}}$ such that $\zeta_{\mathtt{lat}}(s) = (i, j)$ if $Q_{\mathtt{lat}}^\star(s, 0) = i$ and $Q_{\mathtt{lat}}^\star(s, 1) = j$. This satisfies the property of a $Q^\star$-irrelevant abstraction, since $\zeta_{\mathtt{lat}}(s) = \zeta_{\mathtt{lat}}(s') = (i, j)$ implies that $Q_{\mathtt{lat}}^\star(s, 0) = Q_{\mathtt{lat}}^\star(s', 0) = i$ and $Q_{\mathtt{lat}}^\star(s, 1) = Q_{\mathtt{lat}}^\star(s', 1) = j$. This has a constant-sized abstract space ($|\mathcal{Z}| = 4$) and $|\mathcal{A}| = 2$, so Theorem 3.1 precludes statistical modularity with complexity comp.

**Linear Mixture MDP (✗).**

- Latent class $\mathcal{M}_{\mathtt{lat}}$: MDPs $M_{\mathtt{lat}} = (\mathcal{S}, \mathcal{A}, P_{\mathtt{lat}}, R_{\mathtt{lat}}, H)$ such that there is a known feature map $\zeta_{\mathtt{lat}} = \{\zeta_{\mathtt{lat},h} : s', s, a \mapsto \mathbb{R}^d\}_{h=1}^H$ such that

$$\forall h \in [H], \exists \theta_h \in \mathbb{R}^d : \quad P_{\mathtt{lat},h}(s' \mid s, a) = \langle \zeta_{\mathtt{lat},h}(s' \mid s, a), \theta_h \rangle$$

- Latent complexity comp: We take $\mathsf{comp}(\mathcal{M}_{\mathtt{lat}}, \varepsilon, \delta) = \mathtt{poly}(d, H, \varepsilon^{-1}, \log(\delta^{-1}))$, which is attainable by the UCRL-VTR$^+$ algorithm of Zhou et al. [ZGS21]

- Statistical intractability (✗): We take $\mathcal{M}_{\mathtt{lat}} = \{M_{\mathtt{lat}}\}$ to be the construction of Theorem 3.1. Here, there is a single latent model, so this is trivially embeddable with $\zeta_{\mathtt{lat},h}(s' \mid s, a) = P_{\mathtt{lat},h}^\star(s' \mid s, a) \in \mathbb{R}^1$. This has dimension $d = 1$, so Theorem 3.1 precludes statistical modularity with complexity comp.

**Linear $Q^\star/V^\star$ (✗).**

- Latent class $\mathcal{M}_{\mathtt{lat}}$: MDPs $M_{\mathtt{lat}} = (\mathcal{S}, \mathcal{A}, P_{\mathtt{lat}}, R_{\mathtt{lat}}, H)$ such that there are known features maps $\alpha_{\mathtt{lat}} : \mathcal{S} \times \mathcal{A} \to \mathbb{R}^d$ and $\beta_{\mathtt{lat}} : \mathcal{S} \to \mathbb{R}^d$ such that for all $M_{\mathtt{lat}} \in \mathcal{M}_{\mathtt{lat}}$, there exists unknown parameters $\theta_Q, \theta_V \in \mathbb{R}^d$ such that $Q^{M_{\mathtt{lat}},\star}(s, a) = \langle \alpha_{\mathtt{lat}}(s, a), \theta_Q \rangle$ and $V^{M_{\mathtt{lat}},\star}(s) = \langle \beta_{\mathtt{lat}}(s), \theta_V \rangle$.

- Latent complexity comp: We take $\mathsf{comp}(\mathcal{M}_{\mathtt{lat}}, \varepsilon, \delta) = \mathtt{poly}(d, H, \varepsilon^{-1}, \log(\delta^{-1}))$, which is attainable by the BILIN-UCB algorithm of Du et al. [Du+21].

- Statistical intractability (✗): We can take $\mathcal{M}_{\mathtt{lat}}$ to be the latent MDP class from the construction of Theorem 3.1. Since there is a single latent model, this is trivially embeddable with dimension 1, i.e. we can take $\zeta_{\mathtt{lat}}(s, a) = Q_{\mathtt{lat}}^\star(s, a)$ and $\beta_{\mathtt{lat}}(s) = V_{\mathtt{lat}}^\star(s)$. This has dimension $d = 1$, so Theorem 3.1 precludes statistical modularity with complexity comp.

**Low State or State-Action Occupancy ($\forall \pi_M : M \in \mathcal{M}$) (✗).**

- Latent class $\mathcal{M}_{\mathtt{lat}}$: In the Low State Occupancy model, $\mathcal{M}_{\mathtt{lat}} = \{M_{\mathtt{lat}} = (\mathcal{S}, \mathcal{A}, P_{\mathtt{lat}}, R_{\mathtt{lat}}, H)\}$ is a set of MDPs such that there exists a feature map $\zeta_{\mathtt{lat}}^V = \{\zeta_{\mathtt{lat},h} : \mathcal{S} \to \mathbb{R}^d\}_{h=1}^H$ such that for all $\pi \in \{\pi_{M_{\mathtt{lat}}} \mid M_{\mathtt{lat}} \in \mathcal{M}_{\mathtt{lat}}\}$ and for all $M_{\mathtt{lat}} \in \mathcal{M}_{\mathtt{lat}}$, we have

$$\forall h \in [H] \; \exists \theta_h^{M_{\mathtt{lat}},\pi} : \quad d_h^{M_{\mathtt{lat}},\pi}(s) = \left\langle \zeta_{\mathtt{lat},h}^V(s), \theta_h^{M_{\mathtt{lat}},\pi} \right\rangle.$$

For the State-Action Occupancy model, we have that there exists a feature map $\zeta_{\mathtt{lat}}^Q = \{\zeta_{\mathtt{lat},h} : \mathcal{S} \times \mathcal{A} \to \mathbb{R}^d\}_{h=1}^H$ such that for all $\pi \in \{\pi_{M_{\mathtt{lat}}} \mid M_{\mathtt{lat}} \in \mathcal{M}_{\mathtt{lat}}\}$ and for all $M_{\mathtt{lat}} \in \mathcal{M}_{\mathtt{lat}}$, we have

$$\forall h \in [H] \; \exists \theta_h^{M_{\mathtt{lat}},\pi} : \quad d_h^{M_{\mathtt{lat}},\pi}(s, a) = \left\langle \zeta_{\mathtt{lat},h}^Q(s, a), \theta_h^{M_{\mathtt{lat}},\pi} \right\rangle.$$

Note that the feature map does not need to be known in either case.

- Latent complexity comp: We take $\mathsf{comp}(\mathcal{M}_{\mathtt{lat}}, \varepsilon, \delta) = \mathtt{poly}(d, |\mathcal{A}|, H, \log|\mathcal{F}_{\mathtt{lat}}|, \varepsilon^{-1}, \log(\delta^{-1}))$ for the state occupancy case and $\mathsf{comp}(\mathcal{M}_{\mathtt{lat}}, \varepsilon, \delta) = \mathtt{poly}(d, H, \log|\mathcal{M}_{\mathtt{lat}}|, \varepsilon^{-1}, \log(\delta^{-1}))$. Both are attainable by the BILIN-UCB algorithm of Du et al., since i) MDPs with this property have Bilinear rank bounded by $d|\mathcal{A}|$ and $d$ respectively (see Definition 4.3 and Lemma 4.6 of [Du+21]), and ii) one can construct the value function class $\mathcal{F}_{\mathtt{lat}} = \{Q^{M_{\mathtt{lat}},\star} \mid M_{\mathtt{lat}} \in \mathcal{M}_{\mathtt{lat}}\}$ which is realizable and has size $\log|\mathcal{F}_{\mathtt{lat}}| = \log|\mathcal{M}_{\mathtt{lat}}|$.

- Intractability: We can take the construction of Theorem 3.1, which has $|\mathcal{M}_{\mathtt{lat}}| = 1$ and thus is trivially embeddable with dimension 1, i.e. we can take $\zeta_{\mathtt{lat}}^V(s) = d^{M_{\mathtt{lat}}, \pi_{M_{\mathtt{lat}}}}(s)$ and $\zeta_{\mathtt{lat}}^Q(s, a) = d^{M_{\mathtt{lat}}, \pi_{M_{\mathtt{lat}}}}(s, a)$.

**Bisimulation (?)**

- Latent class $\mathcal{M}_{\mathtt{lat}}$: MDPs $M_{\mathtt{lat}} = (\mathcal{S}, \mathcal{A}, P_{\mathtt{lat}}, R_{\mathtt{lat}}, H)$ such that there is a known state abstraction function $\zeta_{\mathtt{lat}} : \mathcal{S} \to \mathcal{Z}$ such that $\zeta_{\mathtt{lat}}(s) = \zeta_{\mathtt{lat}}(\widetilde{s})$ implies that $R_{\mathtt{lat}}(s, a) = R_{\mathtt{lat}}(\widetilde{s}, a)$ for all $a \in \mathcal{A}$ as well as $\sum_{s' : \zeta_{\mathtt{lat}}(s')=z'} P_{\mathtt{lat}}(s' \mid s, a) = \sum_{s' : \zeta_{\mathtt{lat}}(s')=z'} P_{\mathtt{lat}}(s' \mid \widetilde{s}, a)$ for all $z'$.

- Latent complexity comp: We take $\mathsf{comp}(\mathcal{M}_{\mathtt{lat}}, \varepsilon, \delta) = \mathsf{poly}(|\mathcal{Z}|, |\mathcal{A}|, H, \varepsilon^{-1}, \log(\delta^{-1}))$ which is attainable by the OLIVE algorithm of [JKALS17].

- Openness (**?**): A negative result does not follow from existing constructions, since the dynamics from the tree-based construction of Theorem 3.1 are not bisimilar unless $|\mathcal{Z}| = |\mathcal{S}|$, which allows for the application of tabular methods. At the same time, a positive result does not follow from existing methods, since it is non-trivial to extend existing Block MDP methods to use the bisimulation state abstraction in a way that only pays for $|\mathcal{Z}|$.

**Low State-Action Occupancy** ($\forall \pi : \mathcal{S} \to \Delta(\mathcal{A})$) (**?**$^\star$)

- Latent class $\mathcal{M}_{\mathtt{lat}}$: $\mathcal{M}_{\mathtt{lat}} = \{M_{\mathtt{lat}} = (\mathcal{S}, \mathcal{A}, P_{\mathtt{lat}}, R_{\mathtt{lat}}, H)\}$ is a set of MDPs such that there exists a feature map $\zeta_{\mathtt{lat}}^Q = \left\{ \zeta_{\mathtt{lat},h} : \mathcal{S} \times \mathcal{A} \to \mathbb{R}^d \right\}_{h=1}^H$ such that for all $\pi : \mathcal{S} \to \Delta(\mathcal{A})$ and for all $M_{\mathtt{lat}} \in \mathcal{M}_{\mathtt{lat}}$, we have

$$\forall h \in [H] \; \exists \theta_h^{M_{\mathtt{lat}},\pi} : \quad d_h^{M_{\mathtt{lat}},\pi}(s,a) = \left\langle \zeta_{\mathtt{lat},h}^Q(s,a), \theta_h^{M_{\mathtt{lat}},\pi} \right\rangle.$$

  Note that the feature map does not need to be known.

- We take $\mathsf{comp}(\mathcal{M}_{\mathtt{lat}}, \varepsilon, \delta) = \mathsf{poly}(d, H, \log|\mathcal{M}_{\mathtt{lat}}|, \varepsilon^{-1}, \log(\delta^{-1}))$, which is attainable by the BILIN-UCB algorithm of Du et al., since i) MDPs with this property have Bilinear rank bounded by $d$ (see Definition 4.3 and Lemma 4.6 of [Du+21]), and ii) one can construct a realizable value function class of size $\log|\mathcal{F}| = \log|\mathcal{M}|$.

- Openness (**?**): A negative result does not follow from existing constructions, since the dynamics from the tree-based construction of Theorem 3.1 do not have linear occupancies for all $\pi : \mathcal{S} \to \Delta(\mathcal{A})$ unless $d = |\mathcal{S}|$, which allows for the application of tabular methods, and the dynamics from the bandit-based construction Theorem E.1 do not have linear occupancies for all $\pi : \mathcal{S} \to \Delta(\mathcal{A})$ unless $d = |\mathcal{A}|$. At the same time, unlike the low state occupancy case, a positive result does not follow as it is unclear if we can express the observation-space occupancies linearly.

- Statistical tractability with additional (suboptimal) $|\mathcal{A}|$-dependence (**✓**): Note that we can reduce to the Low State Occupancy case (**✓**), since

$$d^\pi(s) = \sum_{a \in \mathcal{A}} d^\pi(s,a) = \left\langle \theta^\pi, \sum_{a \in \mathcal{A}} \zeta_{\mathtt{lat}}^Q(s,a) \right\rangle := \left\langle \theta^\pi, \zeta_{\mathtt{lat}}^V(s) \right\rangle.$$

  However, this blows up the feature norm bound of the feature map $\zeta_{\mathtt{lat}}^V(s)$ by a factor of $|\mathcal{A}|$, which will appear logarithmically in the bound obtained by BILIN-UCB.

**Model class + Coverability** ($\forall \pi : \mathcal{S} \to \Delta(\mathcal{A})$) (**?**).

- Latent class $\mathcal{M}_{\mathtt{lat}}$: $\mathcal{M}_{\mathtt{lat}} = \{M_{\mathtt{lat}} = (\mathcal{S}, \mathcal{A}, P_{\mathtt{lat}}, R_{\mathtt{lat}}, H)\}$ is a set of MDPs that all satisfy coverability with respect to all policies $\pi_{\mathtt{lat}} : \mathcal{S} \to \Delta(\mathcal{A})$, i.e. we have

$$\forall M_{\mathtt{lat}} \in \mathcal{M}_{\mathtt{lat}} : \quad C_{\mathsf{cov}}(M_{\mathtt{lat}}) = \inf_{\mu_h \in \Delta(\mathcal{S} \times \mathcal{A})} \sup_{h \in [H]} \sup_{\pi : \mathcal{S} \to \Delta(\mathcal{A})} \left\| \frac{d_h^{M_{\mathtt{lat}},\pi}}{\mu_h} \right\|_\infty < \infty$$

- Latent complexity comp: We take $\mathsf{comp}(\mathcal{M}_{\mathtt{lat}}, \varepsilon, \delta) = \mathsf{poly}(C_{\mathsf{cov}}, H, \log|\mathcal{M}_{\mathtt{lat}}|, \varepsilon^{-1}, \log(\delta^{-1}))$, which is attainable by the GOLF algorithm via the results of Xie, Foster, Bai, Jiang, and Kakade (see also Lemma F.3). We obtain this by noting that a realizable model class can be used to construct a realizable value function class $\mathcal{F}$ and a complete value function class $\mathcal{G}$ of sizes $\log|\mathcal{F}| = \log|\mathcal{M}|$ and $\log|\mathcal{G}| = \mathcal{O}(\log|\mathcal{M}|)$.

- Openness (**?**): A negative result does not follow from the existing constructions. The tree-based construction of Theorem 3.1 satisfies coverability with $C_{\mathsf{cov}} = \exp(\Omega(H))$ and the bandit-based construction of Theorem E.1 satisfies coverability with $C_{\mathsf{cov}} = |\mathcal{A}|$. In both cases, the lower bounds cannot be used to rule out statistical modularity with the above latent complexity. Similarly, it unclear how to obtain a positive result for the latent-dynamics class $\langle\!\langle M_{\mathtt{lat}}, \Phi \rangle\!\rangle$.

### E.3 Proofs for Lower Bounds (Theorems 3.1 and E.1)

#### E.3.1 Main lower bound (Theorem 3.1)

We will prove the following result.

**Theorem 3.1** (Impossibility of statistical modularity). *For every $N \geq 4$, there exists a decoder class $\Phi$ with $|\Phi| = N$ and a family of base MDPs $\mathcal{M}_{\mathrm{lat}}$ satisfying (i) $|\mathcal{M}_{\mathrm{lat}}| = 1$, (ii) $H \leq \mathcal{O}(\log(N))$, (iii) $|\mathcal{S}| = |\mathcal{X}| \leq N^2$, (iv) $|\mathcal{A}| = 2$, and such that*

*1. For all $\varepsilon, \delta > 0$, we have $\mathsf{comp}(\mathcal{M}_{\mathrm{lat}}, \varepsilon, \delta) = 0$.*

*2. For an absolute constant $c > 0$, $\mathsf{comp}(\langle\!\langle \mathcal{M}_{\mathrm{lat}}, \Phi \rangle\!\rangle, c, c) \geq \Omega(N/\log(N))$.*

**Proof.** Let $N$ be given and assume without loss of generality that it is a power of 2. We first construct the class of latent-dynamics MDPs, following Song et al. [SWFK24].

**Latent MDP.** The construction has a single "known" latent MDP $M_{\mathrm{lat}}$, so that the only uncertainty in the family of latent-dynamics MDPs we construct arises from the emission processes. We set $\mathcal{M}_{\mathrm{lat}} = \{M_{\mathrm{lat}}\}$. Set $H = \log_2(N) + 1$ and $\mathcal{A} = \{0, 1\}$. We define the state space and latent transition dynamics as follows.

- The state space can be partitioned as $\mathcal{S} = \mathcal{S}^1, \ldots, \mathcal{S}^N$.

- Each block $\mathcal{S}^i$ corresponds to a standard depth-$H$ binary tree MDP with deterministic dynamics (e.g., Osband et al.; Domingues et al. [OVR16; DMKV21]). There is a single "root" node at layer $h = 1$, which we denote by $s_{\mathrm{root}}^i$, and $N$ "leaf" nodes at layer $H$, which we denote by $\left\{ s_{\mathrm{leaf}}^{i,j} \right\}_{j \in [N]}$. For each $h = 1, \ldots, H - 1$, choosing action 0 leads to the left successor of the current state deterministically, and choosing action 1 leads to the right sucessor; this process continues until we reach a leaf node at layer $H$.

- The initial state distribution is $P_{\mathrm{lat},1}(\emptyset) = \mathtt{Unif}(s_{\mathrm{root}}^1, \ldots, s_{\mathrm{root}}^N)$.

- There are no rewards for layers $1, \ldots, H - 1$. For layer $H$, the reward is

$$R_H(s_{\mathrm{leaf}}^{i,j}, \cdot) = \mathbb{I}\{j = i\}. \tag{17}$$

This construction can summarized as follows. At layer 1, we draw the index of one of $N$ binary trees uniformly at random, and initialize into the root of the tree. From here, we receive a reward of 1 if we successfully navigate to the leaf node whose index agrees with the index of the tree itself, and receive a reward of 0 otherwise.

Note that the total number of latent states in this construction is $|\mathcal{S}| = N \cdot |\mathcal{S}_1| = N(2N - 1)$

**Observation space and decoder class.** Let us introduce some additional notation. For each block $\mathcal{S}^i$, let $\mathcal{S}_h^i := \{s_h^{i,j}\}_{j \in [2^{h-1}]}$ denote the states in block $i$ that are reachable at layer $h$, so that $\mathcal{S}_1^i = \left\{ s_{\mathrm{root}}^i \right\}$ and $\mathcal{S}_H^i = \{s_{\mathrm{leaf}}^{i,j}\}_{j \in [N]}$. We define $\mathcal{X} = \mathcal{S}$ so that $|\mathcal{X}| \leq 4N^2$, and consider a class of emission processes corresponding to deterministic maps. Let $\Sigma$ denote the set of cyclic permutations on $N$ elements, excluding the identity permutation. That is, each $\sigma_i \in \Sigma$ takes the form

$$\sigma_i : k \mapsto k + i \mod N \quad \text{for } i \in \{1, \ldots, N\}.$$

For each $\sigma \in \Sigma$, we consider the emission process

$$\psi_h^\sigma(\cdot \mid s_h^{(i,j)}) = \mathbb{I}_{s_h^{(\sigma(i),j)}}.$$

That is, $\psi^\sigma$ shifts the index of the binary tree containing $s_h^{(i,j)}$ according to $\sigma$. Let $\Psi = \{\psi^\sigma \mid \sigma \in \Sigma\}$. Consider the decoder class

$$\Phi = \Psi^{-1} := \left\{ s^i \mapsto s^{\psi^{-1}(i)} \mid \psi \in \Psi \right\},$$

which has $|\Phi| = N$. We consider the class of rich-observation MDPs given by

$$\langle\!\langle \mathcal{M}_{\mathrm{lat}}, \Phi \rangle\!\rangle := \left\{ M^i := \langle\!\langle M_{\mathrm{lat}}, \psi^{\sigma_i} \rangle\!\rangle \mid \sigma_i \in \Sigma \right\}. \tag{18}$$

It is clear that this class of rich-observation MDPs satisfies the decodability assumption for emissions $\Psi$.

**Sample complexity lower bound.** To lower bound the sample complexity, we prove a lower bound on the constrained PAC Decision-Estimation Coefficient (DEC) of [FGH23]. For an arbitrary MDP $\overline{M}$ (defined over the space $\mathcal{X}$) and $\varepsilon \in [0, 2^{1/2}]$, define[18]

$$\mathsf{dec}_\varepsilon(\mathcal{M}, \overline{M}) = \inf_{p,q \in \Delta(\Pi)} \sup_{M \in \mathcal{M}} \left\{ \mathbb{E}_{\pi \sim p}[J^M(\pi_M) - J^M(\pi)] \mid \mathbb{E}_{\pi \sim q}\left[D_{\mathsf{H}}^2\big(M(\pi), \overline{M}(\pi)\big)\right] \leq \varepsilon^2 \right\},$$

where $M(\pi)$ denotes the law over trajectories $(x_1, a_1, r_1), \ldots, (x_H, a_H, r_H)$ induced by executing the policy $\pi$ in the MDP $M$, $J^M(\pi)$ denotes the expected reward for policy $\pi$ under $M$, and $\pi_M$ denotes the optimal policy for $M$. We further define

$$\mathsf{dec}_\varepsilon(\mathcal{M}) = \sup_{\overline{M}} \mathsf{dec}_\varepsilon(\mathcal{M}, \overline{M}),$$

where the supremum ranges over all MDPs defined over $\mathcal{X}$ and $\mathcal{A}$. We now appeal to the following technical lemma.

**Lemma E.1.** *For all $\varepsilon^2 \geq 4/N$, we have that* $\mathsf{dec}_\varepsilon(\langle\!\langle \mathcal{M}_{\mathsf{lat}}, \Phi \rangle\!\rangle) \geq \frac{1}{2}$.

In light of Lemma E.1, it follows from Theorem 2.1 in Foster et al. [FGH23][19] that any PAC RL algorithm that uses $T$ episodes of interaction for $T \log(T) \leq c \cdot N$ must have $\mathbb{E}[J^M(\pi_M) - J^M(\widehat{\pi})] \geq c'$ for a worst-case MDP in $\mathcal{M}$, where $c, c' > 0$ are absolute constants. This implies that any PAC RL which has $\mathbb{E}[J^M(\pi_M) - J^M(\widehat{\pi})] \leq c'$ must have $T \log(T) \geq c \cdot N$ and thus $T \geq c \cdot N / \log(N)$.

$\square$

**Proof of Lemma E.1.** Define $\overline{M}_{\mathsf{lat}}$ as the latent-space MDP that has identical dynamics to $M_{\mathsf{lat}}$ but, has zero reward in every state, and define $\overline{M} := \langle\!\langle \overline{M}_{\mathsf{lat}}, \mathsf{id} \rangle\!\rangle$ as the rich-observation MDP obtained by composing $\overline{M}_{\mathsf{lat}}$ with the "identity" emission process $\mathsf{id}$ that sets $x_h = s_h$. Observe that $\overline{M}$ and $M^i$, induce identical dynamics in observation space if rewards are ignored: For all policies $\pi$,

$$\mathbb{P}^{\overline{M},\pi}[(x_1, a_1), \ldots, (x_H, a_H) = \cdot] = \mathbb{P}^{M^i,\pi}[(x_1, a_1), \ldots, (x_H, a_H) = \cdot]. \tag{19}$$

It follows that for each $i$, for all policies $\pi$, we have

$$
\begin{aligned}
& D_{\mathsf{H}}^2\big(M^i(\pi), \overline{M}(\pi)\big) \\
& = D_{\mathsf{H}}^2\big((\langle\!\langle M_{\mathsf{lat}}, \psi_i \rangle\!\rangle)(\pi), (\langle\!\langle \overline{M}_{\mathsf{lat}}, \mathsf{id} \rangle\!\rangle)(\pi)\big) \\
& = \sum_{j=1}^N \mathbb{P}^{\overline{M},\pi}\left[x_H = s_{\mathsf{leaf}}^{(\psi_i(j),j)}\right] \cdot D_{\mathsf{H}}^2(\mathbb{I}_1, \mathbb{I}_0) \\
& = 2 \sum_{j=1}^N \mathbb{P}^{\overline{M},\pi}\left[x_H = s_{\mathsf{leaf}}^{(\psi_i(j),j)}\right] \\
& = \frac{2}{N} \sum_{j=1}^N \mathbb{P}^{\overline{M},\pi}\left[x_H = s_{\mathsf{leaf}}^{(\psi_i(j),j)} \mid x_1 = s_{\mathsf{root}}^{(\psi_i(j))}\right], \tag{20} \\
& = \frac{2}{N} \sum_{j=1}^N \mathbb{P}^{\overline{M},\pi}\left[x_H = s_{\mathsf{leaf}}^{(j,\psi_i^{-1}(j))} \mid x_1 = s_{\mathsf{root}}^{(j)}\right], \tag{21}
\end{aligned}
$$

since the learner receives identical feedback in the MDPs $M^i$ and $\overline{M}$ unless they reach the observation $x_H = s_{\mathsf{leaf}}^{(\psi_i(j),j)}$ for some $j$ (corresponding to latent state $s_{\mathsf{leaf}}^{(j,j)}$ in $M^i$), in which case they receiver reward 1 in $M^i$ but reward 0 in $\overline{M}$. We now claim that for any $q \in \Delta(\Pi)$, there exists a set of at least $N/2$ indices $\mathcal{I}_q \subset [N]$ such that

$$\mathbb{E}_{\pi \sim q}\left[D_{\mathsf{H}}^2\big(M^i(\pi), \overline{M}(\pi)\big)\right] \leq \frac{4}{N} \tag{22}$$

---

[18]For measures $\mathbb{P}$ and $\mathbb{Q}$, we define squared Hellinger distance by $D_{\mathsf{H}}^2(\mathbb{P}, \mathbb{Q}) = \int (\sqrt{d\mathbb{P}} - \sqrt{d\mathbb{Q}})^2$.

[19]Theorem 2.1 in Foster et al. [FGH23] is stated with respect to $\sup_{\overline{M} \in \mathsf{conv}(\mathcal{M})} \mathsf{dec}_\varepsilon(\mathcal{M}, \overline{M})$, but the actual proof (Section 2.2) gives a stronger result that scales with $\sup_{\overline{M}} \mathsf{dec}_\varepsilon(\mathcal{M}, \overline{M})$.

for all $i \in \mathcal{I}_q$. To see this, note that by Eq. (21), we have

$$\mathbb{E}_{i \sim \mathrm{Unif}([N])} \mathbb{E}_{\pi \sim q} \left[ D_{\mathsf{H}}^2 \big( M^i(\pi), \overline{M}(\pi) \big) \right] \leq \mathbb{E}_{\pi \sim q} \left[ \frac{2}{N} \sum_{j=1}^{N} \frac{1}{N} \sum_{i=1}^{N} \mathbb{P}^{\overline{M}, \pi} \left[ x_H = s_{\mathsf{leaf}}^{(j, \psi_i^{-1}(j))} \mid x_1 = s_{\mathsf{root}}^{(j)} \right] \right]$$

$$\leq \mathbb{E}_{\pi \sim q} \left[ \frac{2}{N} \sum_{j=1}^{N} \frac{1}{N} \right] = \frac{2}{N},$$

where the second inequality uses that $\sum_{i=1}^{N} \mathbb{P}^{\overline{M}, \pi} \left[ x_H = s_{\mathsf{leaf}}^{(j, \psi_i^{-1}(j))} \mid x_1 = s_{\mathsf{root}}^{(j)} \right] \leq 1$, as the events in the sum are mutually exclusive (and the event we condition on does not depend on $i$). We conclude by Markov's inequality that $\mathbb{P}_{i \sim \mathrm{Unif}([N])} \left[ \mathbb{E}_{\pi \sim q} \left[ D_{\mathsf{H}}^2 \big( M^i(\pi), \overline{M}(\pi) \big) \right] \geq 4/N \right] \leq 1/2$, giving $\mathcal{I}_q \geq N/2$.

From Eq. (26), we conclude that for all $\varepsilon^2 \geq 4/N$,

$$\mathsf{dec}_\varepsilon(\mathcal{M}, \overline{M}) \geq \inf_{q \in \Delta(\Pi)} \inf_{p \in \Delta(\Pi)} \sup_{i \in \mathcal{I}_q} \left\{ \mathbb{E}_{\pi \sim p} \left[ J^{M^i}(\pi_{M^i}) - J^{M^i}(\pi) \right] \right\}.$$

To lower bound this quantity, observe that for any index $i$ and any policy $\pi$, we have

$$J^{M^i}(\pi_{M^i}) - J^{M^i}(\pi) = \frac{1}{N} \sum_{j=1}^{N} \mathbb{P}^{M^{(i)}, \pi} \left[ x_H \neq s_{\mathsf{leaf}}^{(\psi_i(j), j)} \mid x_1 = s_{\mathsf{root}}^{(\psi_i(j))} \right]$$

$$= 1 - \frac{1}{N} \sum_{j=1}^{N} \mathbb{P}^{M^{(i)}, \pi} \left[ x_H = s_{\mathsf{leaf}}^{(\psi_i(j), j)} \mid x_1 = s_{\mathsf{root}}^{(\psi_i(j))} \right]$$

$$= 1 - \frac{1}{N} \sum_{j=1}^{N} \mathbb{P}^{\overline{M}, \pi} \left[ x_H = s_{\mathsf{leaf}}^{(\psi_i(j), j)} \mid x_1 = s_{\mathsf{root}}^{(\psi_i(j))} \right]$$

$$= 1 - \frac{1}{N} \sum_{j=1}^{N} \mathbb{P}^{\overline{M}, \pi} \left[ x_H = s_{\mathsf{leaf}}^{(j, \psi_i^{-1}(j))} \mid x_1 = s_{\mathsf{root}}^{(j)} \right],$$

where the third inequality uses Eq. (19). We conclude that for any distribution $p, q \in \Delta(\Pi)$,

$$\sup_{i \in \mathcal{I}_q} \left\{ \mathbb{E}_{\pi \sim p} \left[ J^{M^i}(\pi_{M^i}) - J^{M^i}(\pi) \right] \right\}$$

$$\geq \mathbb{E}_{i \sim \mathrm{Unif}(\mathcal{I}_q)} \left\{ \mathbb{E}_{\pi \sim p} \left[ J^{M^i}(\pi_{M^i}) - J^{M^i}(\pi) \right] \right\}$$

$$\geq 1 - \frac{1}{N} \sum_{j=1}^{N} \mathbb{E}_{i \sim \mathrm{Unif}(\mathcal{I}_q)} \mathbb{P}^{\overline{M}, \pi} \left[ x_H = s_{\mathsf{leaf}}^{(j, \psi_i^{-1}(j))} \mid x_1 = s_{\mathsf{root}}^{(j)} \right]$$

$$= 1 - \frac{1}{N} \sum_{j=1}^{N} \frac{1}{|\mathcal{I}_q|} \sum_{i \in \mathcal{I}_q} \mathbb{P}^{\overline{M}, \pi} \left[ x_H = s_{\mathsf{leaf}}^{(j, \psi_i^{-1}(j))} \mid x_1 = s_{\mathsf{root}}^{(j)} \right] \geq 1 - \frac{1}{|\mathcal{I}_q|} \geq \frac{1}{2}$$

as long as $N \geq 4$, where the second-to-last inequality uses that for all $j$, the events $\{ x_H = s_{\mathsf{leaf}}^{(j, \psi_i^{-1}(j))} \mid x_1 = s_{\mathsf{root}}^{(j)} \}$ are disjoint for all $i$. Since this lower bound holds uniformly for all $q, p \in \Delta(\Pi)$, we conclude that

$$\mathsf{dec}_\varepsilon(\langle\!\langle \mathcal{M}_{\mathtt{lat}}, \Phi \rangle\!\rangle, \overline{M}) \geq \frac{1}{2}.$$

$\square$

### E.3.2 Proof of alternative lower bound (Theorem E.1)
We will prove the following result.

**Theorem E.1** (Alternative lower bound). *For every $N \geq 4$, there exists an emission class $\Psi$ and a decoder class $\Phi$ with $|\Psi| = |\Phi| = N$ and a family of latent MDPs $\mathcal{M}_{\mathrm{lat}}$ satisfying (i) $|\mathcal{M}_{\mathrm{lat}}| = 1$, (ii) $H = 1$, (iii) $|\mathcal{S}| = |\mathcal{X}| = N$, (iv) $|\mathcal{A}| = N$, and such that*

*1. For all $\varepsilon, \delta > 0$, we have $\mathsf{comp}(\mathcal{M}_{\mathrm{lat}}, \varepsilon, \delta) = 0$.*

*2. For an absolute constant $c > 0$, $\mathsf{comp}(\langle\!\langle \mathcal{M}_{\mathrm{lat}}, \Phi \rangle\!\rangle, c, c) \geq \Omega(N/\log(N))$.*

**Proof of Theorem E.1.** We repeat more or less repeat the same proof as Theorem 3.1, but with the appropriate modifications to translate from the contextual tree-based construction in Theorem 3.1 to the contextual bandit-based construction in the theorem statement. Let $N$ be given and assume without loss of generality that it is a power of 2.

**Latent MDP.** Our construction has a single "known" latent MDP $M_{\mathrm{lat}}$; that is, the only uncertainty in the family of rich-observation MDPs we construct arises from the emission processes. Set $\mathcal{M}_{\mathrm{lat}} = \{M_{\mathrm{lat}}\}$. Set $H = 1$ and $\mathcal{A} = [N]$. We define the state space and latent transition dynamics as follows.

- The state space can be partitioned as $\mathcal{S} = \mathcal{S}^1, \ldots, \mathcal{S}^N$.

- Each block $\mathcal{S}^i$ corresponds to a single state $s^i$ with $N$ actions denoted by $a^i$, $i \in [N]$.

- The initial state distribution is $P_{\mathrm{lat},1}(\emptyset) = \mathsf{Unif}(s^1, \ldots, s^N)$.

- The reward function is

$$R_1(s^i, a^j) = \mathbb{I}\{j = i\}. \tag{23}$$

Informally, this construction can summarized as a contextual bandit (with uniform context distribution), with a reward of 1 if and only if we play the action corresponding to the index of the context drawn.

Note that the total number of latent states in this construction is $|\mathcal{S}| = N$ and the number of actions is $|\mathcal{A}| = N$.

**Observation space and decoder class.** We define $\mathcal{X} = \mathcal{S}$ so that $|\mathcal{X}| = |\mathcal{S}|$, and consider a class of emission processes corresponding to deterministic maps. Let $\Sigma$ denote the set of cyclic permutations on $N$ elements, excluding the identity permutation. That is, each $\sigma_i \in \Sigma$ takes the form

$$\sigma_i : k \mapsto k + i \mod N, \quad \text{for } i \in \{1, \ldots, N\}.$$

For each $\sigma \in \Sigma$, we consider the emission process

$$\psi^\sigma(\cdot \mid s^i) = \mathbb{I}_{s^{\sigma(i)}}(\cdot)$$

That is, $\psi^\sigma$ shifts the context $s^i$ according to $\sigma$. Let $\Psi = \{\psi^\sigma \mid \sigma \in \Sigma\}$. Consider the decoder class

$$\Phi = \Psi^{-1} := \left\{ s^i \mapsto s^{\psi^{-1}(i)} \mid \psi \in \Psi \right\},$$

which has $|\Phi| = N$. We consider the class of rich-observation MDPs given by

$$\langle\!\langle \mathcal{M}_{\mathrm{lat}}, \Phi \rangle\!\rangle := \left\{ M^i := \langle\!\langle M_{\mathrm{lat}}, \psi^{\sigma_i} \rangle\!\rangle \mid \sigma_i \in \Sigma \right\}. \tag{24}$$

It is clear that this class of rich-observation MDPs satisfies the decodability assumption for emissions $\Psi$.

**Sample complexity lower bound.** To lower bound the sample complexity, we prove a lower bound on the constrained PAC Decision-Estimation Coefficient (DEC) of [FGH23]. For an arbitrary MDP $\overline{M}$ (defined over the space $\mathcal{X}$) and $\varepsilon \in [0, 2^{1/2}]$, define[20]

$$\mathsf{dec}_\varepsilon(\mathcal{M}, \overline{M}) = \inf_{p,q \in \Delta(\Pi)} \sup_{M \in \mathcal{M}} \left\{ \mathbb{E}_{\pi \sim p}[J^M(\pi_M) - J^M(\pi)] \mid \mathbb{E}_{\pi \sim q}\left[D_{\mathsf{H}}^2\big(M(\pi), \overline{M}(\pi)\big)\right] \leq \varepsilon^2 \right\},$$

where $M(\pi)$ denotes the law over observations $(x_1, a_1, r_1)$ induced by executing the policy $\pi$ in the MDP $M$, $J^M(\pi)$ denotes the expected reward for policy $\pi$ under $M$, and $\pi_M$ denotes the optimal policy for $M$. We further define

$$\mathsf{dec}_\varepsilon(\mathcal{M}) = \sup_{\overline{M}} \mathsf{dec}_\varepsilon(\mathcal{M}, \overline{M}),$$

where the supremum ranges over all MDPs defined over $\mathcal{X}$ and $\mathcal{A}$. We now appeal to the following technical lemma.

---

[20]For measures $\mathbb{P}$ and $\mathbb{Q}$, we define squared Hellinger distance by $D_{\mathsf{H}}^2(\mathbb{P}, \mathbb{Q}) = \int(\sqrt{d\mathbb{P}} - \sqrt{d\mathbb{Q}})^2$.

**Lemma E.2.** *For all $\varepsilon^2 \geq 4/N$, we have that $\sup_{\overline{M}} \mathrm{dec}_\varepsilon(\mathcal{M}, \overline{M}) \geq \frac{1}{2}$.*

In light of Lemma E.2, it follows from Theorem 2.1 in Foster et al. [FGH23][21] that any PAC RL algorithm that uses $T$ episodes of interaction for $T \log(T) \leq c \cdot N$ must have $\mathbb{E}[J^M(\pi_M) - J^M(\widehat{\pi})] \geq c'$ for a worst-case MDP in $\mathcal{M}$, where $c, c' > 0$ are absolute constants. This implies that any PAC RL which has $\mathbb{E}[J^M(\pi_M) - J^M(\widehat{\pi})] \leq c'$ must have $T \log(T) \geq c \cdot N$ and thus $T \geq c \cdot N / \log(N)$. $\quad\square$

**Proof of Lemma E.2.** Define $\overline{M}_{\mathrm{lat}}$ as the latent-space MDP that has identical dynamics to $M_{\mathrm{lat}}$ but, has zero reward for every state-action pair, and define $\overline{M} := \langle\!\langle \overline{M}_{\mathrm{lat}}, \mathrm{id} \rangle\!\rangle$ as the rich-observation MDP obtained by composing $\overline{M}_{\mathrm{lat}}$ with the identity emission process that sets $x_h = s_h$. In the rest of the proof, we use the shorthand $\psi_i := \psi^{\sigma_i}$. Observe that $\overline{M}$ and $M^i$, induce identical dynamics in observation space if rewards are ignored, i.e. for all policies $\pi : \mathcal{X} \to \Delta(\mathcal{A})$,

$$\mathbb{P}^{\overline{M}, \pi}[(x_1, a_1) = \cdot] = \mathbb{P}^{M^i, \pi}[(x_1, a_1) = \cdot]. \tag{25}$$

It follows that for each $i$, for all policies $\pi$, we have

$$
\begin{aligned}
&D_{\mathsf{H}}^2\big(M^i(\pi), \overline{M}(\pi)\big) \\
&= D_{\mathsf{H}}^2\big((\langle\!\langle M_{\mathrm{lat}}, \psi_i \rangle\!\rangle)(\pi), (\langle\!\langle \overline{M}_{\mathrm{lat}}, \mathrm{id} \rangle\!\rangle)(\pi)\big) \\
&= \sum_{j=1}^N \mathbb{P}^{\overline{M}, \pi}\Big[x_1 = s^{\psi_i(j)}, a_1 = a^j\Big] \cdot D_{\mathsf{H}}^2(\mathbb{I}_1, \mathbb{I}_0) \\
&= 2\sum_{j=1}^N \mathbb{P}^{\overline{M}, \pi}\Big[x_1 = s^{\psi_i(j)}, a_1 = a^j\Big] \\
&= \frac{2}{N}\sum_{j=1}^N \mathbb{P}^{\overline{M}, \pi}\Big[a_1 = a^j \mid x_1 = s^{\psi_i(j)}\Big] \\
&= \frac{2}{N}\sum_{j=1}^N \mathbb{P}^{\overline{M}, \pi}\Big[a_1 = a^{\psi_i^{-1}(j)} \mid x_1 = s^j\Big]
\end{aligned}
$$

since the learner receives identical feedback in the MDPs $M^i$ and $\overline{M}$ unless they play the action $a_1 = a^j$ given observation $x_1 = s^{\psi_i(j)}$ (corresponding to latent state $s^i$ in $M^i$), in which case they receiver reward 1 in $M^i$ but reward 0 in $\overline{M}$. We now claim that for any $q \in \Delta(\Pi)$, there exists a set of at least $N/2$ indices $\mathcal{I}_q \subset [N]$ such that

$$\mathbb{E}_{\pi \sim q}\big[D_{\mathsf{H}}^2\big(M^i(\pi), \overline{M}(\pi)\big)\big] \leq \frac{4}{N} \tag{26}$$

for all $i \in \mathcal{I}_q$. To see this, note that by Eq. (21), we have

$$\mathbb{E}_{i \sim \mathrm{Unif}([N])} \mathbb{E}_{\pi \sim q}\big[D_{\mathsf{H}}^2\big(M^i(\pi), \overline{M}(\pi)\big)\big] \leq \mathbb{E}_{\pi \sim q}\left[\frac{2}{N}\sum_{j=1}^N \frac{1}{N}\sum_{i=1}^N \mathbb{P}^{\overline{M}, \pi}\Big[a_1 = a^{\psi_i^{-1}(j)} \mid x_1 = j\Big]\right]$$

$$\leq \mathbb{E}_{\pi \sim q}\left[\frac{2}{N}\sum_{j=1}^N \frac{1}{N}\right] = \frac{2}{N}.$$

We conclude by Markov's inequality that $\mathbb{P}_{i \sim \mathrm{Unif}([N])}\big[\mathbb{E}_{\pi \sim q}\big[D_{\mathsf{H}}^2\big(M^i(\pi), \overline{M}(\pi)\big)\big] \geq 4/N\big] \leq 1/2$, giving $\mathcal{I}_q \geq N/2$.

From Eq. (26), we conclude that for all $\varepsilon^2 \geq 4/N$,

$$\mathrm{dec}_\varepsilon(\langle\!\langle \mathcal{M}_{\mathrm{lat}}, \Phi \rangle\!\rangle, \overline{M}) \geq \inf_{q \in \Delta(\Pi)} \inf_{p \in \Delta(\Pi)} \sup_{i \in \mathcal{I}_q}\Big\{\mathbb{E}_{\pi \sim p}\Big[J^{M^i}(\pi_{M^i}) - J^{M^i}(\pi)\Big]\Big\}.$$

---

[21]Theorem 2.1 in Foster et al. [FGH23] is stated with respect to $\sup_{\overline{M} \in \mathrm{conv}(\mathcal{M})} \mathrm{dec}_\varepsilon(\mathcal{M}, \overline{M})$, but the actual proof (Section 2.2) gives a stronger result that scales with $\sup_{\overline{M}} \mathrm{dec}_\varepsilon(\mathcal{M}, \overline{M})$.

To lower bound this quantity, observe that for any index $i$ and any policy $\pi$, we have

$$J^{M^i}(\pi_{M^i}) - J^{M^i}(\pi) = 1 - \frac{1}{N}\sum_{j=1}^{N}\mathbb{P}^{M^{(i)},\pi}[a_1 = a^{(j)} \mid x_1 = s^{(\psi_i(j))}]$$

$$= 1 - \frac{1}{N}\sum_{j=1}^{N}\mathbb{P}^{\overline{M},\pi}[a_1 = a^{(j)} \mid x_1 = s^{(\psi_i(j))}]$$

$$= 1 - \frac{1}{N}\sum_{j=1}^{N}\mathbb{P}^{\overline{M},\pi}\left[a_1 = a^{(\psi_i^{-1}(j))} \mid x_1 = s^{(j)}\right],$$

where the third inequality uses Eq. (25). We conclude that for any distribution $p, q \in \Delta(\Pi)$,

$$\sup_{i \in \mathcal{I}_q}\left\{\mathbb{E}_{\pi \sim p}\left[J^{M^i}(\pi_{M^i}) - J^{M^i}(\pi)\right]\right\}$$

$$\geq \mathbb{E}_{i \sim \mathrm{Unif}(\mathcal{I}_q)}\left\{\mathbb{E}_{\pi \sim p}\left[J^{M^i}(\pi_{M^i}) - J^{M^i}(\pi)\right]\right\}$$

$$\geq 1 - \frac{1}{N}\sum_{j=1}^{N}\mathbb{E}_{i \sim \mathrm{Unif}(\mathcal{I}_q)}\,\mathbb{P}^{\overline{M},\pi}\left[a_1 = a^{(\psi_i^{-1}(j))} \mid x_1 = s^{(j)}\right]$$

$$= 1 - \frac{1}{N}\sum_{j=1}^{N}\frac{1}{|\mathcal{I}_q|}\sum_{i \in \mathcal{I}_q}\mathbb{P}^{\overline{M},\pi}\left[a_1 = a^{(\psi_i^{-1}(j))} \mid x_1 = s^{(j)}\right] \geq 1 - \frac{1}{|\mathcal{I}_q|} \geq \frac{1}{2}$$

as long as $N \geq 4$. Since this lower bound holds uniformly for all $q, p \in \Delta(\Pi)$, we conclude that

$$\mathsf{dec}_\varepsilon(\langle\!\langle \mathcal{M}_{\mathtt{lat}}, \Phi \rangle\!\rangle, \overline{M}) \geq \frac{1}{2}.$$

$\square$

# F Proofs for Section 3.3: Positive Results

This section is dedicated to our upper bound establishing that pushforward-coverable MDPs are statistically modular (Theorem 3.2). We provide a technical overview in Appendix F.1, and provide a full proof in Appendix F.2.

## F.1 Technical Overview: Low-dimensional embeddings for pushforward-coverable MDPs.

The idea behind our positive result is to show that under the conditions of Theorem 3.2, it is possible to construct an (approximately) Bellman-complete value function class for the latent-dynamics MDP $M_{\text{obs}}^\star$, at which point we can apply the GOLF algorithm of Jin et al. [JLM21]. We achieve this via two technical contributions. The first is the introduction of the *mismatch functions* $\Gamma_\phi$, formally defined as follows.

**Definition F.1** (Mismatch functions). *For a decodable emission process $\psi^\star$ and decoder $\phi \in \Phi$, the* mismatch function *for $\phi$, $\Gamma_\phi = \{\Gamma_{\phi,h} : \mathcal{S} \to \Delta(\mathcal{S})\}_{h=1}^H$, is defined, for every $h \in [H]$, as the probability kernel*

$$\Gamma_{\phi,h}(s_h' \mid s_h) := \mathbb{P}_{x_h \sim \psi_h^\star(s_h)}(\phi_h(x_h) = s_h').$$

The mismatch functions allow us to express functions of the decoders as latent objects, and we revisit them in the context of self-predictive estimation (Appendix A). For the present result, we show (Lemma D.7) that the mismatch functions can capture the observation-level Bellman backups for any function of the decoders. That is, for any $x_h, a_h$, letting $s_h = (\psi^\star)^{-1}(x_h)$ denote the true latent state, we have that for any $f_{\text{lat}} : \mathcal{S} \times \mathcal{A} \to \mathbb{R}$ and $\phi \in \Phi$:

$$[\mathcal{T}_h^{M_{\text{obs}}^\star}(f_{\text{lat}} \circ \phi_{h+1})](x_h, a_h) = [\mathcal{T}_h^{M_{\text{lat}}^\star}(\Gamma_{\phi,h+1} \circ V_{f_{\text{lat}}})](s_h, a_h). \tag{27}$$

That is, the Bellman update of $f_{\text{lat}} \circ \phi_{h+1}$ in the latent-dynamics MDP $M_{\text{obs}}^\star$ can be expressed as a Bellman update in the base MDP $M_{\text{lat}}^\star$ for a different (latent) function $\Gamma_{\phi,h+1} \circ V_{f_{\text{lat}}}(s_{h+1}) := \sum_{s_{h+1}'} \Gamma_{\phi,h+1}(s_{h+1}' \mid s_{h+1}) \max_{a'} f_{\text{lat}}(s_{h+1}', a')$.

However, the mismatch functions $\Gamma_\phi$ embed some knowledge of the emission process, and (with only decoder and base model realizability) are unknown to the learner. Our second technical contribution bypasses this by establishing a new structural property for pushforward-coverable MDPs (Lemma F.1): there exist low-dimensional linear embeddings of their transition kernels which can approximate Bellman backups for an arbitrary and *potentially unknown* set of functions, as long as the set is not too large.

**Lemma F.1** (Pushforward-coverable MDPs admit low-dimensional embeddings). *Let $M$ be a known MDP with reward function $r$, transition kernel $P$, and pushforward coverability parameter $C_{\text{push}}$. Let $\mu = \{\mu_h\}_{h \in [H]}$ denote its pushforward coverability distribution (i.e. the minimizer of Definition 3.3) and $\mathcal{F} \subseteq (\mathcal{S} \times [H] \to [0,1])$ be an arbitrary class of functions. Suppose that we sample $W \in \{\pm 1\}^{d \times \mathcal{S}}$ as a matrix of independent Rademacher random variables, and define*

$$\psi_h(s, a) = r_h(s, a) \oplus \frac{1}{\sqrt{d}} W \Big( P_h(\cdot \mid s, a) / \mu_h^{1/2}(\cdot) \Big)_{\cdot \in \mathcal{S}} \in \mathbb{R}^{d+1}.$$

*and*

$$w_{f,h} = 1 \oplus \frac{1}{\sqrt{d}} W \Big( \mu_h^{1/2}(\cdot) f_{h+1}(\cdot) \Big)_{\cdot \in \mathcal{S}} \in \mathbb{R}^{d+1}.$$

*Then for any $\varepsilon_{\text{apx}} \in (0, 1)$, as long as we set*

$$d \geq 2^9 \frac{C_{\text{push}} \log\big(16|\mathcal{F}|H\delta^{-1}/\varepsilon_{\text{apx}}\big)}{\varepsilon_{\text{apx}}},$$

*we have that for all $f \in \mathcal{F}$ and $h \in [H]$, with probability at least $1 - \delta$:*

$$\mathbb{E}_{\mu_h \otimes \text{Unif}(\mathcal{A})}\Big[\big(\texttt{clip}_{[0,2]}[\langle w_{f,h}, \psi_h(s,a)\rangle] - \mathcal{T}_h f_{h+1}(s,a)\big)^2\Big] \leq \varepsilon_{\text{apx}},$$

*as well as $\max_{s,a,h}\|\psi_h(s,a)\|_2^2 \leq C_{\text{push}}(16\log(|\mathcal{S}||\mathcal{A}|H) + 11)$ and $\max_{f,h}\|w_{f,h}\|_2^2 \leq 16\log(|\mathcal{F}|H) + 11$. We emphasize that the feature map $\psi = \{\psi_h\}_{h=1}^H$ is oblivious to $\mathcal{F}$, in the sense that it can be computed directly from $M$ without any knowledge of $\mathcal{F}$.*

We use this property, in conjunction with latent model realizability, to construct linear features that can approximate the right-hand-side of Eq. (27), thus yielding an (approximately) Bellman-complete value function class for the latent-dynamics MDP $M_{\text{obs}}^\star$.

## F.2 Proofs for Latent Model Class + Pushforward Coverability (Theorem 3.2)

In this section, we establish positive results under latent MDP classes which satisfy pushforward coverability. We assume that every model in $\mathcal{M}_{\mathtt{lat}}$ satisfies *pushforward coverability*, defined as follows:

**Definition F.2** (Pushforward coverability). *The pushforward coverability coefficient $C_{\mathtt{push}}$ for an MDP $M$ with transition kernel $P$ is defined by*

$$C_{\mathtt{push}}(M) = \max_{h \in [H]} \inf_{\mu \in \Delta(\mathcal{S})} \sup_{(s,a,s') \in \mathcal{S} \times \mathcal{A} \times \mathcal{S}} \frac{P_{h-1}(s' \mid s, a)}{\mu(s')}. \tag{28}$$

*The pushforward coverability coefficient for an MDP class $\mathcal{M}$ is defined by*

$$C_{\mathtt{push}}(\mathcal{M}) = \max_{M \in \mathcal{M}} C_{\mathtt{push}}(M).$$

Note that for any MDP $M$ we always have

$$C_{\mathtt{cov}}(M, \Pi_{\mathtt{rns}}) \leq C_{\mathtt{push}}(M)|\mathcal{A}|, \tag{29}$$

where $C_{\mathtt{cov}}$ is the state-action coverability coefficient (Definition D.3). Thus, an MDP with low pushforward coverability is also an MDP with low state-action coverability for all policies (upto a dependence on $|\mathcal{A}|$).

We will show the show the following result.

**Theorem 3.2** (Pushforward-coverable MDPs are statistically modular). *Let $\mathcal{M}_{\mathtt{lat}}$ be a base MDP class such that each $M_{\mathtt{lat}} \in \mathcal{M}_{\mathtt{lat}}$ has pushforward coverability bounded by $C_{\mathtt{push}}(M_{\mathtt{lat}}) \leq C_{\mathtt{push}}$. Then, for any decoder class $\Phi$, we have:*

*1. $\mathtt{comp}(\mathcal{M}_{\mathtt{lat}}, \varepsilon, \delta) \leq \mathtt{poly}(C_{\mathtt{push}}, |\mathcal{A}|, H, \log|\mathcal{M}_{\mathtt{lat}}|, \varepsilon^{-1}, \log(\delta^{-1}))$, and*

*2. $\mathtt{comp}(\langle\!\langle \mathcal{M}_{\mathtt{lat}}, \Phi \rangle\!\rangle, \varepsilon, \delta) \leq \mathtt{poly}(C_{\mathtt{push}}, |\mathcal{A}|, H, \log|\mathcal{M}_{\mathtt{lat}}|, \log|\Phi|, \varepsilon^{-1}, \log(\delta^{-1}), \log\log|\mathcal{S}|)$.*

The proof comes in three parts. We will firstly show that MDP that satisfies pushforward coverability admit low-dimensional feature maps that can approximate Bellman backups (Appendix F.2.1), then establish that a regret bound for the GOLF algorithm [XFBJK23] under misspecification (Appendix F.2.2), and then combine these ingredients (Appendix F.2.3).

### F.2.1 A structural result: Pushforward-coverable MDPs are approximately low-rank

Our central technical result for this section is Lemma F.1, which is based on a variant of the Johnson-Lindenstrauss lemma and establishes that under pushforward coverability, we can define a linear feature class which satisfies an approximate form of Bellman completeness. We define the clipping operator via

$$\mathtt{clip}_{[0,2]}(x) := \max\{\min\{x, 2\}, 0\}.$$

We prove the following lemma.

**Lemma F.1** (Pushforward-coverable MDPs admit low-dimensional embeddings). *Let $M$ be a known MDP with reward function $r$, transition kernel $P$, and pushforward coverability parameter $C_{\mathtt{push}}$. Let $\mu = \{\mu_h\}_{h \in [H]}$ denote its pushforward coverability distribution (i.e. the minimizer of Definition 3.3) and $\mathcal{F} \subseteq (\mathcal{S} \times [H] \to [0,1])$ be an arbitrary class of functions. Suppose that we sample $W \in \{\pm 1\}^{d \times \mathcal{S}}$ as a matrix of independent Rademacher random variables, and define*

$$\psi_h(s,a) = r_h(s,a) \oplus \frac{1}{\sqrt{d}} W \left( P_h(\cdot \mid s, a)/\mu_h^{1/2}(\cdot) \right)_{\cdot \in \mathcal{S}} \in \mathbb{R}^{d+1}.$$

*and*

$$w_{f,h} = 1 \oplus \frac{1}{\sqrt{d}} W \left( \mu_h^{1/2}(\cdot) f_{h+1}(\cdot) \right)_{\cdot \in \mathcal{S}} \in \mathbb{R}^{d+1}.$$

*Then for any $\varepsilon_{\mathtt{apx}} \in (0, 1)$, as long as we set*

$$d \geq 2^9 \frac{C_{\mathtt{push}} \log\left(16|\mathcal{F}|H\delta^{-1}/\varepsilon_{\mathtt{apx}}\right)}{\varepsilon_{\mathtt{apx}}},$$

*we have that for all $f \in \mathcal{F}$ and $h \in [H]$, with probability at least $1 - \delta$:*

$$\mathbb{E}_{\mu_h \otimes \text{Unif}(\mathcal{A})}\left[\left(\text{clip}_{[0,2]}[\langle w_{f,h}, \psi_h(s,a)\rangle] - \mathcal{T}_h f_{h+1}(s,a)\right)^2\right] \leq \varepsilon_{\text{apx}},$$

*as well as $\max_{s,a,h}\|\psi_h(s,a)\|_2^2 \leq C_{\text{push}}(16\log(|\mathcal{S}||\mathcal{A}|H) + 11)$ and $\max_{f,h}\|w_{f,h}\|_2^2 \leq 16\log(|\mathcal{F}|H) + 11$. We emphasize that the feature map $\psi = \{\psi_h\}_{h=1}^H$ is oblivious to $\mathcal{F}$, in the sense that it can be computed directly from $M$ without any knowledge of $\mathcal{F}$.*

**Proof of Lemma F.1.** Fix $h \in [H]$, whose dependence we omit for cleanliness. We begin by verifying that, in expectation, $\langle w_f, \psi(s,a)\rangle$ is equal to $\mathcal{T}f(s,a)$. For this, note that

$$\langle w_f, \psi(s,a)\rangle$$

$$= r(s,a) + \frac{1}{d}\sum_{i=1}^d\left(\sum_{s'\in\mathcal{S}}W_{i,s'}\frac{P(s'\mid s,a)}{\mu^{1/2}(s')}\right)\left(\sum_{s''\in\mathcal{S}}W_{i,s''}\mu^{1/2}(s'')f(s'')\right)$$

$$= r(s,a) + \sum_{s'\in\mathcal{S}}P(s'\mid s,a)f(s') + \frac{1}{d}\sum_{i=1}^d\sum_{s'\in\mathcal{S}}\sum_{\substack{s''\in\mathcal{S}\\s''\neq s'}}W_{i,s'}\frac{P(s'\mid s,a)}{\mu^{1/2}(s')}W_{i,s''}\mu^{1/2}(s'')f(s'').$$

Consequently, we have

$$|\mathcal{T}f(s,a) - \langle w_f, \psi(s,a)\rangle| = \left|\frac{1}{d}\sum_{i=1}^d\sum_{s'\in\mathcal{S}}\sum_{\substack{s''\in\mathcal{S}\\s''\neq s'}}W_{i,s'}\frac{P(s'\mid s,a)}{\mu^{1/2}(s')}W_{i,s''}\mu^{1/2}(s'')f(s'')\right|. \quad (30)$$

Note that this remaining noise term is zero-mean – we will show in the sequel that it can be made small by picking $d$ appropriately. We next examine the norms of the vectors $\psi(s,a)$ and $w_f$. Note that we have

$$\|\psi(s,a)\|_2^2 = \frac{1}{d}\sum_{i=1}^d\left(\sum_{s'\in\mathcal{S}}W_{i,s'}\frac{P(s'\mid s,a)}{\mu^{1/2}(s')}\right)^2$$

$$= \sum_{s'\in\mathcal{S}}\frac{P^2(s'\mid s,a)}{\mu(s')} + \frac{1}{d}\sum_{i=1}^d\sum_{s'\in\mathcal{S}}\sum_{\substack{s''\in\mathcal{S}\\s''\neq s'}}W_{i,s'}W_{i,s''}\frac{P(s'\mid s,a)}{\mu^{1/2}(s')}\frac{P(s''\mid s,a)}{\mu^{1/2}(s'')}$$

$$\leq C_{\text{push}} + \frac{1}{d}\sum_{i=1}^d\sum_{s'\in\mathcal{S}}\sum_{\substack{s''\in\mathcal{S}\\s''\neq s'}}W_{i,s'}W_{i,s''}\frac{P(s'\mid s,a)}{\mu^{1/2}(s')}\frac{P(s''\mid s,a)}{\mu^{1/2}(s'')}, \quad (31)$$

where we have used that

$$\sum_{s'\in\mathcal{S}}\frac{P^2(s'\mid s,a)}{\mu(s')} \leq C_{\text{push}}\sum_{s'\in\mathcal{S}}P(s'\mid s,a) = C_{\text{push}}$$

by definition of pushforward coverability. Further note that we have

$$\|w_f\|_2^2 = \frac{1}{d}\sum_{i=1}^d\left(\sum_{s'\in\mathcal{S}}W_{i,s'}\mu^{1/2}(s')f(s')\right)^2$$

$$= \mathbb{E}_{s'\sim\mu}[f(s')] + \frac{1}{d}\sum_{i=1}^d\sum_{s'\in\mathcal{S}}\sum_{\substack{s''\in\mathcal{S}\\s''\neq s'}}W_{i,s'}W_{i,s''}\mu^{1/2}(s')f(s')\cdot\mu^{1/2}(s'')f(s'')$$

$$\leq 1 + \frac{1}{d}\sum_{i=1}^d\sum_{s'\in\mathcal{S}}\sum_{\substack{s''\in\mathcal{S}\\s''\neq s'}}W_{i,s'}W_{i,s''}\mu^{1/2}(s')f(s')\cdot\mu^{1/2}(s'')f(s''). \quad (32)$$

We will now appeal to the following technical lemma to upper bound Eq. (30), Eq. (31), and Eq. (32) by establishing that the Rademacher noise terms concentrate to their expectations. The proof of the lemma will be given in the sequel.

**Lemma F.2.** *Let $u, v \in \mathbb{R}^n$, and let $W \in \{\pm 1\}^{d \times n}$ have independent Rademacher entries. Then with probability at least $1 - \delta$,*

$$\left| \frac{1}{d} \sum_{i \in [d]} \sum_{j \in [n]} \sum_{\substack{k \in [n] \\ k \neq j}} W_{i,j} W_{i,k} u_j v_k \right| \leq \|u\|_2 \|v\|_2 \cdot \sqrt{\frac{32 \log(2\delta^{-1})}{d}} + \|u\|_2^2 \|v\|_2^2 \cdot \frac{64 \log(2\delta^{-1})}{d}. \quad (33)$$

*Furthermore, for any set of vectors $\mathcal{V} \subset \mathbb{R}^n$, we also have*

$$\frac{1}{d} \max_{v \in \mathcal{V}} \sum_{i \in [d]} \sum_{j \in [n]} \sum_{\substack{k \in [n] \\ k \neq j}} W_{i,j} W_{i,k} v_j v_k$$

$$\leq \max_{v \in \mathcal{V}} \|v\|_2^2 (16 \log|\mathcal{V}| + 9) + \max_{v \in \mathcal{V}} \|v\|_2^2 \cdot \sqrt{\frac{32 \log(2\delta^{-1})}{d}} + \max_{v \in \mathcal{V}} \|v\|_2^4 \cdot \frac{64 \log(2\delta^{-1})}{d}.$$

Let $(s, a) \in \mathcal{S} \times \mathcal{A}$ and $f \in \mathcal{F}$. To bound $|\langle \psi(s,a), w_f \rangle - \mathcal{T}f(s,a)|$ (cf. Eq. (30)), we apply the first bound of Lemma F.2 with $u = \left( P(s' \mid s,a)/\mu^{1/2}(s') \right)_{s' \in \mathcal{S}}$ and $v = \left( \mu^{1/2}(s') f(s') \right)_{s' \in \mathcal{S}}$, which gives

$$|\langle \psi(s,a), w_f \rangle - \mathcal{T}f(s,a)| \leq \sqrt{\frac{32 C_{\mathsf{push}} \log(2\delta^{-1})}{d}} + 64 C_{\mathsf{push}} \frac{\log(2\delta^{-1})}{d} := \varepsilon(\delta^{-1}), \quad (34)$$

where we have again used that $\|u\|_2^2 = \sum_{s' \in \mathcal{S}} \frac{P^2(s'|s,a)}{\mu(s')} \leq C_{\mathsf{push}}$ and also that $\|v\|_2^2 = 1$ since $\|f\|_\infty \leq 1$ for all $f \in \mathcal{F}$. To bound Eq. (31), we apply the second bound of Lemma F.2 with $\mathcal{V} = \left\{ \left( \frac{P_{h-1}(s'|s,a)}{\mu_h^{1/2}(s')} \right)_{s' \in \mathcal{S}} \right\}_{\substack{s,a \in \mathcal{S} \times \mathcal{A} \\ h \times [H]}}$, which gives

$$\max_{s,a \in \mathcal{S} \times \mathcal{A}, h \in [H]} \|\psi_h(s,a)\|_2^2 \leq C_{\mathsf{push}} (16 \log|\mathcal{S}||\mathcal{A}|H + 9) + C_{\mathsf{push}} \sqrt{\frac{32 \log(2\delta^{-1})}{d}} + C_{\mathsf{push}}^2 \frac{64 \log(2\delta^{-1})}{d}.$$

Lastly, to bound Eq. (32), we take $\mathcal{V} = \left\{ \left( \mu_h^{1/2}(s') f_h(s') \right)_{s' \in \mathcal{S}} \right\}_{\substack{f \in \mathcal{F} \\ h \in [H]}}$ in Lemma F.2, which establishes that

$$\max_{f \in \mathcal{F}, h \in [H]} \|w_{f,h}\|_2^2 \leq 9 + 16 \log|\mathcal{F}|H + \sqrt{\frac{32 \log(2\delta^{-1})}{d}} + \frac{64 \log(2\delta^{-1})}{d}.$$

Note that Eq. (34) establishes that the Bellman backup $\mathcal{T}f(s,a)$ is well-approximated by $\langle \psi(s,a), w_f \rangle$ only at a single state-action pair $(s,a)$. We can obtain an $L_\infty$-approximation guarantee by taking a union bound over $\mathcal{S}$ and $\mathcal{A}$, which would incur a dependence on $\log|\mathcal{S}|$ in the final sample complexity. Here, we bypass this by instead requiring only an approximation guarantee under the $L_2(\mu \otimes \mathtt{Unif}(\mathcal{A}))$ norm. Via (pushforward) coverability, this will ensure that $\mathbb{E}^\pi \left[ (\langle w_f, \psi(s,a) \rangle - \mathcal{T}f(s,a))^2 \right]$ is well-controlled for all policies $\pi$, which will be sufficient for our downstream sample-complexity analysis of GOLF. However, directly establishing an $L_2(\mu \otimes \mathtt{Unif}(\mathcal{A}))$ approximation guarantee is technically challenging since it would require establishing a fourth-order (rather than second-order) equivalent of Eq. (33). The remainder of the proof will obtain an $L_2(\mu \otimes \mathtt{Unif}(\mathcal{A}))$ approximation guarantee by instead sampling a dataset of size $n$ from $\mu \otimes \mathtt{Unif}(\mathcal{A})$ and taking a union bound over that dataset to ensure a uniform bound on all state-action pairs in that dataset. Via an additional concentration bound, this will ensure that the error is well-behaved under the $L_2(\mu \otimes \mathtt{Unif}(\mathcal{A}))$ norm.

For each $h \in [H]$, sample a dataset $D = \{(s_h^{(i)}, a_h^{(i)})\}_{i=1}^n$ i.i.d. from $\mu_h \otimes \mathtt{Unif}(\mathcal{A})$. By a union bound over $n$, $\mathcal{F}$, and $H$, we have that

$$\forall i \in [n], f \in \mathcal{F}, h \in [H]: \quad \left| \langle \psi_h(s_h^{(i)}, a_h^{(i)}), w_{f,h} \rangle - \mathcal{T}_h f_{h+1}(s_h^{(i)}, a_h^{(i)}) \right| \leq \varepsilon(n|\mathcal{F}|H\delta^{-1}), \quad (35)$$

where we recall the definition of $\varepsilon(\cdot)$ from Eq. (34). Now, let

$$X_{f,h}(s,a) := \left( \mathtt{clip}_{[0,2]}[\langle \psi_h(s,a), w_{f,h} \rangle] - \mathcal{T}_h f_{h+1}(s,a) \right)^2.$$

Note that $|X_{f,h}(s,a)| \leq 4$ and

$$X_{f,h}(s,a) \leq (\langle \psi_h(s,a), w_{f,h} \rangle - \mathcal{T}_h f_{h+1}(s,a))^2,$$

since $\mathcal{T}_h f_{h+1}(s,a) \in [0,2]$ and the clipping operator is 1-Lipshitz. Note that

$$\mathbb{E}_{(s,a) \sim \mu_h \otimes \mathtt{Unif}(\mathcal{A})}[X_{f,h}(s,a)] := \mathbb{E}_{\mu_h \otimes \mathtt{Unif}(\mathcal{A})}\Big[\big(\mathtt{clip}_{[0,2]}[\langle \psi_h(s,a), w_f \rangle] - \mathcal{T}_h f_{h+1}(s,a)\big)^2\Big],$$

where this expectation is only over the sampling of the data point $(s,a)$ (and not the Rademacher matrix $W$). Let

$$X_{i,f,h} := X_{f,h}(s_h^{(i)}, a_h^{(i)}).$$

By boundedness of $X_{f,h}(s,a)$ and Hoeffding's inequality, we have that with probability at least $1 - \delta$:

$$\left| \frac{1}{n} \sum_{i=1}^n X_{i,f,h} - \mathbb{E}_{\mu \otimes \mathtt{Unif}(\mathcal{A})}[X_{f,h}(s,a)] \right| \leq 4\sqrt{\frac{\log(2\delta^{-1})}{n}}.$$

Taking another union bound over $\mathcal{F}$ and $H$ as well as the event in Eq. (35) gives that

$$\forall f \in \mathcal{F}, h \in [H] : \quad \left| \frac{1}{n} \sum_{i=1}^n X_{i,f,h} - \mathbb{E}_{\mu \otimes \mathtt{Unif}(\mathcal{A})}[X_{f,h}(s,a)] \right| \leq 4\sqrt{\frac{\log(2|\mathcal{F}|H\delta^{-1})}{n}}, \tag{36}$$

and $\forall i \in [n], f \in \mathcal{F}, h \in [H] : \quad X_{i,f,h} \leq \varepsilon^2(n|\mathcal{F}|H\delta^{-1}),$ \tag{37}

recalling the definition of $\varepsilon(\cdot)$ from Eq. (34). Then, re-arranging Eq. (36) gives us that

$$\mathbb{E}_{\mu \otimes \mathtt{Unif}(\mathcal{A})}\Big[\big(\mathtt{clip}_{[0,2]}[\langle \psi_h(s_h, a_h), w_f \rangle] - \mathcal{T}_h f_{h+1}(s_h, a_h)\big)^2\Big]$$

$$\leq \frac{1}{n} \sum_{i=1}^n X_{i,f,h} + 4\sqrt{\frac{\log(2|\mathcal{F}|H\delta^{-1})}{n}}$$

$$\leq \varepsilon^2(n|\mathcal{F}|H\delta^{-1}) + 4\sqrt{\frac{\log(2|\mathcal{F}|H\delta^{-1})}{n}}, \tag{38}$$

We now conclude the proof by picking $n$ and $d$ appropriately to ensure that the right-hand-side is bounded by $\varepsilon_{\mathsf{apx}}$, which will ensure the desired claim that

$$\mathbb{E}_{\mu \otimes \mathtt{Unif}(\mathcal{A})}\Big[\big(\mathtt{clip}_{[0,2]}[\langle \psi_h(s_h, a_h), w_f \rangle] - \mathcal{T}_h f_{h+1}(s_h, a_h)\big)^2\Big] \leq \varepsilon_{\mathsf{apx}}.$$

For convenience, we introduce absolute constants $c$ and $c'$ whose precise values may change from line to line. We pick $n = 64 \log\big(2|\mathcal{F}|H\delta^{-1}\big)/\varepsilon_{\mathsf{apx}}^2$. Plugging this into (38) gives

$$\mathbb{E}_{\mu \otimes \mathtt{Unif}(\mathcal{A})}\Big[\big(\mathtt{clip}_{[0,2]}[\langle \psi_h(s_h, a_h), w_f \rangle] - \mathcal{T}_h f_{h+1}(s_h, a_h)\big)^2\Big] \leq \varepsilon^2(n|\mathcal{F}|H\delta^{-1}) + c \cdot \varepsilon \tag{39}$$

Noting that $n \leq 128 \frac{|\mathcal{F}|H\delta^{-1}}{\varepsilon_{\mathsf{apx}}^2}$ and plugging this into $\varepsilon$ (Eq. (34)) gives

$$\varepsilon(n|\mathcal{F}|H\delta^{-1}) \leq C_{\mathsf{push}}^{1/2} \sqrt{\frac{64 \log(16|\mathcal{F}|H\delta^{-1}/\varepsilon_{\mathsf{apx}})}{d}} + C_{\mathsf{push}} \frac{128 \log\big(16|\mathcal{F}|H\delta^{-1}/\varepsilon_{\mathsf{apx}}\big)}{d}. \tag{40}$$

Setting

$$d \geq 2^9 \frac{C_{\mathsf{push}} \log\big(16|\mathcal{F}|H\delta^{-1}/\varepsilon_{\mathsf{apx}}\big)}{\varepsilon_{\mathsf{apx}}}$$

ensures that

$$\varepsilon^2(n|\mathcal{F}|H\delta^{-1}) \leq \varepsilon(n|\mathcal{F}|H\delta^{-1}) \leq \frac{\varepsilon_{\mathsf{apx}}}{2} \tag{41}$$

by Eq. (40). Combining Eq. (38) and Eq. (41), we get

$$\mathbb{E}_{\mu \otimes \mathtt{Unif}(\mathcal{A})}\Big[\big(\mathtt{clip}_{[0,2]}[\langle \psi_h(s_h, a_h), w_f \rangle] - \mathcal{T}_h f_{h+1}(s_h, a_h)\big)^2\Big] \leq \varepsilon_{\mathsf{apx}}, \tag{42}$$

as desired. It only remains to establish the concentration results of Lemma F.2. $\qquad\square$

**Proof of Lemma F.2.** We establish the first claim. Let $i \in [d]$ be fixed, and consider the random variable

$$Z_i := \sum_{j \in [n]} \sum_{\substack{k \in [n] \\ k \neq j}} W_{i,j} W_{i,k} v_j u_k.$$

Note that $\mathbb{E}[Z_i] = 0$ by independence of $W_{i,j}$ and $W_{i,k}$ for every $j \neq k$. By Exercise 6.9 of Boucheron et al. [BLM13], we have that

$$\log \mathbb{E}[\exp(\lambda Z_i)] \leq \frac{16\lambda^2}{2(1 - 64\|u\|_2^2\|v\|_2^2\lambda)} \|u\|_2^2 \|v\|_2^2.$$

Since $Z_i$ are independent, it follows that

$$\log \mathbb{E}\left[\exp\left(\lambda \sum_{i=1}^d Z_i\right)\right] \leq \frac{16\lambda^2}{2(1 - 64\|u\|_2^2\|v\|_2^2\lambda)} \|u\|_2^2 \|v\|_2^2 d.$$

Hence, $\sum_{i=1}^d Z_i$ is a sub-Gamma random variable with parameters $\nu = 16\|u\|_2^2\|v\|_2^2 d$ and $c = 64\|u\|_2^2\|v\|_2^2$, and it follows from Equation (2.5) on page 29 of Boucheron et al. [BLM13] that for all $\varepsilon > 0$,

$$\mathbb{P}\left(\sum_{i=1}^d Z_i \geq \|u\|_2 \|v\|_2 \sqrt{32d\varepsilon} + 64\|u\|_2^2\|v\|_2^2\varepsilon\right) \leq e^{-\varepsilon}.$$

Taking a union bound, and using that the random variable is symmetric, we obtain the desired claim.

We now establish the second claim. Let $\mathcal{V} \subset \mathbb{R}^n$ be a subset of vectors. Let $i \in [d]$ be fixed, and re-consider the random variable

$$Z_i := \max_{v \in \mathcal{V}} \sum_{j \in [n]} \sum_{\substack{k \in [n] \\ k \neq j}} W_{i,j} W_{i,k} v_j v_k.$$

Again appealing to Exercise 6.9 of Boucheron et al. [BLM13], we have that

$$\log \mathbb{E}[\exp(\lambda(Z_i - \mathbb{E}[Z_i]))] \leq \frac{16\lambda^2}{2(1 - 64B\lambda)} \mathbb{E}\left[\max_{v \in \mathcal{V}} \sum_{j \in [n]} \sum_{\substack{k \in [n] \\ k \neq j}} W_{i,j} W_{i,k} v_j^2 v_k^2\right]$$

$$\leq \frac{16\lambda^2}{2(1 - 64B\lambda)} \mathbb{E}\left[\max_{v \in \mathcal{V}} \sum_{j,k=1}^n v_j^2 v_k^2\right]$$

$$= \frac{16\lambda^2}{2(1 - 64B\lambda)} \max_{v \in \mathcal{V}}\|v\|_2^4$$

where $B := \max_{v \in \mathcal{V}}\|v\|_2^4$. Since $Z_i$ are independent, it follows that

$$\log \mathbb{E}\left[\exp\left(\lambda \sum_{i=1}^d (Z_i - \mathbb{E}[Z_i])\right)\right] \leq \frac{16\lambda^2}{2(1 - 64B\lambda)} \max_{v \in \mathcal{V}}\|v\|_2^4 d.$$

Hence, $\sum_{i=1}^d Z_i$ is a sub-Gamma random variable with parameters $\nu = 16 \max_{v \in \mathcal{V}}\|v\|_2^4 d$ and $c = 64 \max_{v \in \mathcal{V}}\|v\|_2^4$, and it follows from Equation (2.5) on page 29 of Boucheron et al. [BLM13] that for all $\varepsilon > 0$,

$$\mathbb{P}\left(\frac{1}{d}\sum_{i=1}^d Z_i \geq \mathbb{E}[Z_i] + \max_{v \in \mathcal{V}}\|v\|_4^2 \sqrt{\frac{32\varepsilon}{d}} + 64 \max_{v \in \mathcal{V}}\|v\|_2^4 \frac{\varepsilon}{d}\right) \leq e^{-\varepsilon}.$$

To conclude, it remains only to show the bound $\mathbb{E}[Z_i] \leq \max_v \|v\|_2^2 (16 \log|\mathcal{V}| + 9)$. This follows by a standard log-sum-exp approach. Below, we abbreviate $\rho_j := W_{i,j}$. We can observe that for any $\lambda > 0$:

$$
\mathbb{E}[Z_i] = \mathbb{E}\left[ \max_{v \in \mathcal{V}} \sum_{j \in [n]} \sum_{\substack{k \in [n] \\ k \neq j}} \rho_j \rho_k v_j v_k \right]
$$

$$
\leq \frac{1}{\lambda} \log\left( \sum_{v \in \mathcal{V}} \mathbb{E}\left[ \exp\left( \lambda \sum_{j \in [n]} \sum_{\substack{k \in [n] \\ k \neq j}} \rho_j \rho_k v_j v_k \right) \right] \right)
$$

$$
\leq \frac{1}{\lambda} \log\left( \sum_{v \in \mathcal{V}} \mathbb{E}\left[ \exp\left( \lambda \left( \sum_{j=1}^n \rho_j v_j \right)^2 \right) \right] \right) \tag{43}
$$

Note that $X := \sum_j \rho_j v_j$ is subGaussian with parameter $\|v\|_2^2$, since:

$$
\mathbb{E}\left[ \exp\left( \lambda \sum_{j=1}^n \rho_j v_j \right) \right] = \prod_{j=1}^n \mathbb{E}[\exp(\lambda \rho_j v_j)] \leq \prod_{j=1}^n \exp\left( \frac{\lambda^2 v_j^2}{2} \right) = \exp\left( \frac{\lambda^2}{2} \|v\|_2^2 \right).
$$

Then, it follows (e.g. Lemma 1.12 of Rigollet et al. [RH23]) that $X^2 - \mathbb{E}[X^2]$ satisfies a sub-exponential MGF bound with parameter $16\|v\|_2^2$, i.e.

$$
\mathbb{E}[\exp(\lambda(X^2 - \mathbb{E}[X^2]))] \leq \exp\left( \frac{256}{2} \lambda^2 \|v\|_2^4 \right) \qquad \forall |\lambda| \leq \frac{1}{16\|v\|_2^2}.
$$

We also note that

$$
\mathbb{E}[X^2] = \sum_{i,j=1}^n v_i v_j \, \mathbb{E}[\varepsilon_i \varepsilon_j] = \|v\|_2^2.
$$

Adding and subtracting $\mathbb{E}[X^2]$ in Eq. (43) gives

$$
\leq \frac{1}{\lambda} \log\left( \sum_{v \in \mathcal{V}} \mathbb{E}\left[ \exp\left( \lambda \left( X^2 - \|v\|_2^2 \right) + \lambda \|v\|_2^2 \right) \right] \right)
$$

$$
= \frac{1}{\lambda} \log\left( \sum_{v \in \mathcal{V}} \mathbb{E}\left[ \exp\left( \lambda \left( X^2 - \|v\|_2^2 \right) \right) \right] \exp\left( \lambda \|v\|_2^2 \right) \right)
$$

$$
\leq \frac{1}{\lambda} \log\left( \sum_{v \in \mathcal{V}} \exp\left( 128 \lambda^2 \|v\|_2^4 + \lambda \|v\|_2^2 \right) \right) \qquad \forall |\lambda| \leq \frac{1}{16 \max_v \|v\|_2^2}
$$

$$
\leq \frac{1}{\lambda} \log|\mathcal{V}| + \max_v 128 \lambda \|v\|_2^4 + \max_v \|v\|_2^2 \qquad \forall |\lambda| \leq \frac{1}{16 \max_v \|v\|_2^2}
$$

Picking $\lambda = \frac{1}{16 \max_v \|v\|_2^2}$ concludes the proof. $\qquad \square$

### F.2.2 GOLF with on-policy misspecification

Consider the version of GOLF [JLM21] in Algorithm 2. We have the following guarantee for the regret of GOLF, which extends Jin et al. [JLM21] to allow for *on-policy* misspecification.

**Lemma F.3.** *Suppose that $Q^{M_{\mathsf{obs}}^\star, \star} \in \mathcal{F}$ and $\mathcal{G}$ satisfies $\varepsilon_{\mathsf{apx}}$-completeness in the sense that for all $h \in [H]$ and $f \in \mathcal{F}_{h+1}$, there exists $g \in \mathcal{G}_h$ such that $\mathbb{E}^\pi \left( g - \mathcal{T}_h^{M_{\mathsf{obs}}^\star} f \right)^2 \leq \varepsilon_{\mathsf{apx}}^2$ for all $\pi \in \Pi_{\mathcal{F}} := \{\pi_f : f \in \mathcal{F}\}$. Let $C_{\mathsf{cov}} := C_{\mathsf{cov}}(M_{\mathsf{obs}}^\star, \Pi_{\mathcal{F}})$ (Definition D.3). Then for an appropriate choice of $\beta$, Algorithm 2 ensures that*

$$
\mathsf{Reg} \leq H \sqrt{C_{\mathsf{cov}} T \log(|\mathcal{F}||\mathcal{G}|HT/\delta)} + HT \sqrt{C_{\mathsf{cov}} \log(T)} \varepsilon_{\mathsf{apx}}.
$$

**Algorithm 2** GOLF [JLM21]

---

**input:** Function classes $\mathcal{F}$ and $\mathcal{G}$, confidence width $\beta > 0$.
**initialize:** $\mathcal{F}^{(0)} \leftarrow \mathcal{F}$, $\mathcal{D}_h^{(0)} \leftarrow \emptyset$ $\forall h \in [H]$.

1: **for** episode $t = 1, 2, \ldots, T$ **do**
2:      Select policy $\pi^{(t)} \leftarrow \pi_{f^{(t)}}$, where $f^{(t)} := \arg\max_{f \in \mathcal{F}^{(t-1)}} f(x_1, \pi_{f,1}(x_1))$.
3:      Execute $\pi^{(t)}$ for one episode and obtain trajectory $(x_1^{(t)}, a_1^{(t)}, r_1^{(t)}), \ldots, (x_H^{(t)}, a_H^{(t)}, r_H^{(t)})$.
4:      Update dataset: $\mathcal{D}_h^{(t)} \leftarrow \mathcal{D}_h^{(t-1)} \cup \left\{ \left( x_h^{(t)}, a_h^{(t)}, x_{h+1}^{(t)} \right) \right\}$ $\forall h \in [H]$.
5:      Compute confidence set:

$$\mathcal{F}^{(t)} \leftarrow \left\{ f \in \mathcal{F} : \mathcal{L}_h^{(t)}(f_h, f_{h+1}) - \min_{g_h \in \mathcal{G}_h} \mathcal{L}_h^{(t)}(g_h, f_{h+1}) \leq \beta \;\; \forall h \in [H] \right\},$$

$$\text{where} \quad \mathcal{L}_h^{(t)}(f, f') := \sum_{(x,a,r,x') \in \mathcal{D}_h^{(t)}} \left( f(x,a) - r - \max_{a' \in \mathcal{A}} f'(x', a') \right)^2, \; \forall f, f' \in \mathcal{F}.$$

6: **end for**
7: Output $\widehat{\pi} = \text{Unif}(\pi^{(1:T)})$.

---

**Proof of Lemma F.3.** For each $f_{h+1} \in \mathcal{F}_{h+1}$, let $\text{apx}[f_h] = \arg\min_{g_h \in \mathcal{G}_h} \sup_{\pi \in \Pi} \mathbb{E}^\pi \left[ (g_h - \mathcal{T}_h f_{h+1})^2 \right]$. Let

$$\delta_h^{(t)}(\cdot, \cdot) := f_h^{(t)}(\cdot, \cdot) - \mathcal{T}_f f_{h+1}^{(t)}(\cdot, \cdot) \quad \& \quad \widetilde{\delta}_h^{(t)}(\cdot, \cdot) := f_h^{(t)}(\cdot, \cdot) - \text{apx}\big[f_{h+1}^{(t)}\big](\cdot, \cdot),$$

and note that by Jensen's inequality we have that for all $\pi$, $\mathbb{E}^\pi\big[\delta_h^{(t)}(\cdot, \cdot)\big] \leq \mathbb{E}^\pi\big[\widetilde{\delta}_h^{(t)}(\cdot, \cdot)\big] + \varepsilon_{\text{apx}}$.
We further adopt the shorthand $d_h^{(t)}(x, a) := d_h^{\pi^{(t)}}(x, a)$ and $\tilde{d}_h^{(t)}(x, a) := \sum_{i < t} d_h^{(t)}(x, a)$. As a consequence of realizability ($Q_{\text{obs},h}^\star \in \mathcal{F}_h$) and approximate Bellman completeness, standard concentration arguments (proved in the sequel) lead to the following result.

**Lemma F.4** (Optimism and small in-sample squared Bellman errors)**.** *With probability at least $1 - \delta$, by taking $\beta = c \log(TH|\mathcal{F}||\mathcal{G}|/\delta) + T\varepsilon_{\text{apx}}$, we have that for all $t \in [T]$,*

$$(i) \;\; Q_{\text{obs},h}^\star \in \mathcal{F}^{(t)}, \quad and \quad (ii) \;\; \sum_{x,a} \tilde{d}_h^{(t)}(x, a) \left( \widetilde{\delta}_h^{(t)}(x, a) \right)^2 \leq \mathcal{O}(\beta).$$

The rest of the proof proceeds similarly to the analysis of Section 3.2 in Xie et al. [XFBJK23]. Namely, by optimism (Lemma F.4) and a standard Bellman error decomposition (Lemma C.6) we have

$$\text{Reg} \leq \sum_{t=1}^{T} \sum_{h=1}^{H} \mathbb{E}_{d_h^{(t)}} \big[ \delta_h^{(t)}(x, a) \big] \leq TH \cdot \varepsilon_{\text{apx}} + \sum_{t=1}^{T} \sum_{h=1}^{H} \mathbb{E}_{d_h^{(t)}} \big[ \widetilde{\delta}_h^{(t)}(x, a) \big].$$

Let us defining the burn-in time

$$\tau_h(x, a) = \min\{t \mid \tilde{d}_h^{(t)}(x, a) \geq C_{\text{cov}} \mu_h^\star(x, a)\},$$

where $\mu_h^\star$ is the coverability distribution for the set of policies $\Pi_{\mathcal{F}}$ (i.e., the distribution $\mu_h^\star$ that achieves the minimum in the coverability definition). Using the same decomposition into "burn-in phase" and "stable phase" in Xie et al. [XFBJK23], we have:

$$\sum_{t=1}^{T} \sum_{h=1}^{H} \mathbb{E}_{d_h^{(t)}} \big[ \widetilde{\delta}_h^{(t)}(x, a) \big] \leq 2HC_{\text{cov}} + \sum_{t=1}^{T} \sum_{h=1}^{H} \mathbb{E}_{d_h^{(t)}} \big[ \widetilde{\delta}_h^{(t)}(x, a) \mathbb{I}\{t \geq \tau_h(x, a)\} \big].$$

Applying a change of measure argument on the second term then gives:

$$\sum_{t=1}^{T}\sum_{h=1}^{H}\mathbb{E}_{d_h^{(t)}}\left[\widetilde{\delta}_h^{(t)}(x,a)\mathbb{I}\{t\geq\tau_h(x,a)\}\right]\leq H\underbrace{\sqrt{\sum_{t=1}^{T}\sum_{x,a}\frac{\left(\mathbb{I}\{t\geq\tau_h(x,a)\}d_h^{(t)}(x,a)\right)^2}{\tilde{d}_h^{(t)}(x,a)}}}_{\text{(A)}}$$

$$\times\underbrace{\sqrt{\sum_{t=1}^{T}\sum_{x,a}\tilde{d}_h^{(t)}(x,a)\left(\widetilde{\delta}_h^{(t)}(x,a)\right)^2}}_{\text{(B)}}$$

By the same reasoning as in Xie et al. [XFBJK23], we have (A) $\leq \mathcal{O}(\sqrt{C_{\mathsf{cov}}\log(T)})$, and by Lemma F.4 we have (B) $\leq \mathcal{O}(\sqrt{\beta T})$. Using that $\beta = \log(TH|\mathcal{F}|/\delta) + T\varepsilon_{\mathsf{apx}}^2$ gives the desired result. It remains to establish the concentration results of Lemma F.4. $\qquad\square$

**Proof of Lemma F.4.** For any function $f$, define a random variable

$$X_t(h,f)=\left(f_h(s_h^{(t)},a_h^{(t)})-r_h^{(t)}-f_{h+1}(s_{h+1}^{(t)})\right)^2-\left(\mathcal{T}_hf_{h+1}(s_h^{(t)},a_h^{(t)})-r_h^{(t)}-f_{h+1}(s_{h+1}^{(t)})\right)^2.$$

Let $\mathfrak{F}_{t,h}=\{s_1^{(i)},a_1^{(i)},r_1^{(i)},\ldots,s_H^{(i)},a_H^{(i)},r_H^{(i)}\}_{i<t}$. Note that

$$\mathbb{E}\left[r_h^{(t)}+f_{h+1}(s_{h+1}^{(t)})\mid\mathfrak{F}_{t,h}\right]=\mathbb{E}^{\pi^{(t)}}[\mathcal{T}_hf(s_h,a_h)]. \tag{44}$$

and thus that

$$\mathbb{E}[X_t(h,f)\mid\mathfrak{F}_{t,h}]=\mathbb{E}^{\pi^{(t)}}\left[(f_h(s_h,a_h)-\mathcal{T}_hf_h(s_h,a_h))^2\right].$$

Next, note that

$$\mathsf{Var}[X_t(h,f)\mid\mathfrak{F}_{t,h}]\leq\mathbb{E}\left[(X_t(h,f))^2\mid\mathfrak{F}_{t,h}\right]$$

$$\leq\mathbb{E}\left[\left(f_h(s_h^{(t)},a_h^{(t)})-\mathcal{T}_hf_h(s_h^{(t)},a_h^{(t)})\right)^2\left(f_h(s_h^{(t)},a_h^{(t)})+\mathcal{T}_hf_h(s_h^{(t)},a_h^{(t)})+2\left(r_h^{(t)}-f_{h+1}(s_{h+1}^{(t)})\right)\right)^2\mid\mathfrak{F}_{t,h}\right]$$

$$\leq16\,\mathbb{E}\left[\left(f_h(s_h^{(t)},a_h^{(t)})-\mathcal{T}_hf_h(s_h^{(t)},a_h^{(t)})\right)^2\mid\mathfrak{F}_{t,h}\right]=16\,\mathbb{E}[X_t(h,f)\mid\mathfrak{F}_{t,h}].$$

By Freedman's inequality (Lemma C.2, Lemma C.3), we have that with probability at least $1-\delta$:

$$\left|\sum_{i<t}X_i(h,f)-\sum_{i<t}\mathbb{E}[X_i(h,f)\mid\mathfrak{F}_{i,h}]\right|\leq\mathcal{O}\left(\sqrt{\log(1/\delta)\sum_{i<t}\mathbb{E}[X_i(h,f)\mid\mathfrak{F}_{i,h}]}+\log(1/\delta)\right)$$

Taking a union bound over $[T]\times[H]\times\mathcal{F}$, we have that for all $t,h,f$, with probability at least $1-\delta$:

$$\left|\sum_{i<t}X_i(h,f)-\sum_{i<t}\mathbb{E}^{\pi^{(i)}}\left[(f_h(s_h,a_h)-\mathcal{T}_hf_h(s_h,a_h))^2\right]\right| \tag{45}$$

$$\leq\mathcal{O}\left(\sqrt{\iota\sum_{i<t}\mathbb{E}^{\pi^{(i)}}\left[(f_h(s_h,a_h)-\mathcal{T}_hf_h(s_h,a_h))^2\right]}+\iota\right), \tag{46}$$

where $\iota=\log(|\mathcal{F}|HT/\delta)$. We now show that

$$\sum_{i<t}X_i(h,f^{(t)})\leq\beta+\mathcal{O}\left(T\varepsilon_{\mathsf{apx}}^2+\iota\right)=\mathcal{O}(\beta), \tag{47}$$

which will imply, from Eq. (46), that

$$\sum_{i<t}\mathbb{E}^{\pi^{(t)}}\left[(f_h(s_h,a_h)-\mathcal{T}_hf_h(s_h,a_h))^2\right]\leq\mathcal{O}(\iota+\beta)=\mathcal{O}(\beta),$$

as desired. To see Eq. (47), let

$$\Delta_t=\sum_{i<t}\left(\mathsf{apx}[\mathcal{T}_hf_{h+1}^{(t)}](s_h^{(i)},a_h^{(i)})-r_h^{(i)}-f_{h+1}^{(t)}(s_{h+1}^{(i)})\right)^2-\left(\mathcal{T}_hf_h^{(t)}(s_h^{(i)},a_h^{(i)})-r_h^{(i)}-f_{h+1}^{(t)}(s_{h+1}^{(i)})\right)^2$$

and then note that:

$$\sum_{i<t} X_i(h, f^{(t)}) = \sum_{i<t}\big(f_h^{(t)}(s_h^{(i)}, a_h^{(i)}) - r_h^{(i)} - f_{h+1}^{(t)}(s_{h+1}^{(i)})\big)^2 - \big(\mathcal{T}_h f_h^{(t)}(s_h^{(i)}, a_h^{(i)}) - r_h^{(i)} - f_{h+1}^{(t)}(s_{h+1}^{(i)})\big)^2$$

$$= \sum_{i<t}\big(f_h^{(t)}(s_h^{(i)}, a_h^{(i)}) - r_h^{(i)} - f_{h+1}^{(i)}(s_{h+1}^{(i)})\big)^2$$

$$- \sum_{i<t}\big(\mathsf{apx}\big[\mathcal{T}_h f_{h+1}^{(t)}\big](s_h^{(i)}, a_h^{(i)}) - r_h^{(i)} - f_{h+1}^{(t)}(s_{h+1}^{(i)})\big)^2 + \Delta_t$$

$$\leq \sum_{i<t}\big(f_h^{(t)}(s_h^{(i)}, a_h^{(i)}) - r_h^{(i)} - f_{h+1}^{(i)}(s_{h+1}^{(i)})\big)^2$$

$$- \inf_{g_h \in \mathcal{G}_h}\sum_{i<t}\big(g(s_h^{(i)}, a_h^{(i)}) - r_h^{(i)} - f_{h+1}^{(t)}(s_{h+1}^{(i)})\big)^2 + \Delta_t$$

$$\leq \beta + \Delta_t.$$

where the second-to-last line follows from $\mathsf{apx}\big[\mathcal{T}_h f_{h+1}^{(t)}\big] \in \mathcal{G}$ and the last line follows from the definition of the confidence set. It remains to show that $\Delta_t \leq \mathcal{O}(T\varepsilon_{\mathsf{apx}}^2 + \iota)$, which we do via a similar concentration argument. Namely, let

$$Y_t(h, f) = \big(\mathsf{apx}[\mathcal{T}_h f_{h+1}](s_h^{(t)}, a_h^{(t)}) - r_h^{(t)} - f_{h+1}^{(k)}(s_{h+1}^{(t)})\big)^2 - \big(\mathcal{T}_h f_h(s_h^{(t)}, a_h^{(t)}) - r_h^{(t)} - f_{h+1}^{(k)}(s_{h+1}^{(t)})\big)^2,$$

and note that, as before,

$$\mathbb{E}[Y_t(h, f) \mid \mathfrak{F}_{t,h}] = \mathbb{E}^{\pi^{(t)}}\Big[\big(\mathsf{apx}[\mathcal{T}_h f_{h+1}](s_h, a_h) - \mathcal{T}_h f_h(s_h, a_h)\big)^2\Big],$$

and

$$\mathsf{Var}[Y_t(h, f) \mid \mathfrak{F}_{t,h}] \leq 16\,\mathbb{E}[Y_t(h, f) \mid \mathfrak{F}_{t,h}],$$

by the same calculation as earlier. Thus, by Freedman's inequality and a union bound, we have that, with probability at least $1 - \delta$,

$$\left|\sum_{i<t} Y_t(h, f) - \sum_{i<t}\mathbb{E}^{\pi^{(t)}}\Big[\big(\mathsf{apx}[\mathcal{T}_h f_{h+1}](s_h, a_h) - \mathcal{T}_h f_h(s_h, a_h)\big)^2\Big]\right| \tag{48}$$

$$\leq \mathcal{O}\left(\sqrt{\iota \sum_{i<t}\mathbb{E}^{\pi^{(t)}}\Big[\big(\mathsf{apx}[\mathcal{T}_h f_{h+1}](s_h, a_h) - \mathcal{T}_h f_h(s_h, a_h)\big)^2\Big]} + \iota\right), \tag{49}$$

where $\iota = \log(|\mathcal{F}|HT/\delta)$. Recalling the misspecification assumption, this implies that

$$\sum_{i<t} Y_t(h, f) \leq \mathcal{O}\big(t\varepsilon_{\mathsf{apx}}^2 + \iota\big),$$

for all $h, f, t$, with high probability. This concludes the result for $(ii)$. For $(i)$, this follows identically to the proof of Lemma 40 in Jin et al. [JLM21], since this only uses the property that $Q^\star \in \mathcal{F}$.

$\square$

### F.2.3 Sample-efficient latent-dynamics RL under pushforward coverability

We conclude by combining the previous two results to obtain the main result for this section.

**Theorem 3.2** (Pushforward-coverable MDPs are statistically modular). *Let $\mathcal{M}_{\mathsf{lat}}$ be a base MDP class such that each $M_{\mathsf{lat}} \in \mathcal{M}_{\mathsf{lat}}$ has pushforward coverability bounded by $C_{\mathsf{push}}(M_{\mathsf{lat}}) \leq C_{\mathsf{push}}$. Then, for any decoder class $\Phi$, we have:*

1. $\mathsf{comp}(\mathcal{M}_{\mathsf{lat}}, \varepsilon, \delta) \leq \mathsf{poly}(C_{\mathsf{push}}, |\mathcal{A}|, H, \log|\mathcal{M}_{\mathsf{lat}}|, \varepsilon^{-1}, \log(\delta^{-1}))$, *and*

2. $\mathsf{comp}(\langle\!\langle\mathcal{M}_{\mathsf{lat}}, \Phi\rangle\!\rangle, \varepsilon, \delta) \leq \mathsf{poly}(C_{\mathsf{push}}, |\mathcal{A}|, H, \log|\mathcal{M}_{\mathsf{lat}}|, \log|\Phi|, \varepsilon^{-1}, \log(\delta^{-1}), \log\log|\mathcal{S}|)$.

**Proof of Theorem 3.2.** Let $M_{\mathsf{obs}}^\star := \langle\!\langle M_{\mathsf{lat}}^\star, \psi^\star\rangle\!\rangle \in \langle\!\langle\mathcal{M}_{\mathsf{lat}}, \Phi\rangle\!\rangle$ be the unknown latent-dynamics MDP. Define observation-level value functions

$$\mathcal{F} = \{Q^{M_{\mathsf{lat}},\star} \circ \phi \mid M_{\mathsf{lat}} \in \mathcal{M}_{\mathsf{lat}}, \phi \in \Phi\},$$

so that $Q^{M_{\mathrm{obs}}^\star,\star} = Q^{M_{\mathrm{lat}}^\star,\star} \circ \phi^\star \in \mathcal{F}$ via decoder and model realizability, and $\log|\mathcal{F}_h| \le \log|\mathcal{M}_{\mathrm{lat}}||\Phi|$. Consider any function class $\mathcal{L} \subseteq \{\mathcal{S} \to [0,1]\}$ and MDP $M_{\mathrm{lat}} = (r_{\mathrm{lat}}, P_{\mathrm{lat}})$. For a given value $\varepsilon_{\mathrm{apx}} > 0$, setting $d$ according to Lemma F.1 implies that there exists a $d$-dimensional feature map $\varphi_{M_{\mathrm{lat}},h}(s,a) \in \mathbb{R}^{d+1}$ such that for all $\ell \in \mathcal{L}$ and $h \in [H]$, there exists $w_{\ell,h} \in \mathbb{R}^{d+1}$ such that

$$\mathbb{E}_{\mu_{M_{\mathrm{lat}}} \otimes \mathrm{Unif}(\mathcal{A})}\left[\left(\mathrm{clip}_{[0,2]}\left[\langle \varphi_{M_{\mathrm{lat}},h}(s,a), w_{\ell,h}\rangle\right] - \mathcal{T}_h^{M_{\mathrm{lat}}}\ell_{h+1}(s,a)\right)^2\right] \le \varepsilon_{\mathrm{apx}}, \qquad (50)$$

where $\mu_{M_{\mathrm{lat}}}$ is the pushforward coverability distribution for $M_{\mathrm{lat}}$. Moreover, the map $\varphi_h$ is explicitly computed as a function of $M_{\mathrm{lat}}$ by a randomized algorithm with success probability $1 - \delta$, with no knowledge of the class $\mathcal{L}$ required. We consider the class

$$\mathcal{L} = \left\{\Gamma_\phi \circ Q^{M_{\mathrm{lat}},\star}(s,a) := \sum_{s' \in \mathcal{S}} \Gamma_\phi(s' \mid s)Q^{M_{\mathrm{lat}},\star}(s',a) \mid \phi \in \Phi, M_{\mathrm{lat}} \in \mathcal{M}_{\mathrm{lat}}\right\}, \qquad (51)$$

where $\Gamma_\phi : \mathcal{S} \to \Delta(\mathcal{S})$ is the mismatch function for decoder $\phi$ and emission $\psi^\star$, defined in Definition F.1. Note that $\mathcal{L}$ has size $\log|\mathcal{L}| \le \log|\mathcal{M}_{\mathrm{lat}}||\Phi|$, and that we have

$$\mathcal{T}_h^{M_{\mathrm{obs}}^\star}(Q_h^{M_{\mathrm{lat}},\star} \circ \phi_h)(x,a) = \mathcal{T}_h^{M_{\mathrm{lat}}^\star}(\Gamma_{\phi,h+1} \circ V_h^{M_{\mathrm{lat}},\star})(\phi_h^\star(x),a)$$

by Lemma D.7. By Lemma D.1 we have that $\mu_{M_{\mathrm{obs}}^\star,h}(x) = \psi_h^\star(x \mid \phi_h^\star(x))\mu_{M_{\mathrm{lat}}^\star,h}(\phi_h^\star(x))$ is the coverability distribution for MDP $M_{\mathrm{obs}}^\star$, and

$$\mathbb{E}_{\mu_{M_{\mathrm{lat}}^\star} \otimes \mathrm{Unif}(\mathcal{A})}[f(s,a)] = \mathbb{E}_{\mu_{M_{\mathrm{obs}}^\star} \otimes \mathrm{Unif}(\mathcal{A})}[f(\phi^\star(x),a)].$$

Now, define

$$\mathcal{G}_{M_{\mathrm{lat}},h} = \left\{(x,a) \mapsto \mathrm{clip}_{[0,2]}[\langle \varphi_{M_{\mathrm{lat}},h}(\phi(x),a), w\rangle] \mid \phi \in \Phi, \|w\|_2^2 \le 11 + 16\log(|\mathcal{M}_{\mathrm{lat}}||\Phi|H)\right\}.$$

Recall the definition of $w_f$ (for $f : \mathcal{S} \times \mathcal{A} \to [0,1]$) from Lemma F.1, and note that by the norm bound $\max_{\ell \in \mathcal{L}}\|w_\ell\|_2^2 \le 11 + 16\log(|\mathcal{M}_{\mathrm{lat}}||\Phi|H)$ given by Lemma F.1, we have $(x,a) \mapsto \langle \varphi_{M_{\mathrm{lat}},h}(\phi(x),a), w_\ell\rangle \in \mathcal{G}_h$ for every $\ell \in \mathcal{L}$. Next, note that by the norm bound $\max_{s,a}\|\psi(s,a)\|_2^2 \le C_{\mathrm{push}}(11 + 16\log(|\mathcal{S}||\mathcal{A}|H))$, given by Lemma F.1, we have every $g_h \in \mathcal{G}_{M_{\mathrm{lat}},h}$ satisfies $\|g_h\|_\infty \le cC_{\mathrm{push}}^{1/2}\log(|\mathcal{M}_{\mathrm{lat}}||\Phi||\mathcal{S}||\mathcal{A}|H) := B$ for some absolute constant $c$. Therefore, $\mathcal{G}_{M_{\mathrm{lat}},h}$ has size $\log|\mathcal{G}_{M_{\mathrm{lat}},h}| \le \widetilde{O}(d \cdot \log(B) + \log|\Phi|) = \widetilde{O}(d\log\log(|\mathcal{S}|) + \log|\Phi|)$, where the $\widetilde{O}$ notation ignores logarithmic factors of $C_{\mathrm{push}}, |\mathcal{A}|, \log|\mathcal{M}_{\mathrm{lat}}|$, and $\log|\Phi|$.[22] Define $\mathcal{G}_h = \cup_{M_{\mathrm{lat}} \in \mathcal{M}_{\mathrm{lat}}}\mathcal{G}_{M_{\mathrm{lat}},h}$, which has size $\log|\mathcal{G}_h| \le \log|M_{\mathrm{lat}}| + (\widetilde{O}(d\log\log(|\mathcal{S}|) + \log|\Phi|))$. Together, these results with Lemma F.1 imply that for all $f_{h+1} \in \mathcal{F}_{h+1}$, there exists $g_h \in \mathcal{G}_h$ such that

$$\mathbb{E}_{\mu_{M_{\mathrm{obs}}^\star,h} \otimes \mathrm{Unif}(\mathcal{A})}\left[\left(g_h(x_h,a_h) - \left[\mathcal{T}_h^{M_{\mathrm{obs}}^\star}f_{h+1}\right](x_h,a_h)\right)^2\right] \le \varepsilon_{\mathrm{apx}}.$$

This, in turn, implies that for all $\pi_{\mathrm{obs}} \in \Pi_{\mathrm{rns}}$ we have

$$\mathbb{E}^{\pi_{\mathrm{obs}}}\left[\left(g_h(x_h,a_h) - \left[\mathcal{T}_h^{M_{\mathrm{obs}}^\star}f_{h+1}\right](x_h,a_h)\right)^2\right] \le C_{\mathrm{push}}|\mathcal{A}|\varepsilon_{\mathrm{apx}},$$

since $\mu_{M_{\mathrm{obs}}^\star,h} \otimes \mathrm{Unif}(\mathcal{A})$ satisfies coverability (Definition D.3) with parameter $C_{\mathrm{cov}}(M_{\mathrm{obs}}^\star, \Pi_{\mathrm{rns}}) \le C_{\mathrm{push}}|\mathcal{A}|$ (Eq. (29)).

Then, it follows by Lemma F.3 that if we run Algorithm 2 with the classes $\mathcal{F}$ and $\mathcal{G}$ we will get

$$\mathrm{Reg} \le H\sqrt{C_{\mathrm{push}}|\mathcal{A}|T\log(|\mathcal{M}_{\mathrm{lat}}||\Phi|HT/\delta)(d\log\log(|\mathcal{S}|) + \log|\Phi|)} + HT\sqrt{C_{\mathrm{push}}^2|\mathcal{A}|^2\log(T)\varepsilon_{\mathrm{apx}}}$$

$$\le H\sqrt{C_{\mathrm{push}}^5|\mathcal{A}|T\log(|\mathcal{M}_{\mathrm{lat}}||\Phi|HT/\delta)\frac{\log(C_{\mathrm{push}}^2|\mathcal{M}_{\mathrm{lat}}||\Phi|^2H\delta^{-1}/\varepsilon_{\mathrm{apx}})\log\log(|\mathcal{S}|)}{\varepsilon_{\mathrm{apx}}}}$$

$$+ HT\sqrt{C_{\mathrm{push}}^2|\mathcal{A}|^2\log(T)\varepsilon_{\mathrm{apx}}}$$

---

[22]Formally, this requires a standard covering number argument; we omit the details.

Choosing $\varepsilon_{\mathsf{apx}} = \frac{1}{\sqrt{T}}$ to balance leads to

$$\mathtt{Reg} \lesssim HT^{3/4}\sqrt{C_{\mathsf{push}}^5|\mathcal{A}|\log(|\mathcal{M}_{\mathtt{lat}}||\Phi|HT/\delta)\log\!\big(C_{\mathsf{push}}^2|\mathcal{M}_{\mathtt{lat}}||\Phi|^2H\delta^{-1}T\big)\log\log(|\mathcal{S}|)}$$

$$+ HT^{3/4}\sqrt{C_{\mathsf{push}}^2|\mathcal{A}|^2\log(T)}$$

$$\lesssim HT^{3/4}\sqrt{C_{\mathsf{push}}^5|\mathcal{A}|^2\log(|\mathcal{M}_{\mathtt{lat}}||\Phi|HT/\delta)\log\!\big(TC_{\mathsf{push}}^2|\mathcal{M}_{\mathtt{lat}}||\Phi|^2H/\delta\big)\log\log(|\mathcal{S}|)},$$

which gives a risk bound of

$$\mathtt{Risk} \lesssim \frac{1}{T^{1/4}}H\sqrt{C_{\mathsf{push}}^5|\mathcal{A}|^2\log(|\mathcal{M}_{\mathtt{lat}}||\Phi|HT/\delta)\log\!\big(TC_{\mathsf{push}}^2|\mathcal{M}_{\mathtt{lat}}||\Phi|^2H/\delta\big)\log\log(|\mathcal{S}|)}.$$

Equating this to $\varepsilon$ gives a sample complexity of

$$T = \mathtt{poly}(C_{\mathsf{push}}, A, H, \log|\mathcal{M}_{\mathtt{lat}}|, \log|\Phi|, \varepsilon^{-1}, \log\!\big(\delta^{-1}\big), \log\log(|\mathcal{S}|)),$$

as desired. Note that we have not made much effort to optimize the rate; in particular, a faster rate is likely possible by using the GOLF.DBR algorithm of Amortila et al. [ACK24], which improves over the GOLF algorithm under the presence of misspecification. $\qquad\square$

# G   Proofs and Additional Information for Section 4.1: Hindsight RL

This appendix contains additional information and proofs related to algorithmic modularity under hindsight observations (Section 4.1), and is organized as follows:

- Appendix G.1 contains the pseudocode and proofs related to the online representation learning oracle EXPWEIGHTS.DR (Lemma 4.1).

- Appendix G.2 contains the proof for our risk bound of the O2L algorithm under hindsight observability (Theorem 4.1).

## G.1   Pseudocode and Proofs for EXPWEIGHTS.DR (Lemma 4.1)

---

**Algorithm 3** Derandomized Exponential Weights (EXPWEIGHTS.DR)

---

**input**: Decoder set $\Phi$
**for** $t = 1, 2, \cdots, T$ **do**
    Get dataset $\{x_h^{(i)}, \phi^\star(x_h^{(i)})\}_{i \in [t-1], h \in [H]}$
    **for** $h = 1, \ldots, H$ **do**
        For $\phi \in \Phi$, compute

$$q_h^{(t)}(\phi_h) \propto \exp\left( -\sum_{i=1}^{t-1} \mathbb{I}\big[\phi_h(x_h^{(i)}) \neq \phi_h^\star(x_h^{(i)})\big] \right),$$

    and set

$$\bar{\phi}_h^{(t)}(x) = \arg\max_{s \in \mathcal{S}} \mathbb{P}_{\phi_h \sim q_h^{(t)}}(\phi_h(x) = s). \tag{52}$$

    **end for**
    Return $\bar{\phi}^{(t)} = \{\bar{\phi}_h^{(t)}\}_{h=1}^H$.
**end for**

---

The main result for this estimator is the following.

**Lemma 4.1** (Online classification via EXPWEIGHTS.DR). *Under decoder realizability* ($\phi^\star \in \Phi$), *EXPWEIGHTS.DR (Algorithm 3) satisfies Assumption 4.2 with*[23]

$$\mathsf{Est}_{\mathsf{class}}(T) = \widetilde{\mathcal{O}}(H \log|\Phi|).$$

**Proof of Lemma 4.1.** For each $h \in [H]$, consider the realizable online classification problem where $x_h^{(t)} \sim d_h^{\pi^{(t)}}$, for $\pi^{(t)}$ chosen adversarially, and $y_h^{(t)} = \phi_h^\star(x_h^{(t)})$. Consider the exponential weights estimator

$$q_h^{(t)}(\phi) \propto \exp\left( -\sum_{i=1}^{t-1} \mathbb{I}\big[\phi(x_h^{(i)}) \neq \phi_h^\star(x_h^{(i)})\big] \right).$$

For every sequence $(x_h^{(t)})_{t=1}^T$, these distributions satisfy the deterministic regret bound

$$\sum_{t=1}^T \mathbb{E}_{\widehat{\phi}_h^{(t)} \sim q_h^{(t)}} \left[ \mathbb{I}\Big[\widehat{\phi}_h^{(t)}(x_h^{(t)}) \neq \phi_h^\star(x_h^{(t)})\Big] \right] \leq 2\log|\Phi|,$$

by Corollary 2.3 of Cesa-Bianchi et al. [CBL06]. Taking conditional expectations over $x_h^{(t)} \sim d_h^{\pi^{(t)}}$ and using Lemma C.3 gives that with probability at least $1 - \delta$:

$$\sum_{t=1}^T \mathbb{E}_{\widehat{\phi}_h^{(t)} \sim q_h^{(t)}} \mathbb{E}^{\pi^{(t)}} \left[ \mathbb{I}\Big[\widehat{\phi}_h^{(t)}(x_h) \neq \phi_h^\star(x_h)\Big] \right] \leq 4\log|\Phi| + 8\log\big(2\delta^{-1}\big).$$

Taking a union bound over $h \in [H]$ and summing over $h \in [H]$ we obtain that with probability at least $1 - \delta$:

$$\sum_{t=1}^T \sum_{h=1}^H \mathbb{E}_{\widehat{\phi}_h^{(t)} \sim q_h^{(t)}} \mathbb{E}^{\pi^{(t)}} \left[ \mathbb{I}\Big[\widehat{\phi}_h^{(t)}(x_h) \neq \phi_h^\star(x_h)\Big] \right] \leq 4H\log|\Phi| + 8H\log\big(2H\delta^{-1}\big).$$

---

[23]In this section, the notations $\widetilde{\mathcal{O}}, \approx$, and $\lesssim$ ignore only constants and logarithmic factors of $H$.

Now, recall that at each time $t$, we define the improper decoder $\bar{\phi}_h^{(t)}$ via:

$$\bar{\phi}_h^{(t)}(x) = \arg\max_{s \in \mathcal{S}} \mathbb{P}_{\phi_h^{(t)} \sim q_h^{(t)}}(\phi_h^{(t)}(x) = s) \tag{53}$$

Let $\ell_h(x_h, q_h^{(t)}) = \mathbb{P}_{\phi_h^{(t)} \sim q_h^{(t)}}(\phi_h^{(t)}(x_h) \neq \phi_h^\star(x_h))$. Note that $\ell$ satisfies

$$\sum_{t=1}^{T} \sum_{h=1}^{H} \mathbb{E}_{\phi_h^{(t)} \sim q_h^{(t)}} \mathbb{E}^{\pi^{(t)}} \left[\mathbb{I}[\phi_h^{(t)}(x_h) \neq \phi_h^\star(x_h)]\right] = \sum_{t=1}^{T} \sum_{h=1}^{H} \mathbb{E}^{\pi^{(t)}} \mathbb{E}_{\phi_h^{(t)} \sim q_h^{(t)}} \left[\mathbb{I}[\phi_h^{(t)}(x_h) \neq \phi_h^\star(x_h)]\right] \tag{54}$$

$$= \sum_{t=1}^{T} \sum_{h=1}^{H} \mathbb{E}^{\pi^{(t)}}[\ell_h(x_h, q_h^{(t)})]. \tag{55}$$

By abuse of notation we also denote $\ell_h(x_h, \bar{\phi}_h) = \mathbb{I}[\bar{\phi}_h(x) \neq \phi^\star(x)]$. We will show that

$$\forall x, t, h : \ell_h(x_h, \bar{\phi}_h^{(t)}) \leq 2\ell_h(x_h, q_h^{(t)}), \tag{56}$$

from which we will obtain that with probability at least $1 - \delta$:

$$\mathsf{Reg}_{\mathsf{class}}(T) = \sum_{t=1}^{T} \sum_{h=1}^{H} \mathbb{E}^{\pi^{(t)}} \left[\mathbb{I}[\bar{\phi}_h^{(t)}(x_h) \neq \phi_h^\star(x_h)]\right] \leq 8H \log|\Phi| + 16H \log\left(2H\delta^{-1}\right).$$

Integrating the high-probability regret bound gives

$$\mathbb{E}[\mathsf{Reg}_{\mathsf{class}}(T)] = \mathcal{O}(H \log(H|\Phi|)),$$

as desired. Towards establishing Eq. (56), let us fix $x$ and let $s_{\max}$ denote the argmax in Eq. (53). There are two cases:

- $\mathbb{P}_{\phi_h^{(t)} \sim q_h^{(t)}}(\phi_h^{(t)}(x) = s_{\max}) \geq \frac{1}{2}$:

$\rightarrow$ If $s_{\max} = \phi^\star(x)$, $\ell(x, \bar{\phi}_h^{(t)}) = 0$ so we are done.

$\rightarrow$ Otherwise, $s_{\max} \neq \phi^\star(x)$ and we have $\ell(x, \bar{\phi}_h^{(t)}) = 1$. However, since $\phi^\star(x) \neq s_{\max}$ we have $\phi_h^{(t)}(x) = s_{\max} \implies \phi_h^{(t)}(x) \neq \phi_h^\star(x)$ and so

$$\mathbb{P}_{\phi_h^{(t)} \sim q_h^{(t)}}(\phi_h^{(t)}(x) \neq \phi_h^\star(x)) \geq \mathbb{P}_{\phi_h^{(t)} \sim q_h^{(t)}}(\phi_h^{(t)}(x) = s_{\max}) \geq \frac{1}{2} = \frac{1}{2}\ell(x, \bar{\phi}_h^{(t)}).$$

- $\mathbb{P}_{\phi_h^{(t)} \sim q_h^{(t)}}(\phi_h^{(t)}(x) = s_{\max}) < \frac{1}{2}$:

$\rightarrow$ If $s_{\max} = \phi_h^\star(x)$, $\ell(x, \bar{\phi}_h^{(t)}) = 0$ so we are done.

$\rightarrow$ Otherwise, $s_{\max} \neq \phi^\star(x)$ and we have $\ell(x, \bar{\phi}_h^{(t)}) = 1$. However, by definition of $s_{\max}$ as the mode we also have

$$\mathbb{P}_{\phi_h^{(t)} \sim q_h^{(t)}}(\phi_h^{(t)}(x) = \phi_h^\star(x)) \leq \mathbb{P}_{\phi_h^{(t)} \sim q_h^{(t)}}(\phi_h^{(t)}(x) = s_{\max}) < \frac{1}{2},$$

so in particular we have

$$\ell(x, q_h^{(t)}) = \mathbb{P}_{\phi_h^{(t)} \sim q_h^{(t)}}(\phi_h^{(t)}(x) \neq \phi_h^\star(x)) > \frac{1}{2} = \frac{1}{2}\ell(x, \bar{\phi}_h^{(t)}).$$

$\square$

### G.2 Proofs for O2L Under Hindsight Observability (Theorem 4.1)

**Theorem 4.1** (Risk bound for O2L under hindsight observability). *Let* $\mathrm{ALG}_{\mathsf{lat}}$ *be a base algorithm with base risk* $\mathsf{Risk}_\star(K)$, *and* $\mathrm{REP}_{\mathsf{class}}$ *a representation learning oracle satisfying Assumption 4.2. Then Algorithm 1, with inputs* $T, K, \Phi$, $\mathrm{REP}_{\mathsf{class}}$, *and* $\mathrm{ALG}_{\mathsf{lat}}$, *has expected risk*

$$\mathbb{E}[\mathsf{Risk}_{\mathsf{obs}}(TK)] \leq \mathsf{Risk}_\star(K) + \frac{2K}{T}\mathsf{Est}_{\mathsf{class}}(T).$$

**Proof of Theorem 4.1.** Let $(\widehat{\phi}^{(t)})_{t\in[T]}$ denote the decoders chosen by $\mathrm{REP}_{\mathrm{class}}$, and let $\rho^{(t)}$ denote the distribution over decoders induced at time $t$ from the interaction of $\mathrm{REP}_{\mathrm{class}}$, $\mathrm{ALG}_{\mathrm{lat}}$, and $M^\star_{\mathrm{obs}}$. Let $\pi^{(t,k)}_{\mathrm{obs}} := \pi^{(t,k)}_{\mathrm{lat}} \circ \widehat{\phi}^{(t)}$ and $p^{(t,k)}_{\mathrm{obs}}$ denote the distribution over (observation-space) policies played at epoch $t$ and episode $k$, induced by the interaction of $\mathrm{REP}_{\mathrm{class}}$, $\mathrm{ALG}_{\mathrm{lat}}$, and $M^\star_{\mathrm{obs}}$. We adopt the notation $\pi^{(t,K+1)}_{\mathrm{lat}} := \widehat{\pi}^{(t)}_{\mathrm{lat}} \sim p^{(t,K+1)}_{\mathrm{lat}}$ for the final policy output by $\mathrm{ALG}_{\mathrm{lat}}$ in epoch $t$ and $(x^{(t,K+1)}_h, a^{(t,K+1)}_h, r^{(t,K+1)}_h)$ for the trajectory collected from that (observation-level) policy $\widehat{\pi}^{(t)}_{\mathrm{lat}} \circ \widehat{\phi}^{(t)}$. We firstly note that by assumption, we have the guarantee

$$\mathbb{E}\left[\sum_{t=1}^{T}\sum_{k=1}^{K+1}\sum_{h=1}^{H}\mathbb{E}_{\pi^{(t,k)}_{\mathrm{obs}}\sim p^{(t,k)}_{\mathrm{obs}}}\mathbb{E}^{\pi^{(t,k)}_{\mathrm{obs}}}\left[\mathbb{I}\left[\widehat{\phi}^{(t)}_h(x_h)\neq\phi^\star_h(x_h)\right]\right]\right] \leq (K+1)\mathsf{Est}_{\mathrm{class}}(T)$$

$$\leq 2K\mathsf{Est}_{\mathrm{class}}(T). \qquad (57)$$

which follows by applying Assumption 4.2 to the distributions $\bar{p}^{(t)}_{\mathrm{obs}} = \frac{1}{(K+1)}\sum_{k=1}^{K+1}p^{(t,k)}_{\mathrm{obs}}$ and noting that

$$\sum_{t=1}^{T}\sum_{h=1}^{H}\frac{1}{K+1}\sum_{k=1}^{K+1}\mathbb{E}_{\pi^{(t,k)}_{\mathrm{obs}}\sim p^{(t,k)}_{\mathrm{obs}}}\mathbb{E}^{\pi^{(t,k)}_{\mathrm{obs}}}\left[\mathbb{I}\left[\widehat{\phi}^{(t)}_h(x_h)\neq\phi^\star_h(x_h)\right]\right]$$

$$=\sum_{t=1}^{T}\sum_{h=1}^{H}\mathbb{E}_{\pi^{(t)}_{\mathrm{obs}}\sim\bar{p}^{(t)}_{\mathrm{obs}}}\mathbb{E}^{\bar{\pi}^{(t)}_{\mathrm{obs}}}\left[\mathbb{I}\left[\widehat{\phi}^{(t)}_h(x_h)\neq\phi^\star_h(x_h)\right]\right] \leq \mathsf{Est}_{\mathrm{class}}(T).$$

Let $\mathsf{Risk}(K,\mathrm{ALG}_{\mathrm{lat}},\phi,M^\star_{\mathrm{obs}}) = J^{M^\star_{\mathrm{obs}}}(\pi^\star_{M^\star_{\mathrm{obs}}}) - J^{M^\star_{\mathrm{obs}}}(\widehat{\pi}_{\mathrm{lat}}\circ\phi)$ be the random variable denoting the risk of the final policy output by $\mathrm{ALG}_{\mathrm{lat}}$ after $K$ rounds of interaction with $M^\star_{\mathrm{obs}}$ when given feature $\phi$ in any epoch $t$. For any $\phi : \mathcal{X} \to \mathcal{S}$, let $\mathbb{E}_\phi$ denote the law over trajectories $(x^{(k)}_h, a^{(k)}_h, r^{(k)}_h)_{k\in[K+1],h\in[H]}$ and policies $(\pi^{(k)}_{\mathrm{lat}}\circ\phi)_{k\in[K+1]}$ generated after $K$ rounds of interaction when $\mathrm{ALG}_{\mathrm{lat}}$ is given feature $\phi$ in any epoch. (Recall that, for all of the above definitions, a new instance of $\mathrm{ALG}_{\mathrm{lat}}$ is initialized at every epoch, so we do not have to specify *which epoch it is*, only the current feature $\phi$). Finally, let $G_t$ be the "good" event

$$G_t = \left\{\forall k\in[K+1],\forall h\in[H]:\ \widehat{\phi}^{(t)}_h(x^{(t,k)}_h) = \phi^\star_h(x^{(t,k)}_h)\right\}.$$

Recall that, in any round $t$, $\mathrm{ALG}_{\mathrm{lat}}$ only observes the latent ("compressed") trajectories $(\widehat{\phi}^{(t)}_h(x^{(t,k)}_h), a^{(t,k)}_h, r^{(t,k)}_h)$ as history for choosing policies. We can therefore conclude that, when $\widehat{\phi}^{(t)}(x^{(t,k)}_h) = \phi^\star(x^{(t,k)}_h)$ for all $k\in[K+1], h\in[H]$, the distribution over final policies $\widehat{\pi}^{(t)}_{\mathrm{lat}}$ chosen by $\mathrm{ALG}_{\mathrm{lat}}$ will be identical as if we had chosen $\phi^\star$ as our decoder. In particular, this implies

$$\mathbb{E}_{\widehat{\phi}^{(t)}}\left[\mathbb{I}\{G_t\}\mathsf{Risk}(K,\mathrm{ALG}_{\mathrm{lat}},\widehat{\phi}^{(t)},M^\star_{\mathrm{obs}})\right] = \mathbb{E}_{\phi^\star}[\mathbb{I}\{G_t\}\mathsf{Risk}(K,\mathrm{ALG}_{\mathrm{lat}},\phi^\star,M^\star_{\mathrm{obs}})]$$

$$\leq \mathsf{Risk}_\star(K), \qquad (58)$$

where the second line simply follows by removing the indicator function, recalling that $\mathsf{Risk}_\star(K) = \mathbb{E}[\mathsf{Risk}(K,\mathrm{ALG}_{\mathrm{lat}},M^\star_{\mathrm{lat}})]$, and using that $\mathsf{Risk}(K,\mathrm{ALG}_{\mathrm{lat}},\phi^\star,M^\star_{\mathrm{obs}}) = \mathsf{Risk}(K,\mathrm{ALG}_{\mathrm{lat}},M^\star_{\mathrm{lat}})$.

Then, we have:

$$\mathbb{E}[\mathsf{Risk}_{\mathrm{obs}}(TK)] = \frac{1}{T}\sum_{t=1}^{T}\mathbb{E}_{\widehat{\phi}^{(t)}\sim\rho^{(t)}}\left[\mathbb{E}_{\widehat{\phi}^{(t)}}\left[\mathsf{Risk}(K,\mathrm{ALG}_{\mathrm{lat}},\widehat{\phi}^{(t)},M^\star_{\mathrm{obs}})\right]\right]$$

$$\leq \frac{1}{T}\sum_{t=1}^{T}\mathbb{E}_{\widehat{\phi}^{(t)}\sim\rho^{(t)}}\left[\mathbb{E}_{\widehat{\phi}^{(t)}}\left[\mathbb{I}\{G_t\}\mathsf{Risk}(K,\mathrm{ALG}_{\mathrm{lat}},\widehat{\phi}^{(t)},M^\star_{\mathrm{obs}})\right]\right]$$

$$+ \frac{1}{T}\sum_{t=1}^{T}\mathbb{E}_{\widehat{\phi}^{(t)}\sim\rho^{(t)}}\left[\mathbb{E}_{\widehat{\phi}^{(t)}}[\mathbb{I}\{\neg G_t\}]\right]$$

$$\leq \frac{1}{T}\sum_{t=1}^{T}\mathsf{Risk}_\star(K) + \frac{1}{T}\sum_{t=1}^{T}\mathbb{P}(\neg G_t)$$

$$= \mathsf{Risk}_\star(K) + \frac{1}{T}\sum_{t=1}^{T}\mathbb{P}(\neg G_t),$$

where the first equality applies the tower rule for conditional expectation, the second equality applies linearity of conditional expectations and the upper bound $\mathrm{Risk}(K, \mathrm{ALG}_{\mathtt{lat}}, \widehat{\phi}^{(t)}, M_{\mathtt{obs}}^\star) \leq 1$, and the third lines applies the upper bound Eq. (58). It remains to bound the last term. Here, note that by a union bound,

$$\mathbb{P}(\neg G_t) \leq \mathbb{E}\left[\sum_{k=1}^{K+1} \sum_{h=1}^{H} \mathbb{E}_{\pi^{(t,k)} \sim p^{(t,k)}} \mathbb{E}^{\pi^{(t,k)}} \mathbb{I}\left\{\widehat{\phi}^{(t)}(x_h^{(t,k)}) \neq \phi^\star(x_h^{(t,k)})\right\}\right],$$

where we have used that trajectory $k$ in round $t$ is sampled from policy $\pi^{(t,k)}$, which is in turn sampled from $p^{(t,k)}$. Summing over $t$ and using the bound in Eq. (57) concludes the proof.

$\square$

# H  Proofs for Appendix A: Self-Predictive Estimation

This appendix contains additional information and proofs related to algorithmic modularity under self-predictive estimation (Appendix A), and is organized as follows:

- Appendix H.1 contains the pseudocode and proofs related to the online representation learning oracle SELFPREDICT.OPT (Lemma A.1).

- Appendix H.2 contains the proof for our risk bound of the O2L algorithm under self-predictive estimation (Theorem A.1).

## H.1  Pseudocode and Proofs for SELFPREDICT.OPT (Lemma A.1)

The pseudocode for our self-predictive estimation procedure is given in Algorithm 4.

---

**Algorithm 4** Optimistic Self-Predictive Latent Model Estimation (SELFPREDICT.OPT)

---

1: **input**: Decoder set $\Phi$, Latent model class $\mathcal{M}_{\text{lat}}$, Mismatch-complete class $\mathcal{L}_{\text{lat}}$, Optimism parameter $\gamma$
2: Set $\beta := \frac{1}{2}\sqrt{C_{\text{cov}} H \log(T)/T}$
3: **for** $t = 1, 2, \cdots, T$ **do**
4:      Get dataset $\mathcal{D}^{(t)} = \{x_h^{(i)}, a_h^{(i)}, r_h^{(i)}, x_{h+1}^{(i)}\}_{i \in [t-1], h \in [H]}$
5:      Compute

$$(\widehat{M}^{(t)}, \widehat{\phi}^{(t)}) = \underset{(M,\phi) \in (\mathcal{M}_{\text{lat}}, \Phi)}{\arg\max} \left\{ (\gamma\beta)^{-1} J^M(\pi_M) + \sum_{h=1}^{H} \sum_{i=1}^{n} \log\big(M_h(r_h^{(i)}, \phi_{h+1}(x_{h+1}^{(i)}) \mid \phi_h(x_h^{(i)}), a_h^{(i)})\big) \right. \tag{59}$$

$$\left. - \max_{(M',\phi') \in (\mathcal{L}_{\text{lat}}, \Phi)} \sum_{i=1}^{n} \log\big(M_h'(r_h^{(i)}, \phi_{h+1}(x_{h+1}^{(i)}) \mid \phi_h'(x_h^{(i)}), a_h^{(i)})\big) \right\}. \tag{60}$$

6:      Return $\widehat{\phi}^{(t)} = \left\{\widehat{\phi}_h^{(t)}\right\}_{h \in [H]}$.
7: **end for**

---

Our main result concerning the SELFPREDICT.OPT estimator for online optimistic self-predictive estimation is the following. We recall our notation for the instantaneous self-prediction error

$$[\Delta_h(M_{\text{lat}}, \phi)](x_h, a_h) := D_{\text{H}}^2\big(M_{\text{lat},h}(\phi_h(x_h), a_h), [\phi_{h+1} \sharp M_{\text{obs},h}^\star](x_h, a_h)\big).$$

**Lemma A.1** (Optimistic self-predictive estimation via SELFPREDICT.OPT). *Let $\Pi_{\text{lat}}$ denote the set of policies played by $\text{ALG}_{\text{lat}}$, and $C_{\text{cov,st}} = C_{\text{cov,st}}(M_{\text{lat}}^\star, \Gamma_\Phi \circ \Pi_{\text{lat}})$ be the state coverability parameter on $M_{\text{lat}}^\star$ over the set of stochastic policies $\Gamma_\Phi \circ \Pi_{\text{lat}}$ (Eq. (9)). Then, for any $\gamma > 0$, under decoder realizability ($\phi^\star \in \Phi$), base model realizability ($M_{\text{lat}}^\star \in \mathcal{M}_{\text{lat}}$), and mismatch function completeness with class $\mathcal{L}_{\text{lat}}$ (Assumption A.2), the estimator in Algorithm 4 with inputs $\Phi, \mathcal{M}_{\text{lat}}, \mathcal{L}_{\text{lat}}$, and $\gamma$ satisfies Assumption A.1 with[24]*

$$\text{Est}_{\text{self;opt}}(T, \gamma) = \widetilde{\mathcal{O}}\bigg(\sqrt{H C_{\text{cov,st}} |\mathcal{A}| T} \log(|\mathcal{M}_{\text{lat}}||\mathcal{L}_{\text{lat}}||\Phi|)\bigg).$$

**Proof of Lemma A.1.** We will firstly establish that the algorithm obtains low *offline* estimation error.

**Lemma H.1** (SELFPREDICT.OPT attains low offline estimation error). *For any $\gamma > 0$, under decoder realizability ($\phi^\star \in \Phi$), model realizability ($M_{\text{lat}}^\star \in \mathcal{M}_{\text{lat}}$), and mismatch function completeness with class $\mathcal{L}_{\text{lat}}$ (Assumption A.2), the estimator in Algorithm 4 with inputs $\Phi, M_{\text{lat}}, \mathcal{L}_{\text{lat}}$, and $\gamma$ satisfies*

---

[24]In this section, the notations $\widetilde{\mathcal{O}}$ and $\lesssim$ ignores constants and logarithmic factors of: $H, C_{\text{cov,st}}, |\mathcal{A}|, T$, and $\log(|\mathcal{M}_{\text{lat}}||\mathcal{L}_{\text{lat}}||\Phi|)$.

*that for all $t \in [T]$, with probability at least $1 - \delta$,*

$$\sum_{h=0}^{H} \sum_{i=1}^{t-1} \mathbb{E}_{\pi^{(i)} \sim p^{(i)}} \mathbb{E}^{\pi^{(i)}} \left[ [\Delta_h(\widehat{M}^{(t)}, \widehat{\phi}^{(t)})](x_h, a_h) \right] + \gamma^{-1} \left( J^{M^\star_{\text{lat}}}(\pi_{M^\star_{\text{lat}}}) - J^{\widehat{M}^{(t)}}(\pi_{\widehat{M}^{(t)}}) \right)$$

$$\leq \mathcal{O}\big( \log\big( |\mathcal{M}_{\text{lat}}||\mathcal{L}_{\text{lat}}||\Phi| H T \delta^{-1} \big) \big). \tag{61}$$

Given this result, we can appeal to offline-to-online conversions to establish the final result. Let $C_{\text{cov}} := C_{\text{cov}}(M^\star_{\text{obs}}, \Pi_{\text{lat}} \circ \Phi)$ denote the (state-action) coverability coefficient in $M^\star_{\text{obs}}$ over the set of policies $\Pi_{\text{lat}} \circ \Phi$. Note that by Lemma D.1 we have $C_{\text{cov,st}}(M^\star_{\text{obs}}, \Pi_{\text{lat}} \circ \Phi) = C_{\text{cov,st}}$ and therefore by Lemma D.4 we have $C_{\text{cov}}(M^\star_{\text{obs}}, \Pi_{\text{lat}} \circ \Phi) \leq C_{\text{cov,st}}|\mathcal{A}|$. Let $\eta > 0$ be a parameter to be chosen later, and $\beta_{\text{off}} = \mathcal{O}\big( \log(|\mathcal{M}_{\text{lat}}||\mathcal{L}_{\text{lat}}||\Phi| H T \delta^{-1}) \big)$ be the offline estimation error guaranteed by Lemma H.1. We abbreviate $\alpha := \sqrt{C_{\text{cov}} H \log(T)}$, $\mathbb{E}^{p^{(t)}}[\cdot] := \mathbb{E}_{\pi^{(t)} \sim p^{(t)}} \mathbb{E}^{\pi^{(t)}}[\cdot]$, and $\mathbb{E}^{\widetilde{p}^{(t)}} := \sum_{i=1}^{t-1} \mathbb{E}_{\pi^{(i)} \sim p^{(i)}} \mathbb{E}^{\pi^{(i)}}[\cdot]$. Then, we have:

$$\sum_{t=1}^{T} \sum_{h=1}^{H} \mathbb{E}^{p^{(t)}} \left[ [\Delta_h(\widehat{M}^{(t)}, \widehat{\phi}^{(t)})](x_h, a_h) \right] + \gamma^{-1} \left( J^{M^\star_{\text{lat}}}(\pi_{M^\star_{\text{lat}}}) - J^{\widehat{M}^{(t)}}(\pi_{\widehat{M}^{(t)}}) \right)$$

$$\leq \alpha \sqrt{\sum_{t=1}^{T} \sum_{h=1}^{H} \mathbb{E}^{\widetilde{p}^{(t)}} \left[ [\Delta_h(\widehat{M}^{(t)}, \widehat{\phi}^{(t)})](x_h, a_h) \right] + \gamma^{-1} \sum_{t=1}^{T} \left( J^{M^\star_{\text{lat}}}(\pi_{M^\star_{\text{lat}}}) - J^{\widehat{M}^{(t)}}(\pi_{\widehat{M}^{(t)}}) \right)}$$

$$+ \mathcal{O}(H C_{\text{cov}})$$

$$\leq \alpha \left( \frac{\eta}{2} \sum_{t=1}^{T} \sum_{h=1}^{H} \mathbb{E}^{\widetilde{p}^{(t)}} \left[ [\Delta_h(\widehat{M}^{(t)}, \widehat{\phi}^{(t)})](x_h, a_h) \right] + \frac{1}{2\eta} \right) + \gamma^{-1} \sum_{t=1}^{T} \left( J^{M^\star_{\text{lat}}}(\pi_{M^\star_{\text{lat}}}) - J^{\widehat{M}^{(t)}}(\pi_{\widehat{M}^{(t)}}) \right)$$

$$+ \mathcal{O}(H C_{\text{cov}})$$

where in the first inequality we have used Lemma C.7 with $g_h^{(t)} = \Delta_h(\widehat{M}^{(t)}, \widehat{\phi}^{(t)})$ and in the second inequality we have used the AM-GM inequality with parameter $\eta$. Collecting terms, we proceed via:

$$= \frac{\alpha \eta}{2} \sum_{t=1}^{T} \left( \sum_{h=1}^{H} \mathbb{E}^{\widetilde{p}^{(t)}} \left[ \Delta_h(\widehat{M}^{(t)}, \widehat{\phi}^{(t)})(x_h, a_h) \right] + \left( \frac{\gamma \eta \alpha}{2} \right)^{-1} \left( J^{M^\star_{\text{lat}}}(\pi_{M^\star_{\text{lat}}}) - J^{\widehat{M}^{(t)}}(\pi_{\widehat{M}^{(t)}}) \right) \right)$$

$$+ \frac{\alpha}{2\eta} + \mathcal{O}(H C_{\text{cov}})$$

$$\leq \frac{\alpha \eta}{2} T \beta_{\text{off}} + \frac{\alpha}{2\eta} + \mathcal{O}(H C_{\text{cov}})$$

$$\leq \mathcal{O}\left( \sqrt{C_{\text{cov,st}}|\mathcal{A}| H \log(T) T} \beta_{\text{off}} + H C_{\text{cov,st}}|\mathcal{A}| \right)$$

$$\leq \mathcal{O}\left( \sqrt{H C_{\text{cov,st}}|\mathcal{A}| T \log(T)} \log\big( |\mathcal{M}_{\text{lat}}||\mathcal{L}_{\text{lat}}||\Phi| H T \delta^{-1} \big) \right),$$

where in the first inequality we have used Lemma H.1 and the definition of $\gamma$ in Algorithm 4 (cf. Eq. (59)) and in the second inequality we have chosen $\eta = 1/\sqrt{T}$ to balance the terms and used the bound $C_{\text{cov}} \leq C_{\text{cov,st}}|\mathcal{A}|$. We convert to an expected regret bound by picking $\delta$ appropriately, which gives the final result. It remains to show Lemma H.1. $\qquad \square$

**Proof of Lemma H.1.** Fix an iteration $t \in [T]$, and abbreviate $\widehat{M} := \widehat{M}^{(t)}$ and $\widehat{\phi} := \widehat{\phi}^{(t)}$. We follow the analysis of maximum likelihood estimation from Geer; Zhang; Agarwal et al. [Gee00; Zha06; AKKS20]. In particular, we quote Lemma 24 of [AKKS20], which in an abstract conditional estimation framework with density class $\mathcal{F}$ states the following.

**Lemma H.2** (Lemma 24 of Agarwal et al. [AKKS20])**.** *Let $D = \{(x_i, y_i)\}$ be a dataset collected with $x_i \sim p^{(i)}(x_{1:i-1}, y_{1:i-1})$ and $y_i \sim f^\star(\cdot \mid x_i)$, $L(f, D) = \sum_{i=1}^{n} \ell(f, (x_i, y_i))$ be any loss function that decomposes additively, $\widehat{f} : D \to \mathcal{F}$ be an estimator, $D'$ be a tangent sequence $D' = \{(\widetilde{x}_i, \widetilde{y}_i)\}$*

*sampled independently via $\widetilde{x}_i \sim p^{(i)}(x_{1:i-1}, y_{1:i-1})$ and $\widetilde{y}_i \sim f^\star(\cdot \mid \widetilde{x}_i)$. Then, with probability at least $1 - \delta$, we have*

$$-\log \mathbb{E}_{D'} \exp\Big(L(\widehat{f}(D), D')\Big) \le -L(\widehat{f}(D), D) + \log\big(|\mathcal{F}|\delta^{-1}\big), \tag{62}$$

For our purposes, we have that $\mathcal{F} = \mathcal{M}_{\text{lat}} \circ \Phi$, the data distribution is collected adaptively (for each $h \in [H]$) via $\pi^{(i)} \sim p^{(i)}$, $x_h^{(i)}, a_h^{(i)} \sim d_h^{M_{\text{obs}}^\star, \pi^{(i)}}$, and $r_h^{(i)}, x_{h+1}^{(i)} \sim M_{\text{obs}}^\star(\cdot \mid x_h^{(i)}, a_h^{(i)})$. For the loss function $L$, we take

$$L((M, \phi), D) = -\sum_{h=0}^{H} \sum_{i=1}^{t} \log\left(\frac{M_{\text{obs}}^\star(r_{h+1}^{(i)}, \phi_{h+1}(x_{h+1}^{(i)}) \mid x_h^{(i)}, a_h^{(i)})}{[M_h \circ \phi_h](r_h^{(i)}, \phi_{h+1}(x_{h+1}^{(i)}) \mid x_h^{(i)}, a_h^{(i)})}\right)$$
$$-\frac{\gamma^{-1}}{2}(J^{M_{\text{lat}}^\star}(\pi_{M_{\text{lat}}^\star}) - J^M(\pi_M)).$$

We begin by upper bounding the quantity $-L((\widehat{M}, \widehat{\phi})(D), D)$ appearing on the right-hand side of Eq. (62), or equivalently lower bounding $L((\widehat{M}, \widehat{\phi})(D), D)$. Let us abbreviate $\widehat{V} = J^{\widehat{M}}(\pi_{\widehat{M}})$ and $V^\star = J^{M_{\text{lat}}^\star}(\pi_{M_{\text{lat}}^\star})$. Towards this, note that

$$L((\widehat{M}, \widehat{\phi})(D), D) = \sum_{h=0}^{H} \sum_{i=1}^{t} \log\Big(\big[\widehat{M}_h \circ \widehat{\phi}_h\big](r_h^{(i)}, \widehat{\phi}_{h+1}(x_{h+1}^{(i)}) \mid x_h^{(i)}, a_h^{(i)})\Big)$$
$$-\sum_{h=0}^{H} \sum_{i=1}^{t} \log\Big(M_{\text{obs}}^\star(r_h^{(i)}, \widehat{\phi}_{h+1}(x_{h+1}^{(i)}) \mid x_h^{(i)}, a_h^{(i)})\Big) + \frac{\gamma^{-1}}{2}(\widehat{V} - V^\star)$$
$$\ge \sum_{h=0}^{H} \sum_{i=1}^{t} \log\Big(\big[\widehat{M}_h \circ \widehat{\phi}_h\big](r_h^{(i)}, \widehat{\phi}_{h+1}(x_{h+1}^{(i)}) \mid x_h^{(i)}, a_h^{(i)})\Big)$$
$$-\sum_{h=0}^{H} \max_{[M' \circ \phi'] \in \mathcal{L}_{\text{lat}} \circ \Phi} \sum_{i=1}^{t} \log\Big([M_h' \circ \phi_h'](r_h^{(i)}, \widehat{\phi}_{h+1}(x_{h+1}^{(i)}) \mid x_h^{(i)}, a_h^{(i)})\Big)$$
$$+\frac{\gamma^{-1}}{2}(\widehat{V} - V^\star)$$
$$\ge \sum_{h=0}^{H} \sum_{i=1}^{t} \log\big(\big[M_{\text{lat},h}^\star \circ \phi_h^\star\big](r_h^{(i)}, \phi_{h+1}^\star(x_{h+1}^{(i)}) \mid x_h^{(i)}, a_h^{(i)})\big)$$
$$-\sum_{h=0}^{H} \max_{[M' \circ \phi'] \in \mathcal{L}_{\text{lat}} \circ \Phi} \sum_{i=1}^{t} \log\big([M_h' \circ \phi_h'](r_h^{(i)}, \phi_{h+1}^\star(x_{h+1}^{(i)}) \mid x_h^{(i)}, a_h^{(i)})\big)$$
$$+\frac{\gamma^{-1}}{2}(V^\star - V^\star)$$
$$= \sum_{h=0}^{H} \sum_{i=1}^{t} \log\big(\big[M_{\text{lat},h}^\star \circ \phi_h^\star\big](r_h^{(i)}, \phi_{h+1}^\star(x_{h+1}^{(i)}) \mid x_h^{(i)}, a_h^{(i)})\big)$$
$$-\sum_{h=0}^{H} \max_{[M' \circ \phi'] \in \mathcal{L}_{\text{lat}} \circ \Phi} \sum_{i=1}^{t} \log\big([M_h' \circ \phi_h'](r_h^{(i)}, \phi_{h+1}^\star(x_{h+1}^{(i)}) \mid x_h^{(i)}, a_h^{(i)})\big),$$

where in the second line we have used Lemma D.8 with Assumption A.2 and in the third line we have used the ERM property of $\widehat{M} \circ \widehat{\phi}$ together with decoder and model realizability. We claim that this implies

$$L((\widehat{M}, \widehat{\phi})(D), D) \ge -\log\big(|\mathcal{L}_{\text{lat}} \circ \Phi|H\delta^{-1}\big) \tag{63}$$

by concentration. Indeed, for each $h \in [H], i \in [t]$, and $[M' \circ \phi'] \in \mathcal{L}_{\text{lat}} \circ \Phi$, let

$$Z_{i,h}^{[M' \circ \phi']} = -\frac{1}{2}\log\left(\frac{M_{\text{obs}}^\star(r_h^{(i)}, \phi_{h+1}^\star(x_{h+1}^{(i)}) \mid x_h^{(i)}, a_h^{(i)})}{[M' \circ \phi'](r_h^{(i)}, \phi_{h+1}^\star(x_{h+1}^{(i)}) \mid x_h^{(i)}, a_h^{(i)})}\right)$$

Applying Lemma C.1, we have that

$$\sum_{i=1}^{t} \log\left( \frac{M_{\mathsf{obs}}^{\star}(r_h^{(i)}, \phi_{h+1}^{\star}(x_{h+1}^{(i)}) \mid x_h^{(i)}, a_h^{(i)})}{[M' \circ \phi'](r_h^{(i)}, \phi_{h+1}^{\star}(x_{h+1}^{(i)}) \mid x_h^{(i)}, a_h^{(i)})} \right)$$
$$\geq \sum_{i=1}^{t} -2\log\left( \mathbb{E}_{\pi^{(i)} \sim p^{(i)}} \mathbb{E}^{\pi^{(i)}} \left[ \exp\left( -\frac{1}{2} \log\left( \frac{M_{\mathsf{obs}}^{\star}(r_h^{(i)}, \phi_{h+1}^{\star}(x_{h+1}^{(i)}) \mid x_h^{(i)}, a_h^{(i)})}{[M' \circ \phi'](r_h^{(i)}, \phi_{h+1}^{\star}(x_{h+1}^{(i)}) \mid x_h^{(i)}, a_h^{(i)})} \right) \right) \right] \right)$$
$$- \log(\delta^{-1}), \tag{64}$$

with probability at least $1 - \delta$, where we have recalled that data is gathered adaptively according to $\pi^{(i)} \sim p^{(i)}$. We now quote the following lemma from Zhang; Agarwal et al. [Zha06; AKKS20].

**Lemma H.3** (Lemma 25 of Agarwal et al. [AKKS20]). *For any $\mathcal{D} \in \Delta(\mathcal{X})$ and $p, q \in [\mathcal{X} \to \Delta(\mathcal{Y})]$, we have*

$$-2\log \mathbb{E}_{x \sim \mathcal{D}, y \sim q(\cdot|x)} \exp\left( -\frac{1}{2} \log(q(y|x)/p(y|x)) \right) \geq \mathbb{E}_{x \sim \mathcal{D}} \left[ D_{\mathsf{H}}^2(q(\cdot \mid x), p(\cdot \mid x)) \right]$$

**Proof of Lemma H.3.** We include the proof for completeness. The result follows via the following steps.

$$-2\log \mathbb{E}_{x \sim \mathcal{D}, y \sim q(\cdot|x)} \exp\left( -\frac{1}{2} \log(q(y|x)/p(y|x)) \right) = -2\log \mathbb{E}_{x \sim \mathcal{D}, y \sim q(\cdot|x)} \sqrt{p(y|x)/q(y|x)}$$
$$\geq 2\left( 1 - \mathbb{E}_{x \sim \mathcal{D}, y \sim q(\cdot|x)} \sqrt{p(y|x)/q(y|x)} \right)$$
$$(\forall x : \log(x) \leq x - 1)$$
$$= \mathbb{E}_{x \sim \mathcal{D}} \left[ 2\left( 1 - \mathbb{E}_{y \sim q(\cdot|x)} \sqrt{p(y|x)/q(y|x)} \right) \right]$$
$$= \mathbb{E}_{x \sim \mathcal{D}} \left[ D_{\mathsf{H}}^2(p(\cdot \mid x), q(\cdot \mid x)) \right]$$

$\square$

By Lemma H.3, we have that the right-hand-side of Eq. (64) is further lower bounded by

$$\sum_{i=1}^{t} \log\left( \frac{M_{\mathsf{obs}}^{\star}(r_h^{(i)}, \phi_{h+1}^{\star}(x_{h+1}^{(i)}) \mid x_h^{(i)}, a_h^{(i)})}{[M' \circ \phi'](r_h^{(i)}, \phi_{h+1}^{\star}(x_{h+1}^{(i)}) \mid x_h^{(i)}, a_h^{(i)})} \right)$$
$$\geq \sum_{i=1}^{t} \mathbb{E}_{\pi^{(i)} \sim p^{(i)}} \mathbb{E}^{\pi^{(i)}} \left[ D_{\mathsf{H}}^2\left( \phi_{h+1}^{\star} \sharp M_{\mathsf{obs}}^{\star}(\cdot \mid x_h^{(i)}, a_h^{(i)}), [M' \circ \phi'](\cdot \mid x_h^{(i)}, a_h^{(i)}) \right) \right] - \log(\delta^{-1})$$
$$\geq -\log(\delta^{-1}),$$

where the last line follows from the non-negativity of squared Hellinger. Taking a union bound over $M' \circ \phi' \in \mathcal{L}_{\mathsf{lat}} \circ \Phi$ and $h \in [H]$ gives the desired lower bound in Eq. (63).

To conclude the proof, it remains to lower bound the left-hand side in Eq. (62). Here, note that:

$$-\log \mathbb{E}_{D'} \exp\left( L((\widehat{M}, \hat{\phi})(D), D') \right) + \frac{\gamma^{-1}}{2}(V^{\star} - \widehat{V})$$
$$= -\log \mathbb{E}_{D'} \left[ \exp\left( -\frac{1}{2} \sum_{h=1}^{H} \sum_{i=1}^{t} \log\left( \frac{M_{\mathsf{obs}}^{\star}(\widetilde{r}_h^{(i)}, \widehat{\phi}_{h+1}(\widetilde{x}_{h+1}^{(i)}) \mid x_h^{(i)}, a_h^{(i)})}{[\widehat{M}_h \circ \widehat{\phi}_h](r_h^{(i)}, \widehat{\phi}_{h+1}(x_{h+1}^{(i)}) \mid x_h^{(i)}, a_h^{(i)})} \right) \right) \right]$$
$$= -\sum_{h=1}^{H} \sum_{i=1}^{t} \log \mathbb{E}_{\pi^{(i)} \sim p^{(i)}} \mathbb{E}^{\pi^{(i)}} \left[ \exp\left( -\frac{1}{2} \log\left( \frac{M_{\mathsf{obs}}^{\star}(r_h^{(i)}, \widehat{\phi}_{h+1}(x_{h+1}^{(i)}) \mid x_h^{(i)}, a_h^{(i)})}{[\widehat{M}_h \circ \widehat{\phi}_h](r_h^{(i)}, \widehat{\phi}_{h+1}(x_{h+1}^{(i)}) \mid x_h^{(i)}, a_h^{(i)})} \right) \right) \right],$$
$$\tag{65}$$

where we have used that in the "tangent sequence" $D'$ the current sample $(\widetilde{r}_h^{(i)}, \widetilde{x}_{h+1}^{(i)})$ is independent of $(r_h^{(i)}, x_{h+1}^{(i)})$. To bound this term, we again appeal to Lemma H.3, concluding that

$$-\sum_{h=1}^{H}\sum_{i=1}^{t} \log \mathbb{E}_{\pi^{(i)}\sim p^{(i)}} \mathbb{E}^{\pi^{(i)}}\left[\exp\left(-\frac{1}{2}\log\left(\frac{M_{\mathsf{obs}}^{\star}(r_h^{(i)}, \widehat{\phi}_{h+1}(x_{h+1}^{(i)}) \mid x_h^{(i)}, a_h^{(i)})}{\left[\widehat{M}_h \circ \widehat{\phi}_h\right](r_h^{(i)}, \widehat{\phi}_{h+1}(x_{h+1}^{(i)}) \mid x_h^{(i)}, a_h^{(i)})}\right)\right)\right]$$

$$\geq \frac{1}{2}\sum_{h=1}^{H}\sum_{i=1}^{t} \mathbb{E}_{\pi^{(i)}\sim p^{(i)}} \mathbb{E}^{\pi^{(i)}}\left[D_{\mathsf{H}}^2\left(\left[\widehat{M}_h \circ \widehat{\phi}_h\right](x_h, a_h), \widehat{\phi}_{h+1}\sharp M_{\mathsf{obs}}^{\star}(x_h, a_h)\right)\right]$$

Combining everything, we have:

$$\frac{1}{2}\left(\sum_{h=1}^{H}\sum_{i=1}^{t} \mathbb{E}_{\pi^{(i)}\sim p^{(i)}} \mathbb{E}^{\pi^{(i)}}\left[D_{\mathsf{H}}^2\left(\left[\widehat{M}_h \circ \widehat{\phi}_h\right](x_h, a_h), \widehat{\phi}_{h+1}\sharp M_{\mathsf{obs}}^{\star}(x_h, a_h)\right)\right] + \gamma^{-1}(V^{\star} - \widehat{V})\right)$$

$$\leq \log\left(|\mathcal{L}_{\mathsf{lat}}||\Phi|H\delta^{-1}\right) + \log\left(|\mathcal{M}_{\mathsf{lat}}||\Phi|\delta^{-1}\right)$$

Taking an additional union bound over $t \in [T]$, we have that with probability at least $1 - \delta$:

$$\sum_{h=1}^{H}\sum_{i=1}^{t} \mathbb{E}_{\pi^{(i)}\sim p_h^{(i)}} \mathbb{E}^{\pi^{(i)}}\left[D_{\mathsf{H}}^2\left(\left[\widehat{M}_h \circ \widehat{\phi}_h\right](x_h, a_h), \widehat{\phi}_{h+1}\sharp M_{\mathsf{obs}}^{\star}(x_h, a_h)\right)\right]$$

$$+ \gamma^{-1}\left(J^{M_{\mathsf{lat}}^{\star}}(\pi_{M_{\mathsf{lat}}^{\star}}) - J^{M^{(t)}}(\pi_{M^{(t)}})\right) \leq \mathcal{O}\left(\log\left(|\mathcal{M}_{\mathsf{lat}}||\mathcal{L}_{\mathsf{lat}}||\Phi|HT\delta^{-1}\right)\right),$$

for all $t \in [T]$, as desired.

$\square$

**Corollary A.1** (Algorithmic modularity via SELFPREDICT.OPT). *Under the same conditions as in Lemma A.1, and for any base algorithm* $\mathrm{ALG}_{\mathsf{lat}}$, *O2L with inputs* $T, K, \Phi$, SELFPREDICT.OPT, *and* $\mathrm{ALG}_{\mathsf{lat}}$ *achieves*

$$\mathbb{E}[\mathsf{Risk}_{\mathsf{obs}}(TK)] \lesssim c_1 \cdot \mathsf{Risk}_{\mathsf{base}}(K) + c_2\gamma \cdot \frac{K}{\sqrt{T}}\sqrt{HC_{\mathsf{cov,st}}|\mathcal{A}|\log(|\mathcal{M}_{\mathsf{lat}}||\mathcal{L}_{\mathsf{lat}}||\Phi|)} + c_3\gamma^{-1} \cdot KH,$$

*for absolute constants* $c_1, c_2, c_3$. *Consequently, for any* $\mathrm{ALG}_{\mathsf{lat}}$ *with base risk* $\mathsf{Risk}_{\mathsf{base}}(K)$, *setting* $\gamma$ *and* $T$ *appropriately gives*

$$\mathbb{E}[\mathsf{Risk}_{\mathsf{obs}}(TK)] \lesssim \mathsf{Risk}_{\mathsf{base}}(K),$$

*with a number of trajectories* $TK = \widetilde{\mathcal{O}}\left(K^5H^3C_{\mathsf{cov,st}}|\mathcal{A}|\log^2(|\mathcal{M}_{\mathsf{lat}}||\mathcal{L}_{\mathsf{lat}}||\Phi|)/(\mathsf{Risk}_{\mathsf{base}}(K))^4\right)$.

**Proof of Corollary A.1.** The first inequality simply follows by plugging the bound of $\mathsf{Est}_{\mathsf{self;opt}}$ from Lemma A.1 into Theorem A.1. For the second inequality, let $\Delta = c_2\sqrt{HC_{\mathsf{cov,st}}|\mathcal{A}|\log(|\mathcal{M}_{\mathsf{lat}}||\mathcal{L}_{\mathsf{lat}}||\Phi|)}$. The result follows by setting $\gamma$ s.t. $c_3\gamma^{-1}HK = \mathsf{Risk}_{\mathsf{base}}(K)$ i.e. $\gamma = c_3\frac{KH}{\mathsf{Risk}_{\mathsf{base}}(K)}$, and $T$ such that $\frac{\gamma K\Delta}{\sqrt{T}} = \mathsf{Risk}_{\mathsf{base}}(K)$ i.e. $T = \frac{K^4\Delta^2\gamma^2}{\mathsf{Risk}_{\mathsf{base}}(K)^2} = \frac{K^4\Delta^2H^2}{(\mathsf{Risk}_{\mathsf{base}}(K))^4}$. Then the result follows by direct substitution and by noting that $\frac{K}{T} \leq 1$ since $\mathsf{Risk}_{\mathsf{base}}(K) \leq 1$.

$\square$

## H.2 Proofs for Main Risk Bound (Theorem A.1)

Our main risk bound (Theorem A.1) follows as a special case of a more general theorem (Theorem H.1), which holds for algorithm that satisfies a property we refer to as CorruptionRobust-ness (Definition I.2). We now state the more general theorem, postponing its proof (and a formal definition of corruption robustness) until Appendix I.

**Theorem H.1** (Risk bound for O2L under self-predictive estimation and `CorruptionRobustness`).
*Assume* $\text{REP}_{\text{self;opt}}$ *satisfies Assumption A.1 with parameter* $\gamma > 0$ *and that* $\mathcal{M}_{\text{lat}}$ *is realizable (i.e.* $M_{\text{lat}}^\star \in \mathcal{M}_{\text{lat}}$*). Furthermore, let* $\text{ALG}_{\text{lat}}$ *be* `CorruptionRobust` *(Definition I.2) with parameter* $\alpha$*. Then,* O2L *(Algorithm 1) with inputs* $T, K, \Phi, \text{ALG}_{\text{lat}}$*, and* $\text{REP}_{\text{self;opt}}$ *has expected risk*

$$\mathbb{E}[\text{Risk}_{\text{obs}}(TK)] \leq c_1 \cdot \text{Risk}_{\text{base}}(K) + c_2 \gamma \cdot \frac{K}{T} \text{Est}_{\text{self;opt}}(T, \gamma) + c_3 \gamma^{-1} \cdot (\alpha^2 + H) \qquad (66)$$

*for absolute constants* $c_1, c_2, c_3 > 0$.

Our main risk bound (Theorem A.1) follows from the following lemma, which establishes that *any* $\text{ALG}_{\text{lat}}$ is `CorruptionRobust` in the sense of Definition I.2 for a sufficiently large corruption robustness parameter. Below, for any POMDP $\widetilde{M}$ over state-action space $\mathcal{S} \times \mathcal{A}$, we write $\widetilde{M}(s_{1:h}, a_{1:h})$ for the conditional probability over reward $r_h$ and $s_{h+1}$ given $s_{1:h}, a_{1:h}$, i.e. $\widetilde{M}_h(s_{1:h}, a_{1:h}) = \widetilde{M}_h(r_h, s_{h+1} = \cdot \mid s_{1:h}, a_{1:h})$.

**Lemma H.4.** *Let* $M^\star$ *be any reference MDP and* $\widetilde{M}$ *be any POMDP with the same state and action space. Then for any algorithm* $\text{ALG}_{\text{lat}}$*, we have*

$$\mathbb{E}^{\widetilde{M}, \text{ALG}_{\text{lat}}}[\text{Risk}_{M^\star}(K)] \leq c_1 \, \mathbb{E}^{M^\star, \text{ALG}_{\text{lat}}}[\text{Risk}_{M^\star}(K)]$$
$$+ c_2 \, \mathbb{E}^{\widetilde{M}, \text{ALG}_{\text{lat}}}\left[\sum_{k=1}^{K} \sum_{h=1}^{H} \mathbb{E}^{\widetilde{M}, \pi^{(k)}}\left[D_{\mathsf{H}}^2\left(M_h^\star(s_h, a_h), \widetilde{M}_h(s_{1:h}, a_{1:h})\right)\right]\right],$$

*where* $c_1, c_2 > 0$ *are absolute constants. In particular,* $\text{ALG}_{\text{lat}}$ *is* `CorruptionRobust` *(Definition I.2) with* $\alpha = c_2 \sqrt{KH}$.

**Proof of Lemma H.4.** Let us abbreviate $\text{ALG} := \text{ALG}_{\text{lat}}$. For $i \in [K]$, let $\tau^{(i)}$ denote the trajectory $(s_1^{(i)}, a_1^{(i)}, r_1^{(i)}, \ldots, s_H^{(i)}, a_H^{(i)}, r_H^{(i)})$. Let $\mathbb{P} := \mathbb{P}^{M^\star, \text{ALG}}$ denote the law of $\{(\pi^{(i)}, \tau^{(i)})\}_{i \in [K]}$ under $\text{ALG}$ in the true MDP $M^\star$, and $\mathbb{Q} := \mathbb{P}^{\widetilde{M}, \text{ALG}}$ denote the law of $\{(\pi^{(i)}, \tau^{(i)})\}_{i \in [K]}$ under $\text{ALG}$ under the POMDP $\widetilde{M}$. Let us write $M^\star(\pi)$ and $\widetilde{M}(\pi)$ for the laws of trajectory $\tau$ sampled from policy $\pi$ in $M^\star$ or $\widetilde{M}$ respectively. Let $\widehat{\pi}$ denote the policy output by the algorithm after $K$ rounds of interaction with the environment. By Lemma C.5 we have

$$\mathbb{E}^{\widetilde{M}, \text{ALG}}\left[J^{M^\star}(\pi_{M^\star}) - J^{M^\star}(\widehat{\pi})\right] \leq 3 \, \mathbb{E}^{M^\star, \text{ALG}}\left[J^{M^\star}(\pi_{M^\star}) - J^{M^\star}(\widehat{\pi})\right] + 4 D_{\mathsf{H}}^2\left(\mathbb{P}^{M^\star, \text{ALG}}, \mathbb{P}^{\widetilde{M}, \text{ALG}}\right).$$

By the subadditivity property for squared Hellinger distance (Lemma C.4) applied to the sequence $\pi^{(1)}, \tau^{(1)}, \ldots, \pi^{(K)}, \tau^{(K)}$, we have

$$D_{\mathsf{H}}^2\left(\mathbb{P}^{M^\star, \text{ALG}}, \mathbb{P}^{\widetilde{M}, \text{ALG}}\right) \leq 7 \, \mathbb{E}^{\widetilde{M}, \text{ALG}}\left[\sum_{k=1}^{K} D_{\mathsf{H}}^2(\mathbb{P}(\pi^{(k)} \mid \pi^{(1:k-1)}, \tau^{(1:k-1)}), \mathbb{Q}(\pi^{(k)} \mid \pi^{(1:k-1)}, \tau^{(1:k-1)})) + \right.$$
$$\left. D_{\mathsf{H}}^2(\mathbb{P}(\tau^{(k)} \mid \pi^{(1:k)}, \tau^{(1:k-1)}), \mathbb{Q}(\tau^{(k)} \mid \pi^{(1:k)}, \tau^{(1:k-1)}))\right]$$

$$= 7 \, \mathbb{E}^{\widetilde{M}, \text{ALG}}\left[\sum_{k=1}^{K} D_{\mathsf{H}}^2(\mathbb{P}(\tau^{(k)} \mid \pi^{(1:k)}, \tau^{(1:k-1)}), \mathbb{Q}(\tau^{(k)} \mid \pi^{(1:k)}, \tau^{(1:k-1)}))\right]$$

$$= 7 \, \mathbb{E}^{\widetilde{M}, \text{ALG}}\left[\sum_{k=1}^{K} D_{\mathsf{H}}^2\left(M^\star(\pi^{(k)}), \widetilde{M}(\pi^{(k)})\right)\right]$$

$$\leq 49 \, \mathbb{E}^{\widetilde{M}, \text{ALG}}\left[\sum_{k=1}^{K} \sum_{h=1}^{H} \mathbb{E}^{\widetilde{M}, \pi^{(k)}}\left[D_{\mathsf{H}}^2\left(M_h^\star(s_h, a_h), \widetilde{M}_h(s_{1:h}, a_{1:h})\right)\right]\right]$$

where in the second step we have used that $\mathbb{P}(\pi^{(k)} \mid \pi^{(1:k)}, \tau^{(1:k-1)}) = \mathbb{Q}(\pi^{(k)} \mid \pi^{(1:k)}, \tau^{(1:k-1)})$ since the histories are equivalent, in the third step we have used that the trajectories are generated by the MDP/PODMP $M^\star$ and $\widetilde{M}$, respectively, in the fourth step we have again applied the subadditivity property of the squared Hellinger distance (Lemma C.4) to the sequence $(s_1, a_1, r_1, \ldots, s_H, a_H, r_H)$. $\qquad \square$

**Theorem A.1** (Risk bound for O2L under self-predictive estimation)**.** *Suppose* $\text{REP}_{\text{self;opt}}$ *satisfies Assumption A.1 with parameter* $\gamma > 0$. *Then Algorithm 1, with inputs* $T, K, \Phi, \text{REP}_{\text{self;opt}}$, *and* $\text{ALG}_{\text{lat}}$ *has expected risk*

$$\mathbb{E}[\text{Risk}_{\text{obs}}(TK)] \leq c_1 \cdot \text{Risk}_{\text{base}}(K) + c_2 \gamma \cdot \frac{K}{T} \text{Est}_{\text{self;opt}}(T, \gamma) + c_3 \gamma^{-1} \cdot KH,$$

*for absolute constants* $c_1, c_2, c_3 > 0$.

**Proof of Theorem A.1.**   This follows from Theorem H.1 as well as Lemma H.4, by taking $\alpha = c_2 \sqrt{KH}$ and simplifying. □

# I   Additional Results for Appendix A: Self-Predictive Estimation

This section contains a more general result for algorithmic modularity under self-predictive estimation (Theorem H.1), from which our main result is derived as a special case, along with associated background, applications, and proofs. This section is organized as follows.

- Appendix I.1 presents: definitions for the $\phi$-compressed POMDP and `CorruptionRobust` algorithms (Appendix I.1.1), statements for properties of the $\phi$-compressed dynamics (Appendix I.1.2). The risk bound for O2L under self-predictive estimation and `CorruptionRobustness` (Theorem H.1) is given in Appendix I.1.3, and a statement that the GOLF algorithm is `CorruptionRobust` (Appendix I.1.4).

- Appendix I.2 presents for the proofs for the properties of the $\phi$-compressed POMDPs.

- Appendix I.3 presents a proof for the risk bound of O2L under self-predictive estimation and `CorruptionRobustness`.

- Appendix I.4 presents a proof that the GOLF algorithm is `CorruptionRobust`.

## I.1   O2L with Self-predictive Estimation and `CorruptionRobust` Base Algorithms

### I.1.1   Definitions: $\phi$-compressed POMDP and `CorruptionRobustness`

Consider iteration $k \in [K]$ of epoch $t \in [T]$ within O2L. Suppose that REPLEARN has chosen decoder $\phi = \phi^{(t)} : \mathcal{X} \to \mathcal{S}$. Then, the latent algorithm has observed the data $\mathcal{D}^{(t,k)} = \{\phi(x_h^{(t,k)}), a_h^{(t,k)}, r_h^{(t,k)}, \phi(x_{h+1}^{(t,k)})\}$ collected from the preceding policies in the epoch: $\pi_{\mathtt{lat}}^{(t,1)} \circ \phi^{(t)}, \dots, \pi_{\mathtt{lat}}^{(t,k-1)} \circ \phi^{(t)}$ (Line 8). Due to possible inaccuracies in the decoder $\phi$, the dataset $\mathcal{D}^{(t,k)}$ may not be generated from a Markovian process and must instead be viewed as being generated from a PODMP, formally defined as follows.

**Definition I.1** ($\phi$-compressed POMDP). *The $\phi$-compressed POMDP $\widetilde{M}_\phi^\star$ induced by $M_{\mathtt{obs}}^\star$ and $\phi$ is defined by:*

1. *Latent state space $\mathcal{X}$*

2. *Action space $\mathcal{A}$*

3. *Observation state space $\mathcal{S}$*

4. *Latent reward functions $R_{\mathtt{obs},h}^\star : \mathcal{X} \times \mathcal{A} \to [0,1]$*

5. *Latent dynamics $P_{\mathtt{obs},h}^\star : \mathcal{X} \times \mathcal{A} \to \Delta(\mathcal{X})$*

6. *(Deterministic) observation function $\mathcal{O}_h : \mathcal{X} \to \mathcal{S}$ defined by $\mathcal{O}_h(x) = \phi_h(x)$,*

7. *Horizon $H$*

8. *Initial latent distribution $P_{\mathtt{obs}}^\star(x_0 \mid \emptyset)$*

Note that the latent space *for the POMDP* is the observation space of the latent-dynamics MDP $M_{\mathtt{obs}}^\star$, and vice-versa; we adopt this terminology because—from the perspective of the base algorithm, the observations $x_h$ can be viewed as a Markovian (yet partially observed process) that generates the learned states $\phi(x_h)$ on which the algorithm acts. We write $\widetilde{P}_\phi^{\pi_{\mathtt{lat}}} := \mathbb{P}^{\widetilde{M}_\phi^\star, \pi_{\mathtt{lat}}}$ for the probability distribution over trajectories $(x_h, s_h, a_h, r_h)_{h \in [H]}$ in the $\phi$-compressed POMDP when playing policy $\pi_{\mathtt{lat}} : \mathcal{S} \times [H] \to \Delta(\mathcal{A})$, where $x_h \in \mathcal{X}$ are the POMDP's latent states, $s_h \in \mathcal{S}$ are the observed states, and $a_h \in \mathcal{A}$ are the actions. We let $\widetilde{\mathbb{E}}_\phi^{\pi_{\mathtt{lat}}} := \mathbb{E}^{\widetilde{M}_\phi^\star, \pi_{\mathtt{lat}}}$ denote the corresponding expectation. We write $\widetilde{P}_{\phi,h}(s_{h+1} \mid s_{1:h}, a_{1:h}) = \widetilde{P}_\phi^{\pi_{\mathtt{lat}}}(s_{h+1} \mid s_{1:h}, a_{1:h})$ and $\tilde{r}_{\phi,h}(r_h \mid s_{1:h}, a_{1:h}) = \widetilde{P}_\phi^{\pi_{\mathtt{lat}}}(r_h \mid s_{1:h}, a_{1:h})$ for the conditional distributions of next states and rewards given the first $h$ state-action pairs, which are policy-independent. We also write $\widetilde{M}_\phi^\star(r_h, s_{h+1} \mid s_{1:h}, a_{1:h}) = \tilde{r}_{\phi,h}(r_h \mid s_{1:h}, a_{1:h})\widetilde{P}_{\phi,h}(s_{h+1} \mid s_{1:h}, a_{1:h})$ for the joint one-step probability. We will abbreviate $\widetilde{M}_\phi^\star(s_{1:h}, a_{1:h}) := \widetilde{M}_\phi^\star(r_h, s_{h+1} = \cdot \mid s_{1:h}, a_{1:h})$.

Note that for any $\pi_{\texttt{lat}}$, $\widetilde{P}_{\phi,h}^{\pi_{\texttt{lat}}}(s_{h+1} \mid s_h, a_h)$ is a well-defined (Markovian, policy-dependent) probability kernel, which is equivalent to

$$\widetilde{P}_{\phi,h}^{\pi_{\texttt{lat}}}(s_{h+1} \mid s_h, a_h) = \sum_{s_{1:h-1}, a_{1:h-1}} \widetilde{P}_{\phi,h}^{\pi_{\texttt{lat}}}(s_{1:h-1}, a_{1:h-1} \mid s_h, a_h) \widetilde{P}_{\phi,h}(s_{h+1} \mid s_{1:h}, a_{1:h}) \quad (67)$$

$$= \widetilde{\mathbb{E}}_{\phi}^{\pi_{\texttt{lat}}} \left[ \widetilde{P}_{\phi,h}(s_{h+1} \mid s_{1:h}, a_{1:h}) \mid s_h, a_h \right] \quad (68)$$

Similarly, $\tilde{r}_{\phi,h}^{\pi_{\texttt{lat}}}(r_h \mid s_h, a_h)$ is a Markovian and policy-dependent reward distribution which is equivalent to

$$\tilde{r}_{\phi,h}^{\pi_{\texttt{lat}}}(r_h \mid s_h, a_h) = \sum_{s_{1:h-1}, a_{1:h-1}} \widetilde{P}_{\phi,h}^{\pi_{\texttt{lat}}}(s_{1:h-1}, a_{1:h-1} \mid s_h, a_h) \tilde{r}_{\phi,h}(r_h \mid s_{1:h}, a_{1:h}) \quad (69)$$

$$= \widetilde{\mathbb{E}}_{\phi}^{\pi_{\texttt{lat}}}[\tilde{r}_{\phi,h}(r_h \mid s_{1:h}, a_{1:h}) \mid s_h, a_h]. \quad (70)$$

Finally, we let

$$\widetilde{M}_{\phi,h}^{\pi_{\texttt{lat}},\star}(r_h, s_{h+1} \mid s_h, a_h) = \widetilde{\mathbb{E}}_{\phi}^{\pi_{\texttt{lat}}} \left[ \widetilde{M}_{\phi}^{\star}(r_h, s_{h+1} \mid s_{1:h}, a_{1:h}) \mid s_h, a_h \right] \quad (71)$$

denote the associated one-step model over joint rewards and transitions.

Our `CorruptionRobustness` condition asserts that the agent—when observing data from the $\phi^{(t)}$-compressed dynamics $\widetilde{M}_{\phi}^{\star}$—attains a risk bound for $M_{\texttt{lat}}$ which is proportional to its risk when observing data from $M_{\texttt{lat}}$ itself, plus a term that captures the degree of misspecification between $\widetilde{M}_{\phi}^{\star}$ and $M_{\texttt{lat}}$.

**Definition I.2** (`CorruptionRobust` algorithm). *We say that* $\text{ALG}_{\texttt{lat}}$ *is* `CorruptionRobust` *with parameters* $\alpha$ *and* $\text{Risk}_{\texttt{base}}$ *if there exists a constant* $c_1$ *such that, for any* $(\phi, M_{\texttt{lat}}) \in \Phi \times \mathcal{M}_{\texttt{lat}}$, *we have*

$$\mathbb{E}^{\widetilde{M}_{\phi}^{\star}, \text{ALG}_{\texttt{lat}}}[\text{Risk}(K, \text{ALG}_{\texttt{lat}}, M_{\texttt{lat}})] \leq c_1 \cdot \text{Risk}_{\texttt{base}}(K)$$

$$+ \alpha \, \mathbb{E}^{\widetilde{M}_{\phi}^{\star}, \text{ALG}_{\texttt{lat}}} \left[ \sqrt{\sum_{k=1}^{K} \sum_{h=1}^{H} \mathbb{E}_{\pi_{\texttt{lat}}^{(k)} \sim p^{(k)}} \widetilde{\mathbb{E}}_{\phi}^{\pi_{\texttt{lat}}^{(k)}} \left[ D_{\mathsf{H}}^2 \left( M_{\texttt{lat},h}(s_h, a_h), \widetilde{M}_{\phi,h}^{\star}(s_{1:h}, a_{1:h}) \right) \right] } \right],$$

*where we recall the definition of the random variable* $\text{Risk}(K, \text{ALG}_{\texttt{lat}}, M_{\texttt{lat}})$ *from Eq. (1), the expectation* $\mathbb{E}^{\widetilde{M}_{\phi}^{\star}, \text{ALG}_{\texttt{lat}}}$ *denotes the interaction protocol of* $\text{ALG}_{\texttt{lat}}$ *in the* $\phi$*-compressed dynamics* $\widetilde{M}_{\phi}^{\star}$, *and* $p^{(k)}$ *denotes the randomization distribution over latent policies that* $\text{ALG}_{\texttt{lat}}$ *plays.*

### I.1.2 Basic properties of the $\phi$-compressed dynamics (Definition I.1)

We establish a number of basic properties for the $\phi$-compressed POMDP and their relation to the self-prediction guarantee obtained by $\text{REP}_{\texttt{self;opt}}$. These properties are proved in Appendix I.2. Firstly, we have the following change-of-measure lemma:

**Lemma I.1** (Change of measure lemma). *For any* $\phi \in \Phi$, $f \in [\mathcal{S} \times \mathcal{A} \to [0,1]]$, $h \in [H]$, *and* $\pi_{\texttt{lat}} \in [\mathcal{S} \times [H] \to \Delta(\mathcal{A})]$, *we have:*

$$\widetilde{\mathbb{E}}_{\phi}^{\pi_{\texttt{lat}}}[f(s_h, a_h)] = \mathbb{E}^{\pi_{\texttt{lat}} \circ \phi}[[f \circ \phi](x_h, a_h)]. \quad (72)$$

The next lemma states that the kernels of the $\phi$-compressed POMDP are well-approximated by the (Markovian) latent model fit by $\text{REP}_{\texttt{self;opt}}$. We recall the instantaneous self-prediction error

$$[\Delta_h(M_{\texttt{lat}}, \phi)](x_h, a_h) := D_{\mathsf{H}}^2 \left( M_{\texttt{lat},h}(\phi_h(x_h), a_h), [\phi_{h+1} \sharp M_{\texttt{obs},h}^{\star}](x_h, a_h) \right).$$

**Lemma I.2** (Near-markovianity of the $\phi$-compressed dynamics). *For any decoder* $\phi$, *base model* $M_{\texttt{lat}}$, *and policy* $\pi_{\texttt{lat}} : \mathcal{S} \times [H] \to \Delta(\mathcal{A})$, *we have:*

$$\sum_{h=0}^{H} \widetilde{\mathbb{E}}_{\phi}^{\pi_{\texttt{lat}}} \left[ D_{\mathsf{H}}^2 \left( M_{\texttt{lat},h}(s_h, a_h), \widetilde{M}_{\phi,h}^{\star}(s_{1:h}, a_{1:h}) \right) \right] \leq \sum_{h=0}^{H} \mathbb{E}^{\pi_{\texttt{lat}} \circ \phi}[[\Delta_h(M_{\texttt{lat}}, \phi)](x_h, a_h)]. \quad (73)$$

*Furthermore, we also have*

$$\sum_{h=0}^{H} \widetilde{\mathbb{E}}_{\phi}^{\pi_{\mathrm{lat}}}\left[D_{\mathsf{H}}^2\left(M_{\mathrm{lat},h}(s_h,a_h), \widetilde{M}_{\phi,h}^{\star,\pi_{\mathrm{lat}}}(s_h,a_h)\right)\right] \leq \sum_{h=0}^{H} \mathbb{E}^{\pi_{\mathrm{lat}}\circ\phi}[[\Delta_h(M_{\mathrm{lat}},\phi)](x_h,a_h)]. \quad (74)$$

A corollary is the following lemma establishing errors between expectations under $M_{\mathrm{lat}}$, the model estimated by $\mathrm{REP}_{\mathrm{self;opt}}$, and those under the $\phi$-compressed POMDP $\widetilde{M}_{\phi}^{\star}$.

**Lemma I.3** (Simulation lemma). *For any latent model $M_{\mathrm{lat}}$ with Markovian transition kernel $\{P_{\mathrm{lat},h}\}_{h\in[H]}$, latent policy $\pi_{\mathrm{lat}} : \mathcal{S} \times [H] \to \Delta(\mathcal{A})$, and decoder $\phi \in \Phi$, we have that for all $f : \mathcal{S} \times \mathcal{A} \to [0,1]$:*

$$|\mathbb{E}^{M_{\mathrm{lat}},\pi_{\mathrm{lat}}}[f(s_h,a_h)] - \widetilde{\mathbb{E}}_{\phi}^{\pi_{\mathrm{lat}}}[f(s_h,a_h)]|$$
$$\leq \sum_{h'<h} \mathbb{E}^{\pi_{\mathrm{lat}}\circ\phi}\left[\left\|[P_{\mathrm{lat}}\circ\phi]_h(x_{h'},a_{h'}) - \phi_{h+1}\sharp P_{\mathrm{obs},h}^{\star}(x_{h'},a_{h'})\right\|_{\mathrm{tv}}\right], \quad (75)$$

*and thus for any sequence of policies $\pi_{\mathrm{lat}}^{(t)}$, latent models $M_{\mathrm{lat}}^{(t)}$, and decoders $\phi^{(t)}$, we have:*

$$\sum_{t=1}^{T}\sum_{h=0}^{H}|\mathbb{E}^{M_{\mathrm{lat}}^{(t)},\pi_{\mathrm{lat}}^{(t)}}[f(s_h,a_h)] - \widetilde{\mathbb{E}}_{\phi^{(t)}}^{\pi_{\mathrm{lat}}^{(t)}}[f(s_h,a_h)]| \quad (76)$$

$$\leq H\sqrt{TH}\sqrt{\sum_{t=1}^{T}\sum_{h=0}^{H}\mathbb{E}^{\pi_{\mathrm{lat}}^{(t)}\circ\phi^{(t)}}\left[[\Delta_h(M_{\mathrm{lat}}^{(t)},\phi^{(t)})](x_h,a_h)\right]}. \quad (77)$$

### I.1.3 Risk bound for O2L under `CorruptionRobustness`

We state the main risk bound for O2L under self-predictive estimation and the above definition of corruption robustness.

**Theorem H.1** (Risk bound for O2L under self-predictive estimation and `CorruptionRobustness`). *Assume $\mathrm{REP}_{\mathrm{self;opt}}$ satisfies [Assumption A.1] with parameter $\gamma > 0$ and that $\mathcal{M}_{\mathrm{lat}}$ is realizable (i.e. $M_{\mathrm{lat}}^{\star} \in \mathcal{M}_{\mathrm{lat}}$). Furthermore, let $\mathrm{ALG}_{\mathrm{lat}}$ be `CorruptionRobust` ([Definition I.2]) with parameter $\alpha$. Then, O2L ([Algorithm 1]) with inputs $T, K, \Phi, \mathrm{ALG}_{\mathrm{lat}}$, and $\mathrm{REP}_{\mathrm{self;opt}}$ has expected risk*

$$\mathbb{E}[\mathrm{Risk}_{\mathrm{obs}}(TK)] \leq c_1 \cdot \mathrm{Risk}_{\mathrm{base}}(K) + c_2\gamma \cdot \frac{K}{T}\mathrm{Est}_{\mathrm{self;opt}}(T,\gamma) + c_3\gamma^{-1} \cdot \left(\alpha^2 + H\right) \quad (66)$$

*for absolute constants $c_1, c_2, c_3 > 0$.*

### I.1.4 Examples of `CorruptionRobust` algorithms

In this section, we establish that the GOLF algorithm satisfies the `CorruptionRobust` definition ([Definition I.2]) with a parameter $\alpha \approx K^{-1/2}$. This improves upon the rate that would be obtained by invoking the generic guarantee in [Lemma H.4]. We expect that several other algorithms can be analyzed in a similar way, thereby leading to tight rates in the same fashion. We restate the pseudocode in [Algorithm 5] for convenience.

Let $M_{\mathrm{lat}} = (r_{\mathrm{lat}}, P_{\mathrm{lat}})$ be given, and we let $Q_{\mathrm{lat}}^{\star} := Q^{M_{\mathrm{lat}},\star}$, and $\mathcal{T}_{\mathrm{lat},h}f(s,a) := r_{\mathrm{lat},h}(s,a) + \mathbb{E}_{s'\sim P_{\mathrm{lat},h}(s,a)}[V_f(s')]$. We assume that the algorithm has a latent function class $\mathcal{F}_{\mathrm{alg}}$ which realizes $Q_{\mathrm{lat}}^{\star}$, as well as a helper class $\mathcal{G}_{\mathrm{alg}}$ which is $\mathcal{T}_{\mathrm{lat}}$-complete for $\mathcal{F}_{\mathrm{alg}}$.

**Assumption I.1** ($\mathcal{T}_{\mathrm{lat}}$-completeness). *We have:*

$$Q_{\mathrm{lat}}^{\star} \in \mathcal{F}_{\mathrm{alg}}, \quad \text{and} \quad \mathcal{T}_{\mathrm{lat}}\mathcal{F}_{\mathrm{alg}} \subseteq \mathcal{G}_{\mathrm{alg}}.$$

For our analysis of GOLF, it is most natural to quantify the corruption levels in the following way.

**Assumption I.2** (Corruption levels of $M_{\mathrm{lat}}$ and $\widetilde{M}_{\phi}^{\star}$). *Let $\varepsilon_{\mathrm{rep}}^2$ be such that, for any sequence of policies $\pi_{\mathrm{lat}}^{(k)}$ played by the algorithm when interacting with the $\phi$-compressed POMDP, we have*

$$\sum_{k=1}^{K}\sum_{h=1}^{H}\mathbb{E}_{\pi_{\mathrm{lat}}^{(k)}\sim p_{\mathrm{lat}}^{(k)}}\widetilde{\mathbb{E}}_{\phi}^{\pi_{\mathrm{lat}}^{(k)}}\left[(r_{\mathrm{lat},h}(s_h,a_h) - \tilde{r}_{\phi,h}^{(k)}(s_h,a_h))^2 + \left\|P_{\mathrm{lat},h}(s_h,a_h) - \widetilde{P}_{\phi,h}^{\pi_{\mathrm{lat}}^{(k)}}(s_h,a_h)\right\|_{\mathrm{tv}}^2\right]$$
$$\leq \varepsilon_{\mathrm{rep}}^2. \quad (78)$$

---

**Algorithm 5** GOLF [JLM21]

---

**input:** Function classes $\mathcal{F}$ and $\mathcal{G}$, confidence width $\beta > 0$.
**initialize:** $\mathcal{F}^{(0)} \leftarrow \mathcal{F}, \mathcal{D}_h^{(0)} \leftarrow \emptyset \;\; \forall h \in [H]$.

1: **for** episode $t = 1, 2, \ldots, T$ **do**
2:       Select policy $\pi^{(t)} \leftarrow \pi_{f^{(t)}}$, where $f^{(t)} := \arg\max_{f \in \mathcal{F}^{(t-1)}} f(x_1, \pi_{f,1}(x_1))$.
3:       Execute $\pi^{(t)}$ for one episode and obtain trajectory $(x_1^{(t)}, a_1^{(t)}, r_1^{(t)}), \ldots, (x_H^{(t)}, a_H^{(t)}, r_H^{(t)})$.
4:       Update dataset: $\mathcal{D}_h^{(t)} \leftarrow \mathcal{D}_h^{(t-1)} \cup \left\{ \left(x_h^{(t)}, a_h^{(t)}, x_{h+1}^{(t)}\right) \right\} \;\; \forall h \in [H]$.
5:       Compute confidence set:

$$\mathcal{F}^{(t)} \leftarrow \left\{ f \in \mathcal{F} : \mathcal{L}_h^{(t)}(f_h, f_{h+1}) - \min_{g_h \in \mathcal{G}_h} \mathcal{L}_h^{(t)}(g_h, f_{h+1}) \leq \beta \;\; \forall h \in [H] \right\},$$

$$\text{where} \quad \mathcal{L}_h^{(t)}(f, f') := \sum_{(x,a,r,x') \in \mathcal{D}_h^{(t)}} \left( f(x,a) - r - \max_{a' \in \mathcal{A}} f'(x', a') \right)^2, \; \forall f, f' \in \mathcal{F}.$$

6: **end for**
7: Output $\widehat{\pi} = \mathtt{Unif}(\pi^{(1:T)})$.

---

We note that

$$\varepsilon_{\mathsf{rep}}^2 \lesssim \sum_{k=1}^{K} \sum_{h=1}^{H} \widetilde{\mathbb{E}}_\phi^{\pi_{\mathsf{lat}}^{(k)}} \left[ D_{\mathsf{H}}^2 \left( M_{\mathsf{lat},h}(s_h, a_h), \widetilde{M}_{\phi,h}^{\star, \pi_{\mathsf{lat}}^{(k)}}(s_h, a_h) \right) \right]$$

$$\leq \sum_{k=1}^{K} \sum_{h=1}^{H} \widetilde{\mathbb{E}}_\phi^{\pi_{\mathsf{lat}}^{(k)}} \left[ D_{\mathsf{H}}^2 \left( M_{\mathsf{lat},h}(s_h, a_h), \widetilde{M}_{\phi,h}^{\star}(s_{1:h}, a_{1:h}) \right) \right]$$

by the data-processing inequality (cf. Eq. (90) and Eq. (80)) and the inequality $\|p - q\|_{\mathsf{tv}}^2 \leq D_{\mathsf{H}}^2(p, q)$, and thus a CorruptionRobustness bound in terms of $\varepsilon_{\mathsf{rep}}$ implies a CorruptionRobustness bound in the sense of Definition I.2.

**Theorem I.1** (Latent GOLF is CorruptionRobust). *Under Assumption I.1 and Assumption I.2, Algorithm 5 with $\beta = c\left(\log\left(|\mathcal{F}||\mathcal{G}|KH\delta^{-1}\right) + \varepsilon_{\mathsf{rep}}\right)$, has regret*

$$\sum_{k=1}^{K} J^{M_{\mathsf{lat}}}(\pi_{M_{\mathsf{lat}}}) - J^{M_{\mathsf{lat}}}(\pi^{(k)}) \leq \mathcal{O}\left( H \sqrt{C_{\mathsf{cov}} K \log(K) \log(|\mathcal{F}||\mathcal{G}|HK/\delta)} \right)$$

$$+ \mathcal{O}\left( H^{3/2} \sqrt{K C_{\mathsf{cov}} \log(K) \varepsilon_{\mathsf{rep}}^2} \right),$$

*and consequently is CorruptionRobust (Definition I.2) with parameters*

$$\alpha = \frac{H^{3/2}}{\sqrt{K}} \sqrt{C_{\mathsf{cov}} \log(K)} \; and \; \mathtt{Risk}_{\mathsf{base}}(K) = \mathcal{O}\left( \frac{H}{\sqrt{K}} \sqrt{C_{\mathsf{cov}} \log(K) \log(|\mathcal{F}||\mathcal{G}|HK)} \right).$$

**Corollary I.1** (GOLF applied in O2L). *Let us suppose that the appropriate assumptions for the estimator in Algorithm 4 to have regret bounded by $\mathtt{Est}_{\mathsf{self}}(T, \gamma) = \mathcal{O}\left(\sqrt{H C_{\mathsf{cov}} T} \log(C_{\mathsf{cov}} |\mathcal{M}_{\mathsf{lat}}||\mathcal{L}_{\mathsf{lat}}||\Phi|HT)\right)$ (Lemma A.1) hold. Then, we can take $\gamma \approx K^{-1/2}$ and $T \approx K^4$, and the bound Theorem H.1 gives an expected risk of $\varepsilon$ with a number of trajectories $TK = \mathtt{poly}(C_{\mathsf{cov}}, H, \log|\mathcal{M}_{\mathsf{lat}}|, \log|\Phi|, \log|\mathcal{L}_{\mathsf{lat}}|) \cdot 1/\varepsilon^{10}$, improving over the $1/\varepsilon^{14}$ rate of the universal result (Corollary A.1).*

### I.2 Proofs for Appendix I.1.2: Properties of $\phi$-compressed POMDPs

**Lemma I.1** (Change of measure lemma). *For any $\phi \in \Phi$, $f \in [\mathcal{S} \times \mathcal{A} \to [0, 1]]$, $h \in [H]$, and $\pi_{\mathsf{lat}} \in [\mathcal{S} \times [H] \to \Delta(\mathcal{A})]$, we have:*

$$\widetilde{\mathbb{E}}_\phi^{\pi_{\mathsf{lat}}}[f(s_h, a_h)] = \mathbb{E}^{\pi_{\mathsf{lat}} \circ \phi}[[f \circ \phi](x_h, a_h)]. \tag{72}$$

**Proof of Lemma I.1.** Recall that $\widetilde{P}_\phi^{\pi_{\mathsf{lat}}}$ denotes the law of $(x_h, s_h, a_h)_{h \in [H]}$ in the $\phi$-compressed POMDP when playing policy $\pi_{\mathsf{lat}}$. For clarity, and to differentiate a random variable from its realization, in the proofs below we will use upper-case notation such as $\{S_h = s_h, A_h = a_h, X_h = x_h\}$ to indicate realizations of random variables in the POMDP.

Let $\tilde{d}_h^{\pi_{\text{lat}}}(s,a) = \widetilde{P}_\phi^{\pi_{\text{lat}}}(S_h = s, A_h = a)$ be the marginalized occupancy measure for in the $\phi$-compressed POMDP $\widetilde{M}_\phi^\star$. We write $d_h^{\pi_{\text{lat}} \circ \phi} := d_h^{M_{\text{obs}}^\star, \pi_{\text{lat}} \circ \phi}$. The left-hand side in Eq. (72) is equal to:

$$\widetilde{\mathbb{E}}_\phi^{\pi_{\text{lat}}}[f(s_h, a_h)] = \sum_{s \in \mathcal{S}, a \in \mathcal{A}} \tilde{d}_h^{\pi_{\text{lat}}}(s,a) f(s,a),$$

Meanwhile, the right-hand side is equal to:

$$\mathbb{E}_h^{\pi_{\text{lat}} \circ \phi}[[f \circ \phi](x_h, a_h)] = \sum_{s \in \mathcal{S}, a \in \mathcal{A}} f(s,a) \sum_{x:\phi(x)=s} d_h^{\pi_{\text{lat}} \circ \phi}(x,a).$$

So it only remains to show that, for each $s \in \mathcal{S}$ and $a \in \mathcal{A}$, we have $\tilde{d}_h^{\pi_{\text{lat}}}(s,a) = \sum_{x:\phi(x)=s} d_h^{\pi_{\text{lat}} \circ \phi}(x,a)$. Firstly, note that it is enough to show that $\sum_{x_h:\phi(x_h)=s_h} d_h^{\pi_{\text{lat}} \circ \phi}(x_h) = \tilde{d}_h^{\pi_{\text{lat}}}(s_h)$, since $\tilde{d}_h^{\pi_{\text{lat}}}(s_h, a_h) = \tilde{d}_h^{\pi_{\text{lat}}}(s_h) \pi_{\text{lat}}(a_h \mid s_h)$ and $\sum_{x_h:\phi(x_h)=s_h} d_h^{\pi_{\text{lat}} \circ \phi}(x_h, a_h) = \sum_{x_h:\phi(x_h)=s_h} d_h^{\pi_{\text{lat}} \circ \phi}(x_h) \pi_{\text{lat}}(a_h \mid \phi(x_h)) = \pi_{\text{lat}}(a_h \mid s_h) \sum_{x_h:\phi(x_h)=s_h} d_h^{\pi_{\text{lat}} \circ \phi}(x_h)$. Toward this, we have:

$$\sum_{x_h:\phi(x_h)=s_h} d_h^{\pi_{\text{lat}} \circ \phi}(x_h) = \sum_{x_h:\phi(x_h)=s_h} \sum_{x_{h-1}, a_{h-1} \in \mathcal{X} \times \mathcal{A}} d_{h-1}^{\pi_{\text{lat}} \circ \phi}(x_{h-1}, a_{h-1}) P_{\text{obs},h}^\star(x_h \mid x_{h-1}, a_{h-1})$$

$$= \sum_{x_{h-1}, a_{h-1} \in \mathcal{X} \times \mathcal{A}} d_{h-1}^{\pi_{\text{lat}} \circ \phi}(x_{h-1}, a_{h-1}) \sum_{x_h:\phi(x_h)=s_h} P_{\text{obs},h}^\star(x_h \mid x_{h-1}, a_{h-1})$$

$$= \sum_{x_{h-1}, a_{h-1} \in \mathcal{X} \times \mathcal{A}} d_{h-1}^{\pi_{\text{lat}} \circ \phi}(x_{h-1}, a_{h-1}) P_{\text{obs},h}^\star(\phi(x_h) = s_h \mid x_{h-1}, a_{h-1})$$

At the same time,

$$\tilde{d}_h^{\pi_{\text{lat}}}(s_h) = \widetilde{P}_\phi^{\pi_{\text{lat}}}(S_h = s_h)$$

$$= \sum_{\tilde{x}, \tilde{a}} \widetilde{P}_\phi^{\pi_{\text{lat}}}(X_{h-1} = \tilde{x}, A_{h-1} = \tilde{a}) \widetilde{P}_\phi^{\pi_{\text{lat}}}(S_h = s_h \mid X_{h-1} = \tilde{x}, A_{h-1} = \tilde{a})$$

$$= \sum_{\tilde{x}, \tilde{a}} \widetilde{P}_\phi^{\pi_{\text{lat}}}(X_{h-1} = \tilde{x}, A_{h-1} = \tilde{a}) P_{\text{obs}}^\star(\phi(x_h) = s_h \mid x_{h-1}, a_{h-1}),$$

where in the second equality we have used the definition of the observation function $s_h = \mathcal{O}(x_h) = \phi(x_h)$.

To conclude, it remains to show that for all $h$, we have:

$$d_h^{\pi_{\text{lat}} \circ \phi}(x_h, a_h) = \widetilde{P}_\phi^{\pi_{\text{lat}}}(X_h = x_h, A_h = a_h).$$

We do this by induction. Again, note that it is sufficient to establish $d_h^{\pi_{\text{lat}} \circ \phi}(x_h) = \widetilde{P}_\phi^{\pi_{\text{lat}}}(X_h = x_h)$. The case $h = 1$ is clear. For the general case, we have:

$$d_h^{\pi_{\text{lat}} \circ \phi}(x_h) = \sum_{x_{h-1}, a_{h-1} \in \mathcal{X} \times \mathcal{A}} d_{h-1}^{\pi_{\text{lat}} \circ \phi}(x_{h-1}, a_{h-1}) P_{\text{obs}}^\star(x_h \mid x_{h-1}, a_{h-1})$$

$$= \sum_{x_{h-1}, a_{h-1} \in \mathcal{X} \times \mathcal{A}} \widetilde{P}_\phi^{\pi_{\text{lat}}}(X_h = x_{h-1}, A_{h-1} = a_{h-1}) P_{\text{obs}}^\star(x_h \mid x_{h-1}, a_{h-1})$$

$$= \sum_{x_{h-1}, a_{h-1} \in \mathcal{X} \times \mathcal{A}} \widetilde{P}_\phi^{\pi_{\text{lat}}}(X_h = x_{h-1}, A_{h-1} = a_{h-1})$$

$$\times \widetilde{P}_\phi^{\pi_{\text{lat}}}(X_h = x_h \mid X_{h-1} = x_{h-1}, A_{h-1} = a_{h-1})$$

$$= \widetilde{P}_\phi^{\pi_{\text{lat}}}(X_h = x_h).$$

$\square$

**Lemma I.2** (Near-markovianity of the $\phi$-compressed dynamics). *For any decoder $\phi$, base model $M_{\mathrm{lat}}$, and policy $\pi_{\mathrm{lat}} : \mathcal{S} \times [H] \to \Delta(\mathcal{A})$, we have:*

$$\sum_{h=0}^{H} \widetilde{\mathbb{E}}_{\phi}^{\pi_{\mathrm{lat}}} \left[ D_{\mathsf{H}}^2 \left( M_{\mathrm{lat},h}(s_h, a_h), \widetilde{M}_{\phi,h}^{\star}(s_{1:h}, a_{1:h}) \right) \right] \leq \sum_{h=0}^{H} \mathbb{E}^{\pi_{\mathrm{lat}} \circ \phi}[[\Delta_h(M_{\mathrm{lat}}, \phi)](x_h, a_h)]. \quad (73)$$

*Furthermore, we also have*

$$\sum_{h=0}^{H} \widetilde{\mathbb{E}}_{\phi}^{\pi_{\mathrm{lat}}} \left[ D_{\mathsf{H}}^2 \left( M_{\mathrm{lat},h}(s_h, a_h), \widetilde{M}_{\phi,h}^{\star, \pi_{\mathrm{lat}}}(s_h, a_h) \right) \right] \leq \sum_{h=0}^{H} \mathbb{E}^{\pi_{\mathrm{lat}} \circ \phi}[[\Delta_h(M_{\mathrm{lat}}, \phi)](x_h, a_h)]. \quad (74)$$

**Proof of Lemma I.2.** We begin with the first event. Note that, for any $\pi_{\mathrm{lat}}$, the PODMP kernel $\widetilde{M}_{\phi,h}^{\star}(r_h, s_{h+1} = \cdot \mid s_{1:h}, a_{1:h})$ can be written as:

$$\widetilde{M}_{\phi,h}^{\star}(r_h, s_{h+1} = \cdot \mid s_{1:h}, a_{1:h}) = \sum_{x_h, a_h \in \mathcal{X} \times \mathcal{A}} \widetilde{P}_{\phi}^{\pi_{\mathrm{lat}}}(r_h, s_{h+1} = \cdot \mid x_h, a_h, s_{1:h}, a_{1:h})$$

$$\times \widetilde{P}_{\phi}^{\pi_{\mathrm{lat}}}(x_h, a_h \mid s_{1:h}, a_{1:h})$$

$$= \sum_{x_h, a_h \in \mathcal{X} \times \mathcal{A}} \widetilde{P}_{\phi}^{\pi_{\mathrm{lat}}}(r_h, s_{h+1} = \cdot \mid x_h, a_h) \widetilde{P}_{\phi}^{\pi_{\mathrm{lat}}}(x_h, a_h \mid s_{1:h}, a_{1:h}),$$

where we have used $\widetilde{M}(r_h, s_{h+1} = \cdot \mid s_{1:h}, a_{1:h}) = \widetilde{P}_{\phi}^{\pi_{\mathrm{lat}}}(r_h, s_{h+1} = \cdot \mid s_{1:h}, a_{1:h})$, the law of total probability, and that $x_h, a_h$ is a sufficient statistic for $r_h$ and $s_{h+1}$. We further note that

$$\widetilde{P}_{\phi}^{\pi_{\mathrm{lat}}}(r_h, s_{h+1} = \cdot \mid x_h, a_h) = M_{\mathrm{obs},h}^{\star}(r_h, \phi_{h+1}(x_{h+1}) = \cdot \mid x_h, a_h), \quad (79)$$

since $s_{h+1} = \mathcal{O}_{h+1}(x_{h+1}) = \phi_{h+1}(x_{h+1})$ is a deterministic function of $x_{h+1}$ and $r_h, x_{h+1} \sim M_{\mathrm{obs},h}^{\star}(x_h, a_h)$. Thus, for a fixed $h$ and $t$, and omitting the $h$ indices on the decoder $\phi$ for cleanliness, the expectation in equation Eq. (73) becomes:

$$\widetilde{\mathbb{E}}_{\phi}^{\pi_{\mathrm{lat}}} \left[ D_{\mathsf{H}}^2 \left( M_{\mathrm{lat},h}(s_h, a_h), \widetilde{M}_{\phi,h}^{\star}(r_h, s_{h+1} = \cdot \mid s_{1:h}, a_{1:h}) \right) \right]$$

$$\leq \sum_{s_{1:h}, a_{1:h} \in (\mathcal{S} \times \mathcal{A})^h} \widetilde{P}_{\phi}^{\pi_{\mathrm{lat}}}(s_{1:h}, a_{1:h}) \sum_{x_h, a_h} \widetilde{P}_{\phi}^{\pi_{\mathrm{lat}}}(x_h, a_h \mid s_{1:h}, a_{1:h})$$

$$\times D_{\mathsf{H}}^2 \left( M_{\mathrm{lat},h}(s_h, a_h), \widetilde{P}_{\phi}^{\pi_{\mathrm{lat}}}(r_h, s_{h+1} = \cdot \mid x_h, a_h) \right) \quad \text{(Jensen)}$$

$$= \sum_{\substack{s_{1:h}, a_{1:h} \in (\mathcal{S} \times \mathcal{A})^h \\ x_h, a_h \in \mathcal{X} \times \mathcal{A}}} \widetilde{P}_{\phi}^{\pi_{\mathrm{lat}}}(s_{1:h}, a_{1:h}) \widetilde{P}_{\phi}^{\pi_{\mathrm{lat}}}(x_h, a_h \mid s_{1:h}, a_{1:h})$$

$$\times D_{\mathsf{H}}^2 \left( M_{\mathrm{lat},h}(\phi(x_h), a_h), \widetilde{P}_{\phi}^{\pi_{\mathrm{lat}}}(r_h, \phi(x_{h+1}) = \cdot \mid x_h, a_h) \right)$$

$$= \sum_{x_h, a_h \in \mathcal{X} \times \mathcal{A}} \widetilde{P}_{\phi}^{\pi_{\mathrm{lat}}}(x_h, a_h) D_{\mathsf{H}}^2 \left( M_{\mathrm{lat}}(\phi(x_h), a_h), M_{\mathrm{obs},h}^{\star}(r_h, \phi(x_{h+1}) = \cdot \mid x_h, a_h) \right)$$

$$\text{(Simplifying \& Eq. (79))}$$

$$= \mathbb{E}^{\pi_{\mathrm{lat}} \circ \phi} \left[ D_{\mathsf{H}}^2 \left( M_{\mathrm{lat}}(\phi(x_h), a_h), M_{\mathrm{obs},h}^{\star}(r_h, \phi(x_{h+1}) = \cdot \mid x_h, a_h) \right) \right]$$

$$\text{(Change of measure (Lemma I.1))}$$

$$= \mathbb{E}^{\pi_{\mathrm{lat}} \circ \phi} \left[ D_{\mathsf{H}}^2 \left( M_{\mathrm{lat}}(\phi(x_h), a_h), \phi \sharp M_{\mathrm{obs},h}^{\star}(x_h, a_h) \right) \right], \quad \text{(By definition of } \phi \sharp M_{\mathrm{obs}}^{\star})$$

as desired. Summing over $h \in [H]$ we obtain the desired bound. The bound Eq. (74) is a consequence of Eq. (73) and the data-processing inequality. Namely, using the definition of $\widetilde{M}_{\phi,h}^{\star, \pi_{\mathrm{lat}}}$ from Eq. (71) and the joint convexity of the squared Hellinger distance we have:

$$D_{\mathsf{H}}^2 \left( M_{\mathrm{lat},h}(\cdot \mid s_h, a_h), \widetilde{M}_{\phi,h}^{\star, \pi_{\mathrm{lat}}}(\cdot \mid s_h, a_h) \right)$$

$$\leq \widetilde{\mathbb{E}}_{\phi}^{\pi_{\mathrm{lat}}} \left[ D_{\mathsf{H}}^2 \left( M_{\mathrm{lat},h}(\cdot \mid s_h, a_h), \widetilde{M}_{\phi,h}^{\star}(\cdot \mid s_{1:h}, a_{1:h}) \right) \mid s_h, a_h \right]. \quad (80)$$

Thus, we have

$$\mathbb{E}_\phi^{\pi_{\text{lat}}}\left[D_{\mathsf{H}}^2\Big(M_{\text{lat},h}(\cdot \mid s_h, a_h), \widetilde{M}_{\phi,h}^{\star,\pi_{\text{lat}}}(\cdot \mid s_h, a_h)\Big)\right]$$

$$\leq \mathbb{E}_\phi^{\pi_{\text{lat}}}\left[\mathbb{E}_\phi^{\pi_{\text{lat}}}\left[D_{\mathsf{H}}^2\Big(M_{\text{lat},h}(\cdot \mid s_h, a_h), \widetilde{M}_{\phi,h}^\star(\cdot \mid s_{1:h}, a_{1:h})\Big) \mid s_h, a_h\right]\right]$$

$$= \mathbb{E}_\phi^{\pi_{\text{lat}}}\left[D_{\mathsf{H}}^2\Big(M_{\text{lat},h}(\cdot \mid s_h, a_h), \widetilde{M}_{\phi,h}^\star(\cdot \mid s_{1:h}, a_{1:h})\Big)\right],$$

as desired. $\qquad\qquad\qquad\qquad\qquad\qquad\qquad\qquad\qquad\qquad\qquad\qquad\qquad\qquad\qquad\qquad\square$

**Lemma I.3** (Simulation lemma). *For any latent model $M_{\text{lat}}$ with Markovian transition kernel $\{P_{\text{lat},h}\}_{h\in[H]}$, latent policy $\pi_{\text{lat}} : \mathcal{S} \times [H] \to \Delta(\mathcal{A})$, and decoder $\phi \in \Phi$, we have that for all $f : \mathcal{S} \times \mathcal{A} \to [0,1]$:*

$$|\mathbb{E}^{M_{\text{lat}},\pi_{\text{lat}}}[f(s_h, a_h)] - \widetilde{\mathbb{E}}_\phi^{\pi_{\text{lat}}}[f(s_h, a_h)]|$$

$$\leq \sum_{h'<h} \mathbb{E}^{\pi_{\text{lat}}\circ\phi}\Big[\big\|[P_{\text{lat}} \circ \phi]_h(x_{h'}, a_{h'}) - \phi_{h+1}\sharp P_{\text{obs},h}^\star(x_{h'}, a_{h'})\big\|_{\text{tv}}\Big], \qquad (75)$$

*and thus for any sequence of policies $\pi_{\text{lat}}^{(t)}$, latent models $M_{\text{lat}}^{(t)}$, and decoders $\phi^{(t)}$, we have:*

$$\sum_{t=1}^T \sum_{h=0}^H |\mathbb{E}^{M_{\text{lat}}^{(t)},\pi_{\text{lat}}^{(t)}}[f(s_h, a_h)] - \widetilde{\mathbb{E}}_{\phi^{(t)}}^{\pi_{\text{lat}}^{(t)}}[f(s_h, a_h)]| \qquad\qquad (76)$$

$$\leq H\sqrt{TH}\sqrt{\sum_{t=1}^T \sum_{h=0}^H \mathbb{E}^{\pi_{\text{lat}}^{(t)}\circ\phi^{(t)}}\big[[\Delta_h(M_{\text{lat}}^{(t)}, \phi^{(t)})](x_h, a_h)\big]}. \qquad (77)$$

**Proof of Lemma I.3.** Firstly note that, from Lemma I.1, the left-hand-side of Eq. (75) is equivalent to

$$|\mathbb{E}^{M_{\text{lat}},\pi_{\text{lat}}}[f(s_h, a_h)] - \widetilde{\mathbb{E}}_\phi^{\pi_{\text{lat}}}[f(s_h, a_h)]| = |\mathbb{E}^{M_{\text{lat}},\pi_{\text{lat}}}[f(s_h, a_h)] - \mathbb{E}^{M_{\text{obs}}^\star,\pi_{\text{lat}}\circ\phi}[[f \circ \phi](x_h, a_h)]|$$
$$(81)$$

For any $\pi_{\text{lat}} : \mathcal{S} \times [H] \to \Delta(\mathcal{A})$, let $d_{\text{lat},h}^{\pi_{\text{lat}}} = d_h^{M_{\text{lat}},\pi_{\text{lat}}}$ denote the occupancy in $M_{\text{lat}}$, and similarly for any $\pi_{\text{obs}} : \mathcal{X} \times [H] \to \Delta(\mathcal{A})$ let $d_{\text{obs},h}^{\pi_{\text{obs}}}(x_h, a_h) = d_h^{M_{\text{obs}}^\star,\pi_{\text{obs}}}(x_h, a_h)$ denote the occupancy in $M_{\text{obs}}^\star$. We overload notation by letting $d_{\text{obs},h}^{\pi_{\text{lat}}\circ\phi}(s, a) := \sum_{x:\phi(x)=s} d_{\text{obs},h}^{\pi\circ\phi}(x, a)$. We will establish the stronger result that

$$\Big\|d_{\text{lat},h}^{\pi_{\text{lat}}}(\cdot) - d_{\text{obs},h}^{\pi_{\text{lat}}\circ\phi}(\cdot)\Big\|_{\text{tv}} \leq \sum_{h'<h} \mathbb{E}^{\pi_{\text{lat}}\circ\phi}\big[\|[P_{\text{lat}} \circ \phi](x_{h'}, a_{h'}) - \phi\sharp P_{\text{obs}}^\star(x_{h'}, a_{h'})\|_{\text{tv}}\big], \quad (82)$$

where the tv norm on the left-hand-side is over $\mathcal{S} \times \mathcal{A}$. Note that this implies the desired bound on Eq. (81) by Holder's inequality. We prove this by induction over $h$. For the base case ($h = 0$), we have:

$$\sum_{s_1,a_1}\left|d_{\text{lat},1}^{\pi_{\text{lat}}}(s_1, a_1) - d_{\text{obs}}^{\pi_{\text{lat}}\circ\phi}(s_1, a_1)\right|$$

$$= \sum_{s_1,a_1}\left|P_{\text{lat},0}(s_1 \mid \emptyset)\pi_{\text{lat}}(a_1 \mid s_1) - \sum_{x_1=\phi(x_1)=s_1} d_{\text{obs}}^{\pi_{\text{lat}}\circ\phi}(x_1, a_1)\right|$$

$$= \sum_{s_1,a_1}\left|P_{\text{lat},0}(s_1 \mid \emptyset)\pi_{\text{lat}}(a_1 \mid s_1) - \sum_{x_1=\phi(x_1)=s_1} P_{\text{obs},0}^\star(x_1 \mid \emptyset)\pi_{\text{lat}}(a_1 \mid \phi(x_1))\right|$$

$$= \sum_{s_1}\left|P_{\text{lat},0}(s_1 \mid \emptyset) - \phi_1\sharp P_{\text{obs},0}^\star(s_1 \mid \emptyset)\right|\sum_{a_1}\pi_{\text{lat}}(a_1 \mid s_1)$$

$$= \big\|P_{\text{lat},0}(\emptyset) - \phi_1\sharp P_{\text{obs},0}^\star(\emptyset)\big\|_{\text{tv}}.$$

For the general case, let us further overload notation by letting $d_{\text{obs},h}^{\pi \circ \phi}(s_h) = \sum_{a_h} d_{\text{obs},h}^{\pi \circ \phi}(s_h, a_h)$ and $P_{\text{obs}}^\star(s_h \mid x_{h-1}, a_{h-1}) = \phi_\sharp P_{\text{obs}}^\star(s_h \mid x_{h-1}, a_{h-1}) = \sum_{x_h : \phi(x_h) = s_h} P_{\text{obs}}^\star(x_h \mid x_{h-1}, a_{h-1})$. Let us also abbreviate $\pi := \pi_{\text{lat}}$. Firstly note that it is sufficient to establish the result for $\sum_{s_h \in \mathcal{S}} \left| d_{\text{lat},h}^\pi(s_h) - d_{\text{obs},h}^{\pi \circ \phi}(s_h) \right|$, since

$$\sum_{s_h, a_h \in \mathcal{S} \times \mathcal{A}} \left| d_{\text{lat},h}^\pi(s_h, a_h) - d_{\text{obs},h}^{\pi \circ \phi}(s_h, a_h) \right| = \sum_{s_h, a_h \in \mathcal{S} \times \mathcal{A}} \left| d_{\text{lat},h}^\pi(s_h) - d_{\text{obs},h}^{\pi \circ \phi}(s_h) \right| \pi(a_h \mid s_h)$$

$$= \sum_{s_h \in \mathcal{S}} \left| d_{\text{lat},h}^\pi(s_h) - d_{\text{obs},h}^{\pi \circ \phi}(s_h) \right|.$$

Below, all summations over $s_h$ (resp. $x_h$) with domain unspecified are over $\mathcal{S}$ (resp. $\mathcal{X}$), and likewise for summations over $s_h, a_h$ or $x_h, a_h$. We have:

$$\sum_{s_h} \left| d_{\text{lat},h}^\pi(s_h) - d_{\text{obs},h}^{\pi \circ \phi}(s_h) \right|$$

$$= \sum_{s_h} \left| d_{\text{lat},h}^\pi(s_h) - \sum_{x_h : \phi(x_h) = s_h} d_{\text{obs},h}^{\pi \circ \phi}(x_h) \right|$$

$$= \sum_{s_h} \left| \sum_{s_{h-1}, a_{h-1}} d_{\text{lat},h}^\pi(s_{h-1}, a_{h-1}) P_{\text{lat},h}(s_h \mid s_{h-1}, a_{h-1}) \right.$$
$$\left. - \sum_{x_h : \phi(x_h) = s_h} \sum_{x_{h-1}, a_{h-1}} d_{\text{obs},h}^{\pi \circ \phi}(x_{h-1}, a_{h-1}) P_{\text{obs},h}^\star(x_h \mid x_{h-1}, a_{h-1}) \right|$$

$$= \sum_{s_h} \left| \sum_{s_{h-1}, a_{h-1}} d_{\text{lat},h}^\pi(s_{h-1}, a_{h-1}) P_{\text{lat},h}(s_h \mid s_{h-1}, a_{h-1}) \right.$$
$$\left. - \sum_{x_{h-1}, a_{h-1}} d_{\text{obs},h}^{\pi \circ \phi}(x_{h-1}, a_{h-1}) P_{\text{obs},h}^\star(s_h \mid x_{h-1}, a_{h-1}) \right|$$

$$= \sum_{s_h} \left| \sum_{s_{h-1}, a_{h-1}} d_{\text{lat},h}^\pi(s_{h-1}, a_{h-1}) P_{\text{lat},h}(s_h \mid s_{h-1}, a_{h-1}) \right.$$
$$- \sum_{x_{h-1}, a_{h-1}} d_{\text{obs},h}^{\pi \circ \phi}(x_{h-1}, a_{h-1}) P_{\text{lat},h}(s_h \mid \phi(x_{h-1}), a_{h-1})$$
$$+ \sum_{x_{h-1}, a_{h-1}} d_{\text{obs},h}^{\pi \circ \phi}(x_{h-1}, a_{h-1}) P_{\text{lat},h}(s_h \mid \phi(x_{h-1}), a_{h-1})$$
$$\left. - \sum_{x_{h-1}, a_{h-1}} d_{\text{obs},h}^{\pi \circ \phi}(x_{h-1}, a_{h-1}) P_{\text{obs},h}^\star(s_h \mid x_{h-1}, a_{h-1}) \right|$$

$$\leq \sum_{s_{h-1}, a_{h-1}} \left| d_{\text{lat},h}^\pi(s_{h-1}, a_{h-1}) - \sum_{x_{h-1} : \phi(x_{h-1}) = s_{h-1}} d_{\text{obs},h}^{\pi \circ \phi}(x_{h-1}, a_{h-1}) \right| \sum_{s_h} P_{\text{lat},h}(s_h \mid s_{h-1}, a_{h-1})$$

$$+ \sum_{s_h} \left| \sum_{x_{h-1}, a_{h-1}} d_{\text{obs},h}^{\pi \circ \phi}(x_{h-1}, a_{h-1}) \big( (P_{\text{lat},h} \circ \phi)(s_h \mid x_{h-1}, a_{h-1}) - P_{\text{obs},h}^\star(s_h \mid x_{h-1}, a_{h-1}) \big) \right|$$

$$\leq \left\| d_{\text{lat},h-1}^\pi(\cdot) - d_{\text{obs},h-1}^{\pi \circ \phi}(\phi^{-1}(\cdot)) \right\|_{\text{tv}}$$

$$+ \sum_{x_{h-1}, a_{h-1}} d_{\text{obs},h}^{\pi \circ \phi}(x_{h-1}, a_{h-1}) \sum_{s_h} \left| (P_{\text{lat},h} \circ \phi)(s_h \mid x_{h-1}, a_{h-1}) - P_{\text{obs},h}^\star(s_h \mid x_{h-1}, a_{h-1}) \right|$$

$$\leq \left\| d_{\text{lat},h-1}^\pi(\cdot) - d_{\text{obs},h-1}^{\pi \circ \phi}(\phi^{-1}(\cdot)) \right\|_{\text{tv}} + \mathbb{E}^{\pi \circ \phi} \left[ \left\| [P_{\text{lat},h} \circ \phi](x_{h-1}, a_{h-1}) - \phi_\sharp P_{\text{obs},h}^\star(x_{h-1}, a_{h-1}) \right\|_{\text{tv}} \right].$$

From which it follows that, for each $h$, we have:

$$\left\| d^{\pi}_{\mathsf{lat},h}(\cdot) - d^{\pi \circ \phi}_{\mathsf{obs},h}(\phi^{-1}(\cdot)) \right\|_{\mathsf{tv}} \le \sum_{h' < h} \mathbb{E}^{\pi \circ \phi} \left[ \left\| [P_{\mathsf{lat}} \circ \phi]_{h'}(x_{h'}, a_{h'}) - \phi_{h'+1} \sharp P^{\star}_{\mathsf{obs},h'}(x_{h'}, a_{h'}) \right\|_{\mathsf{tv}} \right]$$

$$\le \sum_{h' \in [H]} \mathbb{E}^{\pi \circ \phi} \left[ \left\| [P_{\mathsf{lat}} \circ \phi]_{h'}(x_{h'}, a_{h'}) - \phi_{h'+1} \sharp P^{\star}_{\mathsf{obs},h'}(x_{h'}, a_{h'}) \right\|_{\mathsf{tv}} \right].$$

$\square$

### I.3 Proofs for Appendix I.1.3: Risk Bound Under CorruptionRobustness (Theorem H.1)

**Theorem H.1** (Risk bound for O2L under self-predictive estimation and CorruptionRobustness). *Assume* $\mathsf{REP}_{\mathsf{self};\mathsf{opt}}$ *satisfies Assumption A.1 with parameter* $\gamma > 0$ *and that* $\mathcal{M}_{\mathsf{lat}}$ *is realizable (i.e.* $M^{\star}_{\mathsf{lat}} \in \mathcal{M}_{\mathsf{lat}}$). *Furthermore, let* $\mathsf{ALG}_{\mathsf{lat}}$ *be* CorruptionRobust *(Definition I.2) with parameter* $\alpha$. *Then,* O2L *(Algorithm 1) with inputs* $T, K, \Phi, \mathsf{ALG}_{\mathsf{lat}}$, *and* $\mathsf{REP}_{\mathsf{self};\mathsf{opt}}$ *has expected risk*

$$\mathbb{E}[\mathsf{Risk}_{\mathsf{obs}}(TK)] \le c_1 \cdot \mathsf{Risk}_{\mathsf{base}}(K) + c_2 \gamma \cdot \frac{K}{T} \mathsf{Est}_{\mathsf{self};\mathsf{opt}}(T, \gamma) + c_3 \gamma^{-1} \cdot (\alpha^2 + H) \quad (66)$$

*for absolute constants* $c_1, c_2, c_3 > 0$.

**Proof of Theorem H.1.** Let us write $\pi^{(t,K+1)}_{\mathsf{lat}} = \widehat{\pi}^{(t)}_{\mathsf{lat}}$ and, for any $t, k \in [T] \times [K+1]$, $\pi^{(t,k)}_{\mathsf{obs}} := \pi^{(t,k)}_{\mathsf{lat}} \circ \phi^{(t)}$. Let $p^{(t,k)}_{\mathsf{obs}}$ denote the distributions of played policies $\pi^{(t,k)}_{\mathsf{obs}}$ induced by the interaction of $\mathsf{ALG}_{\mathsf{lat}}$ and $\mathsf{REP}_{\mathsf{self};\mathsf{opt}}$ inside the O2L algorithm. Let us write the online sum of self-prediction errors as

$$\varepsilon^2_{\mathsf{rep}} := \sum_{t=1}^{T} \sum_{k=1}^{K+1} \sum_{h=0}^{H} \mathbb{E}_{\pi^{(t,k)}_{\mathsf{obs}} \sim p^{(t,k)}} \mathbb{E}^{\pi^{(t,k)}_{\mathsf{obs}}} \left[ D^2_{\mathsf{H}}\left( [M^{(t)}_{\mathsf{lat}} \circ \phi^{(t)}]_h(x_h, a_h), \phi^{(t)}_{h+1} \sharp M^{\star}_{\mathsf{obs},h}(x_h, a_h) \right) \right] \quad (83)$$

Since the final output policy of O2L satisfies $\widehat{\pi}_{\mathsf{lat}} = \mathsf{Unif}(\widehat{\pi}^{(1)}_{\mathsf{lat}}, \dots, \widehat{\pi}^{(T)}_{\mathsf{lat}})$ (Line 12), we have

$$\mathbb{E}[\mathsf{Risk}_{\mathsf{obs}}(TK)] = \frac{1}{T} \sum_{t=1}^{T} \mathbb{E}\left[ J^{M^{\star}_{\mathsf{obs}}}(\pi^{\star}_{\mathsf{obs}}) - J^{M^{\star}_{\mathsf{obs}}}(\widehat{\pi}^{(t)}_{\mathsf{obs}}) \right].$$

We take the following decomposition on the risk

$$J^{M^{\star}_{\mathsf{obs}}}(\pi^{\star}_{\mathsf{obs}}) - J^{M^{\star}_{\mathsf{obs}}}(\widehat{\pi}^{(t)}_{\mathsf{obs}}) = J^{M^{(t)}_{\mathsf{lat}}}(\pi_{M^{\star}_{\mathsf{lat}}}) - J^{M^{(t)}_{\mathsf{lat}}}(\pi_{M^{(t)}_{\mathsf{lat}}}) + \underbrace{J^{M^{(t)}_{\mathsf{lat}}}(\pi_{M^{(t)}_{\mathsf{lat}}}) - J^{M^{(t)}_{\mathsf{lat}}}(\widehat{\pi}^{(t)}_{\mathsf{lat}})}_{A_t}$$

$$+ \underbrace{J^{M^{(t)}_{\mathsf{lat}}}(\widehat{\pi}^{(t)}_{\mathsf{lat}}) - J^{M^{\star}_{\mathsf{obs}}}(\widehat{\pi}^{(t)}_{\mathsf{obs}})}_{B_t}. \quad (84)$$

We will show that $\mathbb{E}\left[ \sum_{t=1}^{T} A_t \right] \lesssim T\mathsf{Reg}_{\mathsf{base}}(K) + \alpha\sqrt{T}\,\mathbb{E}[\varepsilon_{\mathsf{rep}}]$ and that $\mathbb{E}\left[ \sum_{t=1}^{T} B_t \right] \lesssim \sqrt{TH}\,\mathbb{E}[\varepsilon_{\mathsf{rep}}]$, then return to the first term $J^{M^{\star}_{\mathsf{lat}}}(\pi_{M^{\star}_{\mathsf{lat}}}) - J^{M^{(t)}_{\mathsf{lat}}}(\pi_{M^{(t)}_{\mathsf{lat}}})$ at the end of the proof.

To bound $\mathbb{E}\left[ \sum_{t=1}^{T} A_t \right]$, we note that

$$\sum_{t=1}^{T} \mathbb{E}[A_t] \le c_1 T\mathsf{Risk}_{\mathsf{base}}(K) +$$

$$\alpha \sum_{t=1}^{T} \mathbb{E}\left[ \sqrt{\sum_{k=1}^{K} \sum_{h=1}^{H} \mathbb{E}_{\pi^{(t,k)}_{\mathsf{lat}} \sim p^{(t,k)}_{\mathsf{lat}}} \widetilde{\mathbb{E}}^{\pi^{(t,k)}_{\mathsf{lat}}}_{\phi^{(t)}} \left[ D^2_{\mathsf{H}}\left( M^{(t)}_{\mathsf{lat},h}(s_h, a_h), \widetilde{M}^{\star}_{\phi^{(t)},h}(s_{1:h}, a_{1:h}) \right) \right] } \right]$$

$$\le c_1 T\mathsf{Risk}_{\mathsf{base}}(K) + \alpha \sum_{t=1}^{T} \mathbb{E}\left[ \sqrt{\sum_{k=1}^{K} \sum_{h=1}^{H} \mathbb{E}_{\pi^{(t,k)}_{\mathsf{lat}} \sim p^{(t,k)}_{\mathsf{lat}}} \mathbb{E}^{\pi^{(t,k)}_{\mathsf{lat}} \circ \phi^{(t)}} \left[ [\Delta_h(M^{(t)}_{\mathsf{lat}}, \phi^{(t)})](x_h, a_h) \right] } \right]$$

$$\le c_1 T\mathsf{Risk}_{\mathsf{base}}(K) + \alpha\sqrt{T}\,\mathbb{E}\left[ \sqrt{\sum_{t=1}^{T} \sum_{k=1}^{K} \sum_{h=1}^{H} \mathbb{E}_{\pi^{(t,k)}_{\mathsf{obs}} \sim p^{(t,k)}_{\mathsf{obs}}} \mathbb{E}^{\pi^{(t,k)}_{\mathsf{obs}}} \left[ [\Delta_h(M^{(t)}_{\mathsf{lat}}, \phi^{(t)})](x_h, a_h) \right] } \right]$$

$$\le c_1 T\mathsf{Risk}_{\mathsf{base}}(K) + \alpha\sqrt{T}\,\mathbb{E}[\varepsilon_{\mathsf{rep}}].$$

where the first line follows from the `CorruptionRobust` definition (Definition I.2), the second line follows from Lemma I.2, the third line follows by Cauchy-Schwartz, and the last line recalls the definition of $\varepsilon_{\mathsf{rep}}$ from Eq. (83).

For the term $\sum_{t=1}^{T} B_t$, for any $\pi_{\mathsf{lat}} : \mathcal{S} \times [H] \to \Delta(\mathcal{A})$ we let $Q_{\mathsf{lat}^{(t)},h}^{\pi_{\mathsf{lat}}} = \mathcal{T}_h^{M_{\mathsf{lat}}^{(t)}} Q_{\mathsf{lat}^{(t)},h+1}^{\pi_{\mathsf{lat}}}$ be the $Q^{\pi_{\mathsf{lat}}}$ function of the latent MDP $M_{\mathsf{lat}}^{(t)}$. Note that

$$\sum_{t=1}^{T} \left\{ J^{M_{\mathsf{lat}}^{(t)}}(\widehat{\pi}_{\mathsf{lat}}^{(t)}) - \mathbb{E}^{\widehat{\pi}_{\mathsf{lat}}^{(t)} \circ \phi^{(t)}} \left[ [Q_{\mathsf{lat}^{(t)}}^{\widehat{\pi}^{(t)}} \circ \phi^{(t)}]_1(x_1, a_1) \right] \right\} \tag{85}$$

$$= \sum_{t=1}^{T} \mathbb{E}^{M_{\mathsf{lat}}^{(t)}, \widehat{\pi}_{\mathsf{lat}}^{(t)}} \left[ Q_{\mathsf{lat}^{(t)},1}^{\widehat{\pi}^{(t)}}(s_1, a_1) \right] - \mathbb{E}^{\widehat{\pi}_{\mathsf{lat}}^{(t)} \circ \phi^{(t)}} \left[ [Q_{\mathsf{lat}^{(t)}}^{\widehat{\pi}^{(t)}} \circ \phi^{(t)}]_1(x_1, a_1) \right]$$

$$\leq \sum_{t=1}^{T} \mathbb{E}^{\widehat{\pi}_{\mathsf{lat}}^{(t)} \circ \phi^{(t)}} \left[ \left\| [P_{\mathsf{lat}}^{(t)} \circ \phi^{(t)}]_0(\emptyset) - \phi_1^{(t)} \sharp P_{\mathsf{obs},0}^{\star}(\emptyset) \right\|_{\mathsf{tv}} \right] \qquad \text{(by Lemma I.3)}$$

$$\leq \sum_{t=1}^{T} \sum_{h=0}^{H} \mathbb{E}^{\widehat{\pi}_{\mathsf{lat}}^{(t)} \circ \phi^{(t)}} \left[ \left\| [P_{\mathsf{lat}}^{(t)} \circ \phi^{(t)}]_h(x_h, a_h) - \phi_{h+1}^{(t)} \sharp P_{\mathsf{obs},h}^{\star}(x_h, a_h) \right\|_{\mathsf{tv}} \right]$$

$$\leq \sqrt{TH} \varepsilon_{\mathsf{rep}}, \qquad \text{(by Cauchy-Schwartz)}$$

so it is enough to bound

$$\sum_{t=1}^{T} \left\{ \mathbb{E}^{\widehat{\pi}_{\mathsf{lat}}^{(t)} \circ \phi^{(t)}} \left[ [Q_{\mathsf{lat}^{(t)}}^{\widehat{\pi}_{\mathsf{lat}}^{(t)}} \circ \phi^{(t)}]_1(x_1, a_1) \right] - J^{M_{\mathsf{obs}}^{\star}}(\widehat{\pi}_{\mathsf{obs}}^{(t)}) \right\}.$$

Fix $t$ and $h$, whose indexing we omit below for cleanliness. Note that, for any $\pi_{\mathsf{lat}} : \mathcal{S} \times [H] \to \Delta(\mathcal{A})$, we have:

$$\mathbb{E}^{\pi_{\mathsf{lat}} \circ \phi} \left[ \left( [Q_{\mathsf{lat}}^{\pi_{\mathsf{lat}}} \circ \phi]_h(x_h, a_h) - \mathcal{T}_h^{M_{\mathsf{obs}}^{\star}, \pi_{\mathsf{lat}} \circ \phi} [Q_{\mathsf{lat}}^{\pi_{\mathsf{lat}}} \circ \phi]_{h+1}(x_h, a_h) \right)^2 \right] \tag{86}$$

$$\leq 2 \mathbb{E}^{\pi_{\mathsf{lat}} \circ \phi} \left[ \left( [r_{\mathsf{lat}} \circ \phi]_h - r_{\mathsf{obs},h}^{\star} \right)^2 (x_h, a_h) \right] \tag{87}$$

$$+ 2 \mathbb{E}^{\pi_{\mathsf{lat}} \circ \phi} \left[ \left( \mathbb{E}_{P_{\mathsf{lat},h}(\phi(x_h), a_h)} \left[ Q_{\mathsf{lat},h+1}^{\pi_{\mathsf{lat}}}(\cdot, \pi_{\mathsf{lat}}) \right] - \mathbb{E}_{P_{\mathsf{obs},h}^{\star}(x_h, a_h)} \left[ [Q_{\mathsf{lat}}^{\pi_{\mathsf{lat}}} \circ \phi]_{h+1}(\cdot, \pi_{\mathsf{lat}}) \right] \right)^2 \right] \tag{88}$$

$$\leq 2 \mathbb{E}^{\pi_{\mathsf{lat}} \circ \phi} \left[ \left( [r_{\mathsf{lat}} \circ \phi]_h - r_{\mathsf{obs},h}^{\star} \right)^2 (x_h, a_h) + \left\| P_{\mathsf{lat},h}(\phi(x_h), a_h) - \phi_{h+1} \sharp P_{\mathsf{obs},h}^{\star}(x_h, a_h) \right\|_{\mathsf{tv}}^2 \right] \tag{89}$$

$$\leq 4 \mathbb{E}^{\pi_{\mathsf{lat}} \circ \phi} \left[ D_{\mathsf{H}}^2 \left( M_{\mathsf{lat},h}(\phi_h(x_h), a_h), \phi_{h+1} \sharp M_{\mathsf{obs},h}^{\star}(x_h, a_h) \right) \right], \tag{90}$$

where the final line follows from two applications of the data-processing inequality (since $M_{\mathsf{lat},h}(r_h, s_{h+1} \mid \phi_h(x_h), a_h) = R_{\mathsf{lat},h}(r_h \mid \phi_h(x_h), a_h) P_{\mathsf{lat},h}(s_{h+1} \mid \phi_h(x_h), a_h)$ and $\phi_{h+1} \sharp M_{\mathsf{obs},h}^{\star}(r_h, s_{h+1} \mid x_h, a_h) = R_{\mathsf{obs},h}^{\star}(r_h \mid x_h, a_h) \phi_{h+1} \sharp P_{\mathsf{obs},h}^{\star}(s_{h+1} \mid x_h, a_h))$ as well as the bound $\|p - q\|_{\mathsf{tv}}^2 \leq D_{\mathsf{H}}^2(p, q)$. Summing this over $t, h$ and using a standard decomposition for

regret (Lemma C.6) gives:

$$\sum_{t=1}^{T}\left\{\mathbb{E}^{\widehat{\pi}_{\text{lat}}^{(t)}\circ\phi^{(t)}}\left[[Q_{\text{lat}(t)}^{\widehat{\pi}_{\text{lat}}^{(t)}}\circ\phi^{(t)}]_1(x_1,a_1)\right]-J^{M_{\text{obs}}^{\star}}(\widehat{\pi}_{\text{obs}}^{(t)})\right\}$$

$$=\sum_{t=1}^{T}\sum_{h=1}^{H}\mathbb{E}^{\widehat{\pi}_{\text{lat}}^{(t)}\circ\phi^{(t)}}\left[[Q_{\text{lat}(t)}^{\widehat{\pi}_{\text{lat}}^{(t)}}\circ\phi^{(t)}]_h(x_h,a_h)-\mathcal{T}_h^{M_{\text{obs}}^{\star},\widehat{\pi}_{\text{lat}}^{(t)}\circ\phi^{(t)}}[Q_{\text{lat}(t)}^{\widehat{\pi}_{\text{lat}}^{(t)}}\circ\phi^{(t)}]_{h+1}(x_h,a_h)\right]$$

$$\text{(Lemma C.6)}$$

$$\leq\sqrt{TH}\sqrt{\sum_{t=1}^{T}\sum_{h=1}^{H}\mathbb{E}^{\widehat{\pi}_{\text{lat}}^{(t)}\circ\phi^{(t)}}\left[\left([Q_{\text{lat}(t)}^{\widehat{\pi}_{\text{lat}}^{(t)}}\circ\phi^{(t)}]_h(x_h,a_h)-\mathcal{T}_h^{M_{\text{obs}}^{\star},\widehat{\pi}_{\text{lat}}^{(t)}\circ\phi^{(t)}}[Q_{\text{lat}(t)}^{\widehat{\pi}_{\text{lat}}^{(t)}}\circ\phi^{(t)}]_{h+1}(x_h,a_h)\right)^2\right]}$$

$$\leq\sqrt{4TH}\sqrt{\sum_{t=1}^{T}\sum_{h=1}^{H}\mathbb{E}^{\widehat{\pi}_{\text{lat}}^{(t)}\circ\phi^{(t)}}\left[D_{\mathsf{H}}^2\left(\left[M_{\text{lat},h}^{(t)}\circ\phi_h^{(t)}\right](x_h,a_h),\phi_{h+1}^{(t)}\natural M_{\text{obs},h}^{\star}(x_h,a_h)\right)\right]}$$

$$\text{(By Eq. (90))}$$

$$\leq\sqrt{4TH}\varepsilon_{\text{rep}}.$$

Returning to the decomposition of Eq. (84) and combining everything gives:

$$\mathbb{E}[\mathsf{Risk}_{\text{obs}}]\leq\frac{1}{T}\left\{\sum_{t=1}^{T}\mathbb{E}\left[J^{M_{\text{lat}}^{\star}}(\pi_{M_{\text{lat}}^{\star}})-J^{M_{\text{lat}}^{(t)}}(\pi_{M_{\text{lat}}^{(t)}})\right]\right\}+\frac{1}{T}\left(\alpha\sqrt{T}+4\sqrt{TH}\right)\mathbb{E}[\varepsilon_{\text{rep}}]$$

$$+c_1\cdot\mathsf{Risk}_{\text{base}}(K)$$

$$\leq\frac{1}{T}\left\{\sum_{t=1}^{T}\mathbb{E}\left[J(\pi^{\star})-J^{M_{\text{lat}}^{(t)}}(\pi_{M_{\text{lat}}^{(t)}})+\gamma\varepsilon_{\text{rep}}^2\right]\right\}+\frac{\gamma^{-1}}{T}\left(\alpha\sqrt{T}+4\sqrt{TH}\right)^2$$

$$+c_1\cdot\mathsf{Risk}_{\text{base}}(K)$$

$$\leq\gamma\frac{2K}{T}\mathsf{Est}_{\text{self;opt}}(T,\gamma)+2\gamma^{-1}\left(\alpha^2+16H\right)+c_1\cdot\mathsf{Risk}_{\text{base}}(K),$$

where the second inequality follows by AM-GM applied to the middle term and the third inequality follows from: i) Jensen's inequality, ii) Assumption A.1 applied to the distributions $\bar{p}_{\text{obs}}^{(t)}=\frac{1}{K}\sum_{k=1}^{K}p_{\text{obs}}^{(t,k)}$, iii) the bound $K+1\leq 2K$, and iv) the inequality $(x+y)^2\leq 2(x^2+y^2)$. □

## I.4  Proofs for Appendix I.1.4: Examples of `CorruptionRobust` Algorithms

**Theorem I.1** (Latent GOLF is `CorruptionRobust`). *Under Assumption I.1 and Assumption I.2, Algorithm 5 with $\beta=c\big(\log(|\mathcal{F}||\mathcal{G}|KH\delta^{-1})+\varepsilon_{\text{rep}}\big)$, has regret*

$$\sum_{k=1}^{K}J^{M_{\text{lat}}}(\pi_{M_{\text{lat}}})-J^{M_{\text{lat}}}(\pi^{(k)})\leq\mathcal{O}\left(H\sqrt{C_{\text{cov}}K\log(K)\log(|\mathcal{F}||\mathcal{G}|HK/\delta)}\right)$$

$$+\mathcal{O}\left(H^{3/2}\sqrt{KC_{\text{cov}}\log(K)\varepsilon_{\text{rep}}^2}\right),$$

*and consequently is `CorruptionRobust` (Definition I.2) with parameters*

$$\alpha=\frac{H^{3/2}}{\sqrt{K}}\sqrt{C_{\text{cov}}\log(K)}\ \text{and}\ \mathsf{Risk}_{\text{base}}(K)=\mathcal{O}\left(\frac{H}{\sqrt{K}}\sqrt{C_{\text{cov}}\log(K)\log(|\mathcal{F}||\mathcal{G}|HK)}\right).$$

**Proof of Theorem I.1.**  Recall that the agent is observing data from the $\phi$-compressed POMDP $\widetilde{M}_{\phi}^{\star}$, and thus the datasets are of the form $\mathcal{D}_h^{(k)}=\mathcal{D}_{\phi,h}^{(k)}=\{\phi(x_h^{(i)}),a_h^{(i)},r_h^{(i)},\phi(x_{h+1}^{(i)})\}_{i=1}^{k-1}$. For any $\pi_{\text{lat}}\in\Pi_{\text{lat}}$, we define

$$\widetilde{\mathcal{T}}_{\phi,h}^{\pi_{\text{lat}}}f(s_h,a_h)=\tilde{r}_{\phi,h}^{\pi_{\text{lat}}}(s_h,a_h)+\mathbb{E}_{s'\sim\widetilde{P}_{\phi,h}^{\pi_{\text{lat}}}(s_h,a_h)}[f(s')],$$

where $\tilde{r}_{\phi,h}^{\pi_{\text{lat}}}$ and $\widetilde{P}_{\phi,h}^{\pi_{\text{lat}}}$ are the policy-dependent Markov operators defined in Eq. (67) and Eq. (69).

As a consequence, we observe the following misspecification guarantee for $\mathcal{T}_{\text{lat}}$.

**Lemma I.4** (Misspecification guarantee for $\mathcal{T}_{\text{lat}}$)**.**

$$\forall f : \mathcal{S} \times \mathcal{A} \to [0,1] : \quad \sum_{k=1}^{K} \sum_{h=1}^{H} \widetilde{\mathbb{E}}_{\phi}^{\pi^{(k)}} \left[ \left( \mathcal{T}_{\text{lat},h} f(s_h, a_h) - \widetilde{\mathcal{T}}_{\phi,h}^{\pi^{(k)}} f(s_h, a_h) \right)^2 \right] \leq \mathcal{O}(\varepsilon_{\text{rep}}^2).$$

**Proof of Lemma I.4.** Follows from Assumption I.2 and the definitions of $\widetilde{\mathcal{T}}_{\phi,h}^{\pi^{(k)}}$ and $\mathcal{T}_{\text{lat},h}$. $\qquad\square$

We begin with the following lemmas, which will be proved in the sequel.

**Lemma I.5** (Optimism)**.** *For the choice of $\beta$ in Theorem I.1, with probability at least $1 - \delta$, we have that for all $k \in [K]$:*

$$Q_{\text{lat}}^{\star} \in \mathcal{F}^{(k)}.$$

**Lemma I.6** (Small in-sample squared Bellman errors)**.** *With probability at least $1 - \delta$, we have that for all $k \in [K]$, $h \in [H]$, and $f \in \mathcal{F}^{(k)}$:*

$$\sum_{i=1}^{k-1} \widetilde{\mathbb{E}}_{\phi}^{\pi^{(i)}} \left[ \left( f(s_h, a_h) - \widetilde{\mathcal{T}}_{\phi,h}^{\pi^{(i)}} f(s_h, a_h) \right)^2 \right] \leq \mathcal{O}(\beta).$$

Let us write $\pi_{\text{obs}}^{(k)} := \pi^{(k)} \circ \phi$. Let us introduce the shorthand $\tilde{d}_{\text{obs},h}^{(k)} := \sum_{i=1}^{k-1} d_{\text{obs},h}^{\pi_{\text{obs}}^{(k)}}$, where $d_{\text{obs}}^{\pi}$ is the occupancy for $M_{\text{obs}}^{\star}$, and also the burn-in time

$$\kappa_h(x,a) := \min \left\{ k : \sum_{i=1}^{k-1} d_{\text{obs},h}^{\pi^{(k)}}(x,a) \geq C_{\text{cov}} \mu_h^{\star}(x,a) \right\}.$$

Let us recall, from the analysis of [XFBJK23], that for any $h \in [H]$ and $f : \mathcal{S} \times \mathcal{A} \to [0,1]$ we have

$$\sum_{k=1}^{K} \mathbb{E}^{\pi^{(k)}} [f(s_h, a_h) \mathbb{I}\{k < \kappa_h(s_h, a_h)\}] \leq 2C_{\text{cov}}, \tag{91}$$

as well as

$$\sum_{h=1}^{H} \sum_{k=1}^{K} \sum_{s,a} \frac{(d_h^{\pi_{\text{obs}}^{(k)}}(x,a) \mathbb{I}\{k \geq \kappa_h(x,a)\})^2}{\tilde{d}_h^{(k)}(x,a)} \leq O(H C_{\text{cov}} \log(K)). \tag{92}$$

$$\sum_k J^{M_{\mathsf{lat}}}(\pi_{M_{\mathsf{lat}}}) - J^{M_{\mathsf{lat}}}(\pi^{(k)}) \leq \sum_{k=1}^{K}\sum_{h=1}^{H} \mathbb{E}^{M_{\mathsf{lat}},\pi^{(k)}}[f^{(k)}(s_h,a_h) - \mathcal{T}_{\mathsf{lat}}f^{(k)}(s_h,a_h)]$$

(Optimism (Lemma I.5))

$$\leq \sum_{k=1}^{K}\sum_{h=1}^{H} \widetilde{\mathbb{E}}_{\phi}^{\pi^{(k)}}[f^{(k)}(s_h,a_h) - \mathcal{T}_{\mathsf{lat}}f^{(k)}(s_h,a_h)] + H^{3/2}\sqrt{K\varepsilon_{\mathsf{rep}}^2}$$

(Simulation Lemma Lemma I.3)

$$= \sum_{k=1}^{K}\sum_{h=1}^{H} \mathbb{E}^{\pi^{(k)}\circ\phi}[[(f^{(k)} - \mathcal{T}_{\mathsf{lat}}f^{(k)})\circ\phi](x_h,a_h)] + H^{3/2}\sqrt{K\varepsilon_{\mathsf{rep}}^2}$$

(Change of measure Lemma I.1)

$$\leq \sum_{k=1}^{K}\sum_{h=1}^{H} \mathbb{E}^{\pi^{(k)}\circ\phi}[[(f^{(k)} - \mathcal{T}_{\mathsf{lat}}f^{(k)})\circ\phi](x_h,a_h)\mathbb{I}\{k \geq \kappa_h(x_h,a_h)\}]$$
$$+ 2HC_{\mathsf{cov}} + H^{3/2}\sqrt{K\varepsilon_{\mathsf{rep}}^2} \qquad \text{(Burn-in time Eq. (91))}$$

$$\leq \underbrace{\sum_{k=1}^{K}\sum_{h=1}^{H} \mathbb{E}^{\pi^{(k)}\circ\phi}\left[\left[(f^{(k)} - \widetilde{\mathcal{T}}_{\phi,h}^{\pi^{(k)}}f^{(k)})\circ\phi\right](x_h,a_h)\mathbb{I}\{k \geq \kappa_h(x_h,a_h)\}\right]}_{\text{(I)}}$$

$$+ \underbrace{\sum_{k=1}^{K}\sum_{h=1}^{H} \mathbb{E}^{\pi^{(k)}\circ\phi}\left[\left[(\widetilde{\mathcal{T}}_{\phi,h}^{\pi^{(k)}}f^{(k)} - \mathcal{T}_{\mathsf{lat},h}f^{(k)})\circ\phi\right](x_h,a_h)\right]}_{\text{(II)}}$$

$$+ 2HC_{\mathsf{cov}} + H^{3/2}\sqrt{K\varepsilon_{\mathsf{rep}}^2}$$

Note that, by change of measure (Lemma I.1) and the misspecification guarantee (Lemma I.4), the second term is bounded by:

$$\text{(II)} = \sum_{k=1}^{K}\sum_{h=1}^{H} \widetilde{\mathbb{E}}_{\phi}^{\pi^{(k)}}\left[(\widetilde{\mathcal{T}}_{\phi,h}^{\pi^{(k)}}f^{(k)} - \mathcal{T}_{\mathsf{lat},h}f^{(k)})(s_h,a_h)\right] \leq \sqrt{KH\varepsilon_{\mathsf{rep}}^2}.$$

Turning to the first term, we have:

$$\sum_{h=1}^{H}\sum_{k=1}^{K} \mathbb{E}^{\pi_{\mathsf{obs}}^{(k)}}\left[\left[(f^{(k)} - \widetilde{\mathcal{T}}_{\phi,h}^{\pi^{(k)}}f^{(k)})\circ\phi\right](x_h,a_h)\mathbb{I}\{k \geq \kappa_h(x_h,a_h)\}\right] \tag{93}$$

$$\leq \sqrt{\sum_{h=1}^{H}\sum_{k=1}^{K}\sum_{x,a} \frac{(d_h^{\pi_{\mathsf{obs}}^{(k)}}(x,a)\mathbb{I}\{k \geq \kappa_h(x,a)\})^2}{\tilde{d}_h^{(k)}(x,a)}} \sqrt{\sum_{h=1}^{H}\sum_{k=1}^{K} \mathbb{E}_{\tilde{d}_{\mathsf{obs}}^{(k)}}\left[\left((f^{(k)} - \widetilde{\mathcal{T}}_{\phi,h}^{\pi^{(k)}}f^{(k)})\circ\phi\right)^2(x_h,a_h)\right]} \tag{94}$$

$$\leq \sqrt{HC_{\mathsf{cov}}\log(K)}\sqrt{\sum_{h=1}^{H}\sum_{k=1}^{K} \mathbb{E}_{\tilde{d}_{\mathsf{obs}}^{(k)}}\left[\left((f^{(k)} - \widetilde{\mathcal{T}}_{\phi,h}^{\pi^{(k)}}f^{(k)})\circ\phi\right)^2(x_h,a_h)\right]}$$

(coverability potential Eq. (92))

$$= \sqrt{HC_{\mathsf{cov}}\log(K)}\sqrt{\sum_{h=1}^{H}\sum_{k=1}^{K}\sum_{i=1}^{k-1} \widetilde{\mathbb{E}}_{\phi}^{\pi^{(i)}}\left[\left(f^{(k)}(s_h,a_h) - \widetilde{\mathcal{T}}_{\phi,h}^{\pi^{(k)}}f^{(k)}(s_h,a_h)\right)^2\right]}$$

(change of measure, Lemma I.1)

$$\leq \mathcal{O}\left(H\sqrt{C_{\mathsf{cov}}K\log(K)\beta}\right), \tag{95}$$

where we have used that, from Lemma I.6, we have:

$$\sum_{h=1}^{H}\sum_{k=1}^{K}\sum_{i=1}^{k-1}\widetilde{\mathbb{E}}_{\phi}^{\pi^{(i)}}\left[\left(f^{(k)}(s_h, a_h) - \widetilde{\mathcal{T}}_{\phi}^{\pi^{(i)}}f^{(k)}(s_h, a_h)\right)^2\right] \leq \mathcal{O}(\beta H K).$$

This gives an upper bound on the regret of

$$\sum_{k=1}^{K}J^{M_{\mathrm{lat}}}(\pi_{M_{\mathrm{lat}}}) - J^{M_{\mathrm{lat}}}(\pi^{(k)}) \leq \mathcal{O}\left(H\sqrt{C_{\mathrm{cov}}K\log(K)\beta} + H^{3/2}\sqrt{K\varepsilon_{\mathrm{rep}}^2}\right).$$

Using that $\beta = \mathcal{O}\left(\log\left(\frac{|\mathcal{F}||\mathcal{G}|HK}{\delta}\right) + \varepsilon_{\mathrm{rep}}\right)$ and simplifying gives

$$\sum_{k=1}^{K}J^{M_{\mathrm{lat}}}(\pi_{M_{\mathrm{lat}}}) - J^{M_{\mathrm{lat}}}(\pi^{(k)}) \leq \mathcal{O}\left(H\sqrt{C_{\mathrm{cov}}K\log(K)\log(|\mathcal{F}||\mathcal{G}|HK/\delta)}\right) + \mathcal{O}\left(H^{3/2}\sqrt{KC_{\mathrm{cov}}\log(K)\varepsilon_{\mathrm{rep}}^2}\right),$$

as desired. It only remains to establish the concentrations results.

**Concentration analysis.** We establish the concentration results of Lemma I.5 and Lemma I.6.

**Proof of Lemma I.6.** Let

$$X_k(h, f) = \left(f_h(s_h^{(k)}, a_h^{(k)}) - r_h^{(k)} - f_{h+1}(s_{h+1}^{(k)})\right)^2 - \left(\widetilde{\mathcal{T}}_{\phi}^{\pi^{(k)}}f_h(s_h^{(k)}, a_h^{(k)}) - r_h^{(k)} - f_{h+1}(s_{h+1}^{(k)})\right)^2.$$

Let $\mathfrak{F}_{k,h} = \{s_1^{(i)}, a_1^{(i)}, r_1^{(i)}, \ldots, s_H^{(i)}, a_H^{(i)}, r_H^{(i)}\}_{i=1}^{k}$. Note that

$$\begin{aligned}
\mathbb{E}\left[r_h^{(k)} + f_{h+1}(s_{h+1}^{(k)}) \mid \mathfrak{F}_{k,h}\right] &= \mathbb{E}\left[r_h^{(k)} + f_{h+1}(s_{h+1}^{(k)}) \mid \pi^{(k)}\right] \\
&= \mathbb{E}\left[\mathbb{E}\left[r_h^{(k)} + f_{h+1}(s_{h+1}^{(k)}) \mid s_h^{(k)}, a_h^{(k)}, \pi^{(k)}\right] \mid \pi^{(k)}\right] \\
&= \mathbb{E}\left[\widetilde{\mathcal{T}}_{\phi}^{\pi^{(k)}}f(s_h^{(k)}, a_h^{(k)}) \mid \pi^{(k)}\right] \\
&= \widetilde{\mathbb{E}}_{\phi}^{\pi^{(k)}}\left[\widetilde{\mathcal{T}}_{\phi}^{\pi^{(k)}}f(s_h, a_h)\right],
\end{aligned}$$

and thus that

$$\mathbb{E}[X_k(h, f) \mid \mathfrak{F}_{k,h}] = \widetilde{\mathbb{E}}_{\phi}^{\pi^{(k)}}\left[\left(f_h(s_h, a_h) - \widetilde{\mathcal{T}}_{\phi}^{\pi^{(k)}}f_h(s_h, a_h)\right)^2\right].$$

Next, note that

$$\begin{aligned}
\mathsf{Var}[X_k(h, f) \mid \mathfrak{F}_{k,h}] &\leq \mathbb{E}\left[(X_k(h, f))^2 \mid \mathfrak{F}_{k,h}\right] \\
&\leq 16\,\mathbb{E}\left[\left(f_h(s_h^{(k)}, a_h^{(k)}) - \widetilde{\mathcal{T}}_{\phi}^{\pi^{(k)}}f_h(s_h^{(k)}, a_h^{(k)})\right)^2 \mid \mathfrak{F}_{k,h}\right] \\
&= 16\,\mathbb{E}[X_k(h, f) \mid \mathfrak{F}_{k,h}].
\end{aligned}$$

By Freedman's inequality (Lemma C.2, Lemma C.3), we have that with probability at least $1 - \delta$:

$$\left|\sum_{t<k}X_t(h, f) - \sum_{t<k}\mathbb{E}[X_t(h, f) \mid \mathfrak{F}_{t,h}]\right| \leq \mathcal{O}\left(\sqrt{\log(1/\delta)\sum_{t<k}\mathbb{E}[X_t(h, f) \mid \mathfrak{F}_{t,h}]} + \log(1/\delta)\right)$$

Taking a union bound over $[K] \times [H] \times \mathcal{F}$, we have that for all $k, h, f$, with probability at least $1 - \delta$:

$$\left|\sum_{t<k}X_t(h, f) - \sum_{t<k}\widetilde{\mathbb{E}}_{\phi}^{\pi^{(k)}}\left[\left(f_h(s_h, a_h) - \widetilde{\mathcal{T}}_{\phi}^{\pi^{(k)}}f_h(s_h, a_h)\right)^2\right]\right| \tag{96}$$

$$\leq \mathcal{O}\left(\sqrt{\iota\sum_{t<k}\widetilde{\mathbb{E}}_{\phi}^{\pi^{(k)}}\left[\left(f_h(s_h, a_h) - \widetilde{\mathcal{T}}_{\phi}^{\pi^{(k)}}f_h(s_h, a_h)\right)^2\right]} + \iota\right), \tag{97}$$

where $\iota = \log(|\mathcal{F}|HK/\delta)$. We now show that

$$\sum_{t<k}X_t(h, f^{(k)}) \leq \beta + \mathcal{O}(\varepsilon_{\mathrm{rep}} + \iota) = \mathcal{O}(\beta), \tag{98}$$

which will imply, from Eq. (96), that

$$\sum_{t<k} \widetilde{\mathbb{E}}_\phi^{\pi^{(k)}} \left[ \left( f_h(s_h, a_h) - \widetilde{\mathcal{T}}_\phi^{\pi^{(k)}} f_h(s_h, a_h) \right)^2 \right] \le \mathcal{O}(\iota + \beta) = \mathcal{O}(\beta),$$

as desired. To see Eq. (98), let

$$\Delta_k = \sum_{t<k} \left( \mathcal{T}_{\text{lat}} f_h^{(k)}(s_h^{(t)}, a_h^{(t)}) - r_h^{(t)} - f_{h+1}^{(k)}(s_{h+1}^{(t)}) \right)^2 - \left( \widetilde{\mathcal{T}}_\phi^{\pi^{(t)}} f_h^{(k)}(s_h^{(t)}, a_h^{(t)}) - r_h^{(t)} - f_{h+1}^{(k)}(s_{h+1}^{(t)}) \right)^2$$

and then note that:

$$\begin{aligned}
\sum_{t<k} X_t(h, f^{(k)}) &= \sum_{t<k} \left( f_h^{(k)}(s_h^{(t)}, a_h^{(t)}) - r_h^{(t)} - f_{h+1}^{(k)}(s_{h+1}^{(t)}) \right)^2 - \left( \widetilde{\mathcal{T}}_\phi^{\pi^{(t)}} f_h^{(k)}(s_h^{(t)}, a_h^{(t)}) - r_h^{(t)} - f_{h+1}^{(k)}(s_{h+1}^{(t)}) \right)^2 \\
&= \sum_{t<k} \left( f_h^{(k)}(s_h^{(t)}, a_h^{(t)}) - r_h^{(t)} - f_{h+1}^{(k)}(s_{h+1}^{(t)}) \right)^2 \\
&\quad - \sum_{t<k} \left( \mathcal{T}_{\text{lat}} f_h^{(k)}(s_h^{(t)}, a_h^{(t)}) - r_h^{(t)} - f_{h+1}^{(k)}(s_{h+1}^{(t)}) \right)^2 + \Delta_k \\
&\le \sum_{t<k} \left( f_h^{(k)}(s_h^{(t)}, a_h^{(t)}) - r_h^{(t)} - f_{h+1}^{(k)}(s_{h+1}^{(t)}) \right)^2 \\
&\quad - \inf_{g_h \in \mathcal{G}_h} \sum_{t<k} \left( g(s_h^{(t)}, a_h^{(t)}) - r_h^{(t)} - f_{h+1}^{(k)}(s_{h+1}^{(t)}) \right)^2 + \Delta_k \\
&\le \beta + \Delta_k.
\end{aligned}$$

where the second-to-last line follows from $\mathcal{T}_{\text{lat}} \mathcal{F} \subseteq \mathcal{G}$ and the last line follows from the definition of the confidence set. It remains to show that $\Delta_k \le \mathcal{O}(\varepsilon_{\text{rep}} + \iota)$, which we do via a similar concentration argument. Namely, let

$$Y_t(h, f) = \left( \mathcal{T}_{\text{lat}} f_h(s_h^{(t)}, a_h^{(t)}) - r_h^{(t)} - f_{h+1}^{(k)}(s_{h+1}^{(t)}) \right)^2 - \left( \widetilde{\mathcal{T}}_\phi^{\pi^{(t)}} f_h(s_h^{(t)}, a_h^{(t)}) - r_h^{(t)} - f_{h+1}^{(k)}(s_{h+1}^{(t)}) \right)^2,$$

and note that, as before,

$$\mathbb{E}[Y_t(h, f) \mid \mathfrak{F}_{t,h}] = \widetilde{\mathbb{E}}_\phi^{\pi^{(t)}} \left[ \left( \mathcal{T}_{\text{lat}} f_h(s_h, a_h) - \widetilde{\mathcal{T}}_\phi^{\pi^{(t)}} f_h(s_h, a_h) \right)^2 \right],$$

and

$$\text{Var}[Y_t(h, f) \mid \mathfrak{F}_{t,h}] \le 16 \, \mathbb{E}[Y_t(h, f) \mid \mathfrak{F}_{t,h}],$$

by the same calculation as earlier. Thus, by Freedman's inequality and a union bound, we have that, with probability at least $1 - \delta$,

$$\left| \sum_{t<k} Y_t(h, f) - \sum_{t<k} \widetilde{\mathbb{E}}_\phi^{\pi^{(k)}} \left[ \left( \mathcal{T}_{\text{lat}} f_h(s_h, a_h) - \widetilde{\mathcal{T}}_\phi^{\pi^{(k)}} f_h(s_h, a_h) \right)^2 \right] \right| \tag{99}$$

$$\le \mathcal{O}\left( \sqrt{\iota \sum_{t<k} \widetilde{\mathbb{E}}_\phi^{\pi^{(k)}} \left[ \left( \mathcal{T}_{\text{lat}} f_h(s_h, a_h) - \widetilde{\mathcal{T}}_\phi^{\pi^{(k)}} f_h(s_h, a_h) \right)^2 \right]} + \iota \right), \tag{100}$$

where $\iota = \log(|\mathcal{F}| HK/\delta)$. Recalling the misspecification assumption Lemma I.4, this implies that

$$\sum_{t<k} Y_t(h, f) \le \mathcal{O}(\varepsilon_{\text{rep}} + \iota),$$

for all $h, f, k$, with high probability. Applying this to $f = f^{(k)}$ concludes the result. □

**Proof of Lemma I.5.** We use similar arguments to the preceding lemma. Let $Q_{\text{lat},h}^\star := Q_{M_{\text{lat}},h}^\star$. The aim is to show that, for all $h \in [H], k \in [K], g \in \mathcal{G}$, we have:

$$\sum_{t<k} \left( g(s_h^{(t)}, a_h^{(t)}) - r_h^{(t)} - Q_{\text{lat},h+1}^\star(s_{h+1}^{(t)}) \right)^2 - \left( Q_{\text{lat},h}^\star(s_h^{(t)}, a_h^{(t)}) - r_h^{(t)} - Q_{\text{lat},h}^\star(s_{h+1}^{(t)}) \right)^2 \ge -\beta,$$

from which the conclusion will follow. We show that

$$\sum_{t<k} \underbrace{\left(g(s_h^{(t)}, a_h^{(t)}) - r_h^{(t)} - Q^\star_{\mathrm{lat},h+1}(s_{h+1}^{(t)})\right)^2 - \left(\widetilde{\mathcal{T}}_\phi^{\pi^{(t)}} Q^\star_{\mathrm{lat},h}(s_h^{(t)}, a_h^{(t)}) - r_h^{(t)} - Q^\star_{\mathrm{lat},h}(s_{h+1}^{(t)})\right)^2}_{:=W_t(h,g)} \geq -\beta/2,$$

(101)

and also that

$$\sum_{t<k} \underbrace{\left(\widetilde{\mathcal{T}}_\phi^{\pi^{(t)}} Q^\star_{\mathrm{lat},h}(s_h^{(t)}, a_h^{(t)}) - r_h^{(t)} - Q^\star_{\mathrm{lat},h}(s_{h+1}^{(t)})\right)^2 - \left(Q^\star_{\mathrm{lat},h}(s_h^{(t)}, a_h^{(t)}) - r_h^{(t)} - Q^\star_{\mathrm{lat},h}(s_{h+1}^{(t)})\right)^2}_{:=V_t(h)} \geq -\beta/2.$$

(102)

For Eq. (101), note that

$$\mathbb{E}[W_t(h,g) \mid \mathcal{F}_{t,h}] = \widetilde{\mathbb{E}}_\phi^{\pi^{(t)}} \left[\left(g_h(s_h, a_h) - \widetilde{\mathcal{T}}_{\phi,h}^{\pi^{(t)}} Q^\star_{\mathrm{lat},h}(s_h, a_h)\right)^2\right],$$

(103)

and that $\mathsf{Var}[W_t(h,g) \mid \mathcal{F}_{t,h}] \leq 16\,\mathbb{E}[W_t(h,g) \mid \mathcal{F}_{t,h}]$. By Freedman, this gives

$$\left|\sum_{t<k} W_t(h,g) - \sum_{t<k} \mathbb{E}[W_t(h,g) \mid \mathfrak{F}_{t,h}]\right| \leq \mathcal{O}\left(\sqrt{\iota \sum_{t<k} \mathbb{E}[W_t(h,g) \mid \mathcal{F}_{t,h}]} + \iota\right)$$

$$\leq \frac{1}{2}\,\mathbb{E}[W_t(h,g) \mid \mathcal{F}_{t,h}] + \mathcal{O}(\iota),$$

or in other words

$$\sum_{t<k} W_t(h,g) \geq \frac{1}{2} \sum_{t<k} \mathbb{E}[W_t(h,g) \mid \mathfrak{F}_{t,h}] - \mathcal{O}(\iota) \geq -\mathcal{O}(\iota),$$

using the non-negativity of Eq. (103). For Eq. (102), note that

$$\mathbb{E}[V_t(h) \mid \mathfrak{F}_{t,h}] = -\widetilde{\mathbb{E}}_\phi^{\pi^{(t)}} \left[\left(\mathcal{T}_{\mathrm{lat},h} Q^\star_{\mathrm{lat},h} - \widetilde{\mathcal{T}}_{\phi,h}^{\pi^{(t)}} Q^\star_{\mathrm{lat},h}\right)^2\right] \geq -\varepsilon_{\mathrm{rep}},$$

(104)

and that $\mathsf{Var}[V_t(h) \mid \mathfrak{F}_{t,h}] \leq 16\widetilde{\mathbb{E}}_\phi^{\pi^{(t)}} \left[\left(\mathcal{T}_{\mathrm{lat},h} Q^\star_{\mathrm{lat},h} - \widetilde{\mathcal{T}}_{\phi,h}^{\pi^{(t)}} Q^\star_{\mathrm{lat},h}\right)^2\right]$. By Freedman, this gives

$$\left|\sum_{t<k} V_t(h) - \sum_{t<k} \mathbb{E}[V_t(h) \mid \mathfrak{F}_{t,h}]\right| \tag{105}$$

$$\leq \mathcal{O}\left(\sqrt{\iota \sum_{t<k} \widetilde{\mathbb{E}}_\phi^{\pi^{(t)}} \left[\left(\mathcal{T}_{\mathrm{lat}} Q^\star_{\mathrm{lat},h+1}(s_h, a_h) - \widetilde{\mathcal{T}}_{\phi,h}^{\pi^{(t)}} Q^\star_{\mathrm{lat},h+1}(s_h, a_h)\right)^2\right]} + \iota\right) \tag{106}$$

$$= \mathcal{O}(\varepsilon_{\mathrm{rep}} + \iota), \tag{107}$$

or in other words

$$\sum_{t<k} V_t(h) \geq \sum_{t<k} \mathbb{E}[V_t(h) \mid \mathfrak{F}_{t,h}] - \mathcal{O}(\varepsilon_{\mathrm{rep}} + \iota) \geq -\mathcal{O}(\varepsilon_{\mathrm{rep}} + \iota),$$

where we have used Eq. (104).

$$\square$$

$$\square$$

