# OpenReview forum: "Reinforcement Learning Under Latent Dynamics: Toward Statistical and Algorithmic Modularity"
_NeurIPS.cc/2024/Conference — NeurIPS 2024 oral_

### Official Review · Reviewer_qRgr · 2024-07-12

**Soundness:** 4
**Presentation:** 2
**Contribution:** 4
**Rating:** 6
**Confidence:** 3

**Summary:**

This work investigates representation learning for decision-making in episodic MDPs. They consider a problem setting in which the goal is to through interactive decision-making learn a good policy. The task involves learning a good "decoder", i.e., function that maps the observations to "state" representations. They define a notion of "statistical modularity" in this problem which means that there exists an algorithm that can learn the optimal policy (with $\epsilon$ error) with high probability with a number of episodes that is polynomial in the base MDP and the capacity of the decoder function class. They then prove an impossibility result regarding statistical modularity in this problem in general and prove statistical modularity in MDPs under some condition. They also define a notion of "algorithmic modularity" by introducing an algorithm in a hindsight observability setting in which one can use any decoder of interest and any standard episodic MDP algorithm for decision-making; they prove a regret bound in terms of the quality of the quality of the base MDP algorithm and the quality of the decoder.

**Strengths:**

- I thought the problem the authors worked on was interesting and meaningful. Overall the ideas of introducing the concepts of statistical and algorithmic modularity was interesting.
- The authors provide many theoretical results and the result, especially regarding the algorithmic modularity result in 4.1 seems interesting, intuitive, and rather elegant.

**Weaknesses:**

- I found the notation and terminology used in this paper to be very dense. (I put specific questions / notes about this in the next section). I think I would prefer that the authors to have fewer theoretical results, but more thorough discussion of the results. (E.g., the self-predictive estimation idea seems interesting, but is literally 1 paragraph in the paper.)

**Questions:**

- Could you clarify if you are using the term "latent states" in the sense of partially observed MDPs. Or if you really mean the state is "latent" in the sense that the best representation of the state is unknown? If this is the case, it seems like this is more a representation learning problem rather than a "latent state" problem. Could you add further discussion of how your formulation relates to POMDPs?
- It would be helpful to provide an intuitive definition of decoder earlier in the paper, as it is used in the intro without much context.

**Limitations:**

- There is no empirical evaluation of the algorithm.

---

> ### Author Rebuttal · Authors · 2024-08-05
>
> We thank the reviewer for taking the time to read our paper and for their positive review. We address questions/weaknesses below.
>
> > I found the notation and terminology used in this paper to be very dense. [...]I think I would prefer that the authors to have fewer theoretical results, but more thorough discussion of the results. (E.g., the self-predictive estimation idea seems interesting, but is literally 1 paragraph in the paper.)
>
> We apologize that the reviewer found the paper to be dense, and will strive to revise the text to improve readability. Towards this, we will happily accept any specific recommendations that the reviewer has for which content should be emphasized. In addition, we are happy to use the extra page available for the camera-ready version to expand the discussion around topics like self-predictive estimation.
> > Could you clarify if you are using the term "latent states" in the sense of partially observed MDPs. Or if you really mean the state is "latent" in the sense that the best representation of the state is unknown? If this is the case, it seems like this is more a representation learning problem rather than a "latent state" problem. Could you add further discussion of how your formulation relates to POMDPs?
>
> It is best to think of our use of the term “latent state” as in the sense of the second definition you mention (finding the best representation which remains unknown). However, our use of the term “latent state” is consistent with both of the definitions you mention—in fact, they are the same under the decodability assumption we consider. In detail, our problem formulation studies a restricted class of POMDPs where the emission processes are assumed to be decodable (Definition 2.1 and 2.2). This means that the dynamics are governed by the latent state (which is unobserved, as in POMDPs), but it also means that there exists a representation which can decode the unknown latent state. The decodability assumption also removes any partial observability issues. Thus, we are in the representation learning problem, where the aim is to recover the underlying latent state (of course, as we discuss in the paper, representation learning and exploration must be interleaved in our setting). We are happy to add more discussion to emphasize how our formulation relates to POMDPs, and we thank the author for this suggestion.
>
> >It would be helpful to provide an intuitive definition of decoder earlier in the paper, as it is used in the intro without much context.
>
> We agree that this would be helpful, and thank the reviewer for the suggestion. We will revise the introduction to include a more intuitive explanation.

---

> ### Comment · Reviewer_qRgr · 2024-08-08
>
> Thank you for your comments. Your proposed writing revisions sound good and I think they would improve the presentation!

---

### Official Review · Reviewer_669T · 2024-07-15

**Soundness:** 3
**Presentation:** 3
**Contribution:** 3
**Rating:** 7
**Confidence:** 1

**Summary:**

This paper provides a theoretical analysis of statistical and algorithmic modularity for RL with latent dynamics. Specifically, it offers conditions and theoretical analysis under which RL with latents is tractable. For statistical modularity, both lower and upper bounds are presented. For algorithmic modularity, observation-to-latent reductions are analyzed under two conditions: hindsight observability and self-predictive estimation. Overall, the theory and proofs are technically solid, addressing a critical problem in RL, especially in scenarios where only high-dimensional pixels are observed. Although I am not an expert in RL theory (my focus is more on algorithms and applications), I would give an acceptance rating for this initial review and will be engaged in the discussion.

**Strengths:**

- [**Motivation and Significance**]: The problem of learning from observation for RL is important, and this paper provides fundamental theory on this topic. The statistical and algorithmic guarantees are critical contributions to the field. The theoretical findings, particularly on algorithmic modularity, have the potential to encourage more empirical work on efficiently identifying self-predictive latent states that facilitate RL policy learning.
- [**Technical Soundness**]: Although I am not an expert in RL theory, I reviewed the main paper thoroughly and found the theoretical foundations and proofs to be solid.
- [**Presentation**]: The presentation is clear and accessible, even for readers outside the theory domain.

**Weaknesses:**

Since I am not an expert on RL theory, I have listed most of my questions in this section. The major question from an empirical point of view is how to leverage some of these theoretical results to enhance RL learning from latent dynamics.

Q1: The authors mentioned that block MDP or factored MDP would be a special case of this general framework. Suppose we narrow down the problem to block MDP or factored MDP, can the statistical or algorithmic modularity be easier to achieve?

Q2: Similar to the previous question, what kind of structure (e.g., symmetry, disentanglement) or distribution assumptions in the latent space could mostly benefit the current theoretical framework?

**Questions:**

I listed my questions in the above section.

**Limitations:**

Limitations and discussions are given in the paper. As this is a theoretical work, I do not think it will pose any negative societal impact.

---

> ### Author Rebuttal · Authors · 2024-08-05
>
> We thank the reviewer for their positive review and their thoughtful questions. We address each of the individual questions below.
>
> > The authors mentioned that block MDP or factored MDP would be a special case of this general framework. Suppose we narrow down the problem to block MDP or factored MDP, can the statistical or algorithmic modularity be easier to achieve?
>
> Modularity is indeed easier to achieve for Block MDPs (note that Block MDPs correspond to the case where  $\mathcal{M}_{\mathrm{lat}}$ is tabular, and we indicate that this setting is modular in Figure 1). In particular, for statistical modularity, there are many prior algorithms which achieve the desired sample complexity of $\mathrm{poly}(S,A,H,\log\Phi)$ [Zhang et al., 2022, Mhammedi et al., 2023], which is statistically modular by our definition. Regarding factored MDPs, statistical modularity can be achieved under additional assumptions on the emission process [Misra et al., 21], but the general case remains an interesting open question.
> As for algorithmic modularity, no prior works had studied this desiderata. However our reduction based on self-predictive representation learning (Theorem A.1) can be applied in the tabular (Block MDP) setting to achieve algorithmic modularity, as all the assumptions required by the self-predictive representation learning oracle are satisfied when the latent state space is tabular.
>
> > Similar to the previous question, what kind of structure (e.g., symmetry, disentanglement) or distribution assumptions in the latent space could mostly benefit the current theoretical framework?
>
> We agree that this is an interesting question, and have tackled it in the paper – for example, we have identified that latent pushforward coverability is a general structural condition on the latent space which allows for statistical and algorithmic modularity (this subsumes, for example, the block MDP and latent low-rank MDP results). However, we do not yet have a complete picture of which latent structures or additional parameters are necessary and sufficient, and have posed this as an open question in the conclusion of the paper. We view the introduction of this question, along with partial steps towards addressing it, as one of our main contributions.
>
> **References**
>
> 1. Zhang X, Song Y, Uehara M, Wang M, Agarwal A, Sun W. Efficient reinforcement learning in block mdps: A model-free representation learning approach. InInternational Conference on Machine Learning 2022 Jun 28 (pp. 26517-26547). PMLR.
>
> 2. Mhammedi Z, Foster DJ, Rakhlin A. Representation learning with multi-step inverse kinematics: An efficient and optimal approach to rich-observation rl. InInternational Conference on Machine Learning 2023 Jul 3 (pp. 24659-24700). PMLR.
>
> 3. Misra D, Liu Q, Jin C, Langford J. Provable rich observation reinforcement learning with combinatorial latent states. InInternational Conference on Learning Representations 2021.

---

> > ### Comment · Reviewer_669T · 2024-08-13
> >
> > Thank you for the detailed response. My concerns have been well addressed and I would keep my rating.

---

### Official Review · Reviewer_U7JX · 2024-07-18

**Soundness:** 4
**Presentation:** 3
**Contribution:** 3
**Rating:** 7
**Confidence:** 3

**Summary:**

This paper considers theoretical aspects of reinforcement learning in a certain class of MDPs whose observations are governed by a separate, potentially smaller, MDP. They formalize this class of MDPs and denote them latent MDPs. The authors then consider when such MDPs are statistically learnable, beginning with a negative result: they show that in general, even with known latent dynamics, statistical modularity is impossible. They then highlight that statistical modularity in this setting is in some sense distinct from previous works which assume regularity in the value function, and mention that this is because this structure might be useless without a good learnt representation. The authors then go through a laundry list of MDP formalisms in previous work, and provide for most a result on whether or not they are statistically modular. They finally consider algorithmic results, and introduce a 'meta-algorithm' which balances representation and RL learning (where the underlying RL algorithm is arbitrary). Under some additional assumptions, they prove that the additional representation learning adds sublinear risk.

**Strengths:**

- I believe that this is an important step in bridging RL theory and the issues of RL in practice.
- Balancing representation learning and standard RL learning is an important issue many RL practitioners need to balance. This work paves the way for theoretically-guided answers to those questions.
- I found it rather interesting how the authors demonstrated that much of the structure used in previous work is not amenable to this setting, and that in those cases statistical modularity is not possible.

**Weaknesses:**

- Assuming that latent states can be uniquely decoded from the observations is a rather strong observation.
- There are no experiments in the paper. Of course the contribution of this work is theoretical, but theoretical work can still benefit greatly from some simple experiments which illustrate results in their paper. In particular, doing this allows some readers to better understand the result, and importantly, shows that the results obtained (which are often under unrealistic assumptions) do not break down in practice.

**Questions:**

- Do you believe there are any toy experiments which can be done to illustrate any of your results?

**Limitations:**

They discuss avenues for future work, and are clear on the limitations of their work.

---

> ### Author Rebuttal · Authors · 2024-08-05
>
> We thank the reviewer for their positive review and their helpful comments! Please see responses to individual questions below.
>
>  > Assuming that latent states can be uniquely decoded from the observations is a rather strong observation.
>
> It is true that this imposes stronger assumptions on the observation-space MDP (i.e., there is no partial observability). However, this assumption is well-established in the line of research on Block MDPs and RL with rich observations and permits the design of computationally/statistically efficient algorithms in various cases (e.g. tabular latent MDPs), whereas the analogous POMDP setting would otherwise be intractable (see, e.g., the lower bound in [Krishnamurthy et al., 2016]). Our work addresses the question of generalizing the aforementioned positive results to general latent dynamics, which had remained largely unaddressed, and one of our main contributions is to show that, despite the seemingly nice structure of decodability, strong *negative results* are present. These also imply negative results for the setting *without* decodability. Thus, for many interesting classes of latent dynamics, one cannot hope to remove even this decodability assumption (without placing alternative assumptions). Nonetheless, we hope that by addressing the decodable setting, our work can serve as a starting point toward building a similar understanding for partially observed settings.
>
> > There are no experiments in the paper. Of course the contribution of this work is theoretical, but theoretical work can still benefit greatly from some simple experiments which illustrate results in their paper. [...] Do you believe there are any toy experiments which can be done to illustrate any of your results?
>
> We acknowledge that experiments are an important next step for our results. However, let us emphasize that we believe our theoretical contributions alone are sufficient for publication, and stand on their own merits. Indeed, as is typically the case with theoretically motivated algorithms, developing practical implementations will require non-trivial adaptations and significant implementation effort; given the scope of our theoretical results, we believe it is appropriate to leave a full-scale empirical evaluation for future work.
>
> Regarding toy experiments: a classical toy experiment considered in prior works (for the latent tabular setting) is the “diabolical combination lock”  [Misra et al. ‘20, Zhang et al. ‘22, Mhammedi et al. ‘23], which consists of a small latent combination lock with very high-dimensional observations and which traditional deep RL algorithms fail to solve. For future experiments, since our representation learning oracle allow for sample-efficiency under much more complicated latent dynamics (beyond tabular), it would be interesting to design and test our algorithms on a more complicated version of this domain, which would be unsolvable by both deep RL algorithms as well as prior theoretical latent-dynamics algorithms.
>
> **Refs**
>
> 1. Krishnamurthy A, Agarwal A, Langford J. Pac reinforcement learning with rich observations. Advances in Neural Information Processing Systems. 2016;29.
>
> 2. Misra D, Henaff M, Krishnamurthy A, Langford J. Kinematic state abstraction and provably efficient rich-observation reinforcement learning. InInternational conference on machine learning 2020 Nov 21 (pp. 6961-6971). PMLR.
>
> 3. Zhang X, Song Y, Uehara M, Wang M, Agarwal A, Sun W. Efficient reinforcement learning in block mdps: A model-free representation learning approach. InInternational Conference on Machine Learning 2022 Jun 28 (pp. 26517-26547). PMLR.
>
> 4. Mhammedi Z, Foster DJ, Rakhlin A. Representation learning with multi-step inverse kinematics: An efficient and optimal approach to rich-observation rl. InInternational Conference on Machine Learning 2023 Jul 3 (pp. 24659-24700). PMLR.

---

### Decision · Program_Chairs · 2024-09-25

**Decision:**

Accept (oral)

**Comment:**

This manuscript focused on the function approximation under the context of the reinforcement learning with latent dynamics. The authors discussed on the statistical requirements and algorithmic principals of learning within such scenarios, specifically on the modularities. Here modularities describe if we can decouple the representation learning steps and reinforcement learning under latent dynamics steps from both a statistical and algorithmic perspectives. Statistically, the authors show that if we don’t have a stronger complexity notions (e.g. pushforward coverability in this manuscript), then even we know the exact latent dynamics, we still cannot benefit from it. Algorithmically, the authors show that as long as we have a realizable decoder function class and we have a low-regret reinforcement learning algorithm for the latent MDP, we can have a generic observation-to-latent conversion to decouple the representation learning and reinforcement learning. All reviewers agree this manuscript can be a milestone on bridging the theory and practice of reinforcement learning. I would personally suggest the authors add a discussion on pushforward coverability, especially how it scale on the lower bound instance, to provide more intuition on the statistical modularity.